# TRANSFORMERS GET STABLE: AN END-TO-END SIGNAL PROPAGATION THEORY FOR LANGUAGE MODELS

## ABSTRACT

In spite of their huge success, transformer models remain difficult to scale in depth. In this work, we provide formulae that govern the moments of the forward and backward signal through all transformer components, and develop a unified signal propagation theory for transformers. Our framework can be used to understand and mitigate vanishing/exploding gradients, rank collapse, and instability associated with high attention scores. We also propose DeepScaleLM, an initialization and scaling scheme that conserves unit output/gradient moments throughout the model, enabling the training of very deep models with 100s of layers. We find that transformer models could be much deeper – our deep models improve 1.0 points in perplexity, and 2.2 points in downstream tasks compared to shallow models across multiple model sizes, without any extra parameters, and even outperform larger shallow models using only half the number of parameters.

## 1 INTRODUCTION

Transformer models have become extremely popular across different domains of machine learning. However, deep transformers are plagued with issues of gradient explosion/vanishing (Shleifer et al., 2021; Takase et al., 2022), and of rank collapse (Shi et al., 2022; Zhou et al., 2021; Noci et al., 2022) that adversely affect training stability. Several remedies have been proposed to alleviate these issues, such as changing the location of layernorm to stabilize model training (Xiong et al., 2020), adding extra layernorms after attention (Dehghani et al., 2023), initializing weights as zero (Bachlechner et al., 2020), and changing the scaling of residuals (Wang et al., 2022a; Zhang et al., 2019).

Theoretical analysis of deep transformers, via signal propagation, kernel methods, etc. has led to an improved understanding of these issues. However, these works have inherent simplistic assumptions such as IID inputs, uncorrelated outputs, assuming no effect of attention query/key initialization, simplified treatment of effects of non-linearity, etc (Xu et al., 2019; Davis et al., 2021; Dong et al., 2023). We observed that each one of these assumptions breaks down in a model with real world data, adversely affecting model stability.

Furthermore, these modifications may lead to unintended degradation caused by side effects. For example, changing the position of the Layernorm in Pre-LN can lead to gradient mismatch (Shleifer et al., 2021). Moreover, the optimal initialization/scaling can vary based on data/model characteristics (Zhang et al., 2023; Marion et al., 2022). Detailed discussion of related works is provided in Appendix B. These issues highlight the need for a holistic theoretical framework that can fully explain signal propagation through transformer models with real data, and enable comparative analysis along the stability-performance trade-off curve.

In this paper, we first derive the first and second-order moments (mean and variance) of the outputs and gradients of each of the components of the transformer model – Embeddings, FFN, ReLU/GeLU, LayerNorm, Dropout, Softmax, Single-Head Attention, and provide complete closed-form expressions for the same. Combining these expressions enables us to derive the equations that describe the forward and backward signal flow through the transformer Attention and FFN blocks, and through the entire transformer model for both Pre-LN and Post-LN variants. We validate our theory by empirically verifying the derived equations.

Our theoretical framework can be used to understand multiple training instability issues with very deep transformer – specifically, it demonstrates the vanishing/exploding gradient of deep transform-

ers, rank collapse, and instability caused by high QK values of attention. It can pinpoint the root cause behind these issues, and suggests simple fixes to alleviate them.

Montúfar et al. (2014); Raghu et al. (2017) show that the complexity of Deep Neural Nets increases polynomially with width and exponentially with depth. However, deep transformers are often unstable due to the above mentioned issues Smith et al. (2022); Chowdhery et al. (2022). Our theory further enables us to propose DeepScaleLM, a novel initialization scheme that augments residual/output scaling, and ensures the moments of outputs and gradients to remain fully conserved throughout the model at initialization. The theory also allows practitioners to tune the model stability based on their requirements, at the possible expense of model performance.

DSLM enables us to break the depth barrier and train both Pre-LN and Post-LN models with 100s of layers. We find that transformer models could be much deeper – our 192-layer model outperforms a standard 12-layer model without extra parameters or compute. Similarly, 96 and 384-layer models outperform the 24-layer BERT-large model. Deep DSLM-models are able to beat models with twice the number of parameters. These improvements persist after fine-tuning for downstream tasks.

Our contributions are as follows –

1. We derive and verify the moments for signal propagation through the transformer components, blocks, and the entire model. To the best our knowledge, this is the first work to provide closed-forms for many of these components, for the blocks, and the entire model.

2. We leverage our formulae to tackle vanishing/exploding gradients and outputs, rank collapse, and instability caused by high QK values all under one framework.

3. We propose DeepScaleLM, an initialization and scaling scheme that conserves unit output/gradient throughout the model at initialization, and bounds gradients during training.

4. Our experiments suggest that transformer models could be much deeper (100s of layers) for improved performance with the same number of parameters and compute.

## 2 MOMENTS OF TRANSFORMER MODELS

Table 1: Signal propagation for forward ($\sigma^2_{x_{\text{out}}}$) and backward ($\sigma^2_{g_{\text{in}}}$) passes through components of a transformer. The expressions here are illustrative simplification of full closed form formulae in Appendices C and E.

| Component | $\mu_{\mathbf{x_{out}}}$ | $\sigma^2_{\mathbf{x_{out}}}$ | $\sigma^2_{\mathbf{g_{in}}}$ | $\mathbf{r^l_{x_{out}}}$ | $\mathbf{r^l_{g_{in}}}$ |
|---|---|---|---|---|---|
| Embeddings | 0 | $\sum \sigma^2_{w_{\text{embd}}}$ | - | $\frac{\pi^2}{18*\log(|V|)^2} + \frac{2}{9}$ | 0 |
| Linear ($d_{\text{in}} \rightarrow d_{\text{out}}$) | 0 | $d_{\text{in}}\sigma^2_w(\sigma^2_{x_{\text{in}}} + \mu^2_{x_{\text{in}}})$ | $d_{\text{out}}\sigma^2_w\sigma^2_{g_{\text{out}}}$ | $\frac{r^l_{x_{\text{in}}} + \mu^2_{x_{\text{in}}}/\sigma^2_{x_{\text{in}}}}{1 + \mu^2_{x_{\text{in}}}/\sigma^2_{x_{\text{in}}}}$ | $r^l_{g_{out}}$ |
| ReLU | $\frac{\sigma_{x_{\text{in}}}}{\sqrt{(2\pi)}}$ | $\frac{(\pi-1)}{(2\pi)}\sigma^2_{x_{\text{in}}}$ | $\frac{1}{2}\sigma^2_{g_{\text{out}}}$ | $0.7r^l_{x_{\text{in}}} + 0.3{r^l_{x_{\text{in}}}}^2$ | $(\frac{1}{2} + \frac{\sin^{-1}(r^l_{x_{\text{in}}})}{\pi})\text{r}^l_{g_{\text{out}}}$ |
| LayerNorm ($d$) | 0 | 1 | $\frac{\sigma^2_{g_{\text{out}}}}{\sigma^2_{x_{\text{in}}}}$ | $r^l_{x_{\text{in}}}$ | $r^l_{g_{\text{out}}}$ |
| Dropout ($p$) | $\mu_{x_{in}}$ | $\frac{\sigma^2_{x_{\text{in}}} + p\mu^2_{x_{\text{in}}}}{1-p}$ | $\frac{1}{1-p}\sigma^2_{g_{\text{out}}}$ | $\frac{r^l_{x_{\text{in}}}(1-p)}{1 + p\mu^2_{x_{\text{in}}}/\sigma^2_{x_{\text{in}}}}$ | $(1-p)r^l_{g_{\text{out}}}$ |
| SHA-without V | 0 | $r^l_{x_{\text{in}}}\sigma^2_{x_{\text{in}}}$ | $r^l_{g_{\text{out}}}\sigma^2_{g_{\text{out}}}$ | 1 | 1 |
| Softmax | $\frac{1}{L}$ | $\frac{e^{(1-r^d_{x_{\text{in}}})\sigma^2_{x_{\text{in}}}} - 1}{L^2}$ | $\frac{e^{(1-r^d_{x_{\text{in}}})\sigma^2_{x_{\text{in}}}}}{L^2}\sigma^2_{g_{\text{out}}}$ | - | - |

**Moments of Transformer Components** Following an analysis similar to that of Xavier/Glorot initialization (Glorot & Bengio, 2010), we derive closed-form expressions for the mean and variance

of the output and of the backpropagated gradient for all the components of the transformer model in Table 1 – Embeddings, Linear, ReLU (GeLU in supplementary), LayerNorm, Dropout, Softmax and Single-Head Attention(SHA).

Here $\mu_{x_{\text{in}}}$, $\sigma^2_{x_{\text{in}}}$, $\mu_{x_{\text{out}}}$, $\sigma^2_{x_{\text{out}}}$ are the mean and variance of the input/outputs respectively. $\sigma^2_{g_{\text{out}}}$, $\sigma^2_{g_{\text{in}}}$ are the variance of the gradient back-propagated to and from the component, respectively. $r^l$, $r^d$ are the correlations across sequence length and hidden dimension, respectively. $p$ is the dropout probability, $L$ sequence length, $d_{\text{in}}, d_{\text{out}}$ input/output dimensions of Linear layer, $\sigma^2_w$, $\sigma^2_{w_{\text{embd}}}$ variances of the weights of the Linear layer and the Embeddings table respectively.

Except for SHA, all other derivations of transformer components are fully exact, assuming only normal distribution of inputs, weights and gradients. For LayerNorm and softmax, we assume that the hidden dimension / sequence length are large. Detailed proofs are provided in Appendix C, and all assumptions are also summarized in Appendix L.2. These formulae were also numerically verified by simulations (Section 4.1), and the verification shows that our expectations are tight.

**Moments of Transformer Blocks**  Combining the expressions reported in Table 1, we derive closed-form expressions for the moment transformation during the forward and backward pass of the transformer Attention and FFN blocks. The Attention block refers to the $Q, K, V$ projection, followed by Multi-Head Attention, and Output-Projection Layer. The FFN block refers to the Linear layer, followed by non-linearity (ReLU), and output Linear layer. Table 2 provides our derived equations for these, where $\sigma^2_v$, $\sigma^2_o$, $\sigma^2_{w_1}$, $\sigma^2_{w_2}$ are the variances for $V$ weights, Output-Projection weights, and weights of FFN block Linear layers, and $d$ is model the hidden size.

These results show that considering correlation $r^l$, dropout $p$ and effects of non-linearity are crucial for correctly modelling signal propagation through Transformer blocks. $r^l_{x_{\text{in}}}$ originates from repeated tokens in the input, segment embeddings, and transformer layers. The correlation in embeddings is estimated theoretically by assuming that input tokens follow Zipf (1999) distribution. The equations in Table 2 are simplified from their complete closed form, which can be found in Appendix D.

Table 2: Moment Propagation through the blocks of a transformer layer. Exact closed forms and proofs for these equations are in Appendices D and E.

| Component | $\sigma^2_{\mathbf{x_{out}}}$ | $\mathbf{r}^l_{\mathbf{x_{out}}}$ | $\sigma^2_{\mathbf{g_{in}}}$ |
|---|---|---|---|
| Attention Block | $\dfrac{d^2 \sigma^2_o \sigma^2_v \sigma^2_{x_{\text{in}}} * r^l_{x_{\text{in}}}}{(1-p)}$ | $1 - p$ | $\dfrac{d^2 \sigma^2_o \sigma^2_v * \sigma^2_{g_{\text{out}}}}{L(1-p)}(1 + (L-1)r^l_{g_{\text{out}}})$ |
| FFN Block | $\dfrac{2d^2 \sigma^2_{w_1} \sigma^2_{w_2} \sigma^2_{x_{\text{in}}}}{(1-p)}$ | $(1-p)(\dfrac{1}{\pi} + \dfrac{r^l_{x_{\text{in}}}}{2} + (\dfrac{1}{2} - \dfrac{1}{\pi}){r^l_{x_{\text{in}}}}^2)$ | $\sigma^2_{x_{\text{out}}} * \sigma^2_{g_{\text{out}}}$ |

**Moments of Entire Transformer Model**  By repeatedly applying the expressions in Table 2 for each layer, we calculate the propagation of moments of outputs and gradients through the entire transformer model. We do this for both Pre-LN style transformers, in which the skip connection bypasses the LayerNorm, and for Post-LN style transformers, in which the Layernorm is applied before the skip-connection. The method is fully detailed in Appendix H.1 for Pre-LN and in Appendix H.2 for Post-LN. Our derived moments have remarkably low error even after 192 layers, for both the forward output and backward gradient, as can be seen from Figures 1, 2 and 3 for a 192-layer 256-d model at initialization, initialized with Xavier initialization. These formulae were also verified by simulations, as detailed in Section 4.1.

## 3 APPLICATIONS

### 3.1 EXPLAINING VARIANCE EXPLOSION IN VANILLA TRANSFORMER

Our approach theoretically proves the gradient vanishing/explosion with increasing number of layers, for both Pre-LN and Post-LN transformers, as listed in Table 3.

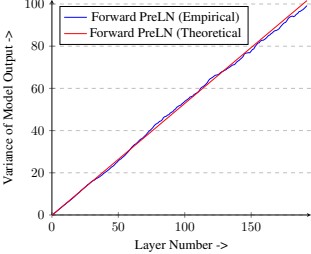

Figure 1: Pre-LN: Forward variance increases linearly with number of layers $N$.

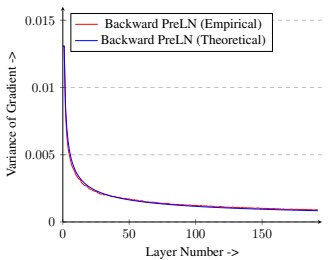

Figure 2: Pre-LN: Backward gradient variance increases hyperbolically with $N$.

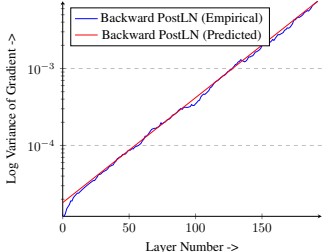

Figure 3: Post-LN: Backward gradient variances vanish exponentially with $N$ (log-scale).

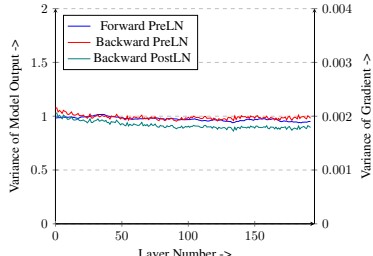

Figure 4: DeepScaleLM: The variances remain conserved for both backward and forwards.

**Exploding Output and Gradient in Pre-LN**    As we prove in Appendix H.1, with increasing depth $N$, the forward output increases linearly for Pre-LN transformer since each layer's output is directly added to the skip connection, as can be seen in Figure 1. For the backward pass, the gradient increases hyperbolically with increasing $N$, as shown in as can be seen in Figure 2. Intuitively, this is because the gradient increases in every layer when a block's gradient is added to the skip connection, and the fractional increase in gradient is inversely proportional to the forward variance (which increases by $N$) because of LayerNorm.

**Vanishing/Exploding Gradient in Post-LN**    While layernorm solves the explosion in the forward pass of networks with residual connections (De & Smith, 2020), it has the opposite impact on the gradient. As proved in Appendix H.2, the moment for the gradient in a Post-LN transformer grows/decays exponentially with the number of layers ( Figure 3).

Intuitively, the gradient is first transformed within the layer and then at the LayerNorm placed before the layer. The multiplicative factor is applied repeatedly, and causes gradient to vanish or explode exponentially, as was also observed in Schoenholz et al. (2017). This exponential decay/explosion also explains why Post-LN models are more challenging to train than Pre-LN for deeper networks (Wang et al., 2022a; Shleifer et al., 2021; Takase et al., 2022).

Table 3: Comparison of maximum theoretical forwards pass and backward pass growth in variance for the entire transformer model across methods. (See Appendix H for proofs)

| Method | Post-LN | | | Pre-LN | | |
|---|---|---|---|---|---|---|
| | **Forward** | **Backward** | **Sensitivity** | **Forward** | **Backward** | **Sensitivity** |
| Vanilla (Xavier/Fixed init) | 1 | $\mathcal{O}(c^{\pm N})$ | $\boldsymbol{\mathcal{O}(N)}$ | $\mathcal{O}(N)$ | $\mathcal{O}(N)$ | $\boldsymbol{\mathcal{O}(log(N))}$ |
| DeepScaleLM (Ours) | 1 | $\boldsymbol{\mathcal{O}(1)}$ | $\mathcal{O}(1)$ | **1** | $\boldsymbol{\mathcal{O}(1)}$ | $\mathcal{O}(1)$ |
| DSInit | 1 | $\mathcal{O}(1)$ | $\mathcal{O}(N^{-1})$ | $\mathcal{O}(1)$ | $\mathcal{O}(1)$ | $\mathcal{O}(N^{-1})$ |
| DeepNet | 1 | $\mathcal{O}(1)$ | $\mathcal{O}(N^{-0.5})$ | - | - | - |

## 3.2 DEEPSCALELM: ENABLING DEEP TRANSFORMERS WITH 100S OF LAYERS

We propose a new initialization and re-scaling scheme, DeepScaleLM (DSLM), that aims to alleviate the explosion issues discussed above.

**Residual/Skip-Connection Scaling**   Let $\sigma_{\text{skip}}^2$, $\sigma_{\text{block}}^2$, $\sigma_{\text{model}}^2$ be the variances of the skip connection, the block, and the output of the final layer of the model, respectively. Let $\sigma_{\text{skip}}^2 = \sigma_{\text{block}}^2$, and we scale them by scalars $\lambda$ and $\beta$ respectively. Then, as has been proven in numerous works (Appendix B.3), if $\lambda^2 + \beta^2 = 1$, this scaling will maintain the variance after addition of the residual.

**Initialization**   However while ensuring $\sigma_{\text{skip}}^2 = \sigma_{\text{block}}^2$ (and equal to the variance of model input) has been done for ResNets (Appendix B.1), it is difficult to achieve theoretically for transformers. By leveraging the equations in Table 2, our theory provides us the tools to achieve this. We modify the initialization of the components of the transformer FFN and Attention blocks such that the variance of their output is 1, as further detailed in Appendix M –

1. We set the variance of embedding weights as $\sigma_e^2 = \frac{1-p}{num_{\text{embd}}}$, where $num_{\text{embd}}$ is the number of embeddings types. As embeddings are followed by a dropout, this ensures the input variance to the model is 1.

2. We set $\sigma_{w_2}^2 = \sigma_{w_1}^2 = \frac{1}{d} * \sqrt{\frac{1-p}{2}}$, to make the output of the FFN block 1.

3. We iteratively calculate layer-by-layer $r_{x_{\text{in}}}^l$, $r_{x_{\text{out}}}^l$ using expressions from Table 2, and calculate the initial variance of the attention block weights to make the output variance 1.

This initialization of transformer blocks, combined with the scaling of the skip connection and residual, and correct initialization of the embeddings, results $\sigma_{\text{model}}^2 = 1$, irrespective of the number of layer $N$. This initialization also preserves the backward gradient, as proved for Pre-LN and Post-LN, in Appendices H.3 and H.4. Empirically, we show the backward gradient being preserved for both Pre-LN and Post-LN even across 192 layers at the start of training (Figure 4).

**Choice of Scaling Parameters**   While any choice of $\beta$ will work at initialization, higher values of $\beta$, for example $\beta^2 = 0.5$ causes gradients to vanish (Figure 10, Table 4). This is because covariance between residual and skip connection increases the forward variance, which causes normalization to decrease backward gradient (De & Smith, 2020).

Similar to other prior works (Appendix B.3), we use $\beta^2 = \frac{k}{N}$ in all our experiments, where $k$ is some small constant. This enables us to bound the fall in gradient (Appendix H.3) for Pre-LN. For Post-LN, $\beta^2 \leq \frac{k}{N^2}$ is theoretically required to bound the gradient (Appendix H.6). In practice, with $\beta^2 = \frac{2}{N}$, even with 768 layers, we empirically observed the final output variance from the model does not exceed 30, and all our models converge. We hence use $\beta^2 = \frac{k}{N}$ (Figure 11), but a practitioner may choose $\beta^2 = \frac{k}{N^\alpha}$, with $\alpha > 1$ if more stability is required at the expense of performance/"sensitivity" (Refer to Section 4.4 and comparison to prior works in Section 4.3). While the above analysis assumes positive covariance (which we always observed experimentally), negative covariance follows a similar reasoning, and will cause gradient explosion instead.

**Simpler Initialization**   Another avenue to handle the covariance between residual and skip connection could be to set $\lambda^2 + \beta^2 < 1$. We therefore also consider a simpler initialization method(Appendix M), in which we modify the initialization of attention value and output matrices to be the same as those of FFN block. This decreases the "effective" $\beta$ of the attention block, but as the attention block has 2x fewer params than FFN, this change in weightage seems reasonable. As we show in Appendices H.5 and H.6 while variances are no longer unit at initialization, they are still bounded. This change does not impact performance significantly, as we show in Table 10.

With the above initialization and skip/residual scaling, our DSLM method enables us to train models with 768 transformer layers for both Pre-LN and Post-LN. As our experiments will show, we find that for the same number of parameters and compute, deeper-narrower models with our method outperform standard-sized models during both pre-training and finetuning.

### 3.3 EXPLAINING IMPACT OF LARGE QK VALUES

In Dehghani et al. (2023), the authors observed large QK values destabilized the training, and solved this empirically by adding a layernorm after attention scores. Prior work on signal propagation in transformers, either ignored the effect of QK initialization (Wang et al., 2022a) or suggest that the backpropagated gradients have a linear relation to QK variance (Noci et al., 2022). Critically, note from our derivations of softmax(Appendix C.7), the backwards gradients from Q/K are exponentially related to their variance. This exponential dependence points out the critical significance of correct initialization of Q/K. For e.g., by initializing them to only 2x the xavier values (keep all other initializations the same), backwards gradients exploded 10000x through a 192 layer model. Our theory explains these empirical observations of the detrimental impact of large QK values, and suggests simple initialization strategy to fix this problem, achieving the same variance on QK without the overhead of LayerNorm.

### 3.4 EXPLAINING AND MITIGATING RANK COLLAPSE

Similar to our work, Noci et al. (2022) also analyze moment propagation through the transformer, and observed the rank collapse of the token's representations at initialization after just a few layers, i.e., all the token representations became the same ($r_x^l \approx 1$ after just 12 layers) at initialization. This has also been reported in Shi et al. (2022); Zhou et al. (2021); Wang et al. (2022b); He et al. (2023); Bachlechner et al. (2020); Zhai et al. (2023), and suggested modifications such as adding a skip connection on attention scores, initializing Q/other weights to 0, or normalizing all FFN weights.

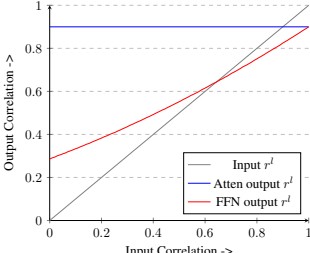

Figure 5: Forward $r^l$ for FFN and Attention blocks with $p = 0.1$. FFN reduces $r^l$ above 0.65, and attention always has $< 1$.

Figure 6: No rank collapse is observed with Xavier initialization with dropout. $r^l$ increases slower with $\beta^2 = \frac{2}{N}$ or for DeepScaleLM.

Our theory suggests a very simple solution – Dropout. As our closed form expressions suggest, both FFN block (because of ReLU) and dropout reduce the correlation(Figure 5). With dropout, our method shows that such a rank collapse will not occur, and $r_x^l$ will quickly reach a stable value $< 1$ (Appendix G), and we verify this empirically in Figure 6.

Alternatively, scaling the block output by $\beta = \frac{1}{\sqrt{N}}$, or equivalently initializing the weights very small in Post-LN will also prevent rank collapse, even without Dropout. For Pre-LN, $\lambda = 1$ slows down increase in $r^l$ compared to $\lambda^2 = 1 - \frac{1}{N}$ (but the same slowdown can be achieved by decreasing $\beta$). While similar to Noci et al. (2022), we highlight some issues in Noci et al. (2022) in Appendix G. For DSLM, applying our block equations iteratively shows that $r_x^l < 1 - \frac{1}{e^2}$ after $N$ layers. This highlights the criticality of correct initialization, dropout and scaling for deep transformer models, as well as the explainability power of our theoretical framework proposed in the paper.

## 4 RESULTS

### 4.1 NUMERICAL VALIDATION OF THEORETICAL RESULTS

We verify the theoretical formulas of transformer components and blocks by running simulations with real and synthetic data, as detailed in Appendix F, over a large range. These simulation results are all fully reproducible using our code released as supplementary material. Even at 99 percentile, no error (other than SHA gradient variance) is larger than 10%, verifying our assumptions.

We also verify our formulae for the entire transformer model, as shown in Figures 1, 2, 3 and 4. Our formulae predict the observed gradient and forward/backward norms with remarkable accuracy. We further vary the model depths $[1 - 768]$, and model dimensions $[128 - 6096]$, and the reported formulae are within $10\%$ error, even across 768 layers of the transformer model.

## 4.2 VALIDITY OF THEORETICAL PREDICTIONS EVEN AFTER TRAINING

Interestingly, our theoretical estimates hold approximately even after the models have been trained for a large number of steps. The model stays in the regime it is initialized with (as has also been shown in Li & Liang (2018); Lee et al. (2019); Jesus et al. (2021); Arora et al. (2019a;b)), highlighting the importance of correct initialization. Further, we analyze forward explosion in a 48-layer PreLN model (after 100k training steps) and gradient explosion in a 64-layer PreLN model (after 150k training steps) and use our theory to predict the moments. Our linear estimation for the forward growth and hyperbolic estimation for the gradient explosion match closely with the observed moments as shown in Appendix in Figures 8 and 9.

## 4.3 DEEPSCALELM: PERPLEXITY IMPROVEMENTS FOR VERY DEEP MODELS

**Implementation Details** We test our method on the Masked Language Modelling task with the BERT Devlin et al. (2019) model. We use Pile-CC dataset Gao et al. (2021) to train our model, and report LM test-set perplexities on the same. We use $k = 2$ for $\beta$, and we use all original hyper-parameters of BERT, except for learning rate (LR). We find that higher LR is needed for our deeper-narrower models (similar to Yang et al. (2021)). Hence, we search for LR for all the models. The training steps were decided based on Chinchilla (Hoffmann et al., 2022). Table 21 provides all hyper-parameter details. When using DSLM, model output was down-scaled by $\sqrt{d}$ before LM-head.

We train different language models with the same number of parameters and compute – while increasing the depth ($N$), we reduce the hidden dimension $d$ keeping number of parameters ($Nd^2$) constant. When changing from 12-layer 1024-d model to 192-layer 256-d model, compute negligibly increases by only $6.6\%$ when keeping $Nd^2$ constant (Table 20).

Table 4: Performance (perplexity) of models with same compute and different shapes. Deep Thin models provide large improvements.

| Model (N,d) | Pre-LN | Post-LN | DSLM |
|---|---|---|---|
| *165M Params* | | | |
| 12, 1024 | 21.7 | 14.2 | 15.6 |
| 48, 512 | 18.0 | 14.8 | 13.1 |
| 192, 256 | 19.8 | 17.1 | **12.9** |
| 768, 128 | 26.9 | diverge | 18.4 |
| *330M Params* | | | |
| 24, 1024 | 19.4 | 13.2 | 14.0 |
| 96, 512 | 24.0 | diverge | **12.2** |
| 384, 256 | 18.5 | diverge | 12.3 |

Table 5: Comparison with prior deep methods.

| Layers | DSInit | DeepNorm | DSLM |
|---|---|---|---|
| 96 | diverge | 13.4 | **12.2** |
| 192 | 15.9 | 14.4 | **12.9** |

Table 6: DSLM with deep Pre-LN.

| Model (N,d) | Baseline | DSLM |
|---|---|---|
| 384, 256 | 18.5 | **17.2** |
| 768, 128 | 26.9 | **25.9** |

**Perplexity Improvements after Pre-Training** In Table 4, we provide the results for two different model sizes, 165M and 330M, with DSLM applied to Post-LN. Post-LN is known to outperform Pre-LN Wang et al. (2022a) (observe Row-1 and Row-4). However, since deeper Post-LN models diverge, most current large LMs (such as GPT3 Brown et al. (2020)) are Pre-LN. Using our method, even a 768 layer Post-LN model (with 2300 Linear and 768 attention layers) converges.

Figure 7 shows that DSLM stabilizes the training of Post-LN models while significantly improving the performance compared to Pre-LN models. Our method is comparable to the baseline for shallow models but starts to outperform as the model gets deeper. Our 192-layer model outperforms the vanilla 12-layer, and our 96 layer outperforms the vanilla 24-layer model. The 165M 192-layer model outperforms the vanilla 24-layer 330M model with $2\times$ params and compute.

**Perplexity Improvements after Pre-training for Pre-LN models**  We also applied DSLM to the deep Pre-LN models reported in Table 4 for both model sizes. Table 7 and Table 6 show that DSLM significantly improves the performance of the Pre-LN model across a range of model depths.

**Comparison with Prior Methods for Deep Transformers**  DSInit and DeepNet stabilize the model training at the expense of reduced "sensitivity" (Section 4.4) by using smaller effective values of $\beta^2$, at $\mathcal{O}(N^{-2})$ and $\mathcal{O}(N^{-1.5})$ respectively. This is also verified by our experiments. Table 6 shows that DSLM outperforms DSInit Zhang et al. (2019) and DeepNet Wang et al. (2022a) across different model depths. Interesting, 96-layer model diverges with DSInit, inspite of DSInit using a smaller $\beta$ asymptotically – this is because the constants hidden in $\mathcal{O}(N^{-2})$ are much larger for DSInit. Our method, by analysing signal propagation, has constants exactly 1.

Table 7: Comparison while increasing depth.

| N | d | Baseline | DSLM |
|---|---|---|---|
| | | *Post-LN* | |
| 192 | 256 | 17.1 | 12.9 |
| 384 | 256 | diverge | **12.3** |
| 48 | 512 | 14.8 | 13.1 |
| 96 | 512 | diverge | **12.2** |
| | | *Pre-LN* | |
| 12 | 512 | 29.4 | 26.0 |
| 96 | 512 | 24.0 | **20.2** |

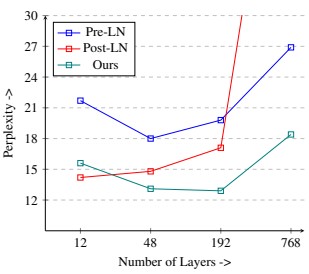

Figure 7: Visualizing performance vs. Depth for 165M models.

**Effect of Increasing Model Size**  Table 7 compares the performance of our approach with the baseline as we vary the model depth ($N$) while keeping the hidden dimension ($d$) constant. The baseline either fails to converge, as observed for Post-LN, or leads to performance degradation, as seen for Pre-LN. By stabilizing the training, DSLM allows training larger models with better performance, for both Pre-LN and Post-LN.

**Downstream Finetuning Results**  We finetune the baseline and DSLM models on the public RACE-M and RACE-H (Lai et al., 2017) datasets. The improvements observed during pre-training (Table 4) are similarly translated into downstream task performance. Table 8 demonstrates the effectiveness of DSLM compared to both Pre-LN and Post-LN baselines.

Table 8: Downstream fine-tuning accuracy on the Middle/High school datasets of RACE benchmark.

| Model Size | Overall | | | RACE-Middle | | | RACE-High | | |
|---|---|---|---|---|---|---|---|---|---|
| | Pre | Post | DSLM | Pre | Post | DSLM | Pre | Post | DSLM |
| 165M | 51.7 | 56.9 | **59.1** | 57.7 | 63.0 | **65.6** | 49.2 | 54.3 | **56.3** |
| 330M | 53.5 | 56.5 | **59.7** | 58.9 | 63.6 | **66.2** | 51.2 | 53.6 | **57.0** |

## 4.4 ANALYSIS OF DEEPSCALELM

**Compute**  Appendix J provides detailed theoretical and wall-clock compute overheads for making models deeper. We observe that up to 200 layers, the theoretical compute is within $6 - 7\%$ of the original shallow model, and wall-clock times also have small overheads less than $15\%$. While our 192-layer 256-d model requires 6% extra compute than the 24-layer 165M parameter model, it manages to outperform the 24-layer 330M model, that has 62.5% extra compute, at equal wall-clock time and at equal number of tokens.

**Ablation of Residual Scaling**  Table 9 provides the results corresponding to the different components of our proposed DSLM scheme for training 96-layer 512-d model Post-LN model. The model fails to converge without the proposed residual scaling. $\beta$ may also be set as learnable (similar to

BatchNorm Ioffe & Szegedy (2015)), after initializing it with $\beta^2 = \frac{2}{N}$. We find that this does not significantly impact performance, and $\beta$ remains within $[0.2 - 5]\times$ of its initialized values.

**Ablation of Initialization**  Table 10 provides ablation results for our proposed initialization. All experiments in Table 10 were conducted for the Pre-LN model with our proposed scaling $(\lambda, \beta)$, since the Post-LN model diverged with Xavier initialization. Xavier initialization performs significantly worse for very deep models. BERT default initialization with $\sigma = 0.02$ also performs worse. Finally, DSLM simpler initialization performs comparably to DSLM.

Table 9: Ablation of various DeepScaleLM components, for a Post-LN model.

| Model | Perf |
|---|---|
| Vanilla Xavier | diverge |
| Xavier + $\beta^2 = 0.5$ | diverge |
| DSLM-Init | diverge |
| DSLM-Init + $\beta^2 = 0.5$ | diverge |
| DSLM-Init + $\beta^2 = \frac{2}{N}$ | 12.2 |

Table 10: Ablation of the initializations.

| Model | Model Size (N,d) | Perf |
|---|---|---|
| Xavier | 165M (192,256) | 38.2 |
| Fixed $\sigma = 0.02$ | 165M (192,256) | 31.6 |
| DSLM | 165M (192,256) | 20.8 |
| DSLM (simple) | 165M (192,256) | **20.7** |
| Fixed $\sigma = 0.02$ | 330M (96,512) | 20.5 |
| DSLM | 330M (96,512) | **20.2** |

**Discussion of Relative Strength**  Existing works have tried to use different values of $\beta$ to stabilize model training (Appendix B.3). In general, for a $\beta$ of the form $\beta^2 = \frac{k}{N^\alpha}$, we can choose from a wide range of values for the constant $k$ and exponent $\alpha$. Intuitively, as $k$ decreases/$\alpha$ increases, the contribution of each layer is reduced, and the observed issues, such as forward growth and gradient explosion/vanishing, are mitigated. However, reducing each layer's weight makes the model more linear, and can affect performance.

Davis et al. (2021) defines "sensitivity" as the variance of relative change in output for small perturbations in parameters, averaged across all parameters. If $\sigma^2_{\text{skip}} = 1$, sensitivity can be shown to be mean across layers of $N * (1/\sigma^2_{\text{block}}) = N * \beta^2$. Mean is not robust to outliers, and hence we suggest median may provide a more robust measure. For e.g., for vanilla pre-LN, Davis et al. (2021)'s definition gives sensitivity as $\mathcal{O}(log(N))$, whereas using median provides a more robust measure as $\mathcal{O}(1)$. But only the first $N/10$ layers have $\mathcal{O}(log(N))$ sensitivity, and the last $9N/10$ layers have $\mathcal{O}(1)$ sensitivity. We will use median in the discussion below.

In Appendix K, we show that the fall in gradient for both pre-LN and post-LN for $\beta^2 = k/N^\alpha$ is $\mathcal{O}(e^{kN^{1-\alpha}})$. The sensitivity is hence $kN^{1-\alpha}$. As we decrease $k$/increase $\alpha$, the gradient fall/growth is reduced by $\mathcal{O}(e^{kN^{1-\alpha}})$, and the training becomes more stable. However, the sensitivity reduces by $\mathcal{O}(kN^{1-\alpha})$.

For DSLM, we chose $\alpha = 1$, that is the sweet spot on the stability-expressivity curve where both the gradient fall bound and sensitivity expressions become independent of model depth. For DS-Init, $\alpha = 2$, and for DeepNet effectively has $\alpha = 1.5$. Although the gradient also becomes stable using $\alpha = 1.5$ or 2, the model expressivity reduces with depth, as shown in Table 3. We conjecture that such models might not be able to extract better results when going deeper, as we indeed verify empirically in the comparison with prior works paragraph in Section 4.3. However, depending on the training landscape for a particular problem, a practitioner might need to increase $\alpha$ (and/or decrease $k$) to stabilize model training and ensure convergence.

## 5 CONCLUSION

We theoretically derive closed forms for the growth of variances for forward and backward pass through individual transformer components as well as the entire transformer model. These formulae enable us to identify and solve the key reasons for vanishing/exploding gradients and rank collapse in very deep transformers. Via scaling and correct initialization, we also enable training very deep transformers up to 768 layers. Our experiments suggest that deeper transformers should be explored - using our method, models with 100s of layers outperform standard models at the same number of parameters and compute.

REPRODUCIBILITY STATEMENT

We release the code for our numerical verification results in the supplementary. Pseudo-code of our proposed DSLM method is provided in Appendix M, and DSLM-simple method can be very easily applied to existing codebases.

Details of implementation are provided in Section 4.3. All hyper-parameters including optimizer, warm-up steps and LR values in Appendix N. Nvidia's Megatron LM (`https://github.com/NVIDIA/Megatron-LM`), at commit 1a26b2910d6b64d8ce6bdebe807739d4ea67f3d7 was used for verification and plots of gradients. Note that while this codebase only supports fixed initialization, modifications to use Xavier instead were minor.

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

# A APPENDIX

CONTENTS

## B   RELATED WORKS

### B.1   INITIALIZATION

Several works, such as (Glorot & Bengio, 2010; He et al., 2015; Brock et al., 2021) improved the initialization of ResNets/ReLU networks, but crucially these works do not consider the impact of correlation in the input, which is large in Transformer models. Schoenholz et al. (2017) initializes weights for networks with bounded activations so that correlation reaches 1 asymptotically.

Some works, such as Mishkin & Matas (2016), sequentially profile each layer empirically by running forward passes through the model, and scaling the weights and/or output to achieve unit variance., and Liu et al. (2020a;b) applied the same method for Transformers. Blake et al. (2023) also tries to achieve unit variance, but does not consider correlation in input or across tokens, and ignores the non-zero mean of ReLU, resulting in incorrect scale. Bachlechner et al. (2020) shows unit variance leads to faster convergence at the start of the training.

We demonstrate that this profiling is unnecessary, and can instead be done theoretically in Deep-ScaleLM. Furthermore, where output or gradient increases in some prior works with more layers (eg. for ADMIN (Liu et al., 2020a), grad and output increase by $\mathcal{O}(log(N))$), our method allows maintaining both unit output and equal gradient across all layers at initialization, and bounded during training.

## B.2 Signal Propagation

Signal propagation has long been studied for ResNets, such as De & Smith (2020); Brock et al. (2021); He et al. (2015); Schoenholz et al. (2017); Anonymous (2022); Labatie et al. (2022); Marion et al. (2022); Klambauer et al. (2017); Balduzzi et al. (2018). Some of these works also compute variances with respect to joint distributions of weights and inputs, resulting in estimates that do not correspond to any single instantiation of the network, as also pointed out by Martens et al. (2021). In our work, we take expectations over inputs for a single instantiation of the network.

For transformers, signal propagation was studied in Xu et al. (2019); Dong et al. (2023); Davis et al. (2021); Noci et al. (2022). Our work also considers previously neglected effects of dropout, input correlation between tokens, non-linearity, $QK$ initialization, and provides closed forms with verifiable correctness of this signal propagation. Ours is the first work to theoretically constrain the output and gradient to almost exactly unit without any profiling passes, showing the validity of our formulae and of our assumptions. See the section Section 3.4 for more discussion on Noci et al. (2022) specifically.

He et al. (2023) extends neural kernel methods of DKS (Martens et al., 2021) to Transformers to model network behaviour, assuming the MLP to be linear in its effect on attention. Q/C maps in kernel methods are similar to signal propagation, as expected moments are equivalent to q and m values of kernels (Martens et al., 2021). Our method relaxes these assumptions, and we show that considering the impact of ReLU/GeLU on correlation is critical to correctly modelling attention.

We also account for cases with non-IID inputs that may occur due to segment/position embeddings or due to non-uniform token distributions in real data (that are distributed approximately per Zipf's law Zipf (1999)) – and find that this strongly affects output variance of the attention block.

## B.3 Moment Control & Residual Scaling

Bounded gradients, or normalizing per-layer gradients, have been shown to results in better/faster convergence (Shen et al., 2020; Yu et al., 2018; You et al., 2017; 2020). Woks such as Takase et al. (2022); Shleifer et al. (2021); Hayou et al. (2019) also achieved improved training by empirically mitigating the gradient explosion.

Scaling with $\lambda^2 + \beta^2 = 1$ to control moments have often been used for ResNets (Balduzzi et al., 2018; Szegedy et al., 2016; Hanin & Rolnick, 2018; Arpit et al., 2019; Zhang et al., 2022; Hoedt et al., 2022). Szegedy et al. (2016) proposed to use any small $\beta$, Balduzzi et al. (2018) proposed to set $\beta^2 = 0.5$, Bachlechner et al. (2020) sets $\beta = 0$ and learnable. De & Smith (2020) showed that $\lambda^2 = 0.5$ is not sufficient to solve vanishing gradients.

$\beta^2 = \frac{k}{N}$ was used to control growth of moments in Arpit et al. (2019); Brock et al. (2021); Marion et al. (2022); Zhang et al. (2023); He et al. (2023); Noci et al. (2022) . $\beta^2 = \frac{k}{n}$, where $n$ is the current layer, was used in De & Smith (2020); Liu et al. (2020a;b); Davis et al. (2021); Blake et al. (2023), but this results in logarithmic bounds instead of constant for forward propagation if $\lambda = 1$ is used, and vanishing gradient for backward propagation otherwise.

Values of $\beta^2 < \frac{k}{N}$, such as (effectively) $\frac{1}{N^2}$ for DSInit (Zhang et al., 2019) or $\frac{1}{N^{1.5}}$ for Deep-Net (Wang et al., 2022a) decrease sensitivity of the model, and may result in the model becoming "too linear". DeepNet shows performance improvements by making the model deeper, but keeping the hidden dimension constant. Our setting is much more strict - we keep the number of parameters (and hence compute) constant, and our method still show performance improves on making the model deeper. For example, DeepNet's 200 layer model is 3.2B params, whereas ours is 330M params.

Sometimes, these $\beta$ values are used in conjunction with $\lambda = 1$, such as in Liu et al. (2020a;b), but as shown in He et al. (2023), fully normalized residual connections with $\lambda^2 + \beta^2 = 1$ often perform better than those with $\lambda = 1$. We also observed lower performance with $\lambda = 1$ in our initial experiments, and hence we fully normalize the residual connections.

Our contribution goes beyond providing an optimal scaling scheme. Using the theoretical framework and closed-form expressions for moment propagation through both Pre-LN and Post-LN developed in this work, practitioners can make informed choices about using any of the scaling factors

above based on the stability-performance tradeoffs, such as using a lower $\beta$ for scenarios with high correlation, or using higher $\beta$ with uncorrelated inputs.

### B.4 OTHER NETWORK MODIFICATIONS FOR DEEP NETWORKS

Shi et al. (2022); Zhou et al. (2021); Wang et al. (2022b); Dong et al. (2023) showed that attention causes rank collapse in deeper models, and Chen et al. (2020); Zhao et al. (2023) showed the same for graphs. Takase et al. (2022) added some extra skip connections from the input of the model, Nguyen & Salazar (2019) modified layernorm slightly, Zhai et al. (2023) normalized all linear layers by their spectral norm, and Shleifer et al. (2021) added extra layer norms. The methods in these works are orthogonal to our approach, and our equations can be easily extended to cover the architectural modifications suggested in these.

## C  MOMENT PROPAGATION THROUGH TRANSFORMER COMPONENTS

We provide detailed proofs of the closed-form expression for each of the transformer component – Linear layer, Dropout, ReLU, GeLU, LayerNorm, and Softmax.

For any component, input is represented as $\mathbf{x}_{\text{in}}$ and $\mathbf{x}_{\text{out}}$ is the output. The gradient flowing in into the component from the output side is represented as $\mathbf{g}_{\text{out}}$ and the backpropagated gradient towards the input is $\mathbf{g}_{\text{in}}$. We switch from vector to matrix notation ($\mathbf{X}_{\text{in}}, \mathbf{X}_{\text{out}}$) whenever needed. We assume that the input is distributed normally $\mathcal{N}(0, \sigma_{x_{in}})$. Further, input is not assumed to be IID and it can have covariance both along the sequence length and hidden dimension. Additional assumptions needed to derive the proofs for softmax and attention can be found in the respective proofs. We derive the forward and backward variants for a specific initialization of weights that corresponds to exactly one network instance - all expectations are taken over inputs for a given instantiation of the network.

### C.1  EMBEDDINGS

We do not assume the input embeddings to be IID. Repetition of same token introduces correlation across the sequence length. We assume that the input tokens have been sampled from a multinomial distribution. The words / token ids are distributed almost according to zipf's law. Assuming we initialize all the embeddings with variance $\sigma_{w_{embd}}^2$, the relevant statistics for word embeddings output $x_{\text{out}_{we}}$ are as follows

$$\mu_{x_{\text{out}_{we}}} = 0$$
$$\sigma_{x_{\text{out}_{we}}}^2 = \sigma_{w_{\text{embd}}}^2$$
$$\text{Cov}^l(x_{\text{out}_{we}}) = \sum \frac{N_i * (N_i - 1)}{L * (L - 1)} * \sigma_{w_{\text{embd}}}^2$$
$$\text{r}^l(x_{\text{out}_{we}}) = \sum \frac{N_i * (N_i - 1)}{L * (L - 1)}$$
$$\text{Cov}^d(x_{\text{out}_{we}}) = 0$$

Assume $i$th word occurs $N_i$ times, it contributes $\frac{N_i*(N_i-1)}{L*(L-1))}$ to the covariance along sequence length. Similarly, we can calculate the correlation for segment-type embeddings output $x_{\text{out}_{se}}$. Zipf's law states that the probability for each token is inversely proportional to its rank. For the word with rank $i$, $p_i = \frac{c}{i}$, where $c = \frac{1}{\sum_i \frac{1}{i}}$. For a sentence of length $L$, the token with probability $p_i$ is expected to occur $p_i.L$ times. Hence, for a given vocabulary size $|V|$, we can calculate the correlation as follows

$$r^l(x_{\text{out}_{we}}) = \sum \frac{N_i * (N_i - 1)}{L * (L - 1)}$$

$$= \sum_i^{|V|} \frac{p_i L * (p_i L - 1)}{L * (L - 1)}$$

$$= \frac{\sum_i p_i^2 * L - 1}{L - 1}$$

$$\approx \frac{\frac{L\pi^2}{6.\log(|V|)^2} - 1}{L - 1}$$

$$\approx \frac{\pi^2}{6.\log(|V|)^2}$$

Similarly, the segment type embeddings have two possible values denoting the sentence order. If first sentence has length $x$, we can consider this as a special case of the analysis performed above with two possible tokens, where $N_1 = x$ and $N_2 = L - x$. Assuming $x$ is distributed uniformly between 0 to $L$, $L - x$ also has the same distribution. Hence,

$$r^l(x_{\text{out}_{se}}, N_1, N_2) = \frac{N_1^2 + N_2^2 - L}{L * (L - 1)}$$

Taking expectation, we get

$$r^l(x_{\text{out}_{se}}) = \frac{\frac{2}{3} * L^2 - L}{L * (L - 1)}$$

$$\approx \frac{2}{3}$$

The correlation from position embeddings is 0. Since the variance is same for all embedding types, the final correlation is the average of the three. Hence

$$r^l(x_{\text{out}}) = \frac{1}{3}\left(r^l(x_{\text{out}_{we}}) + r^l(x_{\text{out}_{se}})\right)$$

$$= \frac{\pi^2}{18 * \log(|V|)^2} + \frac{2}{9}$$

For our case, $|V| = 32000$ and sequence length $L = 256$, the theoretically prediction correlation $r^l_{x_{in}} = 0.247$ which is within 10% of the empirically observed correlation (0.221).

Hence, the final moments for the embedding output are

$$\boxed{\begin{aligned}
\mu_{x_{\text{out}}} &= 0 \\
\sigma^2_{x_{\text{out}}} &= 3 * \sigma^2_{w_{\text{embd}}} \\
\text{Cov}^l_{x_{\text{out}}} &= \left(\frac{\pi^2}{18 * \log(|V|)^2} + \frac{2}{9}\right)\sigma^2_{x_{\text{out}}} \\
\text{Cov}^d_{x_{\text{out}}} &= 0
\end{aligned}}$$

## C.2   LINEAR

For linear layer with $d_{in}$ dimensional input $\mathbf{x}_{\text{in}}$, and $d_{out}$ dimensional output $\mathbf{x}_{\text{out}}$, we can define the forward pass mathematically as,

$$\mathbf{x}_{\text{out}} = \mathbf{x}_{\text{in}}\mathbf{W}$$

$$\implies x_{\text{out}_j} = \sum_{i=1}^{d_{\text{in}}} x_{\text{in}_i} W_{i,j}$$

Similarly, we define the backward pass as,

$$\mathbf{g}_{\text{in}} = \mathbf{g}_{\text{out}}\mathbf{W^T}$$

$$\implies g_{\text{in}_j} = \sum_{i=1}^{d_{\text{out}}} g_{\text{out}_i} W_{j,i}$$

For expectation of output we have,

$$\mathbb{E}[x_{\text{out}_j}] = \mathbb{E}[\sum_{i=1}^{d_{\text{in}}} x_{\text{in}_i} W_{i,j}] = \sum_{i=1}^{d_{\text{in}}} \mathbb{E}[x_{\text{in}_i} W_{i,j}]$$

$$= \sum_{i=1}^{d_{\text{in}}} \mathbb{E}[x_{\text{in}_i}]\mathbb{E}[W_{i,j}] = \mu_{x_{\text{in}}}\mu_w$$

(As weights and input are independent of each other)

$$\boxed{\mu_{x_{\text{out}}} = 0} \qquad\qquad (\forall j)$$

To get variance of the output of forward pass we have,

$$\text{Var}(x_{\text{out}_j}) = \text{Var}(\sum_{i=1}^{d_{\text{in}}} x_{\text{in}_i} W_{i,j})$$

As the weights are initialized independently each term in summation is independent of each other

$$= \sum_{i=1}^{d_{\text{in}}} (\text{Var}(x_{\text{in}_i} W_{i,j}))$$

$$= \sum_{i=1}^{d_{\text{in}}} ((\sigma_{x_{\text{in}}}^2 + \mu_{x_{\text{in}}}^2)(\sigma_w^2 + \mu_w^2) - \mu_{x_{\text{in}}}^2\mu_w^2)$$

(As weights and input are independent of each other)

$$= \sum_{i=1}^{d_{\text{in}}} (\sigma_{x_{\text{in}}}^2 + \mu_{x_{\text{in}}}^2)\sigma_w^2$$

$$\text{Var}(x_{\text{out}_j}) = d_{\text{in}}(\sigma_{x_{\text{in}}}^2 + \mu_{x_{\text{in}}}^2)\sigma_w^2 \qquad\qquad (\forall j)$$

$$\boxed{\sigma_{x_{\text{out}}}^2 = d_{\text{in}}(\sigma_{x_{\text{in}}}^2 + \mu_{x_{\text{in}}}^2)\sigma_w^2}$$

If we have two inputs $\mathbf{x}_{\text{in}}$ and $\mathbf{y}_{\text{in}}$ such that for all $i$ we have $\text{Corr}(x_{\text{in}_i}, y_{\text{in}_i}) = r_{x_{\text{in}}}^l$, and $\mathbf{x}_{\text{out}} = \mathbf{x}_{\text{in}}\mathbf{W}$ and $\mathbf{y}_{\text{out}} = \mathbf{y}_{\text{in}}\mathbf{W}$. Then for any $j$ we have

$$\text{Corr}(x_{\text{out}_j}, y_{\text{out}_j}) = \frac{\mathbb{E}[x_{\text{out}_j}y_{\text{out}_j}] - \mathbb{E}[x_{\text{out}_j}]\mathbb{E}[y_{\text{out}_j}]}{\sqrt{\text{Var}(x_{\text{out}_j})\text{Var}(y_{\text{out}_j})}}$$

$$= \frac{\mathbb{E}[x_{\text{out}_j}y_{\text{out}_j}]}{\sqrt{\sigma_{x_{\text{out}}}^2\sigma_{x_{\text{out}}}^2}}$$

$$= \frac{\mathbb{E}[\sum_{i=1}^{d_{\text{in}}} x_{\text{in}_i} W_{i,j} \sum_{k=1}^{d_{in}} y_{\text{in}_k} W_{k,j}]}{\sigma_{x_{\text{out}}}^2}$$

$$= \frac{\mathbb{E}[\sum_{i=1}^{d_{\text{in}}} x_{\text{in}_i} y_{\text{in}_i} W_{i,j}^2 + \sum_{k=1,k\neq i}^{d_{in}} \sum_{i=1}^{d_{\text{in}}} x_{\text{in}_i} y_{\text{in}_k} W_{i,j} W_{k,j}]}{\sigma_{x_{\text{out}}}^2}$$

In second summation all terms are independent of each other and as the expectation of weights is 0 we have

$$\text{Corr}(x_{\text{out}_j}, y_{\text{out}_j}) = \frac{\mathbb{E}[\sum_{i=1}^{d_{\text{in}}} x_{\text{in}_i} y_{\text{in}_i} W_{i,j}^2]}{\sigma_{x_{\text{out}}}^2}$$

$$= \frac{\sum_{i=1}^{d_{\text{in}}} \mathbb{E}[x_{\text{in}_i} y_{\text{in}_i} W_{i,j}^2]}{\sigma_{x_{\text{out}}}^2} \quad \text{(Independence of weight initialization)}$$

$$= \frac{\sum_{i=1}^{d_{\text{in}}} \mathbb{E}[x_{\text{in}_i} y_{\text{in}_i}] \mathbb{E}[W_{i,j}^2]}{\sigma_{x_{\text{out}}}^2}$$

$$= \frac{\sum_{i=1}^{d_{\text{in}}} (r_{x_{\text{in}}}^l \sigma_{x_{\text{in}}}^2 + \mu_{x_{\text{in}}}^2) \sigma_w^2}{\sigma_{x_{\text{out}}}^2} \quad \text{(Definition of correlation)}$$

$$= \frac{d_{\text{in}} (r_{x_{\text{in}}}^l \sigma_{x_{\text{in}}}^2 + \mu_{x_{\text{in}}}^2) \sigma_w^2}{d_{\text{in}} (\sigma_{x_{\text{in}}}^2 + \mu_{x_{\text{in}}}^2) \sigma_w^2}$$

$$\text{Corr}(x_{\text{out}_j}, y_{\text{out}_j}) = \frac{r_{x_{\text{in}}}^l \sigma_{x_{\text{in}}}^2 + \mu_{x_{\text{in}}}^2}{\sigma_{x_{\text{in}}}^2 + \mu_{x_{\text{in}}}^2}$$

$$\boxed{r_{x_{\text{out}}}^l = \frac{r_{x_{\text{in}}}^l \sigma_{x_{\text{in}}}^2 + \mu_{x_{\text{in}}}^2}{\sigma_{x_{\text{in}}}^2 + \mu_{x_{\text{in}}}^2}}$$

As the backward pass has similar structure, assuming $\mu_{g_{\text{out}}} = 0$ we can use the same analysis to get,

$$\boxed{\begin{array}{c} \mu_{g_{\text{in}}} = 0 \\ \sigma_{g_{\text{in}}}^2 = d_{\text{out}} \sigma_{g_{\text{out}}}^2 \sigma_w^2 \end{array}}$$

## C.3 DROPOUT

We can define Dropout mathematically as,

$$\mathbf{x}_{\text{out}} = \text{Dropout}(\mathbf{x}_{\text{in}})$$

$$\implies x_{\text{out}_i} = \begin{cases} \frac{x_{\text{in}_i}}{(1-p)} & \text{with probability } 1-p \\ 0 & \text{else} \end{cases}$$

To calculate expectation of dropout,

$$\mathbb{E}[x_{\text{out}_i}] = 0 * p + (1-p) * \mathbb{E}[\frac{x_{\text{in}_i}}{(1-p)}]$$

$$\boxed{\mu_{x_{\text{out}}} = \mu_{x_{\text{in}}}}$$

For variance,

$$\text{Var}(x_{\text{out}_i}) = \mathbb{E}[x_{\text{out}_i}^2] - \mathbb{E}[x_{\text{out}_i}]^2$$

$$= 0 * p + (1-p) * \mathbb{E}[\frac{x_{\text{in}_i}^2}{(1-p)^2}] - \mu_{x_{\text{in}}}^2$$

$$= \frac{\mathbb{E}[x_{\text{in}_i}^2]}{(1-p)} - \mu_x^2$$

$$= \frac{\sigma_{x_{\text{in}}}^2 + \mu_{x_{\text{in}}}^2}{(1-p)} - \mu_{x_{\text{in}}}^2$$

$$\boxed{\sigma_{x_{\text{out}}}^2 = \frac{\sigma_{x_{\text{in}}}^2 + p\mu_{x_{\text{in}}}^2}{(1-p)}}$$

If we have two inputs $\mathbf{x}_{\text{in}}$ and $\mathbf{y}_{\text{in}}$ such that for all $i$ we have $\text{Corr}(x_{\text{in}_i}, y_{\text{in}_i}) = r_{x_{\text{in}}}^l$, and $\mathbf{x}_{\text{out}} = \text{Dropout}(\mathbf{x}_{\text{in}})$ and $\mathbf{y}_{\text{out}} = \text{Dropout}(\mathbf{y}_{\text{in}})$. Then for any $j$ we have

$$\text{Corr}(x_{\text{out}_j}, y_{\text{out}_j}) = \frac{\mathbb{E}[x_{\text{out}_j} y_{\text{out}_j}] - \mathbb{E}[x_{\text{out}_j}] \mathbb{E}[y_{\text{out}_j}]}{\sqrt{\text{Var}(x_{\text{out}_j}) \text{Var}(y_{\text{out}_j})}}$$

$$= \frac{\mathbb{E}[x_{\text{out}_j} y_{\text{out}_j}] - \mu_{x_{\text{out}}} \mu_{x_{\text{out}}}}{\sqrt{\sigma_{x_{\text{out}}}^2 \sigma_{x_{\text{out}}}^2}}$$

$$= \frac{p^2 * 0 + 2 * p * (1-p) * 0 + (1-p)^2 * \mathbb{E}[\frac{x_{\text{in}_j} y_{\text{in}_j}}{(1-p)*(1-p)}] - \mu_{x_{\text{out}}}^2}{\sigma_{x_{\text{out}}}^2}$$

$$= \frac{\mathbb{E}[x_{\text{in}_j} y_{\text{in}_j}] - \mu_{x_{\text{out}}}^2}{\sigma_{x_{\text{out}}}^2}$$

$$\boxed{\text{Corr}(x_{\text{out}_j}, y_{\text{out}_j}) = \frac{(r_{x_{\text{in}}}^l \sigma_{x_{\text{in}}}^2)(1-p)}{\sigma_{x_{\text{in}}}^2 + p\mu_{x_{\text{in}}}^2} = r_{x_{\text{out}}}^l}$$

We can define the backward pass of Dropout as,

$$g_{\text{in}_i} = \begin{cases} \frac{g_{\text{out}_i}}{(1-p)} & \text{if } x_i \text{ isn't dropped out (which has probability } (1-p)) \\ 0 & \text{else} \end{cases}$$

Again we can see that backward has similar definition to that of forward pass. Assuming $\mu_{g_{x_{\text{out}}}} = 0$ and using similar analysis we get,

$$\boxed{\begin{aligned} \mu_{g_{\text{in}}} &= 0 \\ \sigma_{g_{\text{in}}}^2 &= \frac{\sigma_{g_{\text{out}}}^2}{(1-p)} \end{aligned}}$$

### C.4  RELU

We can define ReLU mathematically as,

$$\mathbf{x}_{\text{out}} = \text{ReLU}(\mathbf{x}_{\text{in}})$$

$$\implies x_{\text{out}_i} = \begin{cases} x_{\text{in}_i} & \text{if } x_{\text{in}_i} > 0 \\ 0 & \text{else} \end{cases}$$

For getting expectation of output of ReLU for normally distributed input we have,

$$\mathbb{E}[x_{\text{out}_i}] = \int_{-\infty}^{\infty} \frac{\text{ReLU}(x_{\text{in}_i}) \exp\left(\frac{-x_{\text{in}_i}^2}{2\sigma_{x_{\text{in}}}^2}\right)}{\sqrt{2\pi}\sigma_{x_{\text{in}}}} dx_{\text{in}_i}$$

$$= \int_{-\infty}^{0} \frac{0 * \exp\left(\frac{-x_{\text{in}_i}^2}{2\sigma_{x_{\text{in}}}^2}\right)}{\sqrt{2\pi}\sigma_{x_{\text{in}}}} dx_{\text{in}_i} + \int_{0}^{\infty} \frac{x_{\text{in}_i} \exp\left(\frac{-x_{\text{in}_i}^2}{2\sigma_{x_{\text{in}}}^2}\right)}{\sqrt{2\pi}\sigma_{x_{\text{in}}}} dx_{\text{in}_i}$$

$$= \int_{0}^{\infty} \frac{x_{\text{in}_i} \exp\left(\frac{-x_{\text{in}_i}^2}{2\sigma_{x_{\text{in}}}^2}\right)}{\sqrt{2\pi}\sigma_{x_{\text{in}}}} dx_{\text{in}_i}$$

Substituting $t = \frac{x_{\text{in}_i}^2}{2\sigma_{x_{\text{in}}}^2}$ we have $dt = \frac{x_{\text{in}_i} dx_{\text{in}_i}}{\sigma_{x_{\text{in}}}^2}$ we get,

$$\mathbb{E}[x_{\text{out}_i}] = \int_{0}^{\infty} \frac{\sigma_{x_{\text{in}}} \exp\left(-t\right) dt}{\sqrt{2\pi}}$$

$$= \frac{\sigma_{x_{\text{in}}}}{\sqrt{2\pi}}[-\exp\left(-t\right)]_0^{\infty} = \frac{\sigma_{x_{\text{in}}}}{\sqrt{2\pi}}$$

Hence, the mean of output

$$\boxed{\mu_{x_{\text{out}}} = \frac{\sigma_{x_{\text{in}}}}{\sqrt{2\pi}}} \tag{1}$$

Variance of output can be calculated by,

$$\text{Var}(x_{\text{out}_i}) = \mathbb{E}[x_{\text{out}_i}^2] - \mathbb{E}[x_{\text{out}_i}]^2$$

$$= \int_{-\infty}^{\infty} \frac{(\text{ReLU}(x_{\text{in}_i}))^2 \exp\left(\frac{-x_{\text{in}_i}^2}{2\sigma_{x_{\text{in}}}^2}\right)}{\sqrt{2\pi}\sigma_{x_{\text{in}}}} dx_{\text{in}_i} - \frac{\sigma_{x_{\text{in}}}^2}{2\pi}$$

$$= \int_{-\infty}^{0} \frac{0 * \exp\left(\frac{-x_{\text{in}_i}^2}{2\sigma_{x_{\text{in}}}^2}\right)}{\sqrt{2\pi}\sigma_{x_{\text{in}}}} dx_{\text{in}_i} + \int_{0}^{\infty} \frac{x_{\text{in}_i}^2 \exp\left(\frac{-x_{\text{in}_i}^2}{2\sigma_{x_{\text{in}}}^2}\right)}{\sqrt{2\pi}\sigma_{x_{\text{in}}}} dx_{\text{in}_i} - \frac{\sigma_{x_{\text{in}}}^2}{2\pi}$$

$$= \int_{0}^{\infty} \frac{x_{\text{in}_i}^2 \exp\left(\frac{-x_{\text{in}_i}^2}{2\sigma_{x_{\text{in}}}^2}\right)}{\sqrt{2\pi}\sigma_{x_{\text{in}}}} dx_{\text{in}_i} - \frac{\sigma_{x_{\text{in}}}^2}{2\pi}$$

Let $I = \int_{0}^{\infty} \frac{x_{\text{in}_i}^2 \exp\left(\frac{-x_{\text{in}_i}^2}{2\sigma_{x_{\text{in}}}^2}\right)}{\sqrt{2\pi}\sigma_{x_{\text{in}}}} dx_{\text{in}_i}$, then substituting $t = -x_{\text{in}_i}$ we have,

$$I = \int_{0}^{-\infty} \frac{-t^2 \exp\left(\frac{-t^2}{2\sigma_{x_{\text{in}}}^2}\right)}{\sqrt{2\pi}\sigma_{x_{\text{in}}}} dt$$

$$= \int_{-\infty}^{0} \frac{t^2 \exp\left(\frac{-t^2}{2\sigma_{x_{\text{in}}}^2}\right)}{\sqrt{2\pi}\sigma_{x_{\text{in}}}} dt$$

$$\implies I + I = \int_{-\infty}^{0} \frac{t^2 \exp\left(\frac{-t^2}{2\sigma_{x_{\text{in}}}^2}\right)}{\sqrt{2\pi}\sigma_{x_{\text{in}}}} dt + \int_{0}^{\infty} \frac{x_{\text{in}_i}^2 \exp\left(\frac{-x_{\text{in}_i}^2}{2\sigma_{x_{\text{in}}}^2}\right)}{\sqrt{2\pi}\sigma_{x_{\text{in}}}} dx_{\text{in}_i}$$

$$2I = \int_{-\infty}^{\infty} \frac{x_{\text{in}_i}^2 \exp\left(\frac{-x_{\text{in}_i}^2}{2\sigma_{x_{\text{in}}}^2}\right)}{\sqrt{2\pi}\sigma_{x_{\text{in}}}} dx_{\text{in}_i} = \sigma_{x_{\text{in}}}^2$$

$$\implies \text{Var}(x_{\text{out}_i}) = \frac{\sigma_{x_{\text{in}}}^2}{2} - \frac{\sigma_{x_{\text{in}}}^2}{2\pi} = \frac{\sigma_{x_{\text{in}}}^2}{2}\left(1 - \frac{1}{\pi}\right)$$

$$\boxed{\sigma_{x_{\text{out}}}^2 = \frac{\sigma_{x_{\text{in}}}^2}{2}\left(1 - \frac{1}{\pi}\right)}$$

Now for two inputs $\mathbf{x}_{\text{in}}$ and $\mathbf{y}_{\text{in}}$ such that for all $i$ we have $\text{Corr}(x_{\text{in}_i}, y_{\text{in}_i}) = r_{x_{\text{in}}}^l$, and $\mathbf{x}_{\text{out}} = \text{ReLU}(\mathbf{x}_{\text{in}})$ and $\mathbf{y}_{\text{out}} = \text{ReLU}(\mathbf{y}_{\text{in}})$. Then for any $j$ we have,

$$\text{Corr}(x_{\text{out}_j}, y_{\text{out}_j}) = \frac{\mathbb{E}[x_{\text{out}_j} y_{\text{out}_j}] - \mathbb{E}[x_{\text{out}_j}]\mathbb{E}[y_{\text{out}_j}]}{\sqrt{\text{Var}(x_{\text{out}_j})\text{Var}(y_{\text{out}_j})}}$$

$$\mathbb{E}[x_{\text{out}_j} y_{\text{out}_j}] = \int_{0}^{\infty} \int_{0}^{\infty} \frac{x_{\text{in}_j} y_{\text{in}_j}}{2\pi\sigma_{x_{\text{in}}}^2 \sqrt{1 - (r_{x_{\text{in}}}^l)^2}} \exp\left(\frac{-(x_{\text{in}_j}^2 + y_{\text{in}_j}^2 - 2r_{x_{\text{in}}}^l x_{\text{in}_j} y_{\text{in}_j})}{2\sigma_{x_{\text{in}}}^2 (1 - (r_{x_{\text{in}}}^l)^2)}\right) dx_{\text{in}_j} dy_{\text{in}_j}$$

$$= \int_{0}^{\infty} \int_{0}^{\infty} \frac{x_{\text{in}_j} y_{\text{in}_j}}{2\pi\sigma_{x_{\text{in}}}^2 \sqrt{1 - (r_{x_{\text{in}}}^l)^2}} \exp\left(\frac{-(x_{\text{in}_j} - r_{x_{\text{in}}}^l y_{\text{in}_j})^2}{2\sigma_{x_{\text{in}}}^2 (1 - (r_{x_{\text{in}}}^l)^2)}\right) \exp\left(\frac{-y_{\text{in}_j}^2}{2\sigma_{x_{\text{in}}}^2}\right) dx_{\text{in}_j} dy_{\text{in}_j}$$

Substituting $t = x_{\text{in}_j} - r_{x_{\text{in}}}^l y_{\text{in}_j}$, and assuming $y_{\text{in}_j}$ is constant for the inner integral, $dx_{\text{in}_j} = dt$

$$\mathbb{E}[x_{\text{out}_j} y_{\text{out}_j}] =$$

$$= \int_{0}^{\infty} \frac{y_{\text{in}_j} \exp\left(\frac{-y_{\text{in}_j}^2}{2\sigma_{x_{\text{in}}}^2}\right)}{\sqrt{2\pi}\sigma_{x_{\text{in}}}} \int_{-r_{x_{\text{in}}}^l y_{\text{in}_j}}^{\infty} \frac{t + r_{x_{\text{in}}}^l y_{\text{in}_j}}{\sqrt{2\pi}\sigma_{x_{\text{in}}} \sqrt{1 - (r_{x_{\text{in}}}^l)^2}} \exp\left(\frac{-t^2}{2\sigma_{x_{\text{in}}}^2 (1 - (r_{x_{\text{in}}}^l)^2)}\right) dt dy_{\text{in}_j}$$

$$= \int_{0}^{\infty} \frac{y_{\text{in}_j}}{\sqrt{2\pi}\sigma_{x_{\text{in}}}} \exp\left(\frac{-y_{\text{in}_j}^2}{2\sigma_{x_{\text{in}}}^2}\right) \int_{-r_{x_{\text{in}}}^l y_{\text{in}_j}}^{\infty} \frac{t}{\sqrt{2\pi}\sigma_{x_{\text{in}}} \sqrt{1 - (r_{x_{\text{in}}}^l)^2}} \exp\left(\frac{-t^2}{2\sigma_{x_{\text{in}}}^2 (1 - (r_{x_{\text{in}}}^l)^2)}\right) dt dy_{\text{in}_j}$$

$$+ \int_{0}^{\infty} \frac{y_{\text{in}_j}}{\sqrt{2\pi}\sigma_x} \exp\left(\frac{-y_{\text{in}_j}^2}{2\sigma_{x_{\text{in}}}^2}\right) \int_{-r_{x_{\text{in}}}^l y_{\text{in}_j}}^{\infty} \frac{r_{x_{\text{in}}}^l y_{\text{in}_j}}{\sqrt{2\pi}\sigma_{x_{\text{in}}} \sqrt{1 - (r_{x_{\text{in}}}^l)^2}} \exp\left(\frac{-t^2}{2\sigma_{x_{\text{in}}}^2 (1 - (r_{x_{\text{in}}}^l)^2)}\right) dt dy_{\text{in}_j}$$

Let us first define $I_1$ and $I_2$ as:

$$I_1 = \int_0^\infty \frac{y_{\text{in}_j}}{\sqrt{2\pi}\sigma_{x_{\text{in}}}} \exp\left(\frac{-y_{\text{in}_j}^2}{2\sigma_{x_{\text{in}}}^2}\right) \int_{-r_{x_{\text{in}}}^l y_{\text{in}_j}}^\infty \frac{t}{\sqrt{2\pi}\sigma_{x_{\text{in}}}\sqrt{1-(r_{x_{\text{in}}}^l)^2}} \exp\left(\frac{-t^2}{2\sigma_{x_{\text{in}}}^2(1-(r_{x_{\text{in}}}^l)^2)}\right) dt\, dy_{\text{in}_j}$$

$$I_2 = \int_0^\infty \frac{y_{\text{in}_j}}{\sqrt{2\pi}\sigma_x} \exp\left(\frac{-y_{\text{in}_j}^2}{2\sigma_{x_{\text{in}}}^2}\right) \int_{-r_{x_{\text{in}}}^l y_{\text{in}_j}}^\infty \frac{r_{x_{\text{in}}}^l y_{\text{in}_j}}{\sqrt{2\pi}\sigma_{x_{\text{in}}}\sqrt{1-(r_{x_{\text{in}}}^l)^2}} \exp\left(\frac{-t^2}{2\sigma_{x_{\text{in}}}^2(1-(r_{x_{\text{in}}}^l)^2)}\right) dt\, dy_{\text{in}_j}$$

$$I_1 = \int_0^\infty \frac{y_{\text{in}_j}}{\sqrt{2\pi}\sigma_{x_{\text{in}}}} \exp\left(\frac{-y_{\text{in}_j}^2}{2\sigma_{x_{\text{in}}}^2}\right) \int_{-r_{x_{\text{in}}}^l y_{\text{in}_j}}^\infty \frac{t}{\sqrt{2\pi}\sigma_{x_{\text{in}}}\sqrt{1-(r_{x_{\text{in}}}^l)^2}} \exp\left(\frac{-t^2}{2\sigma_{x_{\text{in}}}^2(1-(r_{x_{\text{in}}}^l)^2)}\right) dt\, dy_{\text{in}_j}$$

Substituting $p = \dfrac{t^2}{2\sigma_{x_{\text{in}}}^2(1-(r_{x_{\text{in}}}^l)^2)}$ we have $dp = \dfrac{t\, dt}{\sigma_{x_{\text{in}}}^2(1-(r_{x_{\text{in}}}^l)^2)}$

$$I_1 = \int_0^\infty \frac{y_{\text{in}_j}}{\sqrt{2\pi}\sigma_{x_{\text{in}}}} \exp\left(\frac{-y_{\text{in}_j}^2}{2\sigma_{x_{\text{in}}}^2}\right) \int_{\frac{(r_{x_{\text{in}}}^l y_{\text{in}_j})^2}{2\sigma_{x_{\text{in}}}^2(1-(r_{x_{\text{in}}}^l)^2)}}^\infty \frac{\sigma_{x_{\text{in}}}\sqrt{(1-(r_{x_{\text{in}}}^l)^2)}}{\sqrt{2\pi}} \exp\left(-p\right) dp\, dy_{\text{in}_j}$$

$$= \int_0^\infty \frac{y_{\text{in}_j}}{\sqrt{2\pi}\sigma_{x_{\text{in}}}} \exp\left(\frac{-y_{\text{in}_j}^2}{2\sigma_{x_{\text{in}}}^2}\right) \frac{\sigma_{x_{\text{in}}}\sqrt{(1-(r_{x_{\text{in}}}^l)^2)}}{\sqrt{2\pi}} \exp\left(\frac{-(r_{x_{\text{in}}}^l y_{\text{in}_j})^2}{2\sigma_{x_{\text{in}}}^2(1-(r_{x_{\text{in}}}^l)^2)}\right) dy_{\text{in}_j}$$

$$= \int_0^\infty \frac{y_{\text{in}_j}\sqrt{(1-(r_{x_{\text{in}}}^l)^2)}}{2\pi} \exp\left(\frac{-y_{\text{in}_j}^2}{2\sigma_{x_{\text{in}}}^2(1-(r_{x_{\text{in}}}^l)^2)}\right) dy_{\text{in}_j}$$

Substituting $m = \dfrac{y_{\text{in}_j}^2}{2\sigma_{x_{\text{in}}}^2(1-(r_{x_{\text{in}}}^l)^2)}$, $dm = \dfrac{y_{\text{in}_j}\, dy_{\text{in}_j}}{\sigma_{x_{\text{in}}}^2(1-(r_{x_{\text{in}}}^l)^2)}$,

$$I_1 = \int_0^\infty \frac{\sqrt{(1-(r_{x_{\text{in}}}^l)^2)}}{2\pi}(1-(r_{x_{\text{in}}}^l)^2)\sigma_{x_{\text{in}}}^2 \exp\left(-m\right) dm$$

$$= \frac{(1-(r_{x_{\text{in}}}^l)^2)^{\frac{3}{2}}\sigma_{x_{\text{in}}}^2}{2\pi}$$

$$I_2 = \int_0^\infty \frac{y_{\text{in}_j}}{\sqrt{2\pi}\sigma_{x_{\text{in}}}} \exp\left(\frac{-y_{\text{in}_j}^2}{2\sigma_{x_{\text{in}}}^2}\right) \int_{-r_{x_{\text{in}}}^l y_{\text{in}_j}}^\infty \frac{r_{x_{\text{in}}}^l y_{\text{in}_j}}{\sqrt{2\pi}\sigma_{x_{\text{in}}}\sqrt{1-(r_{x_{\text{in}}}^l)^2}} \exp\left(\frac{-t^2}{2\sigma_{x_{\text{in}}}^2(1-(r_{x_{\text{in}}}^l)^2)}\right) dt\, dy_{\text{in}_j}$$

$$= \int_0^\infty \frac{r_{x_{\text{in}}}^l y_{\text{in}_j}^2}{\sqrt{2\pi}\sigma_{x_{\text{in}}}} \exp\left(\frac{-y_{\text{in}_j}^2}{2\sigma_{x_{\text{in}}}^2}\right) \int_{-r_{x_{\text{in}}}^l y_{\text{in}_j}}^\infty \frac{1}{\sqrt{2\pi}\sigma_{x_{\text{in}}}\sqrt{1-(r_{x_{\text{in}}}^l)^2}} \exp\left(\frac{-t^2}{2\sigma_{x_{\text{in}}}^2(1-(r_{x_{\text{in}}}^l)^2)}\right) dt\, dy_{\text{in}_j}$$

Substituting $p = -t$, where $\Phi$ is CDF of Standard Normal Distribution

$$I_2 = \int_0^\infty \frac{r_{x_{\text{in}}}^l y_{\text{in}_j}^2}{\sqrt{2\pi}\sigma_{x_{\text{in}}}} \exp\left(\frac{-y_{\text{in}_j}^2}{2\sigma_{x_{\text{in}}}^2}\right) \int_{r_{x_{\text{in}}}^l y_{\text{in}_j}}^{-\infty} \frac{-1}{\sqrt{2\pi}\sigma_{x_{\text{in}}}\sqrt{1-(r_{x_{\text{in}}}^l)^2}} \exp\left(\frac{-p^2}{2\sigma_{x_{\text{in}}}^2(1-(r_{x_{\text{in}}}^l)^2)}\right) dp\, dy_{\text{in}_j}$$

$$= \int_0^\infty \frac{r_{x_{\text{in}}}^l y_{\text{in}_j}^2}{\sqrt{2\pi}\sigma_{x_{\text{in}}}} \exp\left(\frac{-y_{\text{in}_j}^2}{2\sigma_{x_{\text{in}}}^2}\right) \int_{-\infty}^{r_{x_{\text{in}}}^l y_{\text{in}_j}} \frac{1}{\sqrt{2\pi}\sigma_{x_{\text{in}}}\sqrt{1-(r_{x_{\text{in}}}^l)^2}} \exp\left(\frac{-p^2}{2\sigma_{x_{\text{in}}}^2(1-(r_{x_{\text{in}}}^l)^2)}\right) dp\, dy_{\text{in}_j}$$

$$= \int_0^\infty \frac{r_{x_{\text{in}}}^l y_{\text{in}_j}^2}{\sqrt{2\pi}\sigma_{x_{\text{in}}}} \exp\left(\frac{-y_{\text{in}_j}^2}{2\sigma_{x_{\text{in}}}^2}\right)\Phi\left(\frac{r_{x_{\text{in}}}^l y_{\text{in}_j}}{\sigma_{x_{\text{in}}}\sqrt{1-(r_{x_{\text{in}}}^l)^2}}\right) dy_{\text{in}_j}$$

$$= \int_0^\infty \frac{r_{x_{\text{in}}}^l y_{\text{in}_j}^2}{\sqrt{2\pi}\sigma_{x_{\text{in}}}} \exp\left(\frac{-y_{\text{in}_j}^2}{2\sigma_{x_{\text{in}}}^2}\right)\left[\frac{1}{2}\left(1+\text{erf}\left(\frac{r_{x_{\text{in}}}^l y_{\text{in}_j}}{\sigma_{x_{\text{in}}}\sqrt{2(1-(r_{x_{\text{in}}}^l)^2)}}\right)\right)\right] dy_{\text{in}_j}$$

$$= \frac{r_{x_{\text{in}}}^l}{2} \int_0^\infty \frac{y_{\text{in}_j}^2}{\sqrt{2\pi}\sigma_{x_{\text{in}}}} \exp\left(\frac{-y_{\text{in}_j}^2}{2\sigma_{x_{\text{in}}}^2}\right) dy_{\text{in}_j} +$$

$$\frac{r_{x_{\text{in}}}^l}{2\sqrt{2\pi}\sigma_{x_{\text{in}}}} \int_0^\infty y_{\text{in}_j}^2 \exp\left(\frac{-y_{\text{in}_j}^2}{2\sigma_{x_{\text{in}}}^2}\right) \text{erf}\left(\frac{r_{x_{\text{in}}}^l y_{\text{in}_j}}{\sigma_{x_{\text{in}}}\sqrt{2(1-(r_{x_{\text{in}}}^l)^2)}}\right) dy_{\text{in}_j}$$

Let us define $I_{2,1}$ and $I_{2,2}$ as

$$I_{2,1} = \frac{r_{x_{\text{in}}}^l}{2} \int_0^\infty \frac{y_{\text{in}_j}^2}{\sqrt{2\pi}\sigma_{x_{\text{in}}}} \exp\left(\frac{-y_{\text{in}_j}^2}{2\sigma_{x_{\text{in}}}^2}\right) dy_{\text{in}_j}$$

$$I_{2,2} = \frac{r_{x_{\text{in}}}^l}{2\sqrt{2\pi}\sigma_{x_{\text{in}}}} \int_0^\infty y_{\text{in}_j}^2 \exp\left(\frac{-y_{\text{in}_j}^2}{2\sigma_{x_{\text{in}}}^2}\right) \text{erf}\left(\frac{r_{x_{\text{in}}}^l y_{\text{in}_j}}{\sigma_{x_{\text{in}}}\sqrt{2(1-(r_{x_{\text{in}}}^l)^2)}}\right) dy_{\text{in}_j}$$

$$I_{2,1} = \frac{r_{x_{\text{in}}}^l}{2} \int_0^\infty \frac{y_{\text{in}_j}^2}{\sqrt{2\pi}\sigma_{x_{\text{in}}}} \exp\left(\frac{-y_{\text{in}_j}^2}{2\sigma_{x_{\text{in}}}^2}\right) dy_{\text{in}_j}$$

$$I_{2,1} = \frac{r_{x_{\text{in}}}^l \sigma_{x_{\text{in}}}^2}{4} \qquad \text{(Same integral as in variance calculation)}$$

From Ng & Geller (1969) we have $\int_0^\infty x^2 \exp\left(-b^2 x^2\right) \text{erf}(ax) dx = \frac{\sqrt{\pi}}{4b^3} - \frac{\tan^{-1}\left(\frac{b}{a}\right)}{2\sqrt{\pi}b^3} + \frac{a}{2\sqrt{\pi}b^2(a^2+b^2)}$.

Hence, putting $a = \frac{r_{x_{\text{in}}}^l}{\sigma_{x_{\text{in}}}\sqrt{2(1-(r_{x_{\text{in}}}^l)^2)}}$ and $b = \frac{1}{\sigma_{x_{\text{in}}}\sqrt{2}}$ we get,

$$I_{2,2} = \frac{r_{x_{\text{in}}}^l}{2\sqrt{2\pi}\sigma_{x_{\text{in}}}}\left[\frac{2\sqrt{2}\sigma_{x_{\text{in}}}^3}{4} - \frac{\tan^{-1}\left(\frac{\sqrt{(1-(r_{x_{\text{in}}}^l)^2)}}{r_{x_{\text{in}}}^l}\right)2\sqrt{2}\sigma_{x_{\text{in}}}^3}{2\sqrt{\pi}} + \frac{\sqrt{2}r_{x_{\text{in}}}^l\sigma_{x_{\text{in}}}^3\sqrt{(1-(r_{x_{\text{in}}}^l)^2)}}{\sqrt{\pi}}\right]$$

$$= \frac{r_{x_{\text{in}}}^l\sigma_{x_{\text{in}}}^2}{4} - \frac{r_{x_{\text{in}}}^l\cos^{-1}(r_{x_{\text{in}}}^l)\sigma_{x_{\text{in}}}^2}{2\pi} + \frac{(r_{x_{\text{in}}}^l)^2\sqrt{(1-(r_{x_{\text{in}}}^l)^2)}\sigma_{x_{\text{in}}}^2}{2\pi}$$

$$\mathbb{E}[x_{\text{out}_j} y_{\text{out}_j}] = I_1 + I_{2,1} + I_{2,2}$$

$$= \frac{(1-(r_{x_{\text{in}}}^l)^2)^{\frac{3}{2}}\sigma_{x_{\text{in}}}^2}{2\pi} + 2 * \frac{r_{x_{\text{in}}}^l\sigma_{x_{\text{in}}}^2}{4} - \frac{r_{x_{\text{in}}}^l\cos^{-1}(r_{x_{\text{in}}}^l)\sigma_{x_{\text{in}}}^2}{2\pi} + \frac{(r_{x_{\text{in}}}^l)^2\sqrt{(1-(r_{x_{\text{in}}}^l)^2)}\sigma_{x_{\text{in}}}^2}{2\pi}$$

$$= \frac{r_{x_{\text{in}}}^l\sigma_{x_{\text{in}}}^2}{2} - \frac{r_{x_{\text{in}}}^l\cos^{-1}(r_{x_{\text{in}}}^l)\sigma_x^2}{2\pi} + \frac{\sqrt{(1-(r_{x_{\text{in}}}^l)^2)}\sigma_{x_{\text{in}}}^2}{2\pi}$$

$$\text{Corr}(x_{\text{out}_j}, y_{\text{out}_j}) = \frac{\mathbb{E}[x_{\text{out}_j} y_{\text{out}_j}] - \mathbb{E}[x_{\text{out}_j}]\mathbb{E}[y_{\text{out}_j}]}{\sqrt{\text{Var}(x_{\text{out}_j})\text{Var}(y_{\text{out}_j})}}$$

$$= \frac{\frac{r_{x_{\text{in}}}^l\sigma_{x_{\text{in}}}^2}{2} - \frac{r_{x_{\text{in}}}^l\cos^{-1}(r_{x_{\text{in}}}^l)\sigma_{x_{\text{in}}}^2}{2\pi} + \frac{\sqrt{(1-(r_{x_{\text{in}}}^l)^2)}\sigma_x^2}{2\pi} - \frac{\sigma_{x_{\text{in}}}^2}{2\pi}}{\frac{\sigma_{x_{\text{in}}}^2}{2}\left(1-\frac{1}{\pi}\right)}$$

$$\boxed{r_{x_{\text{out}}}^l = \frac{\frac{\pi r_{x_{\text{in}}}^l}{2} + r_{x_{\text{in}}}^l\sin^{-1}(r_{x_{\text{in}}}^l) + \sqrt{(1-(r_{x_{\text{in}}}^l)^2)} - 1}{\pi - 1}}$$

Backward pass on ReLU can be defined as,

$$g_{\text{in}_i} = \begin{cases} g_{\text{out}_i} & \text{if } x_{\text{in}_i} > 0 \text{ (which has probability } \frac{1}{2}) \\ 0 & \text{else} \end{cases}$$

Assuming $\mu_{g_{\text{out}}} = 0$,

$$\mathbb{E}[g_{\text{in}_i}] = \frac{1}{2} * 0 + \frac{1}{2} * \mathbb{E}[g_{\text{out}_i}]$$

$$\boxed{\mu_{g_{\text{in}}} = 0}$$

$$\text{Var}(g_{\text{in}_i}) = \mathbb{E}[g_{\text{in}_i}^2] - \mathbb{E}[g_{\text{in}_i}]^2 = \mathbb{E}[g_{\text{in}_i}^2]$$

$$= \frac{1}{2} * 0 + \frac{1}{2} * \mathbb{E}[g_{\text{out}}^2]$$

$$\boxed{\sigma_{g_{\text{in}}}^2 = \frac{\sigma_{g_{\text{out}}}^2}{2}}$$

If for two inputs $\mathbf{x}_{\text{in}}$ and $\mathbf{y}_{\text{in}}$ for all $i$ we have $\text{Corr}(g_{\text{out}_{x_i}}, g_{\text{out}_{y_i}}) = r_{g_{\text{out}}}^l$, and $g_{\text{in}_{x_i}}, g_{\text{in}_{y_i}}$ be the gradient after passing through ReLU layer. Then we have,

$$\mathbb{E}[g_{\text{in}_{x_i}} g_{\text{in}_{y_i}}] = \mathbb{P}(x_{\text{in}_i} > 0, y_{\text{in}_i} > 0)\mathbb{E}[g_{\text{out}_{x_i}} g_{\text{out}_{y_i}}]$$

$$= \mathbb{P}(x_{\text{in}_i} > 0, y_{\text{in}_i} > 0)r_{g_{\text{out}}}^l \sigma_{g_{\text{out}}}^2$$

$$\mathbb{P}(x_{\text{in}_i} > 0, y_{\text{in}_i} > 0) =$$

$$= \int_0^\infty \int_0^\infty \frac{x_{\text{in}_i} y_{\text{in}_i}}{2\pi\sigma_{x_{\text{in}}}^2 \sqrt{1 - (r_{x_{\text{in}}}^l)^2}} \exp\left(\frac{-(x_{\text{in}_i}^2 + y_{\text{in}_i}^2 - 2r_{x_{\text{in}}}^l x_{\text{in}_i} y_{\text{in}_i})}{2\sigma_{x_{\text{in}}}^2(1 - (r_{x_{\text{in}}}^l)^2)}\right)dx_{\text{in}_i} dy_{\text{in}_i}$$

$$= \int_0^\infty \int_0^\infty \frac{x_{\text{in}_i} y_{\text{in}_i}}{2\pi\sigma_{x_{\text{in}}}^2 \sqrt{1 - (r_{x_{\text{in}}}^l)^2}} \exp\left(\frac{-(x_{\text{in}_i} - r_{x_{\text{in}}}^l y_{\text{in}_i})^2}{2\sigma_{x_{\text{in}}}^2(1 - (r_{x_{\text{in}}}^l)^2)}\right) \exp\left(\frac{-y_{\text{in}_i}^2}{2\sigma_{x_{\text{in}}}^2}\right)dx_{\text{in}_i} dy_{\text{in}_i}$$

Substituting $t = x_{\text{in}_i} - r_{x_{\text{in}}}^l y_{\text{in}_i}$, and assuming $y_{\text{in}_i}$ is constant for the inner integral, $dx_{\text{in}_i} = dt$

$$\mathbb{P}(x_{\text{in}_i} > 0, y_{\text{in}_i} > 0) =$$
$$\int_0^\infty \frac{1}{\sqrt{2\pi}\sigma_{x_{\text{in}}}} \exp\left(\frac{-y_{\text{in}_i}^2}{2\sigma_{x_{\text{in}}}^2}\right) \int_{-r_{x_{\text{in}}}^l y_{\text{in}_i}}^\infty \frac{1}{\sqrt{2\pi}\sigma_{x_{\text{in}}}\sqrt{1 - (r_{x_{\text{in}}}^l)^2}} \exp\left(\frac{-t^2}{2\sigma_{x_{\text{in}}}^2(1 - (r_{x_{\text{in}}}^l)^2)}\right)dt dy_{\text{in}_i}$$

Substituting $p = -t$, where $\Phi$ is CDF of Standard Normal Distribution

$$\mathbb{P}(x_{\text{in}_i} > 0, y_{\text{in}_i} > 0) =$$

$$= \int_0^\infty \frac{1}{\sqrt{2\pi}\sigma_{x_{\text{in}}}} \exp\left(\frac{-y_{\text{in}_i}^2}{2\sigma_{x_{\text{in}}}^2}\right) \int_{r_{x_{\text{in}}}^l y_{\text{in}_i}}^{-\infty} \frac{-1}{\sqrt{2\pi}\sigma_{x_{\text{in}}}\sqrt{1 - (r_{x_{\text{in}}}^l)^2}} \exp\left(\frac{-p^2}{2\sigma_{x_{\text{in}}}^2(1 - (r_{x_{\text{in}}}^l)^2)}\right)dp dy_{\text{in}_i}$$

$$= \int_0^\infty \frac{1}{\sqrt{2\pi}\sigma_{x_{\text{in}}}} \exp\left(\frac{-y_{\text{in}_i}^2}{2\sigma_{x_{\text{in}}}^2}\right) \int_{-\infty}^{r_{x_{\text{in}}}^l y_{\text{in}_i}} \frac{1}{\sqrt{2\pi}\sigma_{x_{\text{in}}}\sqrt{1 - (r_{x_{\text{in}}}^l)^2}} \exp\left(\frac{-p^2}{2\sigma_{x_{\text{in}}}^2(1 - (r_{x_{\text{in}}}^l)^2)}\right)dp dy_{\text{in}_i}$$

$$= \int_0^\infty \frac{1}{\sqrt{2\pi}\sigma_{x_{\text{in}}}} \exp\left(\frac{-y_{\text{in}_i}^2}{2\sigma_{x_{\text{in}}}^2}\right)\Phi\left(\frac{r_{x_{\text{in}}}^l y_{\text{in}_i}}{\sigma_{x_{\text{in}}}\sqrt{1 - (r_{x_{\text{in}}}^l)^2}}\right)dy_{\text{in}_i}$$

$$= \int_0^\infty \frac{1}{\sqrt{2\pi}\sigma_{x_{\text{in}}}} \exp\left(\frac{-y_{\text{in}_i}^2}{2\sigma_{x_{\text{in}}}^2}\right)\left[\frac{1}{2}\left(1 + \text{erf}\left(\frac{r_{x_{\text{in}}}^l y_{\text{in}_i}}{\sigma_{x_{\text{in}}}\sqrt{2(1 - (r_{x_{\text{in}}}^l)^2)}}\right)\right)\right]dy_{\text{in}_i}$$

$$= \frac{1}{2}\int_0^\infty \frac{1}{\sqrt{2\pi}\sigma_{x_{\text{in}}}} \exp\left(\frac{-y_{\text{in}_i}^2}{2\sigma_{x_{\text{in}}}^2}\right)dy_{\text{in}_i} + \frac{1}{2\sqrt{2\pi}\sigma_{x_{\text{in}}}}\int_0^\infty \exp\left(\frac{-y_{\text{in}_i}^2}{2\sigma_{x_{\text{in}}}^2}\right)\text{erf}\left(\frac{r_{x_{\text{in}}}^l y_{\text{in}_i}}{\sigma_{x_{\text{in}}}\sqrt{2(1 - (r_{x_{\text{in}}}^l)^2)}}\right)dy_{\text{in}_i}$$

$$= \frac{1}{4} + \frac{1}{2\sqrt{2\pi}\sigma_{x_{\text{in}}}} \int_0^\infty \exp\left(\frac{-y_{\text{in}_i}^2}{2\sigma_{x_{\text{in}}}^2}\right) \text{erf}\left(\frac{r_{x_{\text{in}}}^l y_{\text{in}_i}}{\sigma_{x_{\text{in}}}\sqrt{2(1-(r_{x_{\text{in}}}^l)^2)}}\right) dy_{\text{in}_i}$$

From Ng & Geller (1969) we have $\int_0^\infty \exp\left(-b^2 x^2\right)\text{erf}(ax)dx = \frac{\sqrt{\pi}}{2b} - \frac{1}{b\sqrt{\pi}} \tan^{-1}\left(\frac{b}{a}\right)$

Putting $a = \dfrac{r_{x_{\text{in}}}^l}{\sigma_{x_{\text{in}}}\sqrt{2(1-(r_{x_{\text{in}}}^l)^2)}}$ and $b = \dfrac{1}{\sigma_{x_{\text{in}}}\sqrt{2}}$ we get,

$$\mathbb{P}(x_{\text{in}_i} > 0, y_{\text{in}_i} > 0) = \frac{1}{4} + \frac{1}{2\sqrt{2\pi}\sigma_{x_{\text{in}}}}\left[\frac{\sqrt{\pi}\sigma_{x_{\text{in}}}\sqrt{2}}{2} - \frac{\sigma_{x_{\text{in}}}\sqrt{2}}{\sqrt{\pi}} \tan^{-1}\left(\frac{\sqrt{(1-(r_{x_{\text{in}}}^l)^2)}}{r_{x_{\text{in}}}^l}\right)\right]$$

$$= \frac{1}{4} + \frac{1}{2\pi}\left[\frac{\pi}{2} - \cos^{-1}\left(r_{x_{\text{in}}}^l\right)\right]$$

$$= \frac{1}{4} + \frac{\sin^{-1}\left(r_{x_{\text{in}}}^l\right)}{2\pi}$$

$$\implies \mathbb{E}[g_{\text{in}_{x_i}} g_{\text{in}_{y_i}}] = \left(\frac{1}{4} + \frac{\sin^{-1}\left(r_{x_{\text{in}}}^l\right)}{2\pi}\right) r_{g_{\text{out}}}^l \sigma_{g_{\text{out}}}^2$$

$$\text{Corr}(g_{\text{in}_{x_i}}, g_{\text{in}_{y_i}}) = \frac{\left(\frac{1}{4} + \frac{\sin^{-1}\left(r_{x_{\text{in}}}^l\right)}{2\pi}\right) r_{g_{\text{out}}}^l \sigma_{g_{\text{out}}}^2}{\frac{\sigma_{g_{\text{out}}}^2}{2}}$$

$$\boxed{r_{g_{\text{out}}}^l = \left(\frac{1}{2} + \frac{\sin^{-1}\left(r_{x_{\text{in}}}^l\right)}{\pi}\right) r_{g_{\text{out}}}^l}$$

### C.5 GELU

Forward pass through GeLU is defined as,

$$\mathbf{x}_{\text{out}} = \text{GeLU}(\mathbf{x}_{\text{in}})$$

$$\implies x_{\text{out}_i} = x_{\text{in}_i} \Phi(x_{\text{in}_i})$$

where $\Phi(x)$ is CDF of Standard Normal Distribution at $x$

$$= \frac{x_{\text{in}_i}}{2}\left(1 + \text{erf}\left(\frac{x_{\text{in}_i}}{\sqrt{2}}\right)\right)$$

To get the mean of output of GeLU, we have

$$\mathbb{E}[x_{\text{out}_i}] = \int_{-\infty}^\infty \frac{x_{\text{out}_i}}{\sqrt{2\pi}\sigma_{x_{\text{in}}}} \exp\left(\frac{-x_{\text{in}_i}^2}{2\sigma_{x_{\text{in}}}^2}\right) dx_{\text{in}_i}$$

$$= \int_{-\infty}^\infty \frac{x_{\text{in}_i}\left(1 + \text{erf}\left(\frac{x_{\text{in}_i}}{\sqrt{2}}\right)\right)}{2\sqrt{2\pi}\sigma_{x_{\text{in}}}} \exp\left(\frac{-x_{\text{in}_i}^2}{2\sigma_{x_{\text{in}}}^2}\right) dx_{\text{in}_i}$$

$$= \int_{-\infty}^\infty \frac{x_{\text{in}_i}}{2\sqrt{2\pi}\sigma_{x_{\text{in}}}} \exp\left(\frac{-x_{\text{in}_i}^2}{2\sigma_{x_{\text{in}}}^2}\right) dx_{\text{in}_i} + \int_{-\infty}^\infty \frac{x_{\text{in}_i}\text{erf}\left(\frac{x_{\text{in}_i}}{\sqrt{2}}\right)}{2\sqrt{2\pi}\sigma_{x_{\text{in}}}} \exp\left(\frac{-x_{\text{in}_i}^2}{2\sigma_{x_{\text{in}}}^2}\right) dx_{\text{in}_i}$$

$$= \int_{-\infty}^\infty \frac{x_{\text{in}_i}\text{erf}\left(\frac{x_{\text{in}_i}}{\sqrt{2}}\right)}{2\sqrt{2\pi}\sigma_{x_{\text{in}}}} \exp\left(\frac{-x_{\text{in}_i}^2}{2\sigma_{x_{\text{in}}}^2}\right) dx_{\text{in}_i} \qquad \text{(Integral of odd function)}$$

$$= \frac{1}{2\sqrt{2\pi}\sigma_{x_{\text{in}}}} \int_{-\infty}^\infty x_{\text{in}_i}\text{erf}\left(\frac{x_{\text{in}_i}}{\sqrt{2}}\right) \exp\left(\frac{-x_{\text{in}_i}^2}{2\sigma_{x_{\text{in}}}^2}\right) dx_{\text{in}_i}$$

From 2.6.1.4 of Lipovetsky (2020), $\int_{-\infty}^\infty z\text{erf}(az) \exp\left(-a_1 z^2\right)dz = \dfrac{a}{a_1\sqrt{a^2 + a_1}}$

Substituting, $a = \dfrac{1}{\sqrt{2}}, a_1 = \dfrac{1}{2\sigma_{x_{\text{in}}}^2}$, we have

$$\mathbb{E}[x_{\text{out}_i}] = \frac{1}{2\sqrt{2\pi}\sigma_{x_{\text{in}}}} \frac{\frac{1}{\sqrt{2}}}{\frac{1}{2\sigma_{x_{\text{in}}}^2}\sqrt{\frac{1}{2} + \frac{1}{2\sigma_{x_{\text{in}}}^2}}}$$

$$= \frac{1}{2\sqrt{2\pi}\sigma_{x_{\text{in}}}} \frac{2\sigma_{x_{\text{in}}}^3}{\sqrt{\sigma_{x_{\text{in}}}^2 + 1}}$$

$$\boxed{\mu_{x_{\text{out}}} = \frac{\sigma_{x_{\text{in}}}^2}{\sqrt{2\pi(\sigma_{x_{\text{in}}}^2 + 1)}}}$$

For calculating variance of output,

$$\mathbb{E}[x_{\text{out}_i}^2] = \int_{-\infty}^{\infty} \frac{x_{\text{out}_i}^2}{\sqrt{2\pi}\sigma_{x_{\text{in}}}} \exp\left(\frac{-x_{\text{in}_i}^2}{2\sigma_{x_{\text{in}}}^2}\right) dx_{\text{in}_i}$$

$$= \int_{-\infty}^{\infty} \frac{x_{\text{in}_i}^2 (1 + \text{erf}(\frac{x_{\text{in}_i}}{\sqrt{2}}))^2}{4\sqrt{2\pi}\sigma_{x_{\text{in}}}} \exp\left(\frac{-x_{\text{in}_i}^2}{2\sigma_{x_{\text{in}}}^2}\right) dx_{\text{in}_i}$$

$$= \int_{-\infty}^{\infty} \frac{x_{\text{in}_i}^2}{4\sqrt{2\pi}\sigma_{x_{\text{in}}}} \exp\left(\frac{-x_{\text{in}_i}^2}{2\sigma_{x_{\text{in}}}^2}\right) dx_{\text{in}_i}$$

$$+ \int_{-\infty}^{\infty} \frac{x_{\text{in}_i}^2 \text{erf}(\frac{x_{\text{in}_i}}{\sqrt{2}})}{2\sqrt{2\pi}\sigma_{x_{\text{in}}}} \exp\left(\frac{-x_{\text{in}_i}^2}{2\sigma_{x_{\text{in}}}^2}\right) dx_{\text{in}_i} + \int_{-\infty}^{\infty} \frac{x_{\text{in}_i}^2 \text{erf}^2(\frac{x_{\text{in}_i}}{\sqrt{2}})}{4\sqrt{2\pi}\sigma_{x_{\text{in}}}} \exp\left(\frac{-x_{\text{in}_i}^2}{2\sigma_{x_{\text{in}}}^2}\right) dx_{\text{in}_i}$$

$$= \frac{\sigma_{x_{\text{in}}}^2}{4} + \int_{-\infty}^{\infty} \frac{x_{\text{in}_i}^2 \text{erf}^2(\frac{x_{\text{in}_i}}{\sqrt{2}})}{4\sqrt{2\pi}\sigma_{x_{\text{in}}}} \exp\left(\frac{-x_{\text{in}_i}^2}{2\sigma_{x_{\text{in}}}^2}\right) dx_{\text{in}_i}$$

(Definition of variance, and integral of odd function)

$$= \frac{\sigma_{x_{\text{in}}}^2}{4} + \frac{1}{4\sqrt{2\pi}\sigma_{x_{\text{in}}}} \int_{-\infty}^{\infty} x_{\text{in}_i}^2 \text{erf}^2\left(\frac{x_{\text{in}_i}}{\sqrt{2}}\right) \exp\left(\frac{-x_{\text{in}_i}^2}{2\sigma_{x_{\text{in}}}^2}\right) dx_{\text{in}_i}$$

From 2.7.3.3 of Lipovetsky (2020)

$$\int_{-\infty}^{\infty} z^2 \exp\left(-az^2\right)\text{erf}(a_1 z)\text{erf}(a_2 z) =$$

$$\frac{1}{\sqrt{\pi}}\left(\frac{1}{a\sqrt{a}} \tan^{-1}\left(\frac{a_1 a_2}{\sqrt{a^2 + aa_1^2 + aa_2^2}}\right) + \frac{a_1 a_2 (2a + a_1^2 + a_2^2)}{a\sqrt{a + a_1^2 + a_2^2}(a^2 + aa_1^2 + aa_2^2 + a_1^2 a_2^2)}\right)$$

Substituting $a = \frac{1}{2\sigma_{x_{\text{in}}}^2}, a_1 = a_2 = \frac{1}{\sqrt{2}}$

$$\int_{-\infty}^{\infty} x_{\text{in}_i}^2 \text{erf}^2\left(\frac{x_{\text{in}_i}}{\sqrt{2}}\right) \exp\left(\frac{-x_{\text{in}_i}^2}{2\sigma_{x_{\text{in}}}^2}\right) dx_{\text{in}_i}$$

$$= \frac{1}{\sqrt{\pi}}\left(2\sqrt{2}\sigma_{x_{\text{in}}}^3 \tan^{-1}\left(\frac{\frac{1}{2}}{\sqrt{\frac{1}{4\sigma_{x_{\text{in}}}^4} + \frac{1}{2\sigma_{x_{\text{in}}}^2}}}\right) + \frac{\frac{1}{2}(\frac{1}{\sigma_{x_{\text{in}}}^2} + 1)}{\frac{1}{2\sigma_{x_{\text{in}}}^2}\sqrt{\frac{1}{2\sigma_{x_{\text{in}}}^2} + 1}(\frac{1}{4\sigma_{x_{\text{in}}}^4} + \frac{1}{2\sigma_{x_{\text{in}}}^2} + \frac{1}{4})}\right)$$

$$= \frac{1}{\sqrt{\pi}}\left(2\sqrt{2}\sigma_{x_{\text{in}}}^3 \tan^{-1}\left(\frac{\sigma_{x_{\text{in}}}^2}{\sqrt{(\sigma_{x_{\text{in}}}^2 + 1)^2 - \sigma_{x_{\text{in}}}^4}}\right) + \frac{4\sqrt{2}\sigma_{x_{\text{in}}}^5(\sigma_{x_{\text{in}}}^2 + 1)}{\sqrt{2\sigma_{x_{\text{in}}}^2 + 1}(\sigma_{x_{\text{in}}}^4 + 2\sigma_{x_{\text{in}}}^2 + 1)}\right)$$

$$= \frac{1}{\sqrt{\pi}}\left(2\sqrt{2}\sigma_{x_{\text{in}}}^3 \sin^{-1}\left(\frac{\sigma_{x_{\text{in}}}^2}{\sigma_{x_{\text{in}}}^2 + 1}\right) + \frac{4\sqrt{2}\sigma_{x_{\text{in}}}^5}{\sqrt{2\sigma_{x_{\text{in}}}^2 + 1}(\sigma_{x_{\text{in}}}^2 + 1)}\right)$$

$$= \frac{2\sqrt{2}\sigma_{x_{\text{in}}}^3}{\sqrt{\pi}}\left(\sin^{-1}\left(\frac{\sigma_{x_{\text{in}}}^2}{\sigma_{x_{\text{in}}}^2 + 1}\right) + \frac{2\sigma_{x_{\text{in}}}^2}{\sqrt{2\sigma_{x_{\text{in}}}^2 + 1}(\sigma_{x_{\text{in}}}^2 + 1)}\right)$$

$$\mathbb{E}[x_{\text{out}_i}^2] = \frac{\sigma_{x_{\text{in}}}^2}{4} + \frac{1}{4\sqrt{2\pi}\sigma_{x_{\text{in}}}} \int_{-\infty}^{\infty} x_{\text{in}_i}^2 \operatorname{erf}^2\left(\frac{x_{\text{in}_i}}{\sqrt{2}}\right) \exp\left(\frac{-x_{\text{in}_i}^2}{2\sigma_{x_{\text{in}}}^2}\right) dx_{\text{in}_i}$$

$$= \frac{\sigma_{x_{\text{in}}}^2}{4} + \frac{1}{4\sqrt{2\pi}\sigma_{x_{\text{in}}}} \frac{2\sqrt{2}\sigma_{x_{\text{in}}}^3}{\sqrt{\pi}} \left(\sin^{-1}\left(\frac{\sigma_{x_{\text{in}}}^2}{\sigma_{x_{\text{in}}}^2+1}\right) + \frac{2\sigma_{x_{\text{in}}}^2}{\sqrt{2\sigma_{x_{\text{in}}}^2+1}(\sigma_{x_{\text{in}}}^2+1)}\right)$$

$$\mathbb{E}[x_{\text{out}_i}^2] = \frac{\sigma_{x_{\text{in}}}^2}{4} + \frac{\sigma_{x_{\text{in}}}^2}{2\pi} \left(\sin^{-1}\left(\frac{\sigma_{x_{\text{in}}}^2}{\sigma_{x_{\text{in}}}^2+1}\right) + \frac{2\sigma_{x_{\text{in}}}^2}{\sqrt{2\sigma_{x_{\text{in}}}^2+1}(\sigma_{x_{\text{in}}}^2+1)}\right)$$

$$\operatorname{Var}(x_{\text{out}_i}) = \mathbb{E}[x_{\text{out}_i}^2] - (\mathbb{E}[x_{\text{out}_i}])^2$$

$$\boxed{\sigma_{x_{out}}^2 = \frac{\sigma_{x_{in}}^2}{2\pi}\left(\frac{\pi}{2} - \frac{\sigma_{x_{in}}^2}{1+\sigma_{x_{in}}^2} + \sin^{-1}\left(\frac{\sigma_{x_{in}}^2}{1+\sigma_{x_{in}}^2}\right) + \frac{2\sigma_{x_{in}}^2}{(1+\sigma_{x_{in}}^2)\sqrt{1+2\sigma_{x_{in}}^2}}\right)}$$

Now if we have two inputs $\mathbf{x}_{\text{in}}$ and $\mathbf{y}_{\text{in}}$ such that for all values of $i$, we have $\operatorname{Corr}(x_{\text{in}_i}, y_{\text{in}_i}) = r_{x_{\text{in}}}^l$, then we can calculate the covariance $\operatorname{Cov}(x_{\text{out}_j}, y_{\text{out}_j})$ for any $j$ as,

$$\operatorname{Cov}(x_{\text{out}_j}, y_{\text{out}_j}) = \mathbb{E}[x_{\text{out}_j} y_{\text{out}_j}] - \mathbb{E}[x_{\text{out}_j}]\mathbb{E}[y_{\text{out}_j}]$$

$$\mathbb{E}[x_{\text{out}_j} y_{\text{out}_j}]$$

$$= \iint_{-\infty}^{\infty} \frac{x_{\text{out}_j} y_{\text{out}_j}}{2\pi\sigma_{x_{\text{in}}}^2 \sqrt{(1-(r_{x_{\text{in}}}^l)^2)}} \exp\left(\frac{-x_{\text{in}_j}^2 + 2r_{x_{\text{in}}}^l x_{\text{in}_j} y_{\text{in}_j} - y_{\text{in}_j}^2}{2\sigma_{x_{\text{in}}}^2(1-(r_{x_{\text{in}}}^l)^2)}\right) dx_{\text{in}_j} dy_{\text{in}_j} = I$$

$$= \iint_{-\infty}^{\infty} \frac{x_{\text{in}_j}(1+\operatorname{erf}(\frac{x_{\text{in}_j}}{\sqrt{2}}))y_{\text{in}_j}(1+\operatorname{erf}(\frac{y_{\text{in}_j}}{\sqrt{2}}))}{8\pi\sigma_{x_{\text{in}}}^2 \sqrt{(1-(r_{x_{\text{in}}}^l)^2)}} \exp\left(\frac{-x_{\text{in}_j}^2 + 2r_{x_{\text{in}}}^l x_{\text{in}_j} y_{\text{in}_j} - y_{\text{in}_j}^2}{2\sigma_{x_{\text{in}}}^2(1-(r_{x_{\text{in}}}^l)^2)}\right) dx_{\text{in}_j} dy_{\text{in}_j}$$

$$= \int_{-\infty}^{\infty} \frac{y_{\text{in}_j}(1+\operatorname{erf}(\frac{y_{\text{in}_j}}{\sqrt{2}}))}{8\pi\sigma_{x_{\text{in}}}^2 \sqrt{(1-(r_{x_{\text{in}}}^l)^2)}} \exp\left(\frac{-y_{\text{in}_j}^2}{2\sigma_{x_{\text{in}}}^2(1-(r_{x_{\text{in}}}^l)^2)}\right) I_X \, dy_{\text{in}_j}$$

Where $I_X = \int_{-\infty}^{\infty} x_{\text{in}_j}(1+\operatorname{erf}(\frac{x_{\text{in}_j}}{\sqrt{2}})) \exp\left(\frac{-x_{\text{in}_j}^2 + 2r_{x_{\text{in}}}^l x_{\text{in}_j} y_{\text{in}_j}}{2\sigma_{x_{\text{in}}}^2(1-(r_{x_{\text{in}}}^l)^2)}\right) dx_{\text{in}_j}$

$$I_X = \int_{-\infty}^{\infty} x_{\text{in}_j}(1+\operatorname{erf}(\frac{x_{\text{in}_j}}{\sqrt{2}})) \exp\left(\frac{-x_{\text{in}_j}^2 + 2r_{x_{\text{in}}}^l x_{\text{in}_j} y_{\text{in}_j}}{2\sigma_{x_{\text{in}}}^2(1-(r_{x_{\text{in}}}^l)^2)}\right) dx_{\text{in}_j}$$

$$= \int_{-\infty}^{\infty} x_{\text{in}_j} \exp\left(\frac{-x_{\text{in}_j}^2 + 2r_{x_{\text{in}}}^l x_{\text{in}_j} y_{\text{in}_j}}{2\sigma_{x_{\text{in}}}^2(1-(r_{x_{\text{in}}}^l)^2)}\right) dx_{\text{in}_j} +$$

$$\int_{-\infty}^{\infty} x_{\text{in}_j} \operatorname{erf}(\frac{x_{\text{in}_j}}{\sqrt{2}}) \exp\left(\frac{-x_{\text{in}_j}^2 + 2r_{x_{\text{in}}}^l x_{\text{in}_j} y_{\text{in}_j}}{2\sigma_{x_{\text{in}}}^2(1-(r_{x_{\text{in}}}^l)^2)}\right) dx_{\text{in}_j}$$

Let, $I_{X,1} = \int_{-\infty}^{\infty} x_{\text{in}_j} \exp\left(\frac{-x_{\text{in}_j}^2 + 2r_{x_{\text{in}}}^l x_{\text{in}_j} y_{\text{in}_j}}{2\sigma_{x_{\text{in}}}^2(1-(r_{x_{\text{in}}}^l)^2)}\right) dx_{\text{in}_j}$

$$I_{X,2} = \int_{-\infty}^{\infty} x_{\text{in}_j} \operatorname{erf}(\frac{x_{\text{in}_j}}{\sqrt{2}}) \exp\left(\frac{-x_{\text{in}_j}^2 + 2r_{x_{\text{in}}}^l x_{\text{in}_j} y_{\text{in}_j}}{2\sigma_{x_{\text{in}}}^2(1-(r_{x_{\text{in}}}^l)^2)}\right) dx_{\text{in}_j}$$

$$I_{X,1} = \int_{-\infty}^{\infty} x_{\text{in}_j} \exp\left(\frac{-x_{\text{in}_j}^2 + 2r_{x_{\text{in}}}^l x_{\text{in}_j} y_{\text{in}_j}}{2\sigma_{x_{\text{in}}}^2(1-(r_{x_{\text{in}}}^l)^2)}\right) dx_{\text{in}_j}$$

$$= \int_{-\infty}^{\infty} x_{\text{in}_j} \exp\left(\frac{-x_{\text{in}_j}^2 + 2r_{x_{\text{in}}}^l x_{\text{in}_j} y_{\text{in}_j}}{2\sigma_{x_{\text{in}}}^2(1-(r_{x_{\text{in}}}^l)^2)}\right) \exp\left(\frac{-(r_{x_{\text{in}}}^l)^2 y_{\text{in}_j}^2}{2\sigma_{x_{\text{in}}}^2(1-(r_{x_{\text{in}}}^l)^2)}\right) \exp\left(\frac{(r_{x_{\text{in}}}^l)^2 y_{\text{in}_j}^2}{2\sigma_{x_{\text{in}}}^2(1-(r_{x_{\text{in}}}^l)^2)}\right) dx_{\text{in}_j}$$

$$= \exp\left(\frac{(r_{x_{\text{in}}}^l)^2 y_{\text{in}_j}^2}{2\sigma_{x_{\text{in}}}^2(1-(r_{x_{\text{in}}}^l)^2)}\right) \int_{-\infty}^{\infty} x_{\text{in}_j} \exp\left(\frac{-(x_{\text{in}_j} - r_{x_{\text{in}}}^l y_{\text{in}_j})^2}{2\sigma_{x_{\text{in}}}^2(1-(r_{x_{\text{in}}}^l)^2)}\right) dx_{\text{in}_j}$$

$$= \int_{-\infty}^{\infty} \frac{x_{\text{in}_j}}{\sqrt{2\pi}\sigma_{x_{\text{in}}}\sqrt{(1-(r_{x_{\text{in}}}^l)^2)}} \exp\left(\frac{-(x_{\text{in}_j} - r_{x_{\text{in}}}^l y_{\text{in}_j})^2}{2\sigma_{x_{\text{in}}}^2(1-(r_{x_{\text{in}}}^l)^2)}\right) dx_{\text{in}_j}$$

$$= r_{x_{\text{in}}}^l y_{\text{in}_j} \sqrt{2\pi}\sigma_{x_{\text{in}}}\sqrt{(1-(r_{x_{\text{in}}}^l)^2)} \exp\left(\frac{(r_{x_{\text{in}}}^l)^2 y_{\text{in}_j}^2}{2\sigma_{x_{\text{in}}}^2(1-(r_{x_{\text{in}}}^l)^2)}\right)$$

$$I_{X,2} = \int_{-\infty}^{\infty} x_{\text{in}_j} \operatorname{erf}\left(\frac{x_{\text{in}_j}}{\sqrt{2}}\right) \exp\left(\frac{-x_{\text{in}_j}^2 + 2r_{x_{\text{in}}}^l x_{\text{in}_j} y_{\text{in}_j}}{2\sigma_{x_{\text{in}}}^2(1-(r_{x_{\text{in}}}^l)^2)}\right) dx_{\text{in}_j}$$

From 2.7.2.4 of Lipovetsky (2020),

$$\int_{-\infty}^{\infty} z\operatorname{erf}(a_1 z) \exp\left(-az^2 + bz\right) dz =$$

$$= \frac{\sqrt{\pi}b}{2a\sqrt{a}} \exp\left(\frac{b^2}{4a}\right) \operatorname{erf}\left(\frac{a_1 b}{2\sqrt{a^2 + aa_1^2}}\right) + \frac{a_1}{a\sqrt{a + a_1^2}} \exp\left(\frac{b^2}{4a + 4a_1^2}\right)$$

Substituting $a_1 = \frac{1}{\sqrt{2}}, a = \frac{1}{2\sigma_{x_{\text{in}}}^2(1-(r_{x_{\text{in}}}^l)^2)}, b = \frac{r_{x_{\text{in}}}^l y_{\text{in}_j}}{\sigma_{x_{\text{in}}}^2(1-(r_{x_{\text{in}}}^l)^2)}$, we get

$$I_{X,2} = \frac{\sqrt{\pi}\frac{r_{x_{\text{in}}}^l y_{\text{in}_j}}{\sigma_{x_{\text{in}}}^2(1-(r_{x_{\text{in}}}^l)^2)}}{2\frac{1}{2\sqrt{2}\sigma_{x_{\text{in}}}^3(1-(r_{x_{\text{in}}}^l)^2)^{\frac{3}{2}}}} \exp\left(\frac{\frac{(r_{x_{\text{in}}}^l)^2 y_{\text{in}_j}^2}{\sigma_{x_{\text{in}}}^4(1-(r_{x_{\text{in}}}^l)^2)^2}}{4\frac{1}{2\sigma_{x_{\text{in}}}^2(1-(r_{x_{\text{in}}}^l)^2)}}\right) \operatorname{erf}\left(\frac{\frac{r_{x_{\text{in}}}^l y_{\text{in}_j}}{\sqrt{2}\sigma_{x_{\text{in}}}^2(1-(r_{x_{\text{in}}}^l)^2)}}{2\sqrt{\frac{1}{4\sigma_{x_{\text{in}}}^4(1-(r_{x_{\text{in}}}^l)^2)^2} + \frac{1}{4\sigma_{x_{\text{in}}}^2(1-(r_{x_{\text{in}}}^l)^2)}}}\right)$$

$$+ \frac{\frac{1}{\sqrt{2}}}{\frac{1}{2\sigma_{x_{\text{in}}}^2(1-(r_{x_{\text{in}}}^l)^2)}\sqrt{\frac{1}{2\sigma_{x_{\text{in}}}^2(1-(r_{x_{\text{in}}}^l)^2)} + \frac{1}{2}}} \exp\left(\frac{\frac{(r_{x_{\text{in}}}^l)^2 y_{\text{in}_j}^2}{\sigma_{x_{\text{in}}}^4(1-(r_{x_{\text{in}}}^l)^2)^2}}{4\frac{1}{2\sigma_{x_{\text{in}}}^2(1-(r_{x_{\text{in}}}^l)^2)} + \frac{4}{2}}\right)$$

$$= r_{x_{\text{in}}}^l y_{\text{in}_j} \sqrt{2\pi}\sigma_{x_{\text{in}}}\sqrt{(1-(r_{x_{\text{in}}}^l)^2)} \exp\left(\frac{(r_{x_{\text{in}}}^l)^2 y_{\text{in}_j}^2}{2\sigma_{x_{\text{in}}}^2(1-(r_{x_{\text{in}}}^l)^2)}\right) \operatorname{erf}\left(\frac{r_{x_{\text{in}}}^l y_{\text{in}_j}}{\sqrt{2(\sigma_{x_{\text{in}}}^2(1-(r_{x_{\text{in}}}^l)^2) + 1)}}\right)$$

$$+ \frac{2\sigma_{x_{\text{in}}}^3(1-(r_{x_{\text{in}}}^l)^2)^{\frac{3}{2}}}{\sqrt{\sigma_{x_{\text{in}}}^2(1-(r_{x_{\text{in}}}^l)^2) + 1}} \exp\left(\frac{(r_{x_{\text{in}}}^l)^2 y_{\text{in}_j}^2}{2(\sigma_{x_{\text{in}}}^2(1-(r_{x_{\text{in}}}^l)^2) + 1)\sigma_{x_{\text{in}}}^2(1-(r_{x_{\text{in}}}^l)^2)}\right)$$

Let us define $I_{X,2,1}$ and $I_{X,2,2}$ as:

$$I_{X,2,1} = r_{x_{\text{in}}}^l y_{\text{in}_j} \sqrt{2\pi}\sigma_{x_{\text{in}}}\sqrt{(1-(r_{x_{\text{in}}}^l)^2)} \exp\left(\frac{(r_{x_{\text{in}}}^l)^2 y_{\text{in}_j}^2}{2\sigma_{x_{\text{in}}}^2(1-(r_{x_{\text{in}}}^l)^2)}\right) \operatorname{erf}\left(\frac{r_{x_{\text{in}}}^l y_{\text{in}_j}}{\sqrt{2(\sigma_{x_{\text{in}}}^2(1-(r_{x_{\text{in}}}^l)^2) + 1)}}\right)$$

$$I_{X,2,2} = \frac{2\sigma_{x_{\text{in}}}^3(1-(r_{x_{\text{in}}}^l)^2)^{\frac{3}{2}}}{\sqrt{\sigma_{x_{\text{in}}}^2(1-(r_{x_{\text{in}}}^l)^2) + 1}} \exp\left(\frac{(r_{x_{\text{in}}}^l)^2 y_{\text{in}_j}^2}{2(\sigma_{x_{\text{in}}}^2(1-(r_{x_{\text{in}}}^l)^2) + 1)\sigma_{x_{\text{in}}}^2(1-(r_{x_{\text{in}}}^l)^2)}\right)$$

$$I = \int_{-\infty}^{\infty} \frac{y_{\text{in}_j}(1 + \operatorname{erf}(\frac{y_{\text{in}_j}}{\sqrt{2}}))}{8\pi\sigma_{x_{\text{in}}}^2\sqrt{(1-(r_{x_{\text{in}}}^l)^2)}} \exp\left(\frac{-y_{\text{in}_j}^2}{2\sigma_{x_{\text{in}}}^2(1-(r_{x_{\text{in}}}^l)^2)}\right) I_X \, dy_{\text{in}_j}$$

$$= \int_{-\infty}^{\infty} \frac{y_{\text{in}_j}(1 + \operatorname{erf}(\frac{y_{\text{in}_j}}{\sqrt{2}}))}{8\pi\sigma_{x_{\text{in}}}^2\sqrt{(1-(r_{x_{\text{in}}}^l)^2)}} \exp\left(\frac{-y_{\text{in}_j}^2}{2\sigma_{x_{\text{in}}}^2(1-(r_{x_{\text{in}}}^l)^2)}\right) (I_{X,1} + I_{X,2,1} + I_{X,2,2}) \, dy_{\text{in}_j}$$

$$I_1 = \int_{-\infty}^{\infty} \frac{y_{\text{in}_j}(1 + \operatorname{erf}(\frac{y_{\text{in}_j}}{\sqrt{2}}))}{8\pi\sigma_{x_{\text{in}}}^2\sqrt{(1-(r_{x_{\text{in}}}^l)^2)}} \exp\left(\frac{-y_{\text{in}_j}^2}{2\sigma_{x_{\text{in}}}^2(1-(r_{x_{\text{in}}}^l)^2)}\right) I_{X,1} \, dy_{\text{in}_j}$$

$$I_2 = \int_{-\infty}^{\infty} \frac{y_{\text{in}_j}(1 + \operatorname{erf}(\frac{y_{\text{in}_j}}{\sqrt{2}}))}{8\pi\sigma_{x_{\text{in}}}^2\sqrt{(1-(r_{x_{\text{in}}}^l)^2)}} \exp\left(\frac{-y_{\text{in}_j}^2}{2\sigma_{x_{\text{in}}}^2(1-(r_{x_{\text{in}}}^l)^2)}\right) I_{X,2,1} \, dy_{\text{in}_j}$$

$$I_3 = \int_{-\infty}^{\infty} \frac{y_{\mathrm{in}_j}(1 + \mathrm{erf}(\frac{y_{\mathrm{in}_j}}{\sqrt{2}}))}{8\pi\sigma_{x_{\mathrm{in}}}^2 \sqrt{(1-(r_{x_{\mathrm{in}}}^l)^2)}} \exp\left(\frac{-y_{\mathrm{in}_j}^2}{2\sigma_{x_{\mathrm{in}}}^2(1-(r_{x_{\mathrm{in}}}^l)^2)}\right) I_{X,2,2} dy_{\mathrm{in}_j}$$

We have $I = I_1 + I_2 + I_3$

$$I_1 = \int_{-\infty}^{\infty} \frac{y_{\mathrm{in}_j}(1 + \mathrm{erf}(\frac{y_{\mathrm{in}_j}}{\sqrt{2}}))}{8\pi\sigma_{x_{\mathrm{in}}}^2 \sqrt{(1-(r_{x_{\mathrm{in}}}^l)^2)}} \exp\left(\frac{-y_{\mathrm{in}_j}^2}{2\sigma_{x_{\mathrm{in}}}^2(1-(r_{x_{\mathrm{in}}}^l)^2)}\right) r_{x_{\mathrm{in}}}^l y_{\mathrm{in}_j}$$

$$\sqrt{2\pi}\sigma_{x_{\mathrm{in}}}\sqrt{(1-(r_{x_{\mathrm{in}}}^l)^2)} \exp\left(\frac{(r_{x_{\mathrm{in}}}^l)^2 y_{\mathrm{in}_j}^2}{2\sigma_{x_{\mathrm{in}}}^2(1-(r_{x_{\mathrm{in}}}^l)^2)}\right) dy_{\mathrm{in}_j}$$

$$= \frac{r_{x_{\mathrm{in}}}^l}{4} \int_{-\infty}^{\infty} \frac{y_{\mathrm{in}_j}^2(1 + \mathrm{erf}(\frac{y_{\mathrm{in}_j}}{\sqrt{2}}))}{\sqrt{2\pi}\sigma_{x_{\mathrm{in}}}^2} \exp\left(\frac{-y_{\mathrm{in}_j}^2}{2\sigma_{x_{\mathrm{in}}}^2}\right) dy_{\mathrm{in}_j}$$

$$= \frac{r_{x_{\mathrm{in}}}^l}{4} \int_{-\infty}^{\infty} \frac{y_{\mathrm{in}_j}^2}{\sqrt{2\pi}\sigma_{x_{\mathrm{in}}}^2} \exp\left(\frac{-y_{\mathrm{in}_j}^2}{2\sigma_{x_{\mathrm{in}}}^2}\right) dy_{\mathrm{in}_j} + \frac{r_{x_{\mathrm{in}}}^l}{4} \int_{-\infty}^{\infty} \frac{y_{\mathrm{in}_j}^2 \mathrm{erf}(\frac{y_{\mathrm{in}_j}}{\sqrt{2}})}{\sqrt{2\pi}\sigma_{x_{\mathrm{in}}}^2} \exp\left(\frac{-y_{\mathrm{in}_j}^2}{2\sigma_{x_{\mathrm{in}}}^2}\right) dy_{\mathrm{in}_j}$$

$$= \frac{r_{x_{\mathrm{in}}}^l \sigma_{x_{\mathrm{in}}}^2}{4} \qquad \text{(Definition of variance, and integral of odd function)}$$

$$I_2 = \int_{-\infty}^{\infty} \frac{y_{\mathrm{in}_j}(1 + \mathrm{erf}(\frac{y_{\mathrm{in}_j}}{\sqrt{2}}))}{8\pi\sigma_{x_{\mathrm{in}}}^2 \sqrt{(1-(r_{x_{\mathrm{in}}}^l)^2)}} \exp\left(\frac{-y_{\mathrm{in}_j}^2}{2\sigma_{x_{\mathrm{in}}}^2(1-(r_{x_{\mathrm{in}}}^l)^2)}\right) r_{x_{\mathrm{in}}}^l y_{\mathrm{in}_j}$$

$$\sqrt{2\pi}\sigma_{x_{\mathrm{in}}}\sqrt{(1-(r_{x_{\mathrm{in}}}^l)^2)} \exp\left(\frac{(r_{x_{\mathrm{in}}}^l)^2 y_{\mathrm{in}_j}^2}{2\sigma_{x_{\mathrm{in}}}^2(1-(r_{x_{\mathrm{in}}}^l)^2)}\right) \mathrm{erf}\left(\frac{r_{x_{\mathrm{in}}}^l y_{\mathrm{in}_j}}{\sqrt{2(\sigma_{x_{\mathrm{in}}}^2(1-(r_{x_{\mathrm{in}}}^l)^2)+1)}}\right) dy_{\mathrm{in}_j}$$

$$= \frac{r_{x_{\mathrm{in}}}^l}{4\sqrt{2\pi}\sigma_{x_{\mathrm{in}}}} \int_{-\infty}^{\infty} y_{\mathrm{in}_j}^2(1 + \mathrm{erf}(\frac{y_{\mathrm{in}_j}}{\sqrt{2}})) \exp\left(\frac{-y_{\mathrm{in}_j}^2}{2\sigma_{x_{\mathrm{in}}}^2}\right) \mathrm{erf}\left(\frac{r_{x_{\mathrm{in}}}^l y_{\mathrm{in}_j}}{\sqrt{2(\sigma_{x_{\mathrm{in}}}^2(1-(r_{x_{\mathrm{in}}}^l)^2)+1)}}\right) dy_{\mathrm{in}_j}$$

$$= \frac{r_{x_{\mathrm{in}}}^l}{4\sqrt{2\pi}\sigma_{x_{\mathrm{in}}}} \int_{-\infty}^{\infty} y_{\mathrm{in}_j}^2 \exp\left(\frac{-y_{\mathrm{in}_j}^2}{2\sigma_{x_{\mathrm{in}}}^2}\right) \mathrm{erf}\left(\frac{r_{x_{\mathrm{in}}}^l y_{\mathrm{in}_j}}{\sqrt{2(\sigma_{x_{\mathrm{in}}}^2(1-(r_{x_{\mathrm{in}}}^l)^2)+1)}}\right) dy_{\mathrm{in}_j}$$

$$+ \frac{r_{x_{\mathrm{in}}}^l}{4\sqrt{2\pi}\sigma_{x_{\mathrm{in}}}} \int_{-\infty}^{\infty} y_{\mathrm{in}_j}^2 \mathrm{erf}(\frac{y_{\mathrm{in}_j}}{\sqrt{2}}) \exp\left(\frac{-y_{\mathrm{in}_j}^2}{2\sigma_{x_{\mathrm{in}}}^2}\right) \mathrm{erf}\left(\frac{r_{x_{\mathrm{in}}}^l y_{\mathrm{in}_j}}{\sqrt{2(\sigma_{x_{\mathrm{in}}}^2(1-(r_{x_{\mathrm{in}}}^l)^2)+1)}}\right) dy_{\mathrm{in}_j}$$

$$= \frac{r_{x_{\mathrm{in}}}^l}{4\sqrt{2\pi}\sigma_{x_{\mathrm{in}}}} \int_{-\infty}^{\infty} y_{\mathrm{in}_j}^2 \mathrm{erf}(\frac{y_{\mathrm{in}_j}}{\sqrt{2}}) \exp\left(\frac{-y_{\mathrm{in}_j}^2}{2\sigma_{x_{\mathrm{in}}}^2}\right) \mathrm{erf}\left(\frac{r_{x_{\mathrm{in}}}^l y_{\mathrm{in}_j}}{\sqrt{2(\sigma_{x_{\mathrm{in}}}^2(1-(r_{x_{\mathrm{in}}}^l)^2)+1)}}\right) dy_{\mathrm{in}_j}$$

$$\text{(Integral of Odd function)}$$

From 2.7.3.3 of Lipovetsky (2020),

$$\int_{-\infty}^{\infty} z^2 \exp(-az^2) \mathrm{erf}(a_1 z) \mathrm{erf}(a_2 z) =$$

$$\frac{1}{\sqrt{\pi}} \left( \frac{1}{a\sqrt{a}} \tan^{-1}\left(\frac{a_1 a_2}{\sqrt{a^2 + aa_1^2 + aa_2^2}}\right) + \frac{a_1 a_2(2a + a_1^2 + a_2^2)}{a\sqrt{a + a_1^2 + a_2^2}(a^2 + aa_1^2 + aa_2^2 + a_1^2 a_2^2)} \right)$$

Substituting $a = \frac{1}{2\sigma_{x_{\mathrm{in}}}^2}, a_1 = \frac{1}{\sqrt{2}}, a_2 = \frac{r_{x_{\mathrm{in}}}^l}{\sqrt{2(\sigma_{x_{\mathrm{in}}}^2(1-(r_{x_{\mathrm{in}}}^l)^2)+1)}}$

$$a_1 a_2 = \frac{r_{x_{\mathrm{in}}}^l}{2\sqrt{(\sigma_{x_{\mathrm{in}}}^2(1-(r_{x_{\mathrm{in}}}^l)^2)+1)}}$$

$$a^2 + aa_1^2 + aa_2^2 = \frac{1}{4\sigma_{x_{\mathrm{in}}}^4} + \frac{1}{4\sigma_{x_{\mathrm{in}}}^2} + \frac{(r_{x_{\mathrm{in}}}^l)^2}{4\sigma_{x_{\mathrm{in}}}^2(\sigma_{x_{\mathrm{in}}}^2(1-(r_{x_{\mathrm{in}}}^l)^2)+1)}$$

$$= \frac{\sigma_{x_{\text{in}}}^2(1-(r_{x_{\text{in}}}^l)^2)+1+\sigma_{x_{\text{in}}}^4(1-(r_{x_{\text{in}}}^l)^2)+\sigma_{x_{\text{in}}}^2+(r_{x_{\text{in}}}^l)^2\sigma_{x_{\text{in}}}^2}{4\sigma_{x_{\text{in}}}^4(\sigma_{x_{\text{in}}}^2(1-(r_{x_{\text{in}}}^l)^2)+1)}$$

$$= \frac{\sigma_{x_{\text{in}}}^4+2\sigma_{x_{\text{in}}}^2+1-(r_{x_{\text{in}}}^l)^2\sigma_{x_{\text{in}}}^4}{4\sigma_{x_{\text{in}}}^4(\sigma_{x_{\text{in}}}^2(1-(r_{x_{\text{in}}}^l)^2)+1)} = \frac{(\sigma_{x_{\text{in}}}^2+1)^2-(r_{x_{\text{in}}}^l\sigma_{x_{\text{in}}}^2)^2}{4\sigma_{x_{\text{in}}}^4(\sigma_{x_{\text{in}}}^2(1-(r_{x_{\text{in}}}^l)^2)+1)}$$

$$a+a_1^2+a_2^2 = \frac{a^2+aa_1^2+aa_2^2}{a} = \frac{(\sigma_{x_{\text{in}}}^2+1)^2-(r_{x_{\text{in}}}^l\sigma_{x_{\text{in}}}^2)^2}{4\sigma_{x_{\text{in}}}^4(\sigma_{x_{\text{in}}}^2(1-(r_{x_{\text{in}}}^l)^2)+1)} * 2\sigma_{x_{\text{in}}}^2$$

$$= \frac{(\sigma_{x_{\text{in}}}^2+1)^2-(r_{x_{\text{in}}}^l\sigma_{x_{\text{in}}}^2)^2}{2\sigma_{x_{\text{in}}}^2(\sigma_{x_{\text{in}}}^2(1-(r_{x_{\text{in}}}^l)^2)+1)}$$

$$a^2+aa_1^2+aa_2^2+a_1^2a_2^2 = \frac{(\sigma_{x_{\text{in}}}^2+1)^2-(r_{x_{\text{in}}}^l\sigma_{x_{\text{in}}}^2)^2}{4\sigma_{x_{\text{in}}}^4(\sigma_{x_{\text{in}}}^2(1-(r_{x_{\text{in}}}^l)^2)+1)}+\frac{(r_{x_{\text{in}}}^l)^2}{4(\sigma_{x_{\text{in}}}^2(1-(r_{x_{\text{in}}}^l)^2)+1)}$$

$$= \frac{(\sigma_{x_{\text{in}}}^2+1)^2-(r_{x_{\text{in}}}^l\sigma_{x_{\text{in}}}^2)^2+(r_{x_{\text{in}}}^l)^2\sigma_{x_{\text{in}}}^4}{4\sigma_{x_{\text{in}}}^4(\sigma_{x_{\text{in}}}^2(1-(r_{x_{\text{in}}}^l)^2)+1)} = \frac{(\sigma_{x_{\text{in}}}^2+1)^2}{4\sigma_{x_{\text{in}}}^4(\sigma_{x_{\text{in}}}^2(1-(r_{x_{\text{in}}}^l)^2)+1)}$$

$$2a+a_1^2+a_2^2 = \frac{1}{2\sigma_{x_{\text{in}}}^2}+\frac{(\sigma_{x_{\text{in}}}^2+1)^2-(r_{x_{\text{in}}}^l\sigma_{x_{\text{in}}}^2)^2}{2\sigma_{x_{\text{in}}}^2(\sigma_{x_{\text{in}}}^2(1-(r_{x_{\text{in}}}^l)^2)+1)}$$

$$= \frac{(\sigma_{x_{\text{in}}}^2+1)^2-(r_{x_{\text{in}}}^l\sigma_{x_{\text{in}}}^2)^2+\sigma_{x_{\text{in}}}^2(1-(r_{x_{\text{in}}}^l)^2)+1}{2\sigma_{x_{\text{in}}}^2(\sigma_{x_{\text{in}}}^2(1-(r_{x_{\text{in}}}^l)^2)+1)}$$

$$= \frac{(\sigma_{x_{\text{in}}}^2+1)^2+\sigma_{x_{\text{in}}}^2+1-(r_{x_{\text{in}}}^l\sigma_{x_{\text{in}}}^2)^2-\sigma_{x_{\text{in}}}^2(r_{x_{\text{in}}}^l)^2}{2\sigma_{x_{\text{in}}}^2(\sigma_{x_{\text{in}}}^2(1-(r_{x_{\text{in}}}^l)^2)+1)}$$

$$= \frac{(\sigma_{x_{\text{in}}}^2+1)(\sigma_{x_{\text{in}}}^2+2)-(r_{x_{\text{in}}}^l)^2\sigma_{x_{\text{in}}}^2(\sigma_{x_{\text{in}}}^2+1)}{2\sigma_{x_{\text{in}}}^2(\sigma_{x_{\text{in}}}^2(1-(r_{x_{\text{in}}}^l)^2)+1)}$$

$$= \frac{(\sigma_{x_{\text{in}}}^2+1)(\sigma_{x_{\text{in}}}^2(1-(r_{x_{\text{in}}}^l)^2)+2)}{2\sigma_{x_{\text{in}}}^2(\sigma_{x_{\text{in}}}^2(1-(r_{x_{\text{in}}}^l)^2)+1)}$$

$$I_2 = \frac{r_{x_{\text{in}}}^l}{4\sqrt{2}\pi\sigma_{x_{\text{in}}}}(2\sqrt{2}\sigma_{x_{\text{in}}}^3\tan^{-1}(\frac{\frac{r_{x_{\text{in}}}^l}{2\sqrt{(\sigma_{x_{\text{in}}}^2(1-(r_{x_{\text{in}}}^l)^2)+1)}}}{\sqrt{\frac{(\sigma_{x_{\text{in}}}^2+1)^2-(r_{x_{\text{in}}}^l\sigma_{x_{\text{in}}}^2)^2}{4\sigma_{x_{\text{in}}}^4(\sigma_{x_{\text{in}}}^2(1-(r_{x_{\text{in}}}^l)^2)+1)}}}))$$

$$+\frac{r_{x_{\text{in}}}^l}{4\sqrt{2}\pi\sigma_{x_{\text{in}}}}(\frac{\frac{r_{x_{\text{in}}}^l}{2\sqrt{(\sigma_{x_{\text{in}}}^2(1-(r_{x_{\text{in}}}^l)^2)+1)}}\frac{(\sigma_{x_{\text{in}}}^2+1)(\sigma_{x_{\text{in}}}^2(1-(r_{x_{\text{in}}}^l)^2)+2)}{2\sigma_{x_{\text{in}}}^2(\sigma_{x_{\text{in}}}^2(1-(r_{x_{\text{in}}}^l)^2)+1)}}{\frac{1}{2\sigma_{x_{\text{in}}}^2}\sqrt{\frac{(\sigma_{x_{\text{in}}}^2+1)^2-(r_{x_{\text{in}}}^l\sigma_{x_{\text{in}}}^2)^2}{2\sigma_{x_{\text{in}}}^2(\sigma_{x_{\text{in}}}^2(1-(r_{x_{\text{in}}}^l)^2)+1)}}\frac{(\sigma_{x_{\text{in}}}^2+1)^2}{4\sigma_{x_{\text{in}}}^4(\sigma_{x_{\text{in}}}^2(1-(r_{x_{\text{in}}}^l)^2)+1)}})$$

$$= \frac{r_{x_{\text{in}}}^l}{4\sqrt{2}\pi\sigma_{x_{\text{in}}}}(2\sqrt{2}\sigma_{x_{\text{in}}}^3\tan^{-1}(\frac{r_{x_{\text{in}}}^l\sigma_{x_{\text{in}}}^2}{\sqrt{(\sigma_{x_{\text{in}}}^2+1)^2-(r_{x_{\text{in}}}^l\sigma_{x_{\text{in}}}^2)^2}}))$$

$$+\frac{r_{x_{\text{in}}}^l}{4\sqrt{2}\pi\sigma_{x_{\text{in}}}}(\frac{2\sqrt{2}r_{x_{\text{in}}}^l\sigma_{x_{\text{in}}}^5(\sigma_{x_{\text{in}}}^2(1-(r_{x_{\text{in}}}^l)^2)+2)}{(\sigma_{x_{\text{in}}}^2+1)\sqrt{(\sigma_{x_{\text{in}}}^2+1)^2-(r_{x_{\text{in}}}^l\sigma_{x_{\text{in}}}^2)^2}})$$

$$I_2 = \frac{r_{x_{\text{in}}}^l\sigma_{x_{\text{in}}}^2}{2\pi}(\sin^{-1}(\frac{r_{x_{\text{in}}}^l\sigma_{x_{\text{in}}}^2}{\sigma_{x_{\text{in}}}^2+1})+\frac{r_{x_{\text{in}}}^l\sigma_{x_{\text{in}}}^2(\sigma_{x_{\text{in}}}^2(1-(r_{x_{\text{in}}}^l)^2)+2)}{(\sigma_{x_{\text{in}}}^2+1)\sqrt{(\sigma_{x_{\text{in}}}^2+1)^2-(r_{x_{\text{in}}}^l\sigma_{x_{\text{in}}}^2)^2}})$$

$$I_3 = \int_{-\infty}^{\infty}\frac{y_{\text{in}_j}(1+\text{erf}(\frac{y_{\text{in}_j}}{\sqrt{2}}))}{8\pi\sigma_{x_{\text{in}}}^2\sqrt{(1-(r_{x_{\text{in}}}^l)^2)}}\exp(\frac{-y_{\text{in}_j}^2}{2\sigma_{x_{\text{in}}}^2(1-(r_{x_{\text{in}}}^l)^2)})$$

$$\frac{2\sigma_{x_{\text{in}}}^3(1-(r_{x_{\text{in}}}^l)^2)^{\frac{3}{2}}}{\sqrt{\sigma_{x_{\text{in}}}^2(1-(r_{x_{\text{in}}}^l)^2)+1}}\exp(\frac{(r_{x_{\text{in}}}^l)^2y_{\text{in}_j}^2}{2(\sigma_{x_{\text{in}}}^2(1-(r_{x_{\text{in}}}^l)^2)+1)\sigma_{x_{\text{in}}}^2(1-(r_{x_{\text{in}}}^l)^2)})dy_{\text{in}_j}$$

$$= \int_{-\infty}^{\infty} \frac{\sigma_{x_{\text{in}}}(1-(r_{x_{\text{in}}}^l)^2)y_{\text{in}_j}(1+\text{erf}(\frac{y_{\text{in}_j}}{\sqrt{2}}))}{4\pi\sqrt{\sigma_{x_{\text{in}}}^2(1-(r_{x_{\text{in}}}^l)^2)+1}} \exp\big(\frac{-y_{\text{in}_j}^2(\sigma_{x_{\text{in}}}^2(1-(r_{x_{\text{in}}}^l)^2)+1-(r_{x_{\text{in}}}^l)^2)}{2(\sigma_{x_{\text{in}}}^2(1-(r_{x_{\text{in}}}^l)^2)+1)\sigma_{x_{\text{in}}}^2(1-(r_{x_{\text{in}}}^l)^2)}\big)dy_{\text{in}_j}$$

$$= \int_{-\infty}^{\infty} \frac{\sigma_{x_{\text{in}}}(1-(r_{x_{\text{in}}}^l)^2)y_{\text{in}_j}(1+\text{erf}(\frac{y_{\text{in}_j}}{\sqrt{2}}))}{4\pi\sqrt{\sigma_{x_{\text{in}}}^2(1-(r_{x_{\text{in}}}^l)^2)+1}} \exp\big(\frac{-y_{\text{in}_j}^2(\sigma_{x_{\text{in}}}^2+1)(1-(r_{x_{\text{in}}}^l)^2)}{2(\sigma_{x_{\text{in}}}^2(1-(r_{x_{\text{in}}}^l)^2)+1)\sigma_{x_{\text{in}}}^2(1-(r_{x_{\text{in}}}^l)^2)}\big)dy_{\text{in}_j}$$

$$= \frac{\sigma_{x_{\text{in}}}(1-(r_{x_{\text{in}}}^l)^2)}{4\pi\sqrt{\sigma_{x_{\text{in}}}^2(1-(r_{x_{\text{in}}}^l)^2)+1}} \int_{-\infty}^{\infty} y_{\text{in}_j}(1+\text{erf}(\frac{y_{\text{in}_j}}{\sqrt{2}})) \exp\big(\frac{-y_{\text{in}_j}^2(\sigma_{x_{\text{in}}}^2+1)}{2(\sigma_{x_{\text{in}}}^2(1-(r_{x_{\text{in}}}^l)^2)+1)\sigma_{x_{\text{in}}}^2}\big)dy_{\text{in}_j}$$

$$= \frac{\sigma_{x_{\text{in}}}(1-(r_{x_{\text{in}}}^l)^2)}{4\pi\sqrt{\sigma_{x_{\text{in}}}^2(1-(r_{x_{\text{in}}}^l)^2)+1}} \int_{-\infty}^{\infty} y_{\text{in}_j} \exp\big(\frac{-y_{\text{in}_j}^2(\sigma_{x_{\text{in}}}^2+1)}{2(\sigma_{x_{\text{in}}}^2(1-(r_{x_{\text{in}}}^l)^2)+1)\sigma_{x_{\text{in}}}^2}\big)dy_{\text{in}_j}$$

$$+ \frac{\sigma_{x_{\text{in}}}(1-(r_{x_{\text{in}}}^l)^2)}{4\pi\sqrt{\sigma_{x_{\text{in}}}^2(1-(r_{x_{\text{in}}}^l)^2)+1}} \int_{-\infty}^{\infty} y_{\text{in}_j}\text{erf}(\frac{y_{\text{in}_j}}{\sqrt{2}}) \exp\big(\frac{-y_{\text{in}_j}^2(\sigma_{x_{\text{in}}}^2+1)}{2(\sigma_{x_{\text{in}}}^2(1-(r_{x_{\text{in}}}^l)^2)+1)\sigma_{x_{\text{in}}}^2}\big)dy_{\text{in}_j}$$

$$= \frac{\sigma_{x_{\text{in}}}(1-(r_{x_{\text{in}}}^l)^2)}{4\pi\sqrt{\sigma_{x_{\text{in}}}^2(1-(r_{x_{\text{in}}}^l)^2)+1}} \int_{-\infty}^{\infty} y_{\text{in}_j}\text{erf}(\frac{y_{\text{in}_j}}{\sqrt{2}}) \exp\big(\frac{-y_{\text{in}_j}^2(\sigma_{x_{\text{in}}}^2+1)}{2(\sigma_{x_{\text{in}}}^2(1-(r_{x_{\text{in}}}^l)^2)+1)\sigma_{x_{\text{in}}}^2}\big)dy_{\text{in}_j}$$

(Integral of Odd function)

From 2.6.1.4 of Lipovetsky (2020), $\int_{-\infty}^{\infty} z\,\text{erf}(az)\exp(-a_1 z^2)dz = \dfrac{a}{a_1\sqrt{a^2+a_1}}$

Substituting, $a = \dfrac{1}{\sqrt{2}}, a_1 = \dfrac{(\sigma_{x_{\text{in}}}^2+1)}{2\sigma_{x_{\text{in}}}^2(\sigma_{x_{\text{in}}}^2(1-(r_{x_{\text{in}}}^l)^2)+1)}$, we have

$$I_3 = \frac{\sigma_{x_{\text{in}}}(1-(r_{x_{\text{in}}}^l)^2)}{4\pi\sqrt{\sigma_{x_{\text{in}}}^2(1-(r_{x_{\text{in}}}^l)^2)+1}}\Big(\frac{\frac{1}{\sqrt{2}}}{\frac{(\sigma_{x_{\text{in}}}^2+1)}{2\sigma_{x_{\text{in}}}^2(\sigma_{x_{\text{in}}}^2(1-(r_{x_{\text{in}}}^l)^2)+1)}\sqrt{\frac{1}{2}+\frac{(\sigma_{x_{\text{in}}}^2+1)}{2\sigma_{x_{\text{in}}}^2(\sigma_{x_{\text{in}}}^2(1-(r_{x_{\text{in}}}^l)^2)+1)}}}\Big)$$

$$= \frac{\sigma_{x_{\text{in}}}(1-(r_{x_{\text{in}}}^l)^2)}{4\pi\sqrt{\sigma_{x_{\text{in}}}^2(1-(r_{x_{\text{in}}}^l)^2)+1}}\frac{2\sigma_{x_{\text{in}}}^3(\sigma_{x_{\text{in}}}^2(1-(r_{x_{\text{in}}}^l)^2)+1)^{\frac{3}{2}}}{(\sigma_{x_{\text{in}}}^2+1)\sqrt{\sigma_{x_{\text{in}}}^4(1-(r_{x_{\text{in}}}^l)^2)+\sigma_{x_{\text{in}}}^2+\sigma_{x_{\text{in}}}^2+1}}$$

$$I_3 = \frac{\sigma_{x_{\text{in}}}^4(\sigma_{x_{\text{in}}}^2(1-(r_{x_{\text{in}}}^l)^2)+1)(1-(r_{x_{\text{in}}}^l)^2)}{2\pi(\sigma_{x_{\text{in}}}^2+1)\sqrt{(\sigma_{x_{\text{in}}}^2+1)^2-(r_{x_{\text{in}}}^l\sigma_{x_{\text{in}}}^2)^2}}$$

Finally we have,

$$I = I_1 + I_2 + I_3$$
$$= \frac{r_{x_{\text{in}}}^l\sigma_{x_{\text{in}}}^2}{4} + \frac{r_{x_{\text{in}}}^l\sigma_{x_{\text{in}}}^2}{2\pi}\Big(\sin^{-1}\big(\frac{r_{x_{\text{in}}}^l\sigma_{x_{\text{in}}}^2}{\sigma_{x_{\text{in}}}^2+1}\big) + \frac{r_{x_{\text{in}}}^l\sigma_{x_{\text{in}}}^2(\sigma_{x_{\text{in}}}^2(1-(r_{x_{\text{in}}}^l)^2)+2)}{(\sigma_{x_{\text{in}}}^2+1)\sqrt{(\sigma_{x_{\text{in}}}^2+1)^2-(r_{x_{\text{in}}}^l\sigma_{x_{\text{in}}}^2)^2}}\Big)$$
$$+ \frac{\sigma_{x_{\text{in}}}^4(\sigma_{x_{\text{in}}}^2(1-(r_{x_{\text{in}}}^l)^2)+1)(1-(r_{x_{\text{in}}}^l)^2)}{2\pi(\sigma_{x_{\text{in}}}^2+1)\sqrt{(\sigma_{x_{\text{in}}}^2+1)^2-(r_{x_{\text{in}}}^l\sigma_{x_{\text{in}}}^2)^2}}$$

$$I = \frac{r_{x_{\text{in}}}^l\sigma_{x_{\text{in}}}^2}{4} + \frac{r_{x_{\text{in}}}^l\sigma_{x_{\text{in}}}^2}{2\pi}\sin^{-1}\big(\frac{r_{x_{\text{in}}}^l\sigma_{x_{\text{in}}}^2}{\sigma_{x_{\text{in}}}^2+1}\big) + \frac{\sigma_{x_{\text{in}}}^4(\sigma_{x_{\text{in}}}^2(1-(r_{x_{\text{in}}}^l)^2)+1+(r_{x_{\text{in}}}^l)^2)}{2\pi(\sigma_{x_{\text{in}}}^2+1)\sqrt{(\sigma_{x_{\text{in}}}^2+1)^2-(r_{x_{\text{in}}}^l\sigma_{x_{\text{in}}}^2)^2}}$$

$$I = \frac{\sigma_{x_{\text{in}}}^2}{4}\left[r_{x_{\text{in}}}^l + \frac{2r_{x_{\text{in}}}^l}{\pi}\sin^{-1}\big(\frac{r_{x_{\text{in}}}^l\sigma_{x_{\text{in}}}^2}{\sigma_{x_{\text{in}}}^2+1}\big) + \frac{2\sigma_{x_{\text{in}}}^2(\sigma_{x_{\text{in}}}^2(1-(r_{x_{\text{in}}}^l)^2)+1+(r_{x_{\text{in}}}^l)^2)}{\pi(\sigma_{x_{\text{in}}}^2+1)\sqrt{(\sigma_{x_{\text{in}}}^2+1)^2-(r_{x_{\text{in}}}^l\sigma_{x_{\text{in}}}^2)^2}}\right]$$

We have,

$$\text{Cov}(x_{\text{out}_j}, y_{\text{out}_j}) = I - \mathbb{E}[x_{\text{out}_j}]\mathbb{E}[y_{\text{out}_j}]$$

$$\text{Cov}(x_{\text{out}_j}, y_{\text{out}_j}) = I - \frac{\sigma_{x_{\text{in}}}^4}{2\pi(\sigma_{x_{\text{in}}}^2 + 1)}$$

$$\text{Cov}(x_{\text{out}_j}, y_{\text{out}_j}) = \frac{\sigma_{x_{\text{in}}}^2}{4\pi}(\pi r_{x_{\text{in}}}^l + 2r_{x_{\text{in}}}^l \sin^{-1}\left(\frac{r_{x_{\text{in}}}^l \sigma_{x_{\text{in}}}^2}{\sigma_{x_{\text{in}}}^2 + 1}\right)$$
$$+ \frac{2\sigma_{x_{\text{in}}}^2(\sigma_{x_{\text{in}}}^2(1 - (r_{x_{\text{in}}}^l)^2) + 1 + (r_{x_{\text{in}}}^l)^2)}{(\sigma_{x_{\text{in}}}^2 + 1)\sqrt{(\sigma_{x_{\text{in}}}^2 + 1)^2 - (r_{x_{\text{in}}}^l \sigma_{x_{\text{in}}}^2)^2}} - \frac{2\sigma_{x_{\text{in}}}^2}{(\sigma_{x_{\text{in}}}^2 + 1)})$$

The backward pass through GeLU is defined as,

$$g_{\text{in}_i} = (\Phi(x_{\text{in}_i}) + \frac{x_{\text{in}_i}}{\sqrt{2\pi}}\exp\left(\frac{-x_{\text{in}_i}^2}{2}\right))g_{\text{out}_i}$$

$$= (\frac{1}{2}(1 + \text{erf}(\frac{x_{\text{in}_i}}{\sqrt{2}})) + \frac{x_{\text{in}_i}}{\sqrt{2\pi}}\exp\left(\frac{-x_{\text{in}_i}^2}{2}\right))g_{\text{out}_i}$$

So the mean of gradient is obtained as following,

$$\mathbb{E}[g_{\text{in}_i}] = \mathbb{E}[(\frac{1}{2}(1 + \text{erf}(\frac{x_{\text{in}_i}}{\sqrt{2}})) + \frac{x_{\text{in}_i}}{\sqrt{2\pi}}\exp\left(\frac{-x_{\text{in}_i}^2}{2}\right))g_{\text{out}_i}]$$

$$= \mathbb{E}[(\frac{1}{2}(1 + \text{erf}(\frac{x_{\text{in}_i}}{\sqrt{2}})) + \frac{x_{\text{in}_i}}{\sqrt{2\pi}}\exp\left(\frac{-x_{\text{in}_i}^2}{2}\right))]\mathbb{E}[g_{\text{out}_i}] = 0$$

$$\boxed{\mu_{g_{\text{in}}} = 0}$$

Similarly for variance,

$$\mathbb{E}[g_{\text{in}_i}^2] = \mathbb{E}[(\frac{1}{2}(1 + \text{erf}(\frac{x_{\text{in}_i}}{\sqrt{2}})) + \frac{x_{\text{in}_i}}{\sqrt{2\pi}}\exp\left(\frac{-x_{\text{in}_i}^2}{2}\right))^2 g_{\text{out}_i}^2]$$

$$= \mathbb{E}[(\frac{1}{2}(1 + \text{erf}(\frac{x_{\text{in}_i}}{\sqrt{2}})) + \frac{x_{\text{in}_i}}{\sqrt{2\pi}}\exp\left(\frac{-x_{\text{in}_i}^2}{2}\right))^2]\mathbb{E}[g_{\text{out}_i}^2]$$

$$= \mathbb{E}[(\frac{1}{2}(1 + \text{erf}(\frac{x_{\text{in}_i}}{\sqrt{2}})) + \frac{x_{\text{in}_i}}{\sqrt{2\pi}}\exp\left(\frac{-x_{\text{in}_i}^2}{2}\right))^2]\sigma_{g_{\text{out}}}^2$$

$$I = \mathbb{E}[(\frac{1}{2}(1 + \text{erf}(\frac{x_{\text{in}_i}}{\sqrt{2}})) + \frac{x_{\text{in}_i}}{\sqrt{2\pi}}\exp\left(\frac{-x_{\text{in}_i}^2}{2}\right))^2]$$

$$= \int_{-\infty}^{\infty}(\frac{1}{2}(1 + \text{erf}(\frac{x_{\text{in}_i}}{\sqrt{2}})) + \frac{x_{\text{in}_i}}{\sqrt{2\pi}}\exp\left(\frac{-x_{\text{in}_i}^2}{2}\right))^2 \frac{\exp\left(\frac{-x_{\text{in}_i}^2}{2\sigma_{x_{\text{in}}}^2}\right)}{\sqrt{2\pi}\sigma_{x_{\text{in}}}}dx_{\text{in}_i}$$

$$I = \int_{-\infty}^{\infty}(\frac{1}{4} + \frac{\text{erf}^2(\frac{x_{\text{in}_i}}{\sqrt{2}})}{4} + \frac{x_{\text{in}_i}^2 \exp(-x_{\text{in}_i}^2)}{2\pi} + \frac{\text{erf}(\frac{x_{\text{in}_i}}{\sqrt{2}})}{2} +$$

$$\frac{x_{\text{in}_i}\exp\left(\frac{-x_{\text{in}_i}^2}{2}\right)}{\sqrt{2\pi}} + \frac{x_{\text{in}_i}\exp\left(\frac{-x_{\text{in}_i}^2}{2}\right)\text{erf}(\frac{x_{\text{in}_i}}{\sqrt{2}})}{\sqrt{2\pi}})\frac{\exp\left(\frac{-x_{\text{in}_i}^2}{2\sigma_{x_{\text{in}}}^2}\right)}{\sqrt{2\pi}\sigma_{x_{\text{in}}}}dx_{\text{in}_i}$$

$$I_1 = \int_{-\infty}^{\infty}\frac{1}{4}\frac{\exp\left(\frac{-x_{\text{in}_i}^2}{2\sigma_{x_{\text{in}}}^2}\right)}{\sqrt{2\pi}\sigma_{x_{\text{in}}}}dx_{\text{in}_i}$$

$$I_1 = \frac{1}{4}$$

$$I_2 = \int_{-\infty}^{\infty} \frac{\operatorname{erf}^2(\frac{x_{\text{in}_i}}{\sqrt{2}})}{4} \frac{\exp\left(\frac{-x_{\text{in}_i}^2}{2\sigma_{x_{\text{in}}}^2}\right)}{\sqrt{2\pi}\sigma_{x_{\text{in}}}} dx_{\text{in}_i}$$

$$= \frac{1}{4\sqrt{2\pi}\sigma_{x_{\text{in}}}} \int_{-\infty}^{\infty} \operatorname{erf}^2(\frac{x_{\text{in}_i}}{\sqrt{2}}) \exp\left(\frac{-x_{\text{in}_i}^2}{2\sigma_{x_{\text{in}}}^2}\right) dx_{\text{in}_i}$$

From 2.7.1.3 of Lipovetsky (2020),

$$\int_{-\infty}^{\infty} \operatorname{erf}(a_1 z)\operatorname{erf}(a_2 z) \exp\left(-a z^2\right) dz = \frac{2}{\sqrt{\pi a}} \tan^{-1}\left(\frac{a_1 a_2}{\sqrt{a^2 + a a_1^2 + a a_2^2}}\right)$$

Substituting $a = \frac{1}{2\sigma_{x_{\text{in}}}^2}, a_1 = a_2 = \frac{1}{\sqrt{2}}$

$$I_2 = \frac{1}{4\sqrt{2\pi}\sigma_{x_{\text{in}}}} \frac{2}{\sqrt{\pi \frac{1}{2\sigma_{x_{\text{in}}}^2}}} \tan^{-1}\left(\frac{\frac{1}{2}}{\sqrt{\frac{1}{4\sigma_{x_{\text{in}}}^4} + \frac{1}{4\sigma_{x_{\text{in}}}^2} + \frac{1}{4\sigma_{x_{\text{in}}}^2}}}\right)$$

$$= \frac{1}{2\pi} \tan^{-1}\left(\frac{\sigma_{x_{\text{in}}}^2}{\sqrt{2\sigma_{x_{\text{in}}}^2 + 1}}\right) = \frac{1}{2\pi} \tan^{-1}\left(\frac{\sigma_{x_{\text{in}}}^2}{\sqrt{(\sigma_{x_{\text{in}}}^2 + 1)^2 - \sigma_{x_{\text{in}}}^4}}\right)$$

$$I_2 = \frac{1}{2\pi} \sin^{-1}\left(\frac{\sigma_{x_{\text{in}}}^2}{\sigma_{x_{\text{in}}}^2 + 1}\right)$$

$$I_3 = \int_{-\infty}^{\infty} \frac{x_{\text{in}_i}^2 \exp\left(-x_{\text{in}_i}^2\right)}{2\pi} \frac{\exp\left(\frac{-x_{\text{in}_i}^2}{2\sigma_{x_{\text{in}}}^2}\right)}{\sqrt{2\pi}\sigma_{x_{\text{in}}}} dx_{\text{in}_i}$$

$$= \frac{1}{2\pi\sigma_{x_{\text{in}}}} \int_{-\infty}^{\infty} \frac{x_{\text{in}_i}^2}{\sqrt{2\pi}} \exp\left(\frac{-x_{\text{in}_i}^2(2\sigma_{x_{\text{in}}}^2 + 1)}{2\sigma_{x_{\text{in}}}^2}\right) dx_{\text{in}_i}$$

$$= \frac{1}{2\pi\sigma_{x_{\text{in}}}} \frac{\sigma_{x_{\text{in}}}}{\sqrt{(2\sigma_{x_{\text{in}}}^2 + 1)}} \int_{-\infty}^{\infty} \frac{x_{\text{in}_i}^2}{\sqrt{2\pi}\frac{\sigma_{x_{\text{in}}}}{\sqrt{(2\sigma_{x_{\text{in}}}^2+1)}}} \exp\left(\frac{-x_{\text{in}_i}^2(2\sigma_{x_{\text{in}}}^2 + 1)}{2\sigma_{x_{\text{in}}}^2}\right) dx_{\text{in}_i}$$

$$= \frac{1}{2\pi\sigma_{x_{\text{in}}}} \frac{\sigma_{x_{\text{in}}}}{\sqrt{(2\sigma_{x_{\text{in}}}^2 + 1)}} \frac{\sigma_{x_{\text{in}}}^2}{(2\sigma_{x_{\text{in}}}^2 + 1)} \qquad \text{(Definition of variance)}$$

$$I_3 = \frac{\sigma_{x_{\text{in}}}^2}{2\pi(2\sigma_{x_{\text{in}}}^2 + 1)^{\frac{3}{2}}}$$

$$I_4 = \int_{-\infty}^{\infty} \frac{\operatorname{erf}(\frac{x_{\text{in}_i}}{\sqrt{2}})}{2} \frac{\exp\left(\frac{-x_{\text{in}_i}^2}{2\sigma_{x_{\text{in}}}^2}\right)}{\sqrt{2\pi}\sigma_{x_{\text{in}}}} dx_{\text{in}_i} = 0 \qquad \text{(Integral of odd function)}$$

$$I_5 = \int_{-\infty}^{\infty} \frac{x_{\text{in}_i} \exp\left(\frac{-x_{\text{in}_i}^2}{2}\right)}{\sqrt{2\pi}} \frac{\exp\left(\frac{-x_{\text{in}_i}^2}{2\sigma_{x_{\text{in}}}^2}\right)}{\sqrt{2\pi}\sigma_{x_{\text{in}}}} dx_{\text{in}_i} = 0 \qquad \text{(Integral of odd function)}$$

$$I_6 = \int_{-\infty}^{\infty} \frac{x_{\text{in}_i} \exp\left(\frac{-x_{\text{in}_i}^2}{2}\right)\operatorname{erf}(\frac{x_{\text{in}_i}}{\sqrt{2}})}{\sqrt{2\pi}} \frac{\exp\left(\frac{-x_{\text{in}_i}^2}{2\sigma_{x_{\text{in}}}^2}\right)}{\sqrt{2\pi}\sigma_{x_{\text{in}}}} dx_{\text{in}_i}$$

$$= \frac{1}{2\pi\sigma_{x_{\text{in}}}} \int_{-\infty}^{\infty} x_{\text{in}_i}\operatorname{erf}(\frac{x_{\text{in}_i}}{\sqrt{2}}) \exp\left(\frac{-x_{\text{in}_i}^2(\sigma_{x_{\text{in}}}^2 + 1)}{2\sigma_{x_{\text{in}}}^2}\right) dx_{\text{in}_i}$$

From 2.6.1.4 of Lipovetsky (2020), $\int_{-\infty}^{\infty} z\operatorname{erf}(az) \exp\left(-a_1 z^2\right) dz = \frac{a}{a_1\sqrt{a^2 + a_1}}$

Substituting, $a = \frac{1}{\sqrt{2}}, a_1 = \frac{(\sigma_{x_{\text{in}}}^2 + 1)}{2\sigma_{x_{\text{in}}}^2}$, we have

$$I_6 = \frac{1}{2\pi\sigma_{x_{\text{in}}}} \frac{\frac{1}{\sqrt{2}}}{\frac{(\sigma_{x_{\text{in}}}^2 + 1)}{2\sigma_{x_{\text{in}}}^2}\sqrt{\frac{1}{2} + \frac{(\sigma_{x_{\text{in}}}^2 + 1)}{2\sigma_{x_{\text{in}}}^2}}}$$

$$= \frac{1}{2\pi\sigma_{x_{\text{in}}}} \frac{2\sigma_{x_{\text{in}}}^3}{(\sigma_{x_{\text{in}}}^2+1)\sqrt{2\sigma_{x_{\text{in}}}^2+1}}$$

$$I_6 = \frac{\sigma_{x_{\text{in}}}^2}{\pi(\sigma_{x_{\text{in}}}^2+1)\sqrt{2\sigma_{x_{\text{in}}}^2+1}}$$

$$I = I_1 + I_2 + I_3 + I_4 + I_5 + I_6$$

$$= \frac{1}{4} + \frac{1}{2\pi}\sin^{-1}\left(\frac{\sigma_{x_{\text{in}}}^2}{\sigma_{x_{\text{in}}}^2+1}\right) + \frac{\sigma_{x_{\text{in}}}^2}{2\pi(2\sigma_{x_{\text{in}}}^2+1)^{\frac{3}{2}}} + \frac{\sigma_{x_{\text{in}}}^2}{\pi(\sigma_{x_{\text{in}}}^2+1)\sqrt{2\sigma_{x_{\text{in}}}^2+1}}$$

$$= \frac{1}{4} + \frac{1}{2\pi}\sin^{-1}\left(\frac{\sigma_{x_{\text{in}}}^2}{\sigma_{x_{\text{in}}}^2+1}\right) + \frac{\sigma_{x_{\text{in}}}^2(4\sigma_{x_{\text{in}}}^2+2+\sigma_{x_{\text{in}}}^2+1)}{2\pi(\sigma_{x_{\text{in}}}^2+1)(2\sigma_{x_{\text{in}}}^2+1)^{\frac{3}{2}}}$$

$$I = \frac{1}{4} + \frac{1}{2\pi}\sin^{-1}\left(\frac{\sigma_{x_{\text{in}}}^2}{\sigma_{x_{\text{in}}}^2+1}\right) + \frac{\sigma_{x_{\text{in}}}^2(5\sigma_{x_{\text{in}}}^2+3)}{2\pi(\sigma_{x_{\text{in}}}^2+1)(2\sigma_{x_{\text{in}}}^2+1)^{\frac{3}{2}}}$$

So the variance of gradient of input of GeLU comes out to be

$$\mathbb{E}[g_{\text{in}_i}^2] = I\sigma_{g_{\text{out}}}^2$$

$$\boxed{\sigma_{g_{\text{in}}}^2 = \left[\frac{1}{4} + \frac{1}{2\pi}\sin^{-1}\left(\frac{\sigma_{x_{\text{in}}}^2}{\sigma_{x_{\text{in}}}^2+1}\right) + \frac{\sigma_{x_{\text{in}}}^2(5\sigma_{x_{\text{in}}}^2+3)}{2\pi(\sigma_{x_{\text{in}}}^2+1)(2\sigma_{x_{\text{in}}}^2+1)^{\frac{3}{2}}}\right]\sigma_{g_{\text{out}}}^2}$$

If for two inputs $\mathbf{x}_{\text{in}}$ and $\mathbf{y}_{\text{in}}$ for all $i$ we have $\text{Corr}(g_{\text{out}_{x_i}}, g_{\text{out}_{y_i}}) = r_{g_{\text{out}}}^l$, and $g_{\text{in}_{x_i}}, g_{\text{in}_{y_i}}$ be the gradient after passing through GeLU layer. Then we have,

$$\mathbb{E}[g_{\text{in}_{x_i}}g_{\text{in}_{y_i}}] =$$

$$= \mathbb{E}[(\frac{1}{2}(1+\text{erf}(\frac{x_{\text{in}_i}}{\sqrt{2}})) + \frac{x_{\text{in}_i}}{\sqrt{2\pi}}\exp\left(\frac{-x_{\text{in}_i}^2}{2}\right))g_{\text{out}_{x_i}}(\frac{1}{2}(1+\text{erf}(\frac{y_{\text{in}_i}}{\sqrt{2}})) + \frac{y_{\text{in}_i}}{\sqrt{2\pi}}\exp\left(\frac{-y_{\text{in}_i}^2}{2}\right))g_{\text{out}_{y_i}}]$$

$$\mathbb{E}[g_{\text{in}_{x_i}}g_{\text{in}_{y_i}}] = \mathbb{E}[(\frac{1}{2}(1+\text{erf}(\frac{x_{\text{in}_i}}{\sqrt{2}}))+$$

$$\frac{x_{\text{in}_i}}{\sqrt{2\pi}}\exp\left(\frac{-x_{\text{in}_i}^2}{2}\right))(\frac{1}{2}(1+\text{erf}(\frac{y_{\text{in}_i}}{\sqrt{2}})) + \frac{y_{\text{in}_i}}{\sqrt{2\pi}}\exp\left(\frac{-y_{\text{in}_i}^2}{2}\right))]\mathbb{E}[g_{\text{out}_{x_i}}g_{\text{out}_{y_i}}]$$

$$= \mathbb{E}[(\frac{1}{2}(1+\text{erf}(\frac{x_{\text{in}_i}}{\sqrt{2}}))+$$

$$\frac{x_{\text{in}_i}}{\sqrt{2\pi}}\exp\left(\frac{-x_{\text{in}_i}^2}{2}\right))(\frac{1}{2}(1+\text{erf}(\frac{y_{\text{in}_i}}{\sqrt{2}})) + \frac{y_{\text{in}_i}}{\sqrt{2\pi}}\exp\left(\frac{-y_{\text{in}_i}^2}{2}\right))]r_{g_{\text{out}}}^l\sigma_{g_{\text{out}}}^2$$

$$I = \mathbb{E}[(\frac{1}{2}(1+\text{erf}(\frac{x_{\text{in}_i}}{\sqrt{2}}))+$$

$$\frac{x_{\text{in}_i}}{\sqrt{2\pi}}\exp\left(\frac{-x_{\text{in}_i}^2}{2}\right))(\frac{1}{2}(1+\text{erf}(\frac{y_{\text{in}_i}}{\sqrt{2}})) + \frac{y_{\text{in}_i}}{\sqrt{2\pi}}\exp\left(\frac{-y_{\text{in}_i}^2}{2}\right))]$$

$$= \int_{-\infty}^{\infty}(\frac{1}{2}(1+\text{erf}(\frac{x_{\text{in}_i}}{\sqrt{2}}))+$$

$$\frac{x_{\text{in}_i}}{\sqrt{2\pi}}\exp\left(\frac{-x_{\text{in}_i}^2}{2}\right))(\frac{1}{2}(1+\text{erf}(\frac{y_{\text{in}_i}}{\sqrt{2}})) + \frac{y_{\text{in}_i}}{\sqrt{2\pi}}\exp\left(\frac{-y_{\text{in}_i}^2}{2}\right))p_{x_{\text{in}_i},y_{\text{in}_i}}\,dx_{\text{in}_i}\,dy_{\text{in}_i}$$

Where $p_{x_{\text{in}_i},y_{\text{in}_i}} = \dfrac{1}{2\pi\sigma_{x_{\text{in}}}^2\sqrt{(1-(r_{x_{\text{in}}}^l)^2)}}\exp\left(\dfrac{-x_{\text{in}_i}^2+2r_{x_{\text{in}}}^l x_{\text{in}_i}y_{\text{in}_i}-y_{\text{in}_i}^2}{2\sigma_{x_{\text{in}}}^2(1-(r_{x_{\text{in}}}^l)^2)}\right)$

$$I = \int_{-\infty}^{\infty}\frac{(\frac{1}{2}(1+\text{erf}(\frac{y_{\text{in}_i}}{\sqrt{2}})) + \frac{y_{\text{in}_i}}{\sqrt{2\pi}}\exp\left(\frac{-y_{\text{in}_i}^2}{2}\right))}{2\pi\sigma_{x_{\text{in}}}^2\sqrt{(1-(r_{x_{\text{in}}}^l)^2)}}\exp\left(\frac{-y_{\text{in}_i}^2}{2\sigma_{x_{\text{in}}}^2(1-(r_{x_{\text{in}}}^l)^2)}\right)I_X\,dy_{\text{in}_i}$$

Where,

$$I_X = \int_{-\infty}^{\infty} (\frac{1}{2}(1 + \mathrm{erf}(\frac{x_{\mathrm{in}_i}}{\sqrt{2}})) + \frac{x_{\mathrm{in}_i}}{\sqrt{2\pi}} \exp{(\frac{-x_{\mathrm{in}_i}^2}{2})}) \exp{(\frac{-x_{\mathrm{in}_i}^2 + 2r_{x_{\mathrm{in}}}^l x_{\mathrm{in}_i} y_{\mathrm{in}_i}}{2\sigma_{x_{\mathrm{in}}}^2 (1 - (r_{x_{\mathrm{in}}}^l)^2)})} dx_{\mathrm{in}_i}$$

$$I_{X,1} = \int_{-\infty}^{\infty} \frac{1}{2} \exp{(\frac{-x_{\mathrm{in}_i}^2 + 2r_{x_{\mathrm{in}}}^l x_{\mathrm{in}_i} y_{\mathrm{in}_i}}{2\sigma_{x_{\mathrm{in}}}^2 (1 - (r_{x_{\mathrm{in}}}^l)^2)})} dx_{\mathrm{in}_i}$$

$$= \frac{1}{2} \int_{-\infty}^{\infty} \exp{(\frac{-x_{\mathrm{in}_i}^2 + 2r_{x_{\mathrm{in}}}^l x_{\mathrm{in}_i} y_{\mathrm{in}_i}}{2\sigma_{x_{\mathrm{in}}}^2 (1 - (r_{x_{\mathrm{in}}}^l)^2)})} \exp{(\frac{-(r_{x_{\mathrm{in}}}^l)^2 y_{\mathrm{in}_i}^2}{2\sigma_{x_{\mathrm{in}}}^2 (1 - (r_{x_{\mathrm{in}}}^l)^2)})} \exp{(\frac{(r_{x_{\mathrm{in}}}^l)^2 y_{\mathrm{in}_i}^2}{2\sigma_{x_{\mathrm{in}}}^2 (1 - (r_{x_{\mathrm{in}}}^l)^2)})} dx_{\mathrm{in}_i}$$

$$= \frac{1}{2} \exp{(\frac{(r_{x_{\mathrm{in}}}^l)^2 y_{\mathrm{in}_i}^2}{2\sigma_{x_{\mathrm{in}}}^2 (1 - (r_{x_{\mathrm{in}}}^l)^2)})} \int_{-\infty}^{\infty} \exp{(\frac{-(x_{\mathrm{in}_i} - r_{x_{\mathrm{in}}}^l y_{\mathrm{in}_i})^2}{2\sigma_{x_{\mathrm{in}}}^2 (1 - (r_{x_{\mathrm{in}}}^l)^2)})} dx_{\mathrm{in}_i}$$

$$= \frac{1}{2} \exp{(\frac{(r_{x_{\mathrm{in}}}^l)^2 y_{\mathrm{in}_i}^2}{2\sigma_{x_{\mathrm{in}}}^2 (1 - (r_{x_{\mathrm{in}}}^l)^2)})} \sqrt{2\pi}\sigma_{x_{\mathrm{in}}} \sqrt{(1 - (r_{x_{\mathrm{in}}}^l)^2)} \int_{-\infty}^{\infty} \frac{\exp{(\frac{-(x_{\mathrm{in}_i} - r_{x_{\mathrm{in}}}^l y_{\mathrm{in}_i})^2}{2\sigma_{x_{\mathrm{in}}}^2 (1 - (r_{x_{\mathrm{in}}}^l)^2)})}}{\sqrt{2\pi}\sigma_{x_{\mathrm{in}}} \sqrt{(1 - (r_{x_{\mathrm{in}}}^l)^2)}} dx_{\mathrm{in}_i}$$

$$I_{X,1} = \frac{\sqrt{2\pi}\sigma_{x_{\mathrm{in}}} \sqrt{(1 - (r_{x_{\mathrm{in}}}^l)^2)}}{2} \exp{(\frac{(r_{x_{\mathrm{in}}}^l)^2 y_{\mathrm{in}_i}^2}{2\sigma_{x_{\mathrm{in}}}^2 (1 - (r_{x_{\mathrm{in}}}^l)^2)})}$$

$$I_{X,2} = \int_{-\infty}^{\infty} \frac{\mathrm{erf}(\frac{x_{\mathrm{in}_i}}{\sqrt{2}})}{2} \exp{(\frac{-x_{\mathrm{in}_i}^2 + 2r_{x_{\mathrm{in}}}^l x_{\mathrm{in}_i} y_{\mathrm{in}_i}}{2\sigma_{x_{\mathrm{in}}}^2 (1 - (r_{x_{\mathrm{in}}}^l)^2)})} dx_{\mathrm{in}_i}$$

$$= \frac{1}{2} \int_{-\infty}^{\infty} \mathrm{erf}(\frac{x_{\mathrm{in}_i}}{\sqrt{2}}) \exp{(\frac{-x_{\mathrm{in}_i}^2 + 2r_{x_{\mathrm{in}}}^l x_{\mathrm{in}_i} y_{\mathrm{in}_i}}{2\sigma_{x_{\mathrm{in}}}^2 (1 - (r_{x_{\mathrm{in}}}^l)^2)})} dx_{\mathrm{in}_i}$$

From 2.7.1.6 of Lipovetsky (2020),

$$\int_{-\infty}^{\infty} \mathrm{erf}(a_1 z) \exp{(-az^2 + bz)} dz = \sqrt{\frac{\pi}{a}} \exp{(\frac{b^2}{4a})} \mathrm{erf}(\frac{a_1 b}{2\sqrt{a^2 + aa_1^2}})$$

Substituting $a_1 = \frac{1}{\sqrt{2}}, a = \frac{1}{2\sigma_{x_{\mathrm{in}}}^2 (1 - (r_{x_{\mathrm{in}}}^l)^2)}, b = \frac{r_{x_{\mathrm{in}}}^l y_{\mathrm{in}_i}}{\sigma_{x_{\mathrm{in}}}^2 (1 - (r_{x_{\mathrm{in}}}^l)^2)}$

$$I_{X,2} = \frac{1}{2} \sqrt{\frac{\pi}{\frac{1}{2\sigma_{x_{\mathrm{in}}}^2 (1 - (r_{x_{\mathrm{in}}}^l)^2)}}} \exp{(\frac{\frac{(r_{x_{\mathrm{in}}}^l)^2 y_{\mathrm{in}_i}^2}{\sigma_{x_{\mathrm{in}}}^4 (1 - (r_{x_{\mathrm{in}}}^l)^2)^2}}{4 \frac{1}{2\sigma_{x_{\mathrm{in}}}^2 (1 - (r_{x_{\mathrm{in}}}^l)^2)}})} \mathrm{erf}(\frac{\frac{r_{x_{\mathrm{in}}}^l y_{\mathrm{in}_i}}{\sqrt{2}\sigma_{x_{\mathrm{in}}}^2 (1 - (r_{x_{\mathrm{in}}}^l)^2)}}{2\sqrt{\frac{1}{4\sigma_{x_{\mathrm{in}}}^4 (1 - (r_{x_{\mathrm{in}}}^l)^2)^2} + \frac{1}{4\sigma_{x_{\mathrm{in}}}^2 (1 - (r_{x_{\mathrm{in}}}^l)^2)}}})$$

$$I_{X,2} = \frac{\sqrt{2\pi}\sigma_{x_{\mathrm{in}}} \sqrt{(1 - (r_{x_{\mathrm{in}}}^l)^2)}}{2} \exp{(\frac{(r_{x_{\mathrm{in}}}^l)^2 y_{\mathrm{in}_i}^2}{2\sigma_{x_{\mathrm{in}}}^2 (1 - (r_{x_{\mathrm{in}}}^l)^2)})} \mathrm{erf}(\frac{r_{x_{\mathrm{in}}}^l y_{\mathrm{in}_i}}{\sqrt{2(\sigma_{x_{\mathrm{in}}}^2 (1 - (r_{x_{\mathrm{in}}}^l)^2) + 1)}})$$

$$I_{X,3} = \int_{-\infty}^{\infty} \frac{x_{\mathrm{in}_i}}{\sqrt{2\pi}} \exp{(\frac{-x_{\mathrm{in}_i}^2}{2})} \exp{(\frac{-x_{\mathrm{in}_i}^2 + 2r_{x_{\mathrm{in}}}^l x_{\mathrm{in}_i} y_{\mathrm{in}_i}}{2\sigma_{x_{\mathrm{in}}}^2 (1 - (r_{x_{\mathrm{in}}}^l)^2)})} dx_{\mathrm{in}_i}$$

$$= \int_{-\infty}^{\infty} \frac{x_{\mathrm{in}_i}}{\sqrt{2\pi}} \exp{(\frac{-x_{\mathrm{in}_i}^2 (\sigma_{x_{\mathrm{in}}}^2 (1 - (r_{x_{\mathrm{in}}}^l)^2) + 1) + 2r_{x_{\mathrm{in}}}^l x_{\mathrm{in}_i} y_{\mathrm{in}_i}}{2\sigma_{x_{\mathrm{in}}}^2 (1 - (r_{x_{\mathrm{in}}}^l)^2)})} dx_{\mathrm{in}_i}$$

$$= \int_{-\infty}^{\infty} \frac{x_{\mathrm{in}_i}}{\sqrt{2\pi}} \exp{(\frac{-x_{\mathrm{in}_i}^2 + \frac{2r_{x_{\mathrm{in}}}^l x_{\mathrm{in}_i} y_{\mathrm{in}_i}}{(\sigma_{x_{\mathrm{in}}}^2 (1 - (r_{x_{\mathrm{in}}}^l)^2) + 1)}}{\frac{2\sigma_{x_{\mathrm{in}}}^2 (1 - (r_{x_{\mathrm{in}}}^l)^2)}{(\sigma_{x_{\mathrm{in}}}^2 (1 - (r_{x_{\mathrm{in}}}^l)^2) + 1)}})} dx_{\mathrm{in}_i}$$

$$= \int_{-\infty}^{\infty} \frac{x_{\mathrm{in}_i}}{\sqrt{2\pi}} \exp{(\frac{-x_{\mathrm{in}_i}^2 + \frac{2r_{x_{\mathrm{in}}}^l x_{\mathrm{in}_i} y_{\mathrm{in}_i}}{(\sigma_{x_{\mathrm{in}}}^2 (1 - (r_{x_{\mathrm{in}}}^l)^2) + 1)}}{\frac{2\sigma_{x_{\mathrm{in}}}^2 (1 - (r_{x_{\mathrm{in}}}^l)^2)}{(\sigma_{x_{\mathrm{in}}}^2 (1 - (r_{x_{\mathrm{in}}}^l)^2) + 1)}})} \exp{(\frac{\frac{-(r_{x_{\mathrm{in}}}^l)^2 y_{\mathrm{in}_i}^2}{(\sigma_{x_{\mathrm{in}}}^2 (1 - (r_{x_{\mathrm{in}}}^l)^2) + 1)^2}}{\frac{2\sigma_{x_{\mathrm{in}}}^2 (1 - (r_{x_{\mathrm{in}}}^l)^2)}{(\sigma_{x_{\mathrm{in}}}^2 (1 - (r_{x_{\mathrm{in}}}^l)^2) + 1)}})} *$$

$$\exp{(\frac{\frac{(r_{x_{\mathrm{in}}}^l)^2 y_{\mathrm{in}_i}^2}{(\sigma_{x_{\mathrm{in}}}^2 (1 - (r_{x_{\mathrm{in}}}^l)^2) + 1)^2}}{\frac{2\sigma_{x_{\mathrm{in}}}^2 (1 - (r_{x_{\mathrm{in}}}^l)^2)}{(\sigma_{x_{\mathrm{in}}}^2 (1 - (r_{x_{\mathrm{in}}}^l)^2) + 1)}})} dx_{\mathrm{in}_i}$$

$$= \exp\left(\frac{(r_{x_{\text{in}}}^l)^2 y_{\text{in}_i}^2}{2\sigma_{x_{\text{in}}}^2(1-(r_{x_{\text{in}}}^l)^2)(\sigma_{x_{\text{in}}}^2(1-(r_{x_{\text{in}}}^l)^2)+1)}\right)*$$

$$\int_{-\infty}^{\infty} \frac{x_{\text{in}_i}}{\sqrt{2\pi}} \exp\left(\frac{-\left(x_{\text{in}_i} - \frac{r_{x_{\text{in}}}^l y_{\text{in}_i}}{(\sigma_{x_{\text{in}}}^2(1-(r_{x_{\text{in}}}^l)^2)+1)}\right)^2}{\frac{2\sigma_{x_{\text{in}}}^2(1-(r_{x_{\text{in}}}^l)^2)}{(\sigma_{x_{\text{in}}}^2(1-(r_{x_{\text{in}}}^l)^2)+1)}}\right) dx_{\text{in}_i}$$

$$= \exp\left(\frac{(r_{x_{\text{in}}}^l)^2 y_{\text{in}_i}^2}{2\sigma_{x_{\text{in}}}^2(1-(r_{x_{\text{in}}}^l)^2)(\sigma_{x_{\text{in}}}^2(1-(r_{x_{\text{in}}}^l)^2)+1)}\right) \frac{\sigma_{x_{\text{in}}}\sqrt{1-(r_{x_{\text{in}}}^l)^2}}{\sqrt{(\sigma_{x_{\text{in}}}^2(1-(r_{x_{\text{in}}}^l)^2)+1)}}$$

$$\int_{-\infty}^{\infty} \frac{x_{\text{in}_i}}{\sqrt{2\pi}\frac{\sigma_{x_{\text{in}}}\sqrt{1-(r_{x_{\text{in}}}^l)^2}}{\sqrt{(\sigma_{x_{\text{in}}}^2(1-(r_{x_{\text{in}}}^l)^2)+1)}}} \exp\left(\frac{-\left(x_{\text{in}_i} - \frac{r_{x_{\text{in}}}^l y_{\text{in}_i}}{(\sigma_{x_{\text{in}}}^2(1-(r_{x_{\text{in}}}^l)^2)+1)}\right)^2}{\frac{2\sigma_{x_{\text{in}}}^2(1-(r_{x_{\text{in}}}^l)^2)}{(\sigma_{x_{\text{in}}}^2(1-(r_{x_{\text{in}}}^l)^2)+1)}}\right) dx_{\text{in}_i}$$

$$= \exp\left(\frac{(r_{x_{\text{in}}}^l)^2 y_{\text{in}_i}^2}{2\sigma_{x_{\text{in}}}^2(1-(r_{x_{\text{in}}}^l)^2)(\sigma_{x_{\text{in}}}^2(1-(r_{x_{\text{in}}}^l)^2)+1)}\right).$$

$$\frac{\sigma_{x_{\text{in}}}\sqrt{1-(r_{x_{\text{in}}}^l)^2}}{\sqrt{(\sigma_{x_{\text{in}}}^2(1-(r_{x_{\text{in}}}^l)^2)+1)}} \frac{r_{x_{\text{in}}}^l y_{\text{in}_i}}{(\sigma_{x_{\text{in}}}^2(1-(r_{x_{\text{in}}}^l)^2)+1)}$$

$$I_{X,3} = \frac{r_{x_{\text{in}}}^l y_{\text{in}_i} \sigma_{x_{\text{in}}}\sqrt{1-(r_{x_{\text{in}}}^l)^2}}{(\sigma_{x_{\text{in}}}^2(1-(r_{x_{\text{in}}}^l)^2)+1)^{\frac{3}{2}}} \exp\left(\frac{(r_{x_{\text{in}}}^l)^2 y_{\text{in}_i}^2}{2\sigma_{x_{\text{in}}}^2(1-(r_{x_{\text{in}}}^l)^2)(\sigma_{x_{\text{in}}}^2(1-(r_{x_{\text{in}}}^l)^2)+1)}\right)$$

$$I =$$

$$\int_{-\infty}^{\infty} \frac{\left(\frac{1}{2}(1+\text{erf}(\frac{y_{\text{in}_i}}{\sqrt{2}})) + \frac{y_{\text{in}_i}}{\sqrt{2\pi}}\exp\left(\frac{-y_{\text{in}_i}^2}{2}\right)\right)}{2\pi\sigma_{x_{\text{in}}}^2\sqrt{(1-(r_{x_{\text{in}}}^l)^2)}} \exp\left(\frac{-y_{\text{in}_i}^2}{2\sigma_{x_{\text{in}}}^2(1-(r_{x_{\text{in}}}^l)^2)}\right)(I_{X,1}+I_{X,2}+I_{X,3}) dy_{\text{in}_i}$$

$$I_1 = \int_{-\infty}^{\infty} \frac{\left(\frac{1}{2}(1+\text{erf}(\frac{y_{\text{in}_i}}{\sqrt{2}})) + \frac{y_{\text{in}_i}}{\sqrt{2\pi}}\exp\left(\frac{-y_{\text{in}_i}^2}{2}\right)\right)}{2\pi\sigma_{x_{\text{in}}}^2\sqrt{(1-(r_{x_{\text{in}}}^l)^2)}} \exp\left(\frac{-y_{\text{in}_i}^2}{2\sigma_{x_{\text{in}}}^2(1-(r_{x_{\text{in}}}^l)^2)}\right) I_{X,1} dy_{\text{in}_i}$$

$$= \int_{-\infty}^{\infty} \frac{\left(\frac{1}{2}(1+\text{erf}(\frac{y_{\text{in}_i}}{\sqrt{2}})) + \frac{y_{\text{in}_i}}{\sqrt{2\pi}}\exp\left(\frac{-y_{\text{in}_i}^2}{2}\right)\right)}{2\pi\sigma_{x_{\text{in}}}^2\sqrt{(1-(r_{x_{\text{in}}}^l)^2)}} \exp\left(\frac{-y_{\text{in}_i}^2}{2\sigma_{x_{\text{in}}}^2(1-(r_{x_{\text{in}}}^l)^2)}\right)$$

$$\frac{\sqrt{2\pi}\sigma_{x_{\text{in}}}\sqrt{(1-(r_{x_{\text{in}}}^l)^2)}}{2} \exp\left(\frac{(r_{x_{\text{in}}}^l)^2 y_{\text{in}_i}^2}{2\sigma_{x_{\text{in}}}^2(1-(r_{x_{\text{in}}}^l)^2)}\right) dy_{\text{in}_i}$$

$$= \frac{1}{2}\int_{-\infty}^{\infty} \frac{\left(\frac{1}{2}(1+\text{erf}(\frac{y_{\text{in}_i}}{\sqrt{2}})) + \frac{y_{\text{in}_i}}{\sqrt{2\pi}}\exp\left(\frac{-y_{\text{in}_i}^2}{2}\right)\right)}{\sqrt{2\pi}\sigma_{x_{\text{in}}}} \exp\left(\frac{-y_{\text{in}_i}^2}{2\sigma_{x_{\text{in}}}^2}\right) dy_{\text{in}_i}$$

$$I_{1,1} = \frac{1}{4}\int_{-\infty}^{\infty} \frac{1}{\sqrt{2\pi}\sigma_{x_{\text{in}}}} \exp\left(\frac{-y_{\text{in}_i}^2}{2\sigma_{x_{\text{in}}}^2}\right) dy_{\text{in}_i} = \frac{1}{4}$$

$$I_{1,2} = \frac{1}{4}\int_{-\infty}^{\infty} \frac{\text{erf}(\frac{y_{\text{in}_i}}{\sqrt{2}})}{\sqrt{2\pi}\sigma_{x_{\text{in}}}} \exp\left(\frac{-y_{\text{in}_i}^2}{2\sigma_{x_{\text{in}}}^2}\right) dy_{\text{in}_i} = 0 \qquad \text{(Integral of odd function)}$$

$$I_{1,3} = \frac{1}{2}\int_{-\infty}^{\infty} \frac{y_{\text{in}_i}\exp\left(\frac{-y_{\text{in}_i}^2}{2}\right)}{2\pi\sigma_{x_{\text{in}}}} \exp\left(\frac{-y_{\text{in}_i}^2}{2\sigma_{x_{\text{in}}}^2}\right) dy_{\text{in}_i} = 0 \qquad \text{(Integral of odd function)}$$

$$I_2 = \int_{-\infty}^{\infty} \frac{(\frac{1}{2}(1 + \text{erf}(\frac{y_{\text{in}_i}}{\sqrt{2}})) + \frac{y_{\text{in}_i}}{\sqrt{2\pi}} \exp{(\frac{-y_{\text{in}_i}^2}{2})})}{2\pi\sigma_{x_{\text{in}}}^2 \sqrt{(1 - (r_{x_{\text{in}}}^l)^2)}} \exp{(\frac{-y_{\text{in}_i}^2}{2\sigma_{x_{\text{in}}}^2(1 - (r_{x_{\text{in}}}^l)^2)})} I_{X,2} dy_{\text{in}_i}$$

$$= \int_{-\infty}^{\infty} \frac{(\frac{1}{2}(1 + \text{erf}(\frac{y_{\text{in}_i}}{\sqrt{2}})) + \frac{y_{\text{in}_i}}{\sqrt{2\pi}} \exp{(\frac{-y_{\text{in}_i}^2}{2})})}{2\pi\sigma_{x_{\text{in}}}^2 \sqrt{(1 - (r_{x_{\text{in}}}^l)^2)}} \exp{(\frac{-y_{\text{in}_i}^2}{2\sigma_{x_{\text{in}}}^2(1 - (r_{x_{\text{in}}}^l)^2)})}$$

$$\frac{\sqrt{2\pi}\sigma_{x_{\text{in}}}\sqrt{(1 - (r_{x_{\text{in}}}^l)^2)}}{2} \exp{(\frac{(r_{x_{\text{in}}}^l)^2 y_{\text{in}_i}^2}{2\sigma_{x_{\text{in}}}^2(1 - (r_{x_{\text{in}}}^l)^2)})} \text{erf}(\frac{r_{x_{\text{in}}}^l y_{\text{in}_i}}{\sqrt{2(\sigma_{x_{\text{in}}}^2(1 - (r_{x_{\text{in}}}^l)^2) + 1)}}) dy_{\text{in}_i}$$

$$= \frac{1}{2}\int_{-\infty}^{\infty} \frac{(\frac{1}{2}(1 + \text{erf}(\frac{y_{\text{in}_i}}{\sqrt{2}})) + \frac{y_{\text{in}_i}}{\sqrt{2\pi}} \exp{(\frac{-y_{\text{in}_i}^2}{2})})}{\sqrt{2\pi}\sigma_{x_{\text{in}}}} \exp{(\frac{-y_{\text{in}_i}^2}{2\sigma_{x_{\text{in}}}^2})}.$$

$$\text{erf}(\frac{r_{x_{\text{in}}}^l y_{\text{in}_i}}{\sqrt{2(\sigma_{x_{\text{in}}}^2(1 - (r_{x_{\text{in}}}^l)^2) + 1)}}) dy_{\text{in}_i}$$

$$I_{2,1} = \frac{1}{4}\int_{-\infty}^{\infty} \frac{1}{\sqrt{2\pi}\sigma_{x_{\text{in}}}} \exp{(\frac{-y_{\text{in}_i}^2}{2\sigma_{x_{\text{in}}}^2})} \text{erf}(\frac{r_{x_{\text{in}}}^l y_{\text{in}_i}}{\sqrt{2(\sigma_{x_{\text{in}}}^2(1 - (r_{x_{\text{in}}}^l)^2) + 1)}}) dy_{\text{in}_i} = 0$$

$$\text{(Integral of odd function)}$$

$$I_{2,2} = \frac{1}{4\sqrt{2\pi}\sigma_{x_{\text{in}}}} \int_{-\infty}^{\infty} \text{erf}(\frac{y_{\text{in}_i}}{\sqrt{2}}) \exp{(\frac{-y_{\text{in}_i}^2}{2\sigma_{x_{\text{in}}}^2})} \text{erf}(\frac{r_{x_{\text{in}}}^l y_{\text{in}_i}}{\sqrt{2(\sigma_{x_{\text{in}}}^2(1 - (r_{x_{\text{in}}}^l)^2) + 1)}}) dy_{\text{in}_i}$$

From 2.7.1.3 of Lipovetsky (2020),

$$\int_{-\infty}^{\infty} \text{erf}(a_1 z)\text{erf}(a_2 z)\exp{(-az^2)}dz = \frac{2}{\sqrt{\pi a}}\tan^{-1}{(\frac{a_1 a_2}{\sqrt{a^2 + aa_1^2 + aa_2^2}})}$$

Substituting $a = \frac{1}{2\sigma_{x_{\text{in}}}^2}, a_1 = \frac{1}{\sqrt{2}}, a_2 = \frac{r_{x_{\text{in}}}^l}{\sqrt{2(\sigma_{x_{\text{in}}}^2(1 - (r_{x_{\text{in}}}^l)^2) + 1)}}$

$$I_{2,2} = \frac{1}{4\sqrt{2\pi}\sigma_{x_{\text{in}}}} \frac{2}{\sqrt{\pi\frac{1}{2\sigma_{x_{\text{in}}}^2}}} \tan^{-1}{(\frac{\frac{r_{x_{\text{in}}}^l}{2\sqrt{(\sigma_{x_{\text{in}}}^2(1 - (r_{x_{\text{in}}}^l)^2) + 1)}}}{\sqrt{\frac{1}{4\sigma_{x_{\text{in}}}^4} + \frac{1}{4\sigma_{x_{\text{in}}}^2} + \frac{(r_{x_{\text{in}}}^l)^2}{4\sigma_{x_{\text{in}}}^2(\sigma_{x_{\text{in}}}^2(1 - (r_{x_{\text{in}}}^l)^2) + 1)}}})}$$

$$I_{2,2} = \frac{1}{2\pi}\tan^{-1}{(\frac{r_{x_{\text{in}}}^l \sigma_{x_{\text{in}}}^2}{\sqrt{\sigma_{x_{\text{in}}}^4 + 2\sigma_{x_{\text{in}}}^2 + 1 - (r_{x_{\text{in}}}^l)^2\sigma_{x_{\text{in}}}^4}})} = \frac{1}{2\pi}\tan^{-1}{(\frac{r_{x_{\text{in}}}^l \sigma_{x_{\text{in}}}^2}{\sqrt{(\sigma_{x_{\text{in}}}^2 + 1)^2 - (r_{x_{\text{in}}}^l\sigma_{x_{\text{in}}}^2)^2}})}$$

$$I_{2,2} = \frac{1}{2\pi}\sin^{-1}{(\frac{r_{x_{\text{in}}}^l \sigma_{x_{\text{in}}}^2}{\sigma_{x_{\text{in}}}^2 + 1})}$$

$$I_{2,3} = \frac{1}{4\pi\sigma_{x_{\text{in}}}} \int_{-\infty}^{\infty} y_{\text{in}_i} \exp{(\frac{-y_{\text{in}_i}^2}{2})} \exp{(\frac{-y_{\text{in}_i}^2}{2\sigma_{x_{\text{in}}}^2})} \text{erf}(\frac{r_{x_{\text{in}}}^l y_{\text{in}_i}}{\sqrt{2(\sigma_{x_{\text{in}}}^2(1 - (r_{x_{\text{in}}}^l)^2) + 1)}}) dy_{\text{in}_i}$$

$$= \frac{1}{4\pi\sigma_{x_{\text{in}}}} \int_{-\infty}^{\infty} y_{\text{in}_i} \exp{(\frac{-y_{\text{in}_i}^2(\sigma_{x_{\text{in}}}^2 + 1)}{2\sigma_{x_{\text{in}}}^2})} \text{erf}(\frac{r_{x_{\text{in}}}^l y_{\text{in}_i}}{\sqrt{2(\sigma_{x_{\text{in}}}^2(1 - (r_{x_{\text{in}}}^l)^2) + 1)}}) dy_{\text{in}_i}$$

From 2.6.1.4 of Lipovetsky (2020), $\int_{-\infty}^{\infty} z\,\text{erf}(az)\exp{(-a_1 z^2)}dz = \frac{a}{a_1\sqrt{a^2 + a_1}}$

Substituting, $a = \frac{r_{x_{\text{in}}}^l}{\sqrt{2(\sigma_{x_{\text{in}}}^2(1 - (r_{x_{\text{in}}}^l)^2) + 1)}}, a_1 = \frac{(\sigma_{x_{\text{in}}}^2 + 1)}{2\sigma_{x_{\text{in}}}^2}$, we have

$$I_{2,3} = \frac{1}{4\pi\sigma_{x_{\text{in}}}} \frac{\frac{r_{x_{\text{in}}}^l}{\sqrt{2(\sigma_{x_{\text{in}}}^2(1 - (r_{x_{\text{in}}}^l)^2) + 1)}}}{\frac{(\sigma_{x_{\text{in}}}^2 + 1)}{2\sigma_{x_{\text{in}}}^2}\sqrt{\frac{(r_{x_{\text{in}}}^l)^2}{2(\sigma_{x_{\text{in}}}^2(1 - (r_{x_{\text{in}}}^l)^2) + 1)} + \frac{(\sigma_{x_{\text{in}}}^2 + 1)}{2\sigma_{x_{\text{in}}}^2}}}$$

$$= \frac{r_{x_{\mathrm{in}}}^l \sigma_{x_{\mathrm{in}}}^2}{2\pi(\sigma_{x_{\mathrm{in}}}^2 + 1)\sqrt{\sigma_{x_{\mathrm{in}}}^4 + 2\sigma_{x_{\mathrm{in}}}^2 + 1 - (r_{x_{\mathrm{in}}}^l)^2 \sigma_{x_{\mathrm{in}}}^4}}$$

$$I_{2,3} = \frac{r_{x_{\mathrm{in}}}^l \sigma_{x_{\mathrm{in}}}^2}{2\pi(\sigma_{x_{\mathrm{in}}}^2 + 1)\sqrt{(\sigma_{x_{\mathrm{in}}}^2 + 1)^2 - (r_{x_{\mathrm{in}}}^l \sigma_{x_{\mathrm{in}}}^2)^2}}$$

$$I_3 = \int_{-\infty}^{\infty} \frac{(\frac{1}{2}(1 + \mathrm{erf}(\frac{y_{\mathrm{in}_i}}{\sqrt{2}})) + \frac{y_{\mathrm{in}_i}}{\sqrt{2\pi}}\exp{(\frac{-y_{\mathrm{in}_i}^2}{2})})}{2\pi\sigma_{x_{\mathrm{in}}}^2\sqrt{(1 - (r_{x_{\mathrm{in}}}^l)^2)}}\exp{(\frac{-y_{\mathrm{in}_i}^2}{2\sigma_{x_{\mathrm{in}}}^2(1 - (r_{x_{\mathrm{in}}}^l)^2)})}I_{X,3}dy_{\mathrm{in}_i}$$

$$= \int_{-\infty}^{\infty} \frac{(\frac{1}{2}(1 + \mathrm{erf}(\frac{y_{\mathrm{in}_i}}{\sqrt{2}})) + \frac{y_{\mathrm{in}_i}}{\sqrt{2\pi}}\exp{(\frac{-y_{\mathrm{in}_i}^2}{2})})}{2\pi\sigma_{x_{\mathrm{in}}}^2\sqrt{(1 - (r_{x_{\mathrm{in}}}^l)^2)}}\exp{(\frac{-y_{\mathrm{in}_i}^2}{2\sigma_{x_{\mathrm{in}}}^2(1 - (r_{x_{\mathrm{in}}}^l)^2)})}$$

$$\frac{r_{x_{\mathrm{in}}}^l y_{\mathrm{in}_i}\sigma_{x_{\mathrm{in}}}\sqrt{1 - (r_{x_{\mathrm{in}}}^l)^2}}{(\sigma_{x_{\mathrm{in}}}^2(1 - (r_{x_{\mathrm{in}}}^l)^2) + 1)^{\frac{3}{2}}}\exp{(\frac{(r_{x_{\mathrm{in}}}^l)^2 y_{\mathrm{in}_i}^2}{2\sigma_{x_{\mathrm{in}}}^2(1 - (r_{x_{\mathrm{in}}}^l)^2)(\sigma_{x_{\mathrm{in}}}^2(1 - (r_{x_{\mathrm{in}}}^l)^2) + 1)})}dy_{\mathrm{in}_i}$$

$$= \frac{r_{x_{\mathrm{in}}}^l}{2\pi\sigma_{x_{\mathrm{in}}}(\sigma_{x_{\mathrm{in}}}^2(1 - (r_{x_{\mathrm{in}}}^l)^2) + 1)^{\frac{3}{2}}}\int_{-\infty}^{\infty} y_{\mathrm{in}_i}(\frac{1}{2}(1 + \mathrm{erf}(\frac{y_{\mathrm{in}_i}}{\sqrt{2}})) + \frac{y_{\mathrm{in}_i}}{\sqrt{2\pi}}\exp{(\frac{-y_{\mathrm{in}_i}^2}{2})})$$

$$\exp{(\frac{-y_{\mathrm{in}_i}^2}{2\sigma_{x_{\mathrm{in}}}^2(1 - (r_{x_{\mathrm{in}}}^l)^2)})}\exp{(\frac{(r_{x_{\mathrm{in}}}^l)^2 y_{\mathrm{in}_i}^2}{2\sigma_{x_{\mathrm{in}}}^2(1 - (r_{x_{\mathrm{in}}}^l)^2)(\sigma_{x_{\mathrm{in}}}^2(1 - (r_{x_{\mathrm{in}}}^l)^2) + 1)})}dy_{\mathrm{in}_i}$$

$$= \frac{r_{x_{\mathrm{in}}}^l}{2\pi\sigma_{x_{\mathrm{in}}}(\sigma_{x_{\mathrm{in}}}^2(1 - (r_{x_{\mathrm{in}}}^l)^2) + 1)^{\frac{3}{2}}}\int_{-\infty}^{\infty} y_{\mathrm{in}_i}(\frac{1}{2}(1 + \mathrm{erf}(\frac{y_{\mathrm{in}_i}}{\sqrt{2}})) + \frac{y_{\mathrm{in}_i}}{\sqrt{2\pi}}\exp{(\frac{-y_{\mathrm{in}_i}^2}{2})})$$

$$\exp{(\frac{-y_{\mathrm{in}_i}^2(\sigma_{x_{\mathrm{in}}}^2(1 - (r_{x_{\mathrm{in}}}^l)^2) + 1 - (r_{x_{\mathrm{in}}}^l)^2)}{2(\sigma_{x_{\mathrm{in}}}^2(1 - (r_{x_{\mathrm{in}}}^l)^2) + 1)\sigma_{x_{\mathrm{in}}}^2(1 - (r_{x_{\mathrm{in}}}^l)^2)})}dy_{\mathrm{in}_i}$$

$$= \frac{r_{x_{\mathrm{in}}}^l}{2\pi\sigma_{x_{\mathrm{in}}}(\sigma_{x_{\mathrm{in}}}^2(1 - (r_{x_{\mathrm{in}}}^l)^2) + 1)^{\frac{3}{2}}}\int_{-\infty}^{\infty} y_{\mathrm{in}_i}(\frac{1}{2}(1 + \mathrm{erf}(\frac{y_{\mathrm{in}_i}}{\sqrt{2}})) + \frac{y_{\mathrm{in}_i}}{\sqrt{2\pi}}\exp{(\frac{-y_{\mathrm{in}_i}^2}{2})})$$

$$\exp{(\frac{-y_{\mathrm{in}_i}^2(\sigma_{x_{\mathrm{in}}}^2 + 1)}{2\sigma_{x_{\mathrm{in}}}^2(\sigma_{x_{\mathrm{in}}}^2(1 - (r_{x_{\mathrm{in}}}^l)^2) + 1)})}dy_{\mathrm{in}_i}$$

$$I_{3,1} = \frac{r_{x_{\mathrm{in}}}^l}{4\pi\sigma_{x_{\mathrm{in}}}(\sigma_{x_{\mathrm{in}}}^2(1 - (r_{x_{\mathrm{in}}}^l)^2) + 1)^{\frac{3}{2}}}\int_{-\infty}^{\infty} y_{\mathrm{in}_i}\exp{(\frac{-y_{\mathrm{in}_i}^2(\sigma_{x_{\mathrm{in}}}^2 + 1)}{2\sigma_{x_{\mathrm{in}}}^2(\sigma_{x_{\mathrm{in}}}^2(1 - (r_{x_{\mathrm{in}}}^l)^2) + 1)})}dy_{\mathrm{in}_i} = 0$$

$$\text{(Integral of odd function)}$$

$$I_{3,2} = \frac{r_{x_{\mathrm{in}}}^l}{4\pi\sigma_{x_{\mathrm{in}}}(\sigma_{x_{\mathrm{in}}}^2(1 - (r_{x_{\mathrm{in}}}^l)^2) + 1)^{\frac{3}{2}}}\int_{-\infty}^{\infty} y_{\mathrm{in}_i}\mathrm{erf}(\frac{y_{\mathrm{in}_i}}{\sqrt{2}})\exp{(\frac{-y_{\mathrm{in}_i}^2(\sigma_{x_{\mathrm{in}}}^2 + 1)}{2\sigma_{x_{\mathrm{in}}}^2(\sigma_{x_{\mathrm{in}}}^2(1 - (r_{x_{\mathrm{in}}}^l)^2) + 1)})}dy_{\mathrm{in}_i}$$

From 2.6.1.4 of Lipovetsky (2020), $\int_{-\infty}^{\infty} z\mathrm{erf}(az)\exp{(-a_1 z^2)}dz = \frac{a}{a_1\sqrt{a^2 + a_1}}$

Substituting, $a = \frac{1}{\sqrt{2}}$, $a_1 = \frac{(\sigma_{x_{\mathrm{in}}}^2 + 1)}{2\sigma_{x_{\mathrm{in}}}^2(\sigma_{x_{\mathrm{in}}}^2(1 - (r_{x_{\mathrm{in}}}^l)^2) + 1)}$, we have

$$I_{3,2} = \frac{r_{x_{\mathrm{in}}}^l}{4\pi\sigma_{x_{\mathrm{in}}}(\sigma_{x_{\mathrm{in}}}^2(1 - (r_{x_{\mathrm{in}}}^l)^2) + 1)^{\frac{3}{2}}}\frac{\frac{1}{\sqrt{2}}}{\frac{(\sigma_{x_{\mathrm{in}}}^2 + 1)}{2\sigma_{x_{\mathrm{in}}}^2(\sigma_{x_{\mathrm{in}}}^2(1 - (r_{x_{\mathrm{in}}}^l)^2) + 1)}\sqrt{\frac{1}{2} + \frac{(\sigma_{x_{\mathrm{in}}}^2 + 1)}{2\sigma_{x_{\mathrm{in}}}^2(\sigma_{x_{\mathrm{in}}}^2(1 - (r_{x_{\mathrm{in}}}^l)^2) + 1)}}}$$

$$= \frac{r_{x_{\mathrm{in}}}^l \sigma_{x_{\mathrm{in}}}^2}{2\pi(\sigma_{x_{\mathrm{in}}}^2 + 1)\sqrt{\sigma_{x_{\mathrm{in}}}^4 + 2\sigma_{x_{\mathrm{in}}}^2 + 1 - (r_{x_{\mathrm{in}}}^l)^2 \sigma_{x_{\mathrm{in}}}^4}}$$

$$I_{3,2} = \frac{r_{x_{\mathrm{in}}}^l \sigma_{x_{\mathrm{in}}}^2}{2\pi(\sigma_{x_{\mathrm{in}}}^2 + 1)\sqrt{(\sigma_{x_{\mathrm{in}}}^2 + 1)^2 - (r_{x_{\mathrm{in}}}^l \sigma_{x_{\mathrm{in}}}^2)^2}}$$

$$I_{3,3} = \frac{r_{x_{\mathrm{in}}}^l}{2\pi\sigma_{x_{\mathrm{in}}}(\sigma_{x_{\mathrm{in}}}^2(1 - (r_{x_{\mathrm{in}}}^l)^2) + 1)^{\frac{3}{2}}}.$$

$$\int_{-\infty}^{\infty} \frac{y_{\text{in}_i}^2}{\sqrt{2\pi}} \exp\left(\frac{-y_{\text{in}_i}^2}{2}\right) \exp\left(\frac{-y_{\text{in}_i}^2(\sigma_{x_{\text{in}}}^2+1)}{2\sigma_{x_{\text{in}}}^2(\sigma_{x_{\text{in}}}^2(1-(r_{x_{\text{in}}}^l)^2)+1)}\right) dy_{\text{in}_i}$$

$$= \frac{r_{x_{\text{in}}}^l}{2\pi\sigma_{x_{\text{in}}}(\sigma_{x_{\text{in}}}^2(1-(r_{x_{\text{in}}}^l)^2)+1)^{\frac{3}{2}}} \cdot$$

$$\int_{-\infty}^{\infty} \frac{y_{\text{in}_i}^2}{\sqrt{2\pi}} \exp\left(\frac{-y_{\text{in}_i}^2(\sigma_{x_{\text{in}}}^4+2\sigma_{x_{\text{in}}}^2+1-(r_{x_{\text{in}}}^l)^2\sigma_{x_{\text{in}}}^4)}{2\sigma_{x_{\text{in}}}^2(\sigma_{x_{\text{in}}}^2(1-(r_{x_{\text{in}}}^l)^2)+1)}\right) dy_{\text{in}_i}$$

$$= \frac{r_{x_{\text{in}}}^l}{2\pi\sigma_{x_{\text{in}}}(\sigma_{x_{\text{in}}}^2(1-(r_{x_{\text{in}}}^l)^2)+1)^{\frac{3}{2}}} \int_{-\infty}^{\infty} \frac{y_{\text{in}_i}^2}{\sqrt{2\pi}} \exp\left(\frac{-y_{\text{in}_i}^2((\sigma_{x_{\text{in}}}^2+1)^2-(r_{x_{\text{in}}}^l\sigma_{x_{\text{in}}}^2)^2)}{2\sigma_{x_{\text{in}}}^2(\sigma_{x_{\text{in}}}^2(1-(r_{x_{\text{in}}}^l)^2)+1)}\right) dy_{\text{in}_i}$$

$$= \frac{r_{x_{\text{in}}}^l}{2\pi\sigma_{x_{\text{in}}}(\sigma_{x_{\text{in}}}^2(1-(r_{x_{\text{in}}}^l)^2)+1)^{\frac{3}{2}}} \frac{\sigma_{x_{\text{in}}}\sqrt{(\sigma_{x_{\text{in}}}^2(1-(r_{x_{\text{in}}}^l)^2)+1)}}{\sqrt{(\sigma_{x_{\text{in}}}^2+1)^2-(r_{x_{\text{in}}}^l\sigma_{x_{\text{in}}}^2)^2}}$$

$$\int_{-\infty}^{\infty} \frac{y_{\text{in}_i}^2}{\sqrt{2\pi}\frac{\sigma_{x_{\text{in}}}\sqrt{(\sigma_{x_{\text{in}}}^2(1-(r_{x_{\text{in}}}^l)^2)+1)}}{\sqrt{(\sigma_{x_{\text{in}}}^2+1)^2-(r_{x_{\text{in}}}^l\sigma_{x_{\text{in}}}^2)^2}}} \exp\left(\frac{-y_{\text{in}_i}^2((\sigma_{x_{\text{in}}}^2+1)^2-(r_{x_{\text{in}}}^l\sigma_{x_{\text{in}}}^2)^2)}{2\sigma_{x_{\text{in}}}^2(\sigma_{x_{\text{in}}}^2(1-(r_{x_{\text{in}}}^l)^2)+1)}\right) dy_{\text{in}_i}$$

$$= \frac{r_{x_{\text{in}}}^l}{2\pi\sigma_{x_{\text{in}}}(\sigma_{x_{\text{in}}}^2(1-(r_{x_{\text{in}}}^l)^2)+1)^{\frac{3}{2}}} \frac{\sigma_{x_{\text{in}}}^3(\sigma_{x_{\text{in}}}^2(1-(r_{x_{\text{in}}}^l)^2)+1)^{\frac{3}{2}}}{((\sigma_{x_{\text{in}}}^2+1)^2-(r_{x_{\text{in}}}^l\sigma_{x_{\text{in}}}^2)^2)^{\frac{3}{2}}}$$

$$I_{3,3} = \frac{r_{x_{\text{in}}}^l\sigma_{x_{\text{in}}}^2}{2\pi((\sigma_{x_{\text{in}}}^2+1)^2-(r_{x_{\text{in}}}^l\sigma_{x_{\text{in}}}^2)^2)^{\frac{3}{2}}}$$

$$I = I_1 + I_2 + I_3$$
$$= I_{1,1} + I_{1,2} + I_{1,3} + I_{2,1} + I_{2,2} + I_{2,3} + I_{3,1} + I_{3,2} + I_{3,3}$$

$$I = \frac{1}{4} + \frac{1}{2\pi}\sin^{-1}\left(\frac{r_{x_{\text{in}}}^l\sigma_{x_{\text{in}}}^2}{\sigma_{x_{\text{in}}}^2+1}\right)+$$

$$\frac{2r_{x_{\text{in}}}^l\sigma_{x_{\text{in}}}^2}{2\pi(\sigma_{x_{\text{in}}}^2+1)\sqrt{(\sigma_{x_{\text{in}}}^2+1)^2-(r_{x_{\text{in}}}^l\sigma_{x_{\text{in}}}^2)^2}} + \frac{r_{x_{\text{in}}}^l\sigma_{x_{\text{in}}}^2}{2\pi((\sigma_{x_{\text{in}}}^2+1)^2-(r_{x_{\text{in}}}^l\sigma_{x_{\text{in}}}^2)^2)^{\frac{3}{2}}}$$

$$I = \frac{1}{4} + \frac{1}{2\pi}\sin^{-1}\left(\frac{r_{x_{\text{in}}}^l\sigma_{x_{\text{in}}}^2}{\sigma_{x_{\text{in}}}^2+1}\right) + \frac{r_{x_{\text{in}}}^l\sigma_{x_{\text{in}}}^2((2\sigma_{x_{\text{in}}}^2+3)(\sigma_{x_{\text{in}}}^2+1)-2(r_{x_{\text{in}}}^l\sigma_{x_{\text{in}}}^2)^2)}{2\pi(\sigma_{x_{\text{in}}}^2+1)((\sigma_{x_{\text{in}}}^2+1)^2-(r_{x_{\text{in}}}^l\sigma_{x_{\text{in}}}^2)^2)^{\frac{3}{2}}}$$

We defined $\text{Cov}(g_{\text{in}_{x_i}}, g_{\text{in}_{y_i}})$, as

$$\text{Cov}(g_{\text{in}_{x_i}}, g_{\text{in}_{y_i}}) = I r_{g_{\text{out}}}^l \sigma_{g_{\text{out}}}^2$$

$$\boxed{\begin{aligned}
&\text{Cov}(g_{\text{in}_{x_i}}, g_{\text{in}_{y_i}}) = \\
&\left[\frac{1}{4} + \frac{1}{2\pi}\sin^{-1}\left(\frac{r_{x_{\text{in}}}^l\sigma_{x_{\text{in}}}^2}{\sigma_{x_{\text{in}}}^2+1}\right) + \frac{r_{x_{\text{in}}}^l\sigma_{x_{\text{in}}}^2((2\sigma_{x_{\text{in}}}^2+3)(\sigma_{x_{\text{in}}}^2+1)-2(r_{x_{\text{in}}}^l\sigma_{x_{\text{in}}}^2)^2)}{2\pi(\sigma_{x_{\text{in}}}^2+1)((\sigma_{x_{\text{in}}}^2+1)^2-(r_{x_{\text{in}}}^l\sigma_{x_{\text{in}}}^2)^2)^{\frac{3}{2}}}\right] r_{g_{\text{out}}}^l \sigma_{g_{\text{out}}}^2
\end{aligned}}$$

## C.6 LAYERNORM

For an input $\mathbf{x}_{\text{in}}$ the forward pass of LayerNorm is,

$$\mathbf{x}_{\text{out}} = \text{LayerNorm}(\mathbf{x}_{\text{in}})$$
$$\implies x_{\text{out}_i} = \frac{x_{\text{in}_i} - \bar{x}_{\text{in}}}{\hat{\sigma}_{x_{\text{in}}}}$$

Where

$$\bar{x}_{\text{in}} = \frac{\sum_{i=1}^{d_{\text{in}}} x_{\text{in}_i}}{d_{\text{in}}}$$

$$\hat{\sigma}_{x_{\text{in}}} = \sqrt{\frac{\sum_{i=1}^{d_{\text{in}}}(x_{\text{in}_i} - \bar{x}_{\text{in}})^2}{d_{\text{in}}}}$$

To get expectation of output of LayerNorm,

$$\mathbb{E}[x_{\text{out}_i}] = \mathbb{E}[\frac{x_{\text{in}_i} - \bar{x}_{\text{in}}}{\hat{\sigma}_{x_{\text{in}}}}]$$

$$\sum_{i=1}^{d_{\text{in}}}\mathbb{E}[x_{\text{out}_i}] = \sum_{i=1}^{d_{\text{in}}}\mathbb{E}[\frac{x_{\text{in}_i} - \bar{x}_{\text{in}}}{\hat{\sigma}_{x_{\text{in}}}}]$$

$$= \mathbb{E}[\sum_{i=1}^{d_{\text{in}}}\frac{x_{\text{in}_i} - \bar{x}_{\text{in}}}{\hat{\sigma}_{x_{\text{in}}}}]$$

$$= \mathbb{E}[\frac{\sum_{i=1}^{d_{\text{in}}}(x_{\text{in}_i} - \bar{x}_{\text{in}})}{\hat{\sigma}_{x_{\text{in}}}}]$$

$$\sum_{i=1}^{d_{\text{in}}}\mathbb{E}[x_{\text{out}_i}] = 0$$

By symmetry for any $i, j$ and $i \neq j$ we have $\mathbb{E}[x_{\text{out}_i}] = \mathbb{E}[x_{\text{out}_j}] = \mu_{x_{\text{out}}}$

$$\implies d_{\text{in}}\mu_{x_{\text{out}}} = 0$$

$$\boxed{\mu_{x_{\text{out}}} = 0}$$

Similarly we calculate variance of output by,

$$\text{Var}(x_{\text{out}_i}) = \mathbb{E}[x_{\text{out}_i}^2] - \mathbb{E}[x_{\text{out}_i}]^2 = \mathbb{E}[x_{\text{out}_i}^2]$$

$$\mathbb{E}[x_{\text{out}_i}^2] = \mathbb{E}[\frac{(x_{\text{in}_i} - \bar{x}_{\text{in}})^2}{\hat{\sigma}_{x_{\text{in}}}^2}]$$

$$\sum_{i=1}^{d_{\text{in}}}\mathbb{E}[x_{\text{out}_i}^2] = \sum_{i=1}^{d_{\text{in}}}\mathbb{E}[\frac{(x_{\text{in}_i} - \bar{x}_{\text{in}})^2}{\hat{\sigma}_{x_{\text{in}}}^2}]$$

$$= \mathbb{E}[\sum_{i=1}^{d_{\text{in}}}\frac{(x_{\text{in}_i} - \bar{x}_{\text{in}})^2}{\hat{\sigma}_{x_{\text{in}}}^2}]$$

$$= \mathbb{E}[\frac{\sum_{i=1}^{d_{\text{in}}}(x_{\text{in}_i} - \bar{x}_{\text{in}})^2}{\hat{\sigma}_{x_{\text{in}}}^2}]$$

$$\sum_{i=1}^{d_{\text{in}}}\mathbb{E}[x_{\text{out}_i}^2] = d_{\text{in}}$$

By symmetry for any $i, j$ and $i \neq j$ we have $\mathbb{E}[x_{\text{out}_i}^2] = \mathbb{E}[x_{\text{out}_j}^2] = \sigma_{x_{\text{out}}}^2$

$$\implies d_{\text{in}}\sigma_{x_{\text{out}}}^2 = d_{\text{in}}$$

$$\boxed{\sigma_{x_{\text{out}}}^2 = 1}$$

Now we have $\hat{\sigma}_{x_{\text{in}}} \xrightarrow{a.s} \sigma_{x_{\text{in}}}$ for large $d_{\text{in}}$. So for large values of $d_{\text{in}}$ we can treat $\hat{\sigma}_{x_{\text{in}}}$ as a constant which has value $\sigma_{x_{\text{in}}}$. We use this approximation to get the following results. For two inputs $\mathbf{x}_{\text{in}}$ and $\mathbf{y}_{\text{in}}$ such that for all $i$, $\text{Corr}(x_{\text{in}_i}, y_{\text{in}_i}) = r_{x_{\text{in}}}^l$. For all $j$ we have,

$$\text{Corr}(x_{\text{out}_j}, y_{\text{out}_j}) = \frac{\mathbb{E}[x_{\text{out}_j}y_{\text{out}_j}] - \mathbb{E}[x_{\text{out}_j}]\mathbb{E}[y_{\text{out}_j}]}{\sqrt{\text{Var}(x_{\text{out}_j})\text{Var}(y_{\text{out}_j})}}$$

$$= \frac{\mathbb{E}[x_{\text{out}_j}y_{\text{out}_j}] - \mu_{x_{\text{out}}}\mu_{x_{\text{out}}}}{\sqrt{\sigma_{x_{\text{out}}}^2 \sigma_{x_{\text{out}}}^2}}$$

$$= \frac{\mathbb{E}[x_{\text{out}_j} y_{\text{out}_j}] - 0}{\sqrt{1}}$$

$$= \mathbb{E}[x_{\text{out}_j} y_{\text{out}_j}]$$

$$= \mathbb{E}\left[\frac{(x_{\text{in}_j} - \bar{x}_{\text{in}})(y_{\text{in}_j} - \bar{y}_{\text{in}})}{\hat{\sigma}_{x_{\text{in}}} \hat{\sigma}_{y_{\text{in}}}}\right]$$

$$\approx \mathbb{E}\left[\frac{(x_{\text{in}_j} - \bar{x}_{\text{in}})(y_{\text{in}_j} - \bar{y}_{\text{in}})}{\sigma_{x_{\text{in}}} \sigma_{x_{\text{in}}}}\right]$$

$$= \frac{\mathbb{E}[(x_{\text{in}_j} - \bar{x}_{\text{in}})(y_{\text{in}_j} - \bar{y}_{\text{in}})]}{\sigma^2_{x_{\text{in}}}}$$

$$= \frac{\mathbb{E}\left[\left(x_{\text{in}_j} - \frac{\sum_{k=1}^{d_{\text{in}}} x_{\text{in}_k}}{d_{\text{in}}}\right)\left(y_{\text{in}_j} - \frac{\sum_{l=1}^{d_{\text{in}}} y_{\text{in}_l}}{d_{\text{in}}}\right)\right]}{\sigma^2_{x_{\text{in}}}}$$

$$= \frac{\mathbb{E}\left[x_{\text{in}_j} y_{\text{in}_j} - y_{\text{in}_j} \frac{\sum_{k=1}^{d_{\text{in}}} x_{\text{in}_k}}{d_{\text{in}}} - x_{\text{in}_j} \frac{\sum_{l=1}^{d_{\text{in}}} y_{\text{in}_l}}{d_{\text{in}}} + \frac{\sum_{k=1}^{d_{\text{in}}} x_{\text{in}_k}}{d_{\text{in}}} \frac{\sum_{l=1}^{d_{\text{in}}} y_{\text{in}_l}}{d_{\text{in}}}\right]}{\sigma^2_{x_{\text{in}}}}$$

Elements belonging to different dimensions from $\mathbf{x}_{\text{in}}$ and $\mathbf{y}_{\text{in}}$ are independent of each other and hence for $i, j$ and $i \neq j$ we have $\mathbb{E}[x_{\text{in}_i} y_{\text{in}_j}] = \mu^2_{x_{\text{in}}}$.

$$= \frac{\mathbb{E}[x_{\text{in}_j} y_{\text{in}_j}] - \mathbb{E}\left[y_{\text{in}_j} \frac{\sum_{k=1}^{d_{\text{in}}} x_{\text{in}_k}}{d_{\text{in}}}\right] - \mathbb{E}\left[x_{\text{in}_j} \frac{\sum_{l=1}^{d_{\text{in}}} y_{\text{in}_l}}{d_{\text{in}}}\right] + \mathbb{E}\left[\frac{\sum_{k=1}^{d_{\text{in}}} x_{\text{in}_k}}{d_{\text{in}}} \frac{\sum_{l=1}^{d_{\text{in}}} y_{\text{in}_l}}{d_{\text{in}}}\right]}{\sigma^2_{x_{\text{in}}}}$$

$$= \frac{r^l_{x_{\text{in}}} \sigma^2_{x_{\text{in}}} + \mu^2_{x_{\text{in}}} - \frac{r^l_{x_{\text{in}}} \sigma^2_{x_{\text{in}}} + d_{\text{in}} \mu^2_{x_{\text{in}}}}{d_{\text{in}}} - \frac{r^l_{x_{\text{in}}} \sigma^2_{x_{\text{in}}} + d_{\text{in}} \mu^2_{x_{\text{in}}}}{d_{\text{in}}} + \frac{r^l_{x_{\text{in}}} d_{\text{in}} \sigma^2_{x_{\text{in}}} + d^2_{\text{in}} \mu^2_{x_{\text{in}}}}{d^2_{\text{in}}}}{\sigma^2_{x_{\text{in}}}}$$

$$= \frac{r^l_{x_{\text{in}}} \sigma^2_{x_{\text{in}}} (1 - \frac{1}{d_{\text{in}}})}{\sigma^2_{x_{\text{in}}}}$$

$$\boxed{\text{Corr}(x_{\text{out}_j}, y_{\text{out}_j}) = r^l_{x_{\text{in}}} (1 - \frac{1}{d_{\text{in}}}) \approx r^l_{x_{\text{in}}} = r^l_{x_{\text{out}}}}$$

From Xu et al. (2019) (Eq. 17), the backward pass through LayerNorm is,

$$\mathbf{g}_{\text{in}} = \frac{\mathbf{g}_{\text{out}}}{\hat{\sigma}_{x_{\text{in}}}} \left(\mathbf{I_{d_{in}}} - \frac{\mathbf{1_{d_{in}}^T 1_{d_{in}}} + \mathbf{x_{out}^T x_{out}}}{d_{\text{in}}}\right)$$

$$\approx \frac{\mathbf{g}_{\text{out}}}{\sigma_{x_{\text{in}}}} \left(\mathbf{I_{d_{in}}} - \frac{\mathbf{1_{d_{in}}^T 1_{d_{in}}} + \mathbf{x_{out}^T x_{out}}}{d_{\text{in}}}\right)$$

We have $\lim\limits_{d_{\text{in}} \to \infty} \frac{\mathbf{1_{d_{in}}^T 1_{d_{in}}} + \mathbf{x_{out}^T x_{out}}}{d_{\text{in}}} = \mathbf{O_{d_{in}, d_{in}}}$ where $\mathbf{O_{d_{in}, d_{in}}}$ is zero matrix with shape $d_{\text{in}} \times d_{\text{in}}$

$$\mathbf{g}_{\text{in}} \approx \frac{\mathbf{g}_{\text{out}}}{\sigma_{x_{\text{in}}}} (\mathbf{I_{d_{in}}})$$

$$= \frac{\mathbf{g}_{\text{out}}}{\sigma_{x_{\text{in}}}}$$

$$\implies g_{\text{in}_i} = \frac{g_{\text{out}_i}}{\sigma_{x_{\text{in}}}}$$

If $\mu_{g_{\text{out}}} = 0$,

$$\boxed{\begin{aligned} \mu_{g_{\text{in}}} &= 0 \\ \sigma^2_{g_{\text{in}}} &= \frac{\sigma^2_{g_{\text{out}}}}{\sigma^2_{x_{\text{in}}}} \end{aligned}}$$

## C.7 SOFTMAX

**Assumption**: Other than assuming normally distributed inputs, we also assume that L is large $L >> 1$ to derive softmax variance.

The forward pass of Softmax can be defined as

$$\mathbf{x}_{\text{out}} = \text{Softmax}(\mathbf{x}_{\text{in}})$$

$$x_{\text{out}_i} = \frac{e^{x_{\text{in}_i}}}{\sum_{j=1}^{L} e^{x_{\text{in}_j}}}$$

For calculating mean we can easily see that,

$$\sum_{i=1}^{L} x_{\text{out}_i} = 1$$

Taking expectation both sides, we get

$$\mathbb{E}[\sum_{i=1}^{L} x_{\text{out}_i}] = 1$$

$$\sum_{i=1}^{L} \mathbb{E}[x_{\text{out}_i}] = 1$$

By symmetry we can assume that for any $i, j, i \neq j$, we have $\mathbb{E}[x_{\text{out}_i}] = \mathbb{E}[x_{\text{out}_j}]$

$$L\mathbb{E}[x_{\text{out}_i}] = 1$$

$$\boxed{\mu_{x_{\text{out}}} = \frac{1}{L}}$$

Let us define $z = \sum_j e^{y_j}$ where $y_j = x_j - x_i$ is normally distributed $\mathcal{N}(0, \sigma_j)$. Hence, each $e^{y_j}$ is log-normally distributed, and $z$ is a sum of correlated log-normals. Following (Lo, 2013), this sum of log-normals can be approximated as another log-normal random variable, $Log\mathcal{N}(\mu_z, \sigma_z)$, where $\mu_z$ and $\sigma_z$ are as follows -

$$S_+ = E[\sum_j y_j] = \sum_j e^{\frac{\sigma_j^2}{2}}$$

$$\sigma_z^2 = \frac{1}{S_+^2} \sum_{j,k} corr_{j,k} \sigma_j \sigma_k e^{\frac{1}{2}(\sigma_j^2 + \sigma_k^2)}$$

$$\mu_z = ln(S_+) - \frac{\sigma_z^2}{2}$$

Since the difference of two normals $x_j$ and $x_i$ is also normal, from the M.G.F. of normal distribution, we have $\sigma_j^2 = 2\sigma_{x_{\text{in}}}^2(1 - r_{x_{\text{in}}})$ if $j \neq i$, and $\sigma_j^2 = 0$ if $j = i$.

Also, $corr_{j,k} = 0$ if $j = i$ or $k = i$, else $corr_{j,k} = \frac{1}{2}$.

We can substitute these values in the above equations, to get

$$S_+ = (L-1)e^{\sigma_{x_{\text{in}}}^2(1 - r_{x_{\text{in}}})} + 1$$

$$\sigma_z^2 = \sigma_{x_{\text{in}}}^2(1 - r_{x_{\text{in}}})\frac{L}{L-1}$$

$$\mu_z = ln(S_+) - \frac{\sigma_z^2}{2}$$

Since $z$ is log-normal, $x_{\text{out}} = \frac{1}{z}$ is also log-normal with $Log\mathcal{N}(-\mu_z, \sigma_z)$. The variance of log-normal distribution can be obtained from standard formulae for log-normal distribution as $(e^{\sigma_z^2} - 1)e^{\sigma_z^2 - 2\mu_z}$.

Substituting the values of $\mu_z$ and $\sigma_z$ from above, we get

$$\sigma_{x_{\text{out}}}^2 = \frac{(e^{\sigma_z^2} - 1)e^{2*\sigma_z^2}}{S_+^2}$$
$$= \frac{(e^{\sigma_{x_{\text{in}}}^2(1-r_{x_{\text{in}}})\frac{L}{L-1}} - 1)e^{2\sigma_{x_{\text{in}}}^2(1-r_{x_{\text{in}}})\frac{L}{L-1}}}{((L-1)e^{\sigma_{x_{\text{in}}}^2(1-r_{x_{\text{in}}})} + 1)^2}$$

For large L, we can ignore the 1 in the denominator -

$$\sigma_{x_{\text{out}}}^2 = \frac{(e^{\sigma_{x_{\text{in}}}^2(1-r_{x_{\text{in}}})\frac{L}{L-1}} - 1)}{(L-1)^2}$$

If $L >> 1$ and $\sigma_{x_{\text{in}}}^2$ is small, we get the more simplified formula as -

$$\boxed{\sigma_{x_{\text{out}}}^2 \approx \frac{(e^{(1-r_{x_{\text{in}}}^d)\sigma_{x_{\text{in}}}^2} - 1)}{L^2}} \qquad \text{(Assuming } L >> 1)$$

Using the mean and variances, we can calculate the scale of softmax output as follows-

$$E[x_{\text{out}}^2] = \sigma_{x_{\text{out}}}^2 + \mu_{x_{\text{out}}}^2$$
$$= \frac{(e^{(1-r_{x_{\text{in}}}^d)\sigma_{x_{\text{in}}}^2})}{L^2}$$

The Jacobian of Softmax can be calculated as ((Kim et al., 2021)):

$$J_{i,j} = \begin{cases} x_{\text{out}_i}(1 - x_{\text{out}_i}) & \text{if } i = j \\ -x_{\text{out}_i}x_{\text{out}_j} & \text{else} \end{cases}$$

For large values of $L$ this approximately becomes

$$\mathbf{J} \approx \text{diag}(\mathbf{x}_{\text{out}})$$
$$\mathbf{g}_{\text{in}} = \mathbf{g}_{\text{out}}\mathbf{J}$$
$$g_{\text{in}_i} \approx g_{\text{out}_i}x_{\text{out}_i}$$
$$\mathbb{E}[g_{\text{in}_i}] \approx \mathbb{E}[g_{\text{out}_i}x_{\text{out}_i}]$$
$$= \mathbb{E}[g_{\text{out}_i}]\mathbb{E}[x_{\text{out}_i}] = 0 = \mu_{g_{\text{in}}}$$
$$\mathbb{E}[g_{\text{in}_i}^2] \approx \mathbb{E}[g_{\text{out}_i}^2x_{\text{out}_i}^2]$$
$$= \mathbb{E}[g_{\text{out}_i}^2]\mathbb{E}[x_{\text{out}_i}^2]$$
$$\boxed{\sigma_{g_{\text{in}}}^2 = \sigma_{g_{\text{out}}}^2\frac{(e^{(1-r_{x_{\text{in}}}^d)\sigma_{x_{\text{in}}}^2})}{L^2}}$$

### C.8 SCALED DOT-PRODUCT ATTENTION

**Assumption**: We assume the numerator and denominator of the scaled dot product attention to be independent. These approximations hold true if the denominator has a low variance. The resulting formulae are fairly accurate, as shown in the numerical verification section.

The forward pass of Scaled Dot-Product Attention is

$$\mathbf{X}_{\text{out}} = \text{Dropout}(\text{SoftMax}(\frac{\mathbf{Q}\mathbf{K}^{\mathbf{T}}}{\sqrt{d_k}}))\mathbf{V}$$

Where,

$$\mathbf{Q} = \mathbf{X}_{\text{in}}\mathbf{W}_{\mathbf{Q}}$$
$$\mathbf{K} = \mathbf{X}_{\text{in}}\mathbf{W}_{\mathbf{K}}$$
$$\mathbf{V} = \mathbf{X}_{\text{in}}\mathbf{W}_{\mathbf{V}}$$

$$\mathbf{X}_{\text{out}} = \text{Dropout}(\text{SoftMax}(\frac{\mathbf{X}_{\text{in}}\mathbf{W}_{\mathbf{Q}}\mathbf{W}_{\mathbf{K}}^{\mathbf{T}}\mathbf{X}_{\text{in}}^{\mathbf{T}}}{\sqrt{d_k}}))\mathbf{X}_{\text{in}}\mathbf{W}_{\mathbf{V}}$$

Let,

$$\mathbf{O} = \text{Dropout}(\text{SoftMax}(\frac{\mathbf{X}_{\text{in}}\mathbf{W}_{\mathbf{Q}}\mathbf{W}_{\mathbf{K}}^{\mathbf{T}}\mathbf{X}_{\text{in}}^{\mathbf{T}}}{\sqrt{d_k}}))\mathbf{X}_{\text{in}}$$

$$\mathbf{W} = \frac{\mathbf{X}_{\text{in}}\mathbf{W}_{\mathbf{Q}}\mathbf{W}_{\mathbf{K}}^{\mathbf{T}}}{\sqrt{d_k}}$$

$$\mathbf{O} = \text{Dropout}(\text{SoftMax}(\mathbf{W}\mathbf{X}_{\text{in}}^{\mathbf{T}}))\mathbf{X}_{\text{in}}$$

Using results from Linear Layer we have $\sigma_w^2 = d_{\text{in}}\sigma_{x_{\text{in}}}^2\sigma_q^2\sigma_k^2 = d_{\text{in}}\sigma_{x_{\text{in}}}^2\sigma_{qk}^2$

$$
\begin{aligned}
O_{i,j} &= \sum_{k=1}^{L} \text{Dropout}(\text{SoftMax}(\mathbf{W}\mathbf{X}_{\text{in}}^{\mathbf{T}}))_{i,k} X_{\text{in}_{k,j}} \\
&= \sum_{k=1}^{L} \text{Dropout}(\frac{\exp\left((\mathbf{W}\mathbf{X}_{\text{in}}^{\mathbf{T}})_{i,k}\right)}{\sum_{m=1}^{L} \exp\left((\mathbf{W}\mathbf{X}_{\text{in}}^{\mathbf{T}})_{i,m}\right)}) X_{\text{in}_{k,j}} \\
&= \sum_{k=1}^{L} \frac{\text{Dropout}(\exp\left((\mathbf{W}\mathbf{X}_{\text{in}}^{\mathbf{T}})_{i,k}\right))}{\sum_{m=1}^{L} \exp\left((\mathbf{W}\mathbf{X}_{\text{in}}^{\mathbf{T}})_{i,m}\right)} X_{\text{in}_{k,j}} \\
&= \frac{\sum_{k=1}^{L} \text{Dropout}(\exp\left((\mathbf{W}\mathbf{X}_{\text{in}}^{\mathbf{T}})_{i,k}\right)) X_{\text{in}_{k,j}}}{\sum_{m=1}^{L} \exp\left((\mathbf{W}\mathbf{X}_{\text{in}}^{\mathbf{T}})_{i,m}\right)} \\
&= \frac{\sum_{k=1}^{L} \text{Dropout}(\exp\left(\sum_{l=1}^{d_{\text{in}}} W_{i,l} X_{\text{in}_{k,l}}\right)) X_{\text{in}_{k,j}}}{\sum_{m=1}^{L} \exp\left(\sum_{n=1}^{d_{\text{in}}} W_{i,n} X_{\text{in}_{m,n}}\right)} \\
&= \frac{\sum_{k=1}^{L} \text{Dropout}(\exp\left(\sum_{l=1}^{d_{\text{in}}} W_{i,l} X_{\text{in}_{k,l}}\right) X_{\text{in}_{k,j}})}{\sum_{m=1}^{L} \exp\left(\sum_{n=1}^{d_{\text{in}}} W_{i,n} X_{\text{in}_{m,n}}\right)}
\end{aligned}
$$

Let,

$$\text{Num}(O_{i,j}) = \sum_{k=1}^{L} \text{Dropout}(\exp\big(\sum_{l=1}^{d_\text{in}} W_{i,l} X_{\text{in}_{k,l}}\big) X_{\text{in}_{k,j}})$$

$$\text{Den}(O_{i,j}) = \sum_{m=1}^{L} \exp\big(\sum_{n=1}^{d_\text{in}} W_{i,n} X_{\text{in}_{m,n}}\big)$$

To get the expectation and variance of $O_{i,j}$ we make the following assumptions, that we found to be reasonably accurate for large sequence lengths. This approximation is reasonable as long as the mean of the denominator elements are much larger than their variance, and the correlation is small between numerator and denominator, which is true for Large sequence lengths L:

$$\mathbb{E}[O_{i,j}] \approx \frac{\mathbb{E}[\text{Num}(O_{i,j})]}{\mathbb{E}[\text{Den}(O_{i,j})]}$$

$$\mathbb{E}[O_{i,j}^2] \approx \frac{\mathbb{E}[(\text{Num}(O_{i,j}))^2]}{\mathbb{E}[(\text{Den}(O_{i,j}))^2]}$$

Then to get expectation we have,

$$\mathbb{E}[\text{Num}(O_{i,j})] = \mathbb{E}[\sum_{k=1}^{L} \text{Dropout}(\exp\big(\sum_{l=1}^{d_\text{in}} W_{i,l} X_{\text{in}_{k,l}}\big) X_{\text{in}_{k,j}})]$$

$$= \sum_{k=1}^{L} \mathbb{E}[\text{Dropout}(\exp\big(\sum_{l=1}^{d_\text{in}} W_{i,l} X_{\text{in}_{k,l}}\big) X_{\text{in}_{k,j}})]$$

$$= \sum_{k=1}^{L} \mathbb{E}[\exp\big(\sum_{l=1}^{d_\text{in}} W_{i,l} X_{\text{in}_{k,l}}\big) X_{\text{in}_{k,j}}]$$

(Dropout doesn't change expectation)

$$\mathbb{E}[\exp\big(\sum_{l=1}^{d_\text{in}} W_{i,l} X_{\text{in}_{k,l}}\big) X_{\text{in}_{k,j}}] = \mathbb{E}[(\exp\big(W_{i,j} X_{\text{in}_{k,j}}\big) X_{\text{in}_{k,j}}) \prod_{l=1,l\neq j}^{d_\text{in}} \exp\big(W_{i,l} X_{\text{in}_{k,l}}\big)]$$

As weights are initialized independently,

$$\mathbb{E}[\exp\big(\sum_{l=1}^{d_\text{in}} W_{i,l} X_{\text{in}_{k,l}}\big) X_{\text{in}_{k,j}}] = \mathbb{E}[\exp\big(W_{i,j} X_{\text{in}_{k,j}}\big) X_{\text{in}_{k,j}}] \prod_{l=1,l\neq j}^{d_\text{in}} \mathbb{E}[\exp\big(W_{i,l} X_{\text{in}_{k,l}}\big)]$$

$$\mathbb{E}[\exp\big(W_{i,j} X_{\text{in}_{k,j}}\big) X_{\text{in}_{k,j}}] =$$

$$\int_{-\infty}^{\infty} \frac{X_{\text{in}_{k,j}}}{\sqrt{2\pi}\sigma_{x_\text{in}}} \exp\big(\frac{-X_{\text{in}_{k,j}}^2}{2\sigma_{x_\text{in}}^2}\big) dX_{\text{in}_{k,j}} \int_{-\infty}^{\infty} \frac{\exp\big(W_{i,j} X_{\text{in}_{k,j}}\big)}{\sqrt{2\pi}\sigma_w} \exp\big(\frac{-W_{i,j}^2}{2\sigma_w^2}\big) dW_{i,j}$$

$$= \int_{-\infty}^{\infty} \frac{X_{\text{in}_{k,j}}}{\sqrt{2\pi}\sigma_{x_\text{in}}} \exp\big(\frac{-X_{\text{in}_{k,j}}^2}{2\sigma_{x_\text{in}}^2}\big) \exp\big(\frac{X_{\text{in}_{k,j}}^2 \sigma_w^2}{2}\big) dX_{\text{in}_{k,j}}$$

(Using MGF of Normal Distribution)

$$\mathbb{E}[\exp\big(W_{i,j} X_{\text{in}_{k,j}}\big) X_{\text{in}_{k,j}}] = 0 \qquad \text{(Integral of an Odd function from } -\infty \text{ to } \infty)$$

$$\mathbb{E}[\exp\big(\sum_{l=1}^{d_\text{in}} W_{i,l} X_{\text{in}_{k,l}}\big) X_{\text{in}_{k,j}}] = 0$$

$$\mathbb{E}[\text{Num}(O_{i,j})] = 0$$

$$\mathbb{E}[O_{i,j}] = 0 = \mu_{x_\text{out}}$$

Similarly for variance (Drop signifies Dropout),

$$\mathbb{E}[(\text{Num}(O_{i,j}))^2] = \mathbb{E}[(\sum_{k=1}^{L} \text{Drop}(\exp\big(\sum_{l=1}^{d_\text{in}} W_{i,l} X_{\text{in}_{k,l}}\big) X_{\text{in}_{k,j}}))^2]$$

$$= \mathbb{E}[\sum_{k=1}^{L}(\text{Drop}(\exp{(\sum_{l=1}^{d_{\text{in}}}W_{i,l}X_{\text{in}_{k,l}})X_{\text{in}_{k,j}}))^2]+$$

$$\mathbb{E}[\sum_{k=1}^{L}\sum_{m=1,m\neq k}^{L}(\text{Drop}(\exp{(\sum_{l=1}^{d_{\text{in}}}W_{i,l}X_{\text{in}_{k,l}})X_{\text{in}_{k,j}}))(\text{Drop}(\exp{(\sum_{l=1}^{d_{\text{in}}}W_{i,l}X_{\text{in}_{m,l}})X_{\text{in}_{m,j}}))]$$

$$= \sum_{k=1}^{L}\mathbb{E}[(\text{Drop}(\exp{(\sum_{l=1}^{d_{\text{in}}}W_{i,l}X_{\text{in}_{k,l}})X_{\text{in}_{k,j}}))^2]+$$

$$\sum_{k=1}^{L}\sum_{m=1,m\neq k}^{L}\mathbb{E}[(\text{Drop}(\exp{(\sum_{l=1}^{d_{\text{in}}}W_{i,l}X_{\text{in}_{k,l}})X_{\text{in}_{k,j}}))(\text{Drop}(\exp{(\sum_{l=1}^{d_{\text{in}}}W_{i,l}X_{\text{in}_{m,l}})X_{\text{in}_{m,j}}))]$$

$$= \sum_{k=1}^{L}(1-p)\mathbb{E}[\frac{(\exp{(\sum_{l=1}^{d_{\text{in}}}W_{i,l}X_{\text{in}_{k,l}})X_{\text{in}_{k,j}})^2}{(1-p)^2}]+$$

$$\sum_{k=1}^{L}\sum_{m=1,m\neq k}^{L}(1-p)^2\mathbb{E}[\frac{(\exp{(\sum_{l=1}^{d_{\text{in}}}W_{i,l}X_{\text{in}_{k,l}})X_{\text{in}_{k,j}})(\exp{(\sum_{l=1}^{d_{\text{in}}}W_{i,l}X_{\text{in}_{m,l}})X_{\text{in}_{m,j}})}{(1-p)^2}]$$

$$= \sum_{k=1}^{L}\frac{1}{(1-p)}\mathbb{E}[(\exp{(\sum_{l=1}^{d_{\text{in}}}W_{i,l}X_{\text{in}_{k,l}})X_{\text{in}_{k,j}})^2]+$$

$$\sum_{k=1}^{L}\sum_{m=1,m\neq k}^{L}\mathbb{E}[(\exp{(\sum_{l=1}^{d_{\text{in}}}W_{i,l}X_{\text{in}_{k,l}})X_{\text{in}_{k,j}})\exp{(\sum_{l=1}^{d_{\text{in}}}W_{i,l}X_{\text{in}_{m,l}})X_{\text{in}_{m,j}}]$$

$$= \sum_{k=1}^{L}\frac{1}{(1-p)}\mathbb{E}[\exp{(\sum_{l=1}^{d_{\text{in}}}2W_{i,l}X_{\text{in}_{k,l}})X_{\text{in}_{k,j}}^2]+$$

$$\sum_{k=1}^{L}\sum_{m=1,m\neq k}^{L}\mathbb{E}[\exp{(\sum_{l=1}^{d_{\text{in}}}W_{i,l}(X_{\text{in}_{k,l}}+X_{\text{in}_{m,l}}))X_{\text{in}_{k,j}}X_{\text{in}_{m,j}}]$$

$$\mathbb{E}[\exp{(\sum_{l=1}^{d_{\text{in}}}2W_{i,l}X_{\text{in}_{k,l}})X_{\text{in}_{k,j}}^2] = \mathbb{E}[(\exp{(2W_{i,j}X_{\text{in}_{k,j}})X_{\text{in}_{k,j}}^2})\prod_{l=1,l\neq j}^{d_{\text{in}}}\exp{(2W_{i,l}X_{\text{in}_{k,l}})}]$$

$$= \mathbb{E}[\exp{(2W_{i,j}X_{\text{in}_{k,j}})X_{\text{in}_{k,j}}^2}]\prod_{l=1,l\neq j}^{d_{\text{in}}}\mathbb{E}[\exp{(2W_{i,l}X_{\text{in}_{k,l}})}]$$

$$\mathbb{E}[\exp{(2W_{i,j}X_{\text{in}_{k,j}})X_{\text{in}_{k,j}}^2}] =$$

$$\int_{-\infty}^{\infty}\frac{X_{\text{in}_{k,j}}^2}{\sqrt{2\pi}\sigma_{x_{\text{in}}}}\exp{(\frac{-X_{\text{in}_{k,j}}^2}{2\sigma_{x_{\text{in}}}^2})}dX_{\text{in}_{k,j}}\int_{-\infty}^{\infty}\frac{\exp{(2W_{i,j}X_{\text{in}_{k,j}})}}{\sqrt{2\pi}\sigma_{w}}\exp{(\frac{-W_{i,j}^2}{2\sigma_{w}^2})}dW_{i,j}$$

$$= \int_{-\infty}^{\infty}\frac{X_{\text{in}_{k,j}}^2}{\sqrt{2\pi}\sigma_{x_{\text{in}}}}\exp{(\frac{-X_{\text{in}_{k,j}}^2}{2\sigma_{x_{\text{in}}}^2})}\exp{(2X_{\text{in}_{k,j}}^2\sigma_{w}^2)}dX_{\text{in}_{k,j}} \quad \text{(Using MGF of Normal Distribution)}$$

$$= \int_{-\infty}^{\infty}\frac{X_{\text{in}_{k,j}}^2}{\sqrt{2\pi}\sigma_{x_{\text{in}}}}\exp{(\frac{-X_{\text{in}_{k,j}}^2(1-4\sigma_{x_{\text{in}}}^2\sigma_{w}^2)}{2\sigma_{x_{\text{in}}}^2})}dX_{\text{in}_{k,j}}$$

$$= \int_{-\infty}^{\infty}\frac{X_{\text{in}_{k,j}}^2}{\sqrt{2\pi}\frac{\sigma_{x_{\text{in}}}}{\sqrt{(1-4\sigma_{x_{\text{in}}}^2\sigma_{w}^2)}}\sqrt{(1-4\sigma_{x_{\text{in}}}^2\sigma_{w}^2)}}\exp{(\frac{-X_{\text{in}_{k,j}}^2(1-4\sigma_{x_{\text{in}}}^2\sigma_{w}^2)}{2\sigma_{x_{\text{in}}}^2})}dX_{\text{in}_{k,j}}$$

$$= \frac{1}{\sqrt{(1 - 4\sigma_{x_{\text{in}}}^2 \sigma_w^2)}} \int_{-\infty}^{\infty} \frac{X_{\text{in}_{k,j}}^2}{\sqrt{2\pi} \frac{\sigma_{x_{\text{in}}}}{\sqrt{(1 - 4\sigma_{x_{\text{in}}}^2 \sigma_w^2)}}} \exp\left(\frac{-X_{\text{in}_{k,j}}^2 (1 - 4\sigma_{x_{\text{in}}}^2 \sigma_w^2)}{2\sigma_{x_{\text{in}}}^2}\right) dX_{\text{in}_{k,j}}$$

$$= \frac{1}{\sqrt{(1 - 4\sigma_{x_{\text{in}}}^2 \sigma_w^2)}} \frac{\sigma_{x_{\text{in}}}^2}{(1 - 4\sigma_{x_{\text{in}}}^2 \sigma_w^2)}$$

$$= \frac{\sigma_{x_{\text{in}}}^2}{(1 - 4\sigma_{x_{\text{in}}}^2 \sigma_w^2)^{\frac{3}{2}}}$$

$$\mathbb{E}[\exp(2W_{i,l} X_{\text{in}_{k,l}})] =$$

$$\int_{-\infty}^{\infty} \frac{1}{\sqrt{2\pi}\sigma_{x_{\text{in}}}} \exp\left(\frac{-X_{\text{in}_{k,j}}^2}{2\sigma_{x_{\text{in}}}^2}\right) dX_{\text{in}_{k,j}} \int_{-\infty}^{\infty} \frac{\exp(2W_{i,j} X_{\text{in}_{k,j}})}{\sqrt{2\pi}\sigma_w} \exp\left(\frac{-W_{i,j}^2}{2\sigma_w^2}\right) dW_{i,j}$$

$$= \int_{-\infty}^{\infty} \frac{1}{\sqrt{2\pi}\sigma_{x_{\text{in}}}} \exp\left(\frac{-X_{\text{in}_{k,j}}^2}{2\sigma_{x_{\text{in}}}^2}\right) \exp(2X_{\text{in}_{k,j}}^2 \sigma_w^2) dX_{\text{in}_{k,j}} \quad \text{(Using MGF of Normal Distribution)}$$

$$= \int_{-\infty}^{\infty} \frac{1}{\sqrt{2\pi}\sigma_{x_{\text{in}}}} \exp\left(\frac{-X_{\text{in}_{k,j}}^2 (1 - 4\sigma_{x_{\text{in}}}^2 \sigma_w^2)}{2\sigma_{x_{\text{in}}}^2}\right) dX_{\text{in}_{k,j}}$$

$$= \int_{-\infty}^{\infty} \frac{1}{\sqrt{2\pi} \frac{\sigma_{x_{\text{in}}}}{\sqrt{(1 - 4\sigma_{x_{\text{in}}}^2 \sigma_w^2)}}} \sqrt{(1 - 4\sigma_{x_{\text{in}}}^2 \sigma_w^2)} \exp\left(\frac{-X_{\text{in}_{k,j}}^2 (1 - 4\sigma_{x_{\text{in}}}^2 \sigma_w^2)}{2\sigma_{x_{\text{in}}}^2}\right) dX_{\text{in}_{k,j}}$$

$$= \frac{1}{\sqrt{(1 - 4\sigma_{x_{\text{in}}}^2 \sigma_w^2)}} \int_{-\infty}^{\infty} \frac{1}{\sqrt{2\pi} \frac{\sigma_{x_{\text{in}}}}{\sqrt{(1 - 4\sigma_{x_{\text{in}}}^2 \sigma_w^2)}}} \exp\left(\frac{-X_{\text{in}_{k,j}}^2 (1 - 4\sigma_{x_{\text{in}}}^2 \sigma_w^2)}{2\sigma_{x_{\text{in}}}^2}\right) dX_{\text{in}_{k,j}}$$

$$= \frac{1}{\sqrt{(1 - 4\sigma_{x_{\text{in}}}^2 \sigma_w^2)}}$$

$$\exp\left(\sum_{l=1}^{d_{\text{in}}} W_{i,l}(X_{\text{in}_{k,l}} + X_{\text{in}_{m,l}})\right) X_{\text{in}_{k,j}} X_{\text{in}_{m,j}} = (\exp(W_{i,j}(X_{\text{in}_{k,j}} + X_{\text{in}_{m,j}})) X_{\text{in}_{k,j}} X_{\text{in}_{m,j}})$$

$$\prod_{l=1,l\neq j}^{d_{\text{in}}} \exp(W_{i,l}(X_{\text{in}_{k,l}} + X_{\text{in}_{m,l}}))$$

$$\mathbb{E}[\exp\left(\sum_{l=1}^{d_{\text{in}}} W_{i,l}(X_{\text{in}_{k,l}} + X_{\text{in}_{m,l}})\right) X_{\text{in}_{k,j}} X_{\text{in}_{m,j}}] = \mathbb{E}[\exp(W_{i,j}(X_{\text{in}_{k,j}} + X_{\text{in}_{m,j}})) X_{\text{in}_{k,j}} X_{\text{in}_{m,j}}]$$

$$\prod_{l=1,l\neq j}^{d_{\text{in}}} \mathbb{E}[\exp(W_{i,l}(X_{\text{in}_{k,l}} + X_{\text{in}_{m,l}}))]$$

Let $W_{i,j} = w$, $X_{\text{in}_{k,j}} = x_k$, $X_{\text{in}_{m,j}} = x_m$, and $r_{x_{\text{in}}}^l = r$. Then,

$$\mathbb{E}[\exp(w(x_k + x_m)) x_k x_m] =$$
$$\iint_{-\infty}^{\infty} \frac{x_k x_m}{2\sigma_{x_{\text{in}}}^2 \sqrt{1 - r^2}} \exp\left(\frac{-(x_k^2 + x_m^2 - 2r x_k x_m)}{2\sigma_{x_{\text{in}}}^2 (1 - r^2)}\right) dx_k dx_m \cdot$$

$$\int_{-\infty}^{\infty} \frac{\exp(w(x_k + x_m))}{2\sigma_w^2} \exp\left(\frac{-w^2}{2\sigma_w^2}\right) dw$$

$$= \iint_{-\infty}^{\infty} \frac{x_k x_m}{2\sigma_{x_{\text{in}}}^2 \sqrt{1 - r^2}} \exp\left(\frac{-(x_k^2 + x_m^2 - 2r x_k x_m)}{2\sigma_{x_{\text{in}}}^2 (1 - r^2)}\right) \exp\left(\frac{\sigma_w^2 (x_k + x_m)^2}{2}\right) dx_k dx_m$$
$$\text{(Using MGF of Normal Distribution)}$$

$$= \iint_{-\infty}^{\infty} \frac{x_k x_m}{2\sigma_{x_{\text{in}}}^2 \sqrt{1-r^2}} \exp\left(\frac{-(x_k^2(1-c)+x_m^2(1-c)-2(r+c)x_k x_m)}{2\sigma_{x_{\text{in}}}^2(1-r^2)}\right) dx_k dx_m$$

$$\text{(Let } c = (1-r^2)\sigma_w^2 \sigma_{x_{\text{in}}}^2)$$

Let $r\prime = \dfrac{r+c}{1-c}$, and $\sigma\prime^2(1-r\prime^2) = \dfrac{\sigma_{x_{\text{in}}}^2(1-r^2)}{1-c}$

$$= \iint_{-\infty}^{\infty} \frac{x_k x_m}{2\sigma_{x_{\text{in}}}^2 \sqrt{1-r^2}} \exp\left(\frac{-(x_k^2+x_m^2-2r\prime x_k x_m)}{2\sigma\prime^2(1-r\prime^2)}\right) dx_k dx_m$$

$$= \frac{\sigma\prime^2 \sqrt{1-r\prime^2}}{\sigma_{x_{\text{in}}}^2 \sqrt{1-r^2}} \iint_{-\infty}^{\infty} \frac{x_k x_m}{2\sigma\prime^2 \sqrt{1-r\prime^2}} \exp\left(\frac{-(x_k^2+x_m^2-2r\prime x_k x_m)}{2\sigma\prime^2(1-r\prime^2)}\right) dx_k dx_m$$

$$= \frac{\sigma\prime^2 \sqrt{1-r\prime^2}}{\sigma_{x_{\text{in}}}^2 \sqrt{1-r^2}} r\prime \sigma\prime^2$$

$$= \frac{r\prime \sigma\prime^4 \sqrt{1-r\prime^2}}{\sigma_{x_{\text{in}}}^2 \sqrt{1-r^2}}$$

$$= \frac{r\prime \sigma_{x_{\text{in}}}^4 (1-r^2)^2 \sqrt{1-r\prime^2}}{\sigma_{x_{\text{in}}}^2 (1-c)^2 (1-r\prime^2)^2 \sqrt{1-r^2}}$$

$$= \frac{(r+c)\sigma_{x_{\text{in}}}^2 (1-r^2)^{\frac{3}{2}}}{(1-c)^3 (1-r\prime^2)^{\frac{3}{2}}}$$

$$= (r+c)\sigma_{x_{\text{in}}}^2 \left(\frac{(1-r^2)}{(1-c)^2(1-r\prime^2)}\right)^{\frac{3}{2}}$$

$$= (r+c)\sigma_{x_{\text{in}}}^2 \left(\frac{(1-r^2)}{(1-c)^2(1-(\frac{r+c}{1-c})^2)}\right)^{\frac{3}{2}}$$

$$= (r+c)\sigma_{x_{\text{in}}}^2 \left(\frac{(1-r^2)}{(1-c)^2-(r+c)^2}\right)^{\frac{3}{2}}$$

$$= (r+c)\sigma_{x_{\text{in}}}^2 \left(\frac{(1-r)(1+r)}{(1+r)(1-r-2c)}\right)^{\frac{3}{2}}$$

$$= \frac{(r+c)\sigma_{x_{\text{in}}}^2}{(1-\frac{2c}{(1-r)})^{\frac{3}{2}}}$$

$$= \frac{(r+(1-r^2)\sigma_{x_{\text{in}}}^2 \sigma_w^2)\sigma_{x_{\text{in}}}^2}{(1-2(1+r)\sigma_{x_{\text{in}}}^2 \sigma_w^2)^{\frac{3}{2}}} \qquad (c = (1-r^2)\sigma_w^2 \sigma_{x_{\text{in}}}^2)$$

$$= \frac{(r_{x_{\text{in}}}^l+(1-(r_{x_{\text{in}}}^l)^2)\sigma_{x_{\text{in}}}^2 \sigma_w^2)\sigma_{x_{\text{in}}}^2}{(1-2(1+r_{x_{\text{in}}}^l)\sigma_{x_{\text{in}}}^2 \sigma_w^2)^{\frac{3}{2}}}$$

Let $W_{i,l} = w$, $r_{x_{\text{in}}}^l = r$, $X_{\text{in}_{k,l}} = x_k$, and $X_{\text{in}_{m,l}} = x_m$

$$\mathbb{E}[\exp(W_{i,l}(X_{\text{in}_{k,l}}+X_{\text{in}_{m,l}}))] =$$

$$\iint_{-\infty}^{\infty} \frac{1}{2\sigma_{x_{\text{in}}}^2 \sqrt{1-r^2}} \exp\left(\frac{-(x_k^2+x_m^2-2rx_k x_m)}{2\sigma_{x_{\text{in}}}^2(1-r^2)}\right) dx_k dx_m.$$

$$\int_{-\infty}^{\infty} \frac{\exp(w(x_k+x_m))}{2\sigma_w^2} \exp\left(\frac{-w^2}{2\sigma_w^2}\right) dw$$

$$= \iint_{-\infty}^{\infty} \frac{1}{2\sigma_{x_{\text{in}}}^2 \sqrt{1-r^2}} \exp\left(\frac{-(x_k^2+x_m^2-2rx_k x_m)}{2\sigma_{x_{\text{in}}}^2(1-r^2)}\right) \exp\left(\frac{\sigma_w^2(x_k+x_m)^2}{2}\right) dx_k dx_m$$

$$\text{(Using MGF of Normal Distribution)}$$

$$= \iint_{-\infty}^{\infty} \frac{1}{2\sigma_{x_{\text{in}}}^2 \sqrt{1-r^2}} \exp\left(\frac{-(x_k^2(1-c)+x_m^2(1-c)-2(r+c)x_k x_m)}{2\sigma_{x_{\text{in}}}^2(1-r^2)}\right) dx_k dx_m$$

$$\text{(Let } c = (1-r^2)\sigma_w^2 \sigma_{x_{\text{in}}}^2)$$

Let $r\prime = \dfrac{r+c}{1-c}$, and $\sigma\prime^2(1-r\prime^2) = \dfrac{\sigma_{x_{\text{in}}}^2(1-r^2)}{1-c}$

$$= \iint_{-\infty}^{\infty} \frac{1}{2\sigma_{x_{\text{in}}}^2\sqrt{1-r^2}} \exp\left(\frac{-(x_k^2 + x_m^2 - 2r\prime x_k x_m)}{2\sigma\prime^2(1-r\prime^2)}\right) dx_k dx_m$$

$$= \frac{\sigma\prime^2\sqrt{1-r\prime^2}}{\sigma_{x_{\text{in}}}^2\sqrt{1-r^2}} \iint_{-\infty}^{\infty} \frac{1}{2\sigma\prime^2\sqrt{1-r\prime^2}} \exp\left(\frac{-(x_k^2 + x_m^2 - 2r\prime x_k x_m)}{2\sigma\prime^2(1-r\prime^2)}\right) dx_k dx_m$$

$$= \frac{\sigma\prime^2\sqrt{1-r\prime^2}}{\sigma_{x_{\text{in}}}^2\sqrt{1-r^2}}$$

$$= \frac{\sigma_{x_{\text{in}}}^2(1-r^2)\sqrt{1-r\prime^2}}{\sigma_{x_{\text{in}}}^2(1-c)(1-r\prime^2)\sqrt{1-r^2}}$$

$$= \frac{\sqrt{1-r^2}}{(1-c)\sqrt{1-r\prime^2}}$$

$$= \sqrt{\frac{1-r^2}{(1-c)^2(1-r\prime^2)}}$$

$$= \sqrt{\frac{1-r^2}{(1-c)^2(1-(\frac{r+c}{1-c})^2)}}$$

$$= \sqrt{\frac{1-r^2}{(1-c)^2 - (r+c)^2}}$$

$$= \sqrt{\frac{(1+r)(1-r)}{(1+r)(1-r-2c)}}$$

$$= \frac{1}{\sqrt{(1-\frac{2c}{(1-r)})}}$$

$$= \frac{1}{\sqrt{(1-2(1+r)\sigma_{x_{\text{in}}}^2\sigma_w^2)}} \qquad (c = (1-r^2)\sigma_w^2\sigma_{x_{\text{in}}}^2)$$

$$= \frac{1}{\sqrt{(1-2(1+r_{x_{\text{in}}}^l)\sigma_{x_{\text{in}}}^2\sigma_w^2)}}$$

Using these results we have,

$$\mathbb{E}[(\text{Num}(O_{i,j}))^2] = L\frac{\sigma_{x_{\text{in}}}^2}{(1-p)(1-4\sigma_{x_{\text{in}}}^2\sigma_w^2)^{\frac{d_{\text{in}}}{2}+1}} + L(L-1)\frac{(r_{x_{\text{in}}}^l + (1-(r_{x_{\text{in}}}^l)^2)\sigma_{x_{\text{in}}}^2\sigma_w^2)\sigma_{x_{\text{in}}}^2}{(1-2(1+r_{x_{\text{in}}}^l)\sigma_{x_{\text{in}}}^2\sigma_w^2)^{\frac{d_{\text{in}}}{2}+1}}$$

For denominator,

$$\mathbb{E}[(\text{Den}(O_{i,j}))^2] = \mathbb{E}[(\sum_{m=1}^{L}\exp\left(\sum_{n=1}^{d_{\text{in}}} W_{i,n}X_{\text{in}_{m,n}}\right))^2]$$

$$= \mathbb{E}[\sum_{m=1}^{L}\exp\left(2\sum_{n=1}^{d_{\text{in}}} W_{i,n}X_{\text{in}_{m,n}}\right) + \sum_{m_1=1}^{L}\sum_{m_2=1,m_2\neq m_1}^{L}\exp\left(\sum_{n=1}^{d_{\text{in}}} W_{i,n}(X_{\text{in}_{m_1,n}} + X_{\text{in}_{m_2,n}})\right)]$$

$$= \mathbb{E}[\sum_{m=1}^{L}\exp\left(2\sum_{n=1}^{d_{\text{in}}} W_{i,n}X_{\text{in}_{m,n}}\right)] + \mathbb{E}[\sum_{m_1=1}^{L}\sum_{m_2=1,m_2\neq m_1}^{L}\exp\left(\sum_{n=1}^{d_{\text{in}}} W_{i,n}(X_{\text{in}_{m_1,n}} + X_{\text{in}_{m_2,n}})\right)]$$

$$= \sum_{m=1}^{L}\mathbb{E}[\exp\left(2\sum_{n=1}^{d_{\text{in}}} W_{i,n}X_{\text{in}_{m,n}}\right)] + \sum_{m_1=1}^{L}\sum_{m_2=1,m_2\neq m_1}^{L}\mathbb{E}[\exp\left(\sum_{n=1}^{d_{\text{in}}} W_{i,n}(X_{\text{in}_{m_1,n}} + X_{\text{in}_{m_2,n}})\right)]$$

$$\mathbb{E}[\exp{(2\sum_{n=1}^{d_{\text{in}}} W_{i,n}X_{\text{in}_{m,n}})}] = \mathbb{E}[\prod_{n=1}^{d_{\text{in}}} \exp{(2W_{i,n}X_{\text{in}_{m,n}})}]$$

$$= \prod_{n=1}^{d_{\text{in}}} \mathbb{E}[\exp{(2W_{i,n}X_{\text{in}_{m,n}})}]$$

$$= \prod_{n=1}^{d_{\text{in}}} \frac{1}{\sqrt{(1 - 4\sigma_{x_{\text{in}}}^2 \sigma_w^2)}}$$

$$= \frac{1}{(1 - 4\sigma_{x_{\text{in}}}^2 \sigma_w^2)^{\frac{d_{\text{in}}}{2}}}$$

$$\mathbb{E}[\exp{(\sum_{n=1}^{d_{\text{in}}} W_{i,n}(X_{\text{in}_{m_1,n}} + X_{\text{in}_{m_2,n}}))}] = \mathbb{E}[\prod_{n=1}^{d_{\text{in}}} \exp{(W_{i,n}(X_{\text{in}_{m_1,n}} + X_{\text{in}_{m_2,n}}))}]$$

$$= \prod_{n=1}^{d_{\text{in}}} \mathbb{E}[\exp{(W_{i,n}(X_{\text{in}_{m_1,n}} + X_{\text{in}_{m_2,n}}))}]$$

$$= \prod_{n=1}^{d_{\text{in}}} \frac{1}{\sqrt{(1 - 2(1 + r_{x_{\text{in}}}^l)\sigma_{x_{\text{in}}}^2 \sigma_w^2)}}$$

$$= \frac{1}{(1 - 2(1 + r_{x_{\text{in}}}^l)\sigma_{x_{\text{in}}}^2 \sigma_w^2)^{\frac{d_{\text{in}}}{2}}}$$

So we have,

$$\mathbb{E}[(\text{Den}(O_{i,j}))^2] = \frac{L}{(1 - 4\sigma_{x_{\text{in}}}^2 \sigma_w^2)^{\frac{d_{\text{in}}}{2}}} + \frac{L(L-1)}{(1 - 2(1 + r_{x_{\text{in}}}^l)\sigma_{x_{\text{in}}}^2 \sigma_w^2)^{\frac{d_{\text{in}}}{2}}}$$

From our assumption,

$$\mathbb{E}[O_{i,j}^2] \approx \frac{\mathbb{E}[(\text{Num}(O_{i,j}))^2]}{\mathbb{E}[(\text{Den}(O_{i,j}))^2]}$$

$$= \frac{L\frac{\sigma_{x_{\text{in}}}^2}{(1-p)(1-4\sigma_{x_{\text{in}}}^2 \sigma_w^2)^{\frac{d_{\text{in}}}{2}+1}} + L(L-1)\frac{(r_{x_{\text{in}}}^l + (1 - (r_{x_{\text{in}}}^l)^2)\sigma_{x_{\text{in}}}^2 \sigma_w^2)\sigma_{x_{\text{in}}}^2}{(1-2(1+r_{x_{\text{in}}}^l)\sigma_{x_{\text{in}}}^2 \sigma_w^2)^{\frac{d_{\text{in}}}{2}+1}}}{\frac{L}{(1-4\sigma_{x_{\text{in}}}^2 \sigma_w^2)^{\frac{d_{\text{in}}}{2}}} + \frac{L(L-1)}{(1-2(1+r_{x_{\text{in}}}^l)\sigma_{x_{\text{in}}}^2 \sigma_w^2)^{\frac{d_{\text{in}}}{2}}}}$$

$$= \frac{\frac{\sigma_{x_{\text{in}}}^2 c_1^{\frac{-d_{\text{in}}}{2}}}{(1-p)(1-4\sigma_{x_{\text{in}}}^2 \sigma_w^2)} + (L-1)\frac{(r_{x_{\text{in}}}^l + (1 - (r_{x_{\text{in}}}^l)^2)\sigma_{x_{\text{in}}}^2 \sigma_w^2)\sigma_{x_{\text{in}}}^2}{(1-2(1+r_{x_{\text{in}}}^l)\sigma_{x_{\text{in}}}^2 \sigma_w^2)}}{c_1^{\frac{-d_{\text{in}}}{2}} + (L-1)}$$

Where $c_1 = \frac{1 - 4d_{\text{in}}\sigma_{x_{\text{in}}}^4 \sigma_{qk}^2}{(1 - 2(1 + r_{x_{\text{in}}}^l)d_{\text{in}}\sigma_{x_{\text{in}}}^4 \sigma_{qk}^2)}$

$$\boxed{\begin{aligned}
&\text{Var}(O_{i,j}) = \\
&\frac{\sigma_{x_{\text{in}}}^2}{c_1^{\frac{-d_{\text{in}}}{2}} + (L-1)}\left(\frac{c_1^{\frac{-d_{\text{in}}}{2}}}{(1-p)(1 - 4d_{\text{in}}\sigma_{x_{\text{in}}}^4 \sigma_{qk}^2)} + (L-1)\frac{(r_{x_{\text{in}}}^l + (1 - (r_{x_{\text{in}}}^l)^2)d_{\text{in}}\sigma_{x_{\text{in}}}^4 \sigma_{qk}^2)}{(1 - 2(1 + r_{x_{\text{in}}}^l)d_{\text{in}}\sigma_{x_{\text{in}}}^4 \sigma_{qk}^2)}\right)
\end{aligned}}$$

Now for covariance we have,

$$O_{i,j}O_{m,j} = \frac{\text{Num}(O_{i,j})\text{Num}(O_{m,j})}{\text{Den}(O_{i,j})\text{Den}(O_{m,j})}$$

We again make the approximation that,

$$\mathbb{E}[O_{i,j}O_{m,j}] \approx \frac{\mathbb{E}[\text{Num}(O_{i,j})\text{Num}(O_{m,j})]}{\mathbb{E}[\text{Den}(O_{i,j})\text{Den}(O_{m,j})]}$$

$$\text{Num}(O_{i,j})\text{Num}(O_{m,j}) =$$

$$(\sum_{k_1=1}^{L} \text{Drop}(\exp\left(\sum_{l=1}^{d_{\text{in}}} W_{i,l}X_{\text{in}_{k_1,l}}\right)X_{\text{in}_{k_1,j}}))(\sum_{k_2=1}^{L} \text{Drop}(\exp\left(\sum_{l=1}^{d_{\text{in}}} W_{m,l}X_{\text{in}_{k_2,l}}\right)X_{\text{in}_{k_2,j}}))$$

$$\text{Num}(O_{i,j})\text{Num}(O_{m,j}) =$$

$$\sum_{k=1}^{L} \text{Drop}(\exp\left(\sum_{l=1}^{d_{\text{in}}} W_{i,l}X_{\text{in}_{k,l}}\right)X_{\text{in}_{k,j}})\text{Drop}(\exp\left(\sum_{l=1}^{d_{\text{in}}} W_{m,l}X_{\text{in}_{k,l}}\right)X_{\text{in}_{k,j}})+$$

$$\sum_{k_1=1}^{L}\sum_{k_2=1,k_2\neq k_1}^{L} \text{Drop}(\exp\left(\sum_{l=1}^{d_{\text{in}}} W_{i,l}X_{\text{in}_{k_1,l}}\right)X_{\text{in}_{k_1,j}})\text{Drop}(\exp\left(\sum_{l=1}^{d_{\text{in}}} W_{m,l}X_{\text{in}_{k_2,l}}\right)X_{\text{in}_{k_2,j}})$$

Thus we have,

$$\mathbb{E}[\text{Num}(O_{i,j})\text{Num}(O_{m,j})] =$$

$$\mathbb{E}[\sum_{k=1}^{L} \text{Drop}(\exp\left(\sum_{l=1}^{d_{\text{in}}} W_{i,l}X_{\text{in}_{k,l}}\right)X_{\text{in}_{k,j}})\text{Drop}(\exp\left(\sum_{l=1}^{d_{\text{in}}} W_{m,l}X_{\text{in}_{k,l}}\right)X_{\text{in}_{k,j}})+$$

$$\sum_{k_1=1}^{L}\sum_{k_2=1,k_2\neq k_1}^{L} \text{Drop}(\exp\left(\sum_{l=1}^{d_{\text{in}}} W_{i,l}X_{\text{in}_{k_1,l}}\right)X_{\text{in}_{k_1,j}})\text{Drop}(\exp\left(\sum_{l=1}^{d_{\text{in}}} W_{m,l}X_{\text{in}_{k_2,l}}\right)X_{\text{in}_{k_2,j}})]$$

$$= \mathbb{E}[\sum_{k=1}^{L} \text{Drop}(\exp\left(\sum_{l=1}^{d_{\text{in}}} W_{i,l}X_{\text{in}_{k,l}}\right)X_{\text{in}_{k,j}})\text{Drop}(\exp\left(\sum_{l=1}^{d_{\text{in}}} W_{m,l}X_{\text{in}_{k,l}}\right)X_{\text{in}_{k,j}})]$$

$$+ \mathbb{E}[\sum_{k_1=1}^{L}\sum_{k_2=1,k_2\neq k_1}^{L} \text{Drop}(\exp\left(\sum_{l=1}^{d_{\text{in}}} W_{i,l}X_{\text{in}_{k_1,l}}\right)X_{\text{in}_{k_1,j}})\text{Drop}(\exp\left(\sum_{l=1}^{d_{\text{in}}} W_{m,l}X_{\text{in}_{k_2,l}}\right)X_{\text{in}_{k_2,j}})]$$

$$= \sum_{k=1}^{L} \mathbb{E}[\text{Drop}(\exp\left(\sum_{l=1}^{d_{\text{in}}} W_{i,l}X_{\text{in}_{k,l}}\right)X_{\text{in}_{k,j}})\text{Drop}(\exp\left(\sum_{l=1}^{d_{\text{in}}} W_{m,l}X_{\text{in}_{k,l}}\right)X_{\text{in}_{k,j}})]+$$

$$\sum_{k_1=1}^{L}\sum_{k_2=1,k_2\neq k_1}^{L} \mathbb{E}[\text{Drop}(\exp\left(\sum_{l=1}^{d_{\text{in}}} W_{i,l}X_{\text{in}_{k_1,l}}\right)X_{\text{in}_{k_1,j}})\text{Drop}(\exp\left(\sum_{l=1}^{d_{\text{in}}} W_{m,l}X_{\text{in}_{k_2,l}}\right)X_{\text{in}_{k_2,j}})]$$

$$\mathbb{E}[\text{Dropout}(\exp\left(\sum_{l=1}^{d_{\text{in}}} W_{i,l}X_{\text{in}_{k,l}}\right)X_{\text{in}_{k,j}})\text{Dropout}(\exp\left(\sum_{l=1}^{d_{\text{in}}} W_{m,l}X_{\text{in}_{k,l}}\right)X_{\text{in}_{k,j}})]$$

$$= (1-p)^2\mathbb{E}[\frac{(\exp\left(\sum_{l=1}^{d_{\text{in}}} W_{i,l}X_{\text{in}_{k,l}}\right)X_{\text{in}_{k,j}})}{(1-p)}\frac{(\exp\left(\sum_{l=1}^{d_{\text{in}}} W_{m,l}X_{\text{in}_{k,l}}\right)X_{\text{in}_{k,j}})}{(1-p)}]$$

$$= \mathbb{E}[(\exp\left(\sum_{l=1}^{d_{\text{in}}} W_{i,l}X_{\text{in}_{k,l}}\right)X_{\text{in}_{k,j}})(\exp\left(\sum_{l=1}^{d_{\text{in}}} W_{m,l}X_{\text{in}_{k,l}}\right)X_{\text{in}_{k,j}})]$$

$$= \mathbb{E}[\exp\left(\sum_{l=1}^{d_{\text{in}}} (W_{i,l}+W_{m,l})X_{\text{in}_{k,l}}\right)X_{\text{in}_{k,j}}^2]$$

$$= \mathbb{E}[\exp\left((W_{i,j}+W_{m,j})X_{\text{in}_{k,j}}\right)X_{\text{in}_{k,j}}^2 \prod_{l=1,l\neq j}^{d_{\text{in}}} \exp\left((W_{i,l}+W_{m,l})X_{\text{in}_{k,l}}\right)]$$

$$= \mathbb{E}[\exp\left((W_{i,j}+W_{m,j})X_{\text{in}_{k,j}}\right)X_{\text{in}_{k,j}}^2] \prod_{l=1,l\neq j}^{d_{\text{in}}} \mathbb{E}[\exp\left((W_{i,l}+W_{m,l})X_{\text{in}_{k,l}}\right)]$$

$$\mathbb{E}[\exp\left((W_{i,j}+W_{m,j})X_{\text{in}_{k,j}}\right)X_{\text{in}_{k,j}}^2] =$$

$$\int_{-\infty}^{\infty} \frac{x^2}{\sqrt{2\pi}\sigma_{x_{\text{in}}}} \exp\left(\frac{-x^2}{2\sigma_{x_{\text{in}}}^2}\right)dx \int_{-\infty}^{\infty} \frac{\exp(w_1 x)}{\sqrt{2\pi}\sigma_w} \exp\left(\frac{-w_1^2}{2\sigma_w^2}\right)dw_1 \int_{-\infty}^{\infty} \frac{\exp(w_2 x)}{\sqrt{2\pi}\sigma_w} \exp\left(\frac{-w_2^2}{2\sigma_w^2}\right)dw_2$$

Where $W_{i,j} = w_1, W_{m,j} = w_2, X_{\text{in}_{k,j}} = x$

$$= \int_{-\infty}^{\infty} \frac{x^2}{\sqrt{2\pi}\sigma_{x_{\text{in}}}} \exp\left(\frac{-x^2}{2\sigma_{x_{\text{in}}}^2}\right) \exp\left(\frac{x^2\sigma_w^2}{2}\right) \exp\left(\frac{x^2\sigma_w^2}{2}\right)dx \qquad \text{(By MGF of Normal Distribution)}$$

$$= \int_{-\infty}^{\infty} \frac{x^2}{\sqrt{2\pi}\sigma_{x_{\text{in}}}} \exp\left(\frac{-x^2(1 - 2\sigma_{x_{\text{in}}}^2\sigma_w^2)}{2\sigma_{x_{\text{in}}}^2}\right)dx$$

$$= \frac{1}{\sqrt{(1 - 2\sigma_{x_{\text{in}}}^2\sigma_w^2)}} \int_{-\infty}^{\infty} \frac{x^2}{\sqrt{2\pi}\frac{\sigma_{x_{\text{in}}}}{\sqrt{(1-2\sigma_{x_{\text{in}}}^2\sigma_w^2)}}} \exp\left(\frac{-x^2(1 - 2\sigma_{x_{\text{in}}}^2\sigma_w^2)}{2\sigma_{x_{\text{in}}}^2}\right)dx$$

$$= \frac{1}{\sqrt{(1 - 2\sigma_{x_{\text{in}}}^2\sigma_w^2)}} \frac{\sigma_{x_{\text{in}}}^2}{(1 - 2\sigma_{x_{\text{in}}}^2\sigma_w^2)}$$

$$= \frac{\sigma_{x_{\text{in}}}^2}{(1 - 2\sigma_{x_{\text{in}}}^2\sigma_w^2)^{\frac{3}{2}}}$$

$$\mathbb{E}[\exp((W_{i,l} + W_{m,l})X_{\text{in}_{k,l}})] = \int_{-\infty}^{\infty} \frac{1}{\sqrt{2\pi}\sigma_{x_{\text{in}}}} \exp\left(\frac{-x^2}{2\sigma_{x_{\text{in}}}^2}\right)dx \int_{-\infty}^{\infty} \frac{\exp(w_1 x)}{\sqrt{2\pi}\sigma_w} \exp\left(\frac{-w_1^2}{2\sigma_w^2}\right)dw_1$$

$$\int_{-\infty}^{\infty} \frac{\exp(w_2 x)}{\sqrt{2\pi}\sigma_w} \exp\left(\frac{-w_2^2}{2\sigma_w^2}\right)dw_2$$

Where $W_{i,l} = w_1, W_{m,l} = w_2, X_{\text{in}_{k,l}} = x$

$$= \int_{-\infty}^{\infty} \frac{1}{\sqrt{2\pi}\sigma_{x_{\text{in}}}} \exp\left(\frac{-x^2}{2\sigma_{x_{\text{in}}}^2}\right) \exp\left(\frac{x^2\sigma_w^2}{2}\right) \exp\left(\frac{x^2\sigma_w^2}{2}\right)dx \qquad \text{(By MGF of Normal Distribution)}$$

$$= \int_{-\infty}^{\infty} \frac{1}{\sqrt{2\pi}\sigma_{x_{\text{in}}}} \exp\left(\frac{-x^2(1 - 2\sigma_{x_{\text{in}}}^2\sigma_w^2)}{2\sigma_{x_{\text{in}}}^2}\right)dx$$

$$= \frac{1}{\sqrt{(1 - 2\sigma_{x_{\text{in}}}^2\sigma_w^2)}} \int_{-\infty}^{\infty} \frac{1}{\sqrt{2\pi}\frac{\sigma_{x_{\text{in}}}}{\sqrt{(1-2\sigma_{x_{\text{in}}}^2\sigma_w^2)}}} \exp\left(\frac{-x^2(1 - 2\sigma_{x_{\text{in}}}^2\sigma_w^2)}{2\sigma_{x_{\text{in}}}^2}\right)dx$$

$$= \frac{1}{\sqrt{(1 - 2\sigma_{x_{\text{in}}}^2\sigma_w^2)}}$$

$$\mathbb{E}[\text{Dropout}(\exp\left(\sum_{l=1}^{d_{\text{in}}} W_{i,l}X_{\text{in}_{k_1,l}}\right)X_{\text{in}_{k_1,j}})\text{Dropout}(\exp\left(\sum_{l=1}^{d_{\text{in}}} W_{m,l}X_{\text{in}_{k_2,l}}\right)X_{\text{in}_{k_2,j}})]$$

$$= (1-p)^2 \mathbb{E}\left[\frac{(\exp\left(\sum_{l=1}^{d_{\text{in}}} W_{i,l}X_{\text{in}_{k_1,l}}\right)X_{\text{in}_{k_1,j}})}{(1-p)} \frac{(\exp\left(\sum_{l=1}^{d_{\text{in}}} W_{m,l}X_{\text{in}_{k_2,l}}\right)X_{\text{in}_{k_2,j}})}{(1-p)}\right]$$

$$= \mathbb{E}[(\exp\left(\sum_{l=1}^{d_{\text{in}}} W_{i,l}X_{\text{in}_{k_1,l}}\right)X_{\text{in}_{k_1,j}})(\exp\left(\sum_{l=1}^{d_{\text{in}}} W_{m,l}X_{\text{in}_{k_2,l}}\right)X_{\text{in}_{k_2,j}})]$$

$$= \mathbb{E}[\exp\left(\sum_{l=1}^{d_{\text{in}}} (W_{i,l}X_{\text{in}_{k_1,l}} + W_{m,l}X_{\text{in}_{k_2,l}})\right)X_{\text{in}_{k_1,j}}X_{\text{in}_{k_2,j}}]$$

$$= \mathbb{E}[\exp(W_{i,j}X_{\text{in}_{k_1,j}} + W_{m,j}X_{\text{in}_{k_2,j}})X_{\text{in}_{k_1,j}}X_{\text{in}_{k_2,j}} \prod_{l=1, l\neq j}^{d_{\text{in}}} \exp(W_{i,l}X_{\text{in}_{k_1,l}} + W_{m,l}X_{\text{in}_{k_2,l}})]$$

$$= \mathbb{E}[\exp(W_{i,j}X_{\text{in}_{k_1,j}} + W_{m,j}X_{\text{in}_{k_2,j}})X_{\text{in}_{k_1,j}}X_{\text{in}_{k_2,j}}] \prod_{l=1, l\neq j}^{d_{\text{in}}} \mathbb{E}[\exp(W_{i,l}X_{\text{in}_{k_1,l}} + W_{m,l}X_{\text{in}_{k_2,l}})]$$

$$\mathbb{E}[\exp{(W_{i,j}X_{\text{in}_{k_1},j} + W_{m,j}X_{\text{in}_{k_2},j})}X_{\text{in}_{k_1},j}X_{\text{in}_{k_2},j}] =$$

$$\iint_{-\infty}^{\infty} \frac{x_1 x_2}{2\pi\sigma_{x_{\text{in}}}^2\sqrt{(1-r^2)}}\exp{-(\frac{x_1^2 + x_2^2 - 2rx_1x_2}{2\sigma_{x_{\text{in}}}^2(1-r^2)})}dx_1 dx_2.$$

$$\int_{-\infty}^{\infty}\frac{\exp{(w_1 x_1)}}{\sqrt{2\pi}\sigma_w}\exp{(\frac{-w_1^2}{2\sigma_w^2})}dw_1\int_{-\infty}^{\infty}\frac{\exp{(w_2 x_2)}}{\sqrt{2\pi}\sigma_w}\exp{(\frac{-w_2^2}{2\sigma_w^2})}dw_2$$

Where $W_{i,j} = w_1, W_{m,j} = w_2, X_{\text{in}_{k_1},j} = x_1, X_{\text{in}_{k_2},j} = x_2, r_{x_{\text{in}}}^l = r$

$$= \iint_{-\infty}^{\infty}\frac{x_1 x_2}{2\pi\sigma_{x_{\text{in}}}^2\sqrt{(1-r^2)}}\exp{-(\frac{x_1^2 + x_2^2 - 2rx_1x_2}{2\sigma_{x_{\text{in}}}^2(1-r^2)})}\exp{(\frac{x_1^2\sigma_w^2}{2})}\exp{(\frac{x_2^2\sigma_w^2}{2})}dx_1 dx_2$$

$$= \iint_{-\infty}^{\infty}\frac{x_1 x_2}{2\pi\sigma_{x_{\text{in}}}^2\sqrt{(1-r^2)}}\exp{-(\frac{x_1^2(1-c) + x_2^2(1-c) - 2rx_1x_2}{2\sigma_{x_{\text{in}}}^2(1-r^2)})}dx_1 dx_2$$

$$(\text{Let } c = (1-r^2)\sigma_{x_{\text{in}}}^2\sigma_w^2)$$

Let $r\prime = \frac{r}{1-c}$, and $\sigma\prime^2(1-r\prime^2) = \frac{\sigma_{x_{\text{in}}}^2(1-r^2)}{1-c}$

$$= \iint_{-\infty}^{\infty}\frac{x_1 x_2}{2\sigma_{x_{\text{in}}}^2\sqrt{1-r^2}}\exp{(\frac{-(x_1^2 + x_2^2 - 2r\prime x_1 x_2)}{2\sigma\prime^2(1-r\prime^2)})}dx_1 dx_2$$

$$= \frac{\sigma\prime^2\sqrt{1-r\prime^2}}{\sigma_{x_{\text{in}}}^2\sqrt{1-r^2}}\iint_{-\infty}^{\infty}\frac{x_1 x_2}{2\sigma\prime^2\sqrt{1-r\prime^2}}\exp{(\frac{-(x_1^2 + x_2^2 - 2r\prime x_1 x_2)}{2\sigma\prime^2(1-r\prime^2)})}dx_1 dx_2$$

$$= \frac{\sigma\prime^2\sqrt{1-r\prime^2}}{\sigma_{x_{\text{in}}}^2\sqrt{1-r^2}}r\prime\sigma\prime^2$$

$$= \frac{r\prime\sigma\prime^4\sqrt{1-r\prime^2}}{\sigma_{x_{\text{in}}}^2\sqrt{1-r^2}}$$

$$= \frac{r\prime\sigma_{x_{\text{in}}}^4(1-r^2)^2\sqrt{1-r\prime^2}}{\sigma_{x_{\text{in}}}^2(1-c)^2(1-r\prime^2)^2\sqrt{1-r^2}}$$

$$= \frac{r\sigma_{x_{\text{in}}}^2(1-r^2)^{\frac{3}{2}}}{(1-c)^3(1-r\prime^2)^{\frac{3}{2}}}$$

$$= r\sigma_{x_{\text{in}}}^2\left(\frac{(1-r^2)}{(1-c)^2(1-r\prime^2)}\right)^{\frac{3}{2}}$$

$$= r\sigma_{x_{\text{in}}}^2\left(\frac{(1-r^2)}{(1-c)^2(1-(\frac{r}{1-c})^2)}\right)^{\frac{3}{2}}$$

$$= r\sigma_{x_{\text{in}}}^2\left(\frac{(1-r^2)}{(1-c)^2 - r^2}\right)^{\frac{3}{2}}$$

$$= r\sigma_{x_{\text{in}}}^2\left(\frac{(1-r)(1+r)}{(1+r-c)(1-r-c)}\right)^{\frac{3}{2}}$$

$$= r\sigma_{x_{\text{in}}}^2\left(\frac{1}{(1-\frac{c}{1+r})(1-\frac{c}{1-r})}\right)^{\frac{3}{2}}$$

$$= \frac{r\sigma_{x_{\text{in}}}^2}{[(1-(1-r)\sigma_{x_{\text{in}}}^2\sigma_w^2)(1-(1+r)\sigma_{x_{\text{in}}}^2\sigma_w^2)]^{\frac{3}{2}}} \qquad (c = (1-r^2)\sigma_w^2\sigma_{x_{\text{in}}}^2)$$

$$= \frac{r_{x_{\text{in}}}^l\sigma_{x_{\text{in}}}^2}{[(1-(1-r_{x_{\text{in}}}^l)\sigma_{x_{\text{in}}}^2\sigma_w^2)(1-(1+r_{x_{\text{in}}}^l)\sigma_{x_{\text{in}}}^2\sigma_w^2)]^{\frac{3}{2}}}$$

$$\mathbb{E}[\exp\left(W_{i,l}X_{\text{in}_{k_1,l}} + W_{m,l}X_{\text{in}_{k_2,l}}\right)] = \iint_{-\infty}^{\infty} \frac{\exp-\left(\frac{x_1^2+x_2^2-2rx_1x_2}{2\sigma_{x_{\text{in}}}^2(1-r^2)}\right)}{2\pi\sigma_{x_{\text{in}}}^2\sqrt{(1-r^2)}}dx_1dx_2.$$

$$\int_{-\infty}^{\infty}\frac{\exp\left(w_1x_1\right)}{\sqrt{2\pi}\sigma_w}\exp\left(\frac{-w_1^2}{2\sigma_w^2}\right)dw_1 \int_{-\infty}^{\infty}\frac{\exp\left(w_2x_2\right)}{\sqrt{2\pi}\sigma_w}\exp\left(\frac{-w_2^2}{2\sigma_w^2}\right)dw_2$$

Where $W_{i,l} = w_1, W_{m,l} = w_2, X_{\text{in}_{k_1,l}} = x_1, X_{\text{in}_{k_2,l}} = x_2, r_{x_{\text{in}}}^l = r$

$$= \iint_{-\infty}^{\infty}\frac{1}{2\pi\sigma_{x_{\text{in}}}^2\sqrt{(1-r^2)}}\exp-\left(\frac{x_1^2+x_2^2-2rx_1x_2}{2\sigma_{x_{\text{in}}}^2(1-r^2)}\right)\exp\left(\frac{x_1^2\sigma_w^2}{2}\right)\exp\left(\frac{x_2^2\sigma_w^2}{2}\right)dx_1dx_2$$

$$= \iint_{-\infty}^{\infty}\frac{1}{2\pi\sigma_{x_{\text{in}}}^2\sqrt{(1-r^2)}}\exp-\left(\frac{x_1^2(1-c)+x_2^2(1-c)-2rx_1x_2}{2\sigma_{x_{\text{in}}}^2(1-r^2)}\right)dx_1dx_2$$

$$\text{(Let } c = (1-r^2)\sigma_{x_{\text{in}}}^2\sigma_w^2)$$

Let $r\prime = \dfrac{r}{1-c}$, and $\sigma\prime^2(1-r\prime^2) = \dfrac{\sigma_{x_{\text{in}}}^2(1-r^2)}{1-c}$

$$= \iint_{-\infty}^{\infty}\frac{1}{2\sigma_{x_{\text{in}}}^2\sqrt{1-r^2}}\exp\left(\frac{-(x_1^2+x_2^2-2r\prime x_1x_2)}{2\sigma\prime^2(1-r\prime^2)}\right)dx_1dx_2$$

$$= \frac{\sigma\prime^2\sqrt{1-r\prime^2}}{\sigma_{x_{\text{in}}}^2\sqrt{1-r^2}}\iint_{-\infty}^{\infty}\frac{1}{2\sigma\prime^2\sqrt{1-r\prime^2}}\exp\left(\frac{-(x_1^2+x_2^2-2r\prime x_1x_2)}{2\sigma\prime^2(1-r\prime^2)}\right)dx_1dx_2$$

$$= \frac{\sigma\prime^2\sqrt{1-r\prime^2}}{\sigma_{x_{\text{in}}}^2\sqrt{1-r^2}}$$

$$= \frac{\sigma_{x_{\text{in}}}^2(1-r^2)\sqrt{1-r\prime^2}}{\sigma_{x_{\text{in}}}^2(1-c)(1-r\prime^2)\sqrt{1-r^2}}$$

$$= \frac{\sqrt{1-r^2}}{(1-c)\sqrt{1-r\prime^2}}$$

$$= \sqrt{\frac{1-r^2}{(1-c)^2 1-r\prime^2}}$$

$$= \sqrt{\frac{1-r^2}{(1-c)^2(1-(\frac{r}{1-c})^2)}}$$

$$= \sqrt{\frac{1-r^2}{(1-c)^2-r^2}}$$

$$= \sqrt{\frac{(1+r)(1-r)}{(1+r-c)(1-r-c)}}$$

$$= \frac{1}{\sqrt{(1-\frac{c}{1+r})(1-\frac{c}{1-r})}}$$

$$= \frac{1}{\sqrt{(1-(1-r)\sigma_{x_{\text{in}}}^2\sigma_w^2)(1-(1+r)\sigma_{x_{\text{in}}}^2\sigma_w^2)}} \qquad (c = (1-r^2)\sigma_w^2\sigma_{x_{\text{in}}}^2)$$

$$= \frac{1}{\sqrt{(1-(1-r_{x_{\text{in}}}^l)\sigma_{x_{\text{in}}}^2\sigma_w^2)(1-(1+r_{x_{\text{in}}}^l)\sigma_{x_{\text{in}}}^2\sigma_w^2)}}$$

So we have,

$$\mathbb{E}[\text{Num}(O_{i,j})\text{Num}(O_{m,j})] =$$

$$L\frac{\sigma_{x_{\text{in}}}^2}{(1-2\sigma_{x_{\text{in}}}^2\sigma_w^2)^{\frac{d_{\text{in}}}{2}+1}} + L(L-1)\frac{r_{x_{\text{in}}}^l\sigma_{x_{\text{in}}}^2}{[(1-(1-r_{x_{\text{in}}}^l)\sigma_{x_{\text{in}}}^2\sigma_w^2)(1-(1+r_{x_{\text{in}}}^l)\sigma_{x_{\text{in}}}^2\sigma_w^2)]^{\frac{d_{\text{in}}}{2}+1}}$$

$$\text{Den}(O_{i,j})\text{Den}(O_{m,j}) = \left(\sum_{k_1=1}^{L} \exp\left(\sum_{l=1}^{d_{\text{in}}} W_{i,l}X_{\text{in}_{k_1,l}}\right)\right)\left(\sum_{k_2=1}^{L} \exp\left(\sum_{l=1}^{d_{\text{in}}} W_{m,l}X_{\text{in}_{k_2,l}}\right)\right)$$

$$= \sum_{k=1}^{L} \exp\left(\sum_{l=1}^{d_{\text{in}}}(W_{i,l}+W_{m,l})X_{\text{in}_{k,l}}\right)$$

$$+ \sum_{k_1=1}^{L}\sum_{k_2=1,k_2\neq k_1}^{L} \exp\left(\sum_{l=1}^{d_{\text{in}}}(W_{i,l}X_{\text{in}_{k_1,l}}+W_{m,l}X_{\text{in}_{k_2,l}})\right)$$

$$= \sum_{k=1}^{L}\prod_{l=1}^{d_{\text{in}}} \exp\left((W_{i,l}+W_{m,l})X_{\text{in}_{k,l}}\right)$$

$$+ \sum_{k_1=1}^{L}\sum_{k_2=1,k_2\neq k_1}^{L}\prod_{l=1}^{d_{\text{in}}} \exp\left((W_{i,l}X_{\text{in}_{k_1,l}}+W_{m,l}X_{\text{in}_{k_2,l}})\right)$$

$$\mathbb{E}[\text{Den}(O_{i,j})\text{Den}(O_{m,j})] = \mathbb{E}\left[\sum_{k=1}^{L}\prod_{l=1}^{d_{\text{in}}} \exp\left((W_{i,l}+W_{m,l})X_{\text{in}_{k,l}}\right)\right.$$

$$\left.+ \sum_{k_1=1}^{L}\sum_{k_2=1,k_2\neq k_1}^{L}\prod_{l=1}^{d_{\text{in}}} \exp\left((W_{i,l}X_{\text{in}_{k_1,l}}+W_{m,l}X_{\text{in}_{k_2,l}})\right)\right]$$

$$= \mathbb{E}\left[\sum_{k=1}^{L}\prod_{l=1}^{d_{\text{in}}} \exp\left((W_{i,l}+W_{m,l})X_{\text{in}_{k,l}}\right)\right]$$

$$+ \mathbb{E}\left[\sum_{k_1=1}^{L}\sum_{k_2=1,k_2\neq k_1}^{L}\prod_{l=1}^{d_{\text{in}}} \exp\left((W_{i,l}X_{\text{in}_{k_1,l}}+W_{m,l}X_{\text{in}_{k_2,l}})\right)\right]$$

$$= \sum_{k=1}^{L}\mathbb{E}\left[\prod_{l=1}^{d_{\text{in}}} \exp\left((W_{i,l}+W_{m,l})X_{\text{in}_{k,l}}\right)\right]$$

$$+ \sum_{k_1=1}^{L}\sum_{k_2=1,k_2\neq k_1}^{L}\mathbb{E}\left[\prod_{l=1}^{d_{\text{in}}} \exp\left((W_{i,l}X_{\text{in}_{k_1,l}}+W_{m,l}X_{\text{in}_{k_2,l}})\right)\right]$$

$$= \sum_{k=1}^{L}\prod_{l=1}^{d_{\text{in}}}\mathbb{E}\left[\exp\left((W_{i,l}+W_{m,l})X_{\text{in}_{k,l}}\right)\right]$$

$$+ \sum_{k_1=1}^{L}\sum_{k_2=1,k_2\neq k_1}^{L}\prod_{l=1}^{d_{\text{in}}}\mathbb{E}\left[\exp\left((W_{i,l}X_{\text{in}_{k_1,l}}+W_{m,l}X_{\text{in}_{k_2,l}})\right)\right]$$

Again using earlier results we get,

$$\mathbb{E}[\text{Den}(O_{i,j})\text{Den}(O_{m,j})] =$$

$$L\frac{1}{(1-2\sigma_{x_{\text{in}}}^2\sigma_w^2)^{\frac{d_{\text{in}}}{2}}} + L(L-1)\frac{1}{[(1-(1-r_{x_{\text{in}}}^l)\sigma_{x_{\text{in}}}^2\sigma_w^2)(1-(1+r_{x_{\text{in}}}^l)\sigma_{x_{\text{in}}}^2\sigma_w^2)]^{\frac{d_{\text{in}}}{2}}}$$

So we get the covariance as,

$$\mathbb{E}[O_{i,j}O_{m,j}] \approx \frac{\mathbb{E}[\text{Num}(O_{i,j})\text{Num}(O_{m,j})]}{\mathbb{E}[\text{Den}(O_{i,j})\text{Den}(O_{m,j})]}$$

$$= \frac{L\frac{\sigma_{x_{\text{in}}}^2}{(1-2\sigma_{x_{\text{in}}}^2\sigma_w^2)^{\frac{d_{\text{in}}}{2}+1}} + L(L-1)\frac{r_{x_{\text{in}}}^l\sigma_{x_{\text{in}}}^2}{[(1-(1-r_{x_{\text{in}}}^l)\sigma_{x_{\text{in}}}^2\sigma_w^2)(1-(1+r_{x_{\text{in}}}^l)\sigma_{x_{\text{in}}}^2\sigma_w^2)]^{\frac{d_{\text{in}}}{2}+1}}}{L\frac{1}{(1-2\sigma_{x_{\text{in}}}^2\sigma_w^2)^{\frac{d_{\text{in}}}{2}}} + L(L-1)\frac{1}{[(1-(1-r_{x_{\text{in}}}^l)\sigma_{x_{\text{in}}}^2\sigma_w^2)(1-(1+r_{x_{\text{in}}}^l)\sigma_{x_{\text{in}}}^2\sigma_w^2)]^{\frac{d_{\text{in}}}{2}}}}$$

$$= \frac{\frac{\sigma_{x_{\text{in}}}^2 c_2^{\frac{-d_{\text{in}}}{2}}}{(1-2\sigma_{x_{\text{in}}}^2 \sigma_w^2)} + (L-1)\frac{r_{x_{\text{in}}}^l \sigma_{x_{\text{in}}}^2}{(1-(1-r_{x_{\text{in}}}^l)\sigma_{x_{\text{in}}}^2 \sigma_w^2)(1-(1+r_{x_{\text{in}}}^l)\sigma_{x_{\text{in}}}^2 \sigma_w^2)}}{c_2^{\frac{-d_{\text{in}}}{2}} + (L-1)}$$

Where $c_2 = \dfrac{1 - 2d_{\text{in}}\sigma_{x_{\text{in}}}^4 \sigma_{qk}^2}{(1-(1-r_{x_{\text{in}}}^l)d_{\text{in}}\sigma_{x_{\text{in}}}^4 \sigma_{qk}^2)(1-(1+r_{x_{\text{in}}}^l)d_{\text{in}}\sigma_{x_{\text{in}}}^4 \sigma_{qk}^2)}$

$$\boxed{\begin{aligned} \text{Cov}_O^l = \qquad\qquad\qquad\qquad\qquad\qquad\qquad\qquad\qquad \\ \frac{\sigma_{x_{\text{in}}}^2}{c_2^{\frac{-d_{\text{in}}}{2}} + (L-1)}\left( \frac{c_2^{\frac{-d_{\text{in}}}{2}}}{(1-2d_{\text{in}}\sigma_{x_{\text{in}}}^4 \sigma_{qk}^2)} + \frac{(L-1)r_{x_{\text{in}}}^l}{(1-(1-r_{x_{\text{in}}}^l)d_{\text{in}}\sigma_{x_{\text{in}}}^4 \sigma_{qk}^2)(1-(1+r_{x_{\text{in}}}^l)d_{\text{in}}\sigma_{x_{\text{in}}}^4 \sigma_{qk}^2)} \right) \end{aligned}}$$

Now it's easy to see both these constants are always less than 1.

If they are significantly smaller than 1 (which happens if $\sigma_{qk}$ is of considerable value) and $d_{\text{in}}$ is also sufficiently large, the approximations for variance and covariance become,

$$\text{Var}(O_{i,j}) \approx \frac{\sigma_{x_{\text{in}}}^2}{(1-p)(1-4d_{\text{in}}\sigma_{x_{\text{in}}}^4 \sigma_{qk}^2)} \qquad\qquad (c_1^{\frac{-d_{\text{in}}}{2}} \gg L)$$

$$\text{Cov}_O^l \approx \frac{\sigma_{x_{\text{in}}}^2}{(1-2d_{\text{in}}\sigma_{x_{\text{in}}}^4 \sigma_{qk}^2)} \qquad\qquad (c_2^{\frac{-d_{\text{in}}}{2}} \gg L)$$

The above situation corresponds to scenarios where one of the input to Softmax is extremely large compared to the others, hence resulting in degenerate attention only attending to one token. This can also be observed experimentally by setting $\sigma_{qk}$ considerably large, such as by initializing them to a few times larger than the standard Xavier initialization.

To avoid this degenerate attention, we choose smaller values of $\sigma_q, \sigma_k$, resulting in values of $c_1$ and $c_2$ almost equal to 1. In that scenario, the approximate value for variance and covariance are,

$$\boxed{\begin{aligned} \text{Var}(O_{i,j}) \approx r_{x_{\text{in}}}^l \sigma_{x_{\text{in}}}^2 \\ \text{Cov}_O^l \approx r_{x_{\text{in}}}^l \sigma_{x_{\text{in}}}^2 \end{aligned}} \qquad\qquad \begin{aligned} (c_1^{\frac{-d_{\text{in}}}{2}} \approx 1, L \gg 1, \sigma_{qk} \ll 1) \\ (c_2^{\frac{-d_{\text{in}}}{2}} \approx 1, L \gg 1, \sigma_{qk} \ll 1) \end{aligned}$$

To get the final variance and covariance we can use results of Linear layer to account for $\mathbf{W_V}$. If we initialize $\sigma_q$ and $\sigma_k$ to be small, in initial phase of training the output of Softmax layer can be treated as being a constant $= \frac{\mathbf{1_L^T 1_L}}{\mathbf{L}}$. Using this assumption we have,

$$\mathbf{X}_{\text{out}} \approx \text{Dropout}(\frac{\mathbf{1_L^T 1_L}}{\mathbf{L}})\mathbf{X}_{\text{in}}\mathbf{W_V}$$

$$\implies \mathbf{g_{x_{\text{in}}}} \approx \text{Dropout}(\frac{\mathbf{1_L^T 1_L}}{\mathbf{L}})^T \mathbf{g_{x_{\text{out}}}}\mathbf{W_V}^T$$

$$= \text{Dropout}(\frac{\mathbf{1_L^T 1_L}}{\mathbf{L}})\mathbf{g_{x_{\text{out}}}}\mathbf{W_V}^T$$

$$\boxed{\begin{aligned} \mu_{g_{\text{in}}} &= 0 \\ \sigma_{g_{\text{in}}}^2 &= \frac{\sigma_{g_{\text{out}}}^2 d\sigma_v^2}{L(1-p)}(1 + (L-1)r_{g_{\text{out}}}^l(1-p)) \\ \text{Cov}_{g_{\text{in}}}^l &= \frac{\sigma_{g_{\text{out}}}^2 d\sigma_v^2}{L}(1 + (L-1)r_{g_{\text{out}}}^l) \end{aligned}}$$

# D MOMENT PROPAGATION THROUGH TRANSFORMER BLOCKS

## D.1 TRANSFORMER ATTENTION BLOCK

A forward pass through attention block consists of LayerNorm, followed by Scaled Dot-Product Attention, followed by an output projection layer (a Linear Layer), and finally a Dropout. Using the results from above we get,

$$\mu_{x_{\text{out}}} = 0 * 0 * 0 * 0 = 0$$

$$\sigma^2_{x_{\text{out}}}$$

$$= \frac{\sigma^2_{x_{\text{in}}} \cdot d_{\text{in}} \sigma^2_v}{c_1^{\frac{-d_{\text{in}}}{2}} + (L-1)} \left( \frac{c_1^{\frac{-d_{\text{in}}}{2}}}{(1-p)(1-4d_{\text{in}}\sigma^2_q\sigma^2_k)} + (L-1)\frac{(r^l_{x_{\text{in}}} + (1-(r^l_{x_{\text{in}}})^2)d_{\text{in}}\sigma^2_q\sigma^2_k)}{(1-2(1+r^l_{x_{\text{in}}})d_{\text{in}}\sigma^2_q\sigma^2_k)} \right) \cdot \frac{d_{\text{in}}\sigma^2_o}{(1-p)}$$

$$= \frac{d_{\text{in}}^2\sigma^2_o\sigma^2_v\sigma^2_{x_{\text{in}}}}{(1-p)(c_1^{\frac{-d_{\text{in}}}{2}} + (L-1))} \left( \frac{c_1^{\frac{-d_{\text{in}}}{2}}}{(1-p)(1-4d_{\text{in}}\sigma^2_q\sigma^2_k)} + (L-1)\frac{(r^l_{x_{\text{in}}} + (1-(r^l_{x_{\text{in}}})^2)d_{\text{in}}\sigma^2_q\sigma^2_k)}{(1-2(1+r^l_{x_{\text{in}}})d_{\text{in}}\sigma^2_q\sigma^2_k)} \right)$$

$$\text{Cov}^l_{x_{\text{out}}}$$

$$= \frac{\sigma^2_{x_{\text{in}}} \cdot d_{\text{in}}\sigma^2_v}{c_2^{\frac{-d_{\text{in}}}{2}} + (L-1)} \left( \frac{c_2^{\frac{-d_{\text{in}}}{2}}}{(1-2d_{\text{in}}\sigma^2_q\sigma^2_k)} + \frac{(L-1)r^l_{x_{\text{in}}}}{(1-(1-r^l_{x_{\text{in}}})d_{\text{in}}\sigma^2_q\sigma^2_k)(1-(1+r^l_{x_{\text{in}}})d_{\text{in}}\sigma^2_q\sigma^2_k)} \right) \cdot d_{\text{in}}\sigma^2_o \cdot 1$$

$$= \frac{d_{\text{in}}^2\sigma^2_o\sigma^2_v\sigma^2_{x_{\text{in}}}}{c_2^{\frac{-d_{\text{in}}}{2}} + (L-1)} \left( \frac{c_2^{\frac{-d_{\text{in}}}{2}}}{(1-2d_{\text{in}}\sigma^2_q\sigma^2_k)} + (L-1)\frac{r^l_{x_{\text{in}}}}{(1-(1-r^l_{x_{\text{in}}})d_{\text{in}}\sigma^2_q\sigma^2_k)(1-(1+r^l_{x_{\text{in}}})d_{\text{in}}\sigma^2_q\sigma^2_k)} \right)$$

$$\sigma^2_{g_{\text{in}}} = \sigma^2_{g_{\text{out}}} * \frac{1}{(1-p)} * d_{\text{in}}\sigma^2_o * \frac{d_{\text{in}}\sigma^2_v}{L(1-p)}(1 + (L-1)r^l_{g_{\text{out}}}(1-p))$$

$$= \frac{d_{\text{in}}^2\sigma^2_{g_{\text{out}}}\sigma^2_v\sigma^2_o}{L(1-p)^2}(1 + (L-1)r^l_{g_{\text{out}}}(1-p))$$

$$\text{Cov}^l_{g_{\text{in}}} = \sigma^2_{g_{\text{out}}} * 1 * d_{\text{in}}\sigma^2_o * \frac{d_{\text{in}}\sigma^2_v}{L}(1 + (L-1)r^l_{g_{\text{out}}})$$

$$= \frac{d_{\text{in}}^2\sigma^2_{g_{\text{out}}}\sigma^2_v\sigma^2_o}{L}(1 + (L-1)r^l_{g_{\text{out}}})$$

## D.2 TRANSFORMER FFN BLOCK

A forward pass through the FFN block of a transfer has a LayerNorm, then a Linear layer from $d$ to $4d$, which is then passed through a ReLU gate, the output of which is the projected back to $d$ dimension using another Linear layer, and eventually passed through a Dropout. Again using the results from above we get,

$$\mu_{x_{\text{out}}} = 0 \qquad\qquad \text{(Last Linear Layer makes it 0)}$$

$$\sigma^2_{x_{\text{out}}} = 1 * d_{\text{in}}\sigma^2_{w_1} * (\frac{\pi - 1}{2\pi} + \frac{1}{2\pi}) * 4d_{\text{in}}\sigma^2_{w_2} * \frac{1}{(1-p)} * \sigma^2_{x_{\text{in}}}$$

$$= \frac{2d_{\text{in}}^2\sigma^2_{w_1}\sigma^2_{w_2}}{(1-p)}\sigma^2_{x_{\text{in}}}$$

$$\text{Cov}^l_{x_{\text{out}}} = d_{\text{in}}\sigma^2_{w_1} * (\frac{r^l_{x_{\text{in}}}}{4} + \frac{(1-(r^l_{x_{\text{in}}})^2)^{0.5}}{2\pi} + \frac{r^l_{x_{\text{in}}}\sin^{-1}(r^l_{x_{\text{in}}})}{2\pi} - \frac{1}{2\pi} + \frac{1}{2\pi}) * 4d_{\text{in}}\sigma^2_{w_2} * \sigma^2_{x_{\text{in}}}$$

$$= 4d_{\text{in}}^2\sigma^2_{w_1}\sigma^2_{w_2}\sigma^2_{x_{\text{in}}}(\frac{r^l_{x_{\text{in}}}}{4} + \frac{(1-(r^l_{x_{\text{in}}})^2)^{0.5}}{2\pi} + \frac{r^l_{x_{\text{in}}}\sin^{-1}(r^l_{x_{\text{in}}})}{2\pi})$$

$$r^l_{x_{\text{out}}} = 2 * (1 - p) * (\frac{r^l_{x_{\text{in}}}}{4} + \frac{(1 - (r^l_{x_{\text{in}}})^2)^{0.5}}{2\pi} + \frac{r^l_{x_{\text{in}}} \sin^{-1}(r^l_{x_{\text{in}}})}{2\pi})$$

$$\approx (1 - p) * (\frac{r^l_{x_{\text{in}}}}{2} + \frac{1}{\pi} + (\frac{1}{2} - \frac{1}{\pi})r^l_{x_{\text{in}}}{}^2) \qquad \text{(Fitting a 2-nd order polynomial)}$$

$$\sigma^2_{g_{\text{in}}} = \sigma^2_{g_{\text{out}}} * \frac{1}{(1 - p)} * d_{\text{in}}\sigma^2_{w_2} * \frac{1}{2} * 4d_{\text{in}}\sigma^2_{w_1}$$

$$= \frac{2d^2_{\text{in}}\sigma^2_{w_1}\sigma^2_{w_2}\sigma^2_{g_{\text{out}}}}{(1 - p)}$$

$$\text{Cov}^l_{g_{\text{in}}} = \text{Cov}^l_{g_{\text{out}}} * 1 * d_{\text{in}}\sigma^2_{w_2} * (\frac{1}{4} + \frac{\sin^{-1}(r^l_{x_{\text{in}}})}{2\pi}) * 4d_{\text{in}}\sigma^2_{w_1}$$

$$= 4d^2_{\text{in}}\sigma^2_{w_1}\sigma^2_{w_2}\text{Cov}^l_{g_{\text{out}}}(\frac{1}{4} + \frac{\sin^{-1}(r^l_{x_{\text{in}}})}{2\pi})$$

# E    SUMMARY TABLE OF MOMENT PROPAGATION THROUGH TRANSFORMER COMPONENTS

In Table 11, Table 12, Table 13, Table 14, Table 15 and Table 16, we summarize the signal propagation formulae for all the transformer components.

Table 11: Moment Propagation (mean) during forward pass through components of transformer model.

| Component | $\mu_{x_{\text{out}}}$ |
|---|---|
| Embeddings | 0 |
| FFN ($d_1.d_2$) | 0 |
| ReLU | $\frac{\sigma_{x_{\text{in}}}}{\sqrt{(2\pi)}}$ |
| GeLU | $\frac{\sigma^2_{x_{\text{in}}}}{\sqrt{2\pi(\sigma^2_{x_{\text{in}}} + 1)}}$ |
| LayerNorm ($d$) | 0 |
| Dropout ($p$) | $\mu_{x_{\text{in}}}$ |
| Softmax | $\frac{1}{L}$ |
| SHA Block (without V) | 0 |
| Attn Block | 0 |
| FFN Block | 0 |

Table 12: Moment Propagation (variance) during forward pass through components of transformer model.

| Component | $\sigma^2_{x_{\text{out}}}$ |
|---|---|
| Embeddings | $0$ |
| FFN $(d_1.d_2)$ | $d_1\sigma_w^2(\sigma^2_{x_{\text{in}}} + \mu^2_{x_{\text{in}}})$ |
| ReLU | $\dfrac{(\pi-1)}{(2\pi)}\sigma^2_{x_{\text{in}}}$ |
| GeLU | $\dfrac{\sigma^2_{x_{\text{in}}}}{2\pi}\left(\dfrac{\pi}{2} - \dfrac{\sigma^2_{x_{\text{in}}}}{1+\sigma^2_{x_{\text{in}}}} + \sin^{-1}\left(\dfrac{\sigma^2_{x_{\text{in}}}}{1+\sigma^2_{x_{\text{in}}}}\right) + \dfrac{2\sigma^2_{x_{\text{in}}}}{(1+\sigma^2_{x_{\text{in}}})\sqrt{1+2\sigma^2_{x_{\text{in}}}}}\right)$ |
| Layer Norm $(d)$ | $1$ |
| Dropout $(p)$ | $\dfrac{\sigma^2_{x_{\text{in}}} + p\mu^2_{x_{\text{in}}}}{1-p}$ |
| Softmax | $\dfrac{(e^{\sigma^2_{x_{\text{in}}}(1-r^l_{x_{\text{in}}})\frac{L}{L-1}}-1)e^{2\sigma^2_{x_{\text{in}}}(1-r^l_{x_{\text{in}}})\frac{L}{L-1}}}{((L-1)e^{\sigma^2_{x_{\text{in}}}(1-r^l_{x_{\text{in}}})}+1)^2}$ |
| SHA (without V) | $\dfrac{d_{\text{in}}\sigma^2_{x_{\text{in}}}}{(1-p)(c_1^{\frac{-d_{\text{in}}}{2}}+(L-1))}\left(\dfrac{c_1^{\frac{-d_{\text{in}}}{2}}}{(1-p)(1-4d_{\text{in}}\sigma_q^2\sigma_k^2)} + (L-1)\dfrac{(r^l_{x_{\text{in}}}+(1-(r^l_{x_{\text{in}}})^2)d_{\text{in}}\sigma_q^2\sigma_k^2)}{(1-2(1+r^l_{x_{\text{in}}})d_{\text{in}}\sigma_q^2\sigma_k^2)}\right)$ |
| Attn Block (Approx) | $\dfrac{d_{\text{in}}^2\sigma_o^2\sigma_v^2\sigma^2_{x_{\text{in}}}}{(1-p)(c_1^{\frac{-d_{\text{in}}}{2}}+(L-1))}\left(\dfrac{c_1^{\frac{-d_{\text{in}}}{2}}}{(1-p)(1-4d_{\text{in}}\sigma_q^2\sigma_k^2)} + (L-1)\dfrac{(r^l_{x_{\text{in}}}+(1-(r^l_{x_{\text{in}}})^2)d_{\text{in}}\sigma_q^2\sigma_k^2)}{(1-2(1+r^l_{x_{\text{in}}})d_{\text{in}}\sigma_q^2\sigma_k^2)}\right)$ |
| FFN Block | $\dfrac{2d_{\text{in}}^2\sigma_{w_1}^2\sigma_{w_2}^2\sigma^2_{x_{\text{in}}}}{(1-p)}$ |

Table 13: Moment Propagation (variance) during backwards pass through components of transformer model.

| Component | $\sigma^2_{g_{\text{in}}}$ |
|---|---|
| Embeddings | - |
| FFN $(d_1.d_2)$ | $d_2\sigma^2_w\sigma^2_{g_{\text{out}}}$ |
| ReLU | $\dfrac{1}{2}\sigma^2_{g_{\text{out}}}$ |
| GeLU | $\left[\frac{1}{4} + \frac{1}{2\pi}\sin^{-1}\left(\frac{\sigma^2_{x_{\text{in}}}}{\sigma^2_{x_{\text{in}}}+1}\right) + \frac{\sigma^2_{x_{\text{in}}}(5\sigma^2_{x_{\text{in}}}+3)}{2\pi(\sigma^2_{x_{\text{in}}}+1)(2\sigma^2_{x_{\text{in}}}+1)^{\frac{3}{2}}}\right]\sigma^2_{g_{\text{out}}}$ |
| LayerNorm $(d)$ | $\dfrac{\sigma^2_{g_{\text{out}}}}{\sigma^2_{x_{\text{in}}}}$ |
| Dropout $(p)$ | $\dfrac{1}{1-p}\sigma^2_{g_{\text{out}}}$ |
| Softmax | $\left(\dfrac{(e^{\sigma^2_{x_{\text{in}}}(1-r^l_{x_{\text{in}}})\frac{L}{L-1}}-1)e^{2\sigma^2_{x_{\text{in}}}(1-r^l_{x_{\text{in}}})\frac{L}{L-1}}}{((L-1)e^{\sigma^2_{x_{\text{in}}}(1-r^l_{x_{\text{in}}})}+1)^2} + \dfrac{1}{L^2}\right)\sigma^2_{g_{\text{out}}}$ |
| SHA Block (without V) | $\dfrac{d_{\text{in}}\sigma^2_{g_{\text{out}}}}{L(1-p)^2}(1 + (L-1)r^l_{g_{\text{out}}}(1-p))$ |
| Attn Block (Approx) | $\dfrac{d^2_{\text{in}}\sigma^2_{g_{\text{out}}}\sigma^2_v\sigma^2_o}{L(1-p)^2}(1 + (L-1)r^l_{g_{\text{out}}}(1-p))$ |
| FFN Block | $\dfrac{2d^2_{\text{in}}\sigma^2_{w_1}\sigma^2_{w_2}\sigma^2_{g_{\text{out}}}}{(1-p)}$ |

Table 14: Covariance (along sequence length) propagation through the components of transformer model.

| Component | $\mathrm{Cov}^l_{x_{\mathrm{out}}}$ |
|---|---|
| Embeddings | $\sum \dfrac{N_i * (N_i - 1)}{L * (L-1))} * \sigma^2_{w_{\mathrm{embd}}}$ |
| FFN $(d_1.d_2)$ | $d_1 \sigma^2_w (\mathrm{Cov}^l_{x_{\mathrm{in}}} + \mu^2_{x_{\mathrm{in}}})$ |
| ReLU | $(\dfrac{1}{4} + \dfrac{\sin^{-1}(r^l_{x_{\mathrm{in}}})}{2\pi})\mathrm{Cov}^l_{x_{\mathrm{in}}} - (1 - \sqrt{(1-(r^l_{x_{\mathrm{in}}})^2)})\dfrac{\sigma^2_{x_{\mathrm{in}}}}{2\pi}$ |
| GeLU | $\dfrac{\sigma^2_{x_{\mathrm{in}}}}{4\pi}(\pi r^l_{x_{\mathrm{in}}} + 2r^l_{x_{\mathrm{in}}}\sin^{-1}(\dfrac{r^l_{x_{\mathrm{in}}}\sigma^2_{x_{\mathrm{in}}}}{\sigma^2_{x_{\mathrm{in}}}+1}) + \dfrac{2\sigma^2_{x_{\mathrm{in}}}(\sigma^2_{x_{\mathrm{in}}}(1-(r^l_{x_{\mathrm{in}}})^2)+1+(r^l_{x_{\mathrm{in}}})^2)}{(\sigma^2_{x_{\mathrm{in}}}+1)\sqrt{(\sigma^2_{x_{\mathrm{in}}}+1)^2-(r^l_{x_{\mathrm{in}}}\sigma^2_{x_{\mathrm{in}}})^2}} - \dfrac{2\sigma^2_{x_{\mathrm{in}}}}{(\sigma^2_{x_{\mathrm{in}}}+1)})$ |
| LayerNorm $(d)$ | $(1 - \dfrac{1}{d})\dfrac{\mathrm{Cov}^l_{x_{\mathrm{in}}}}{\sigma^2_{x_{\mathrm{in}}}}$ |
| Dropout $(p)$ | $\mathrm{Cov}^l_{x_{\mathrm{in}}}$ |
| SHA (without V) | $\dfrac{d_{\mathrm{in}}\sigma^2_{x_{\mathrm{in}}}}{c_2^{\frac{-d_{\mathrm{in}}}{2}} + (L-1)}\left(\dfrac{c_2^{\frac{-d_{\mathrm{in}}}{2}}}{(1-2d_{\mathrm{in}}\sigma^2_q\sigma^2_k)} + (L-1)\dfrac{r^l_{x_{\mathrm{in}}}}{(1-(1-r^l_{x_{\mathrm{in}}})d_{\mathrm{in}}\sigma^2_q\sigma^2_k)(1-(1+r^l_{x_{\mathrm{in}}})d_{\mathrm{in}}\sigma^2_q\sigma^2_k)}\right)$ |
| Attn Block (Approx) | $\dfrac{d^2_{\mathrm{in}}\sigma^2_o\sigma^2_v\sigma^2_{x_{\mathrm{in}}}}{c_2^{\frac{-d_{\mathrm{in}}}{2}} + (L-1)}\left(\dfrac{c_2^{\frac{-d_{\mathrm{in}}}{2}}}{(1-2d_{\mathrm{in}}\sigma^2_q\sigma^2_k)} + (L-1)\dfrac{r^l_{x_{\mathrm{in}}}}{(1-(1-r^l_{x_{\mathrm{in}}})d_{\mathrm{in}}\sigma^2_q\sigma^2_k)(1-(1+r^l_{x_{\mathrm{in}}})d_{\mathrm{in}}\sigma^2_q\sigma^2_k)}\right)$ |
| FFN Block | $4d_{\mathrm{in}}\sigma^2_{w_1}\sigma^2_{w_2}\sigma^2_{x_{\mathrm{in}}}(\dfrac{r^l_{x_{\mathrm{in}}}}{4} + \dfrac{\sqrt{(1-(r^l_{x_{\mathrm{in}}})^2}}{2\pi} + \dfrac{r^l_{x_{\mathrm{in}}}\sin^{-1}(r^l_{x_{\mathrm{in}}})}{2\pi})$ |

Table 15: Covariance (hidden dimension) propagation through the components of transformer model.

| Component | $\mathrm{Cov}^d_{x_{\mathrm{out}}}$ |
|---|---|
| Embeddings | $0$ |
| FFN $(d_1.d_2)$ | $0$ |
| ReLU | $(\dfrac{1}{4} + \dfrac{\sin^{-1}(r^d_{x_{\mathrm{in}}})}{2\pi})\mathrm{Cov}^d_{x_{\mathrm{in}}} - (1 - \sqrt{(1-(r^d_{x_{\mathrm{in}}})^2)})\dfrac{\sigma^2_{x_{\mathrm{in}}}}{2\pi}$ |
| GeLU | |
| LayerNorm $(d)$ | $-\dfrac{1}{d-1}$ |
| Dropout $(p)$ | $\mathrm{Cov}^d_{x_{\mathrm{in}}}$ |
| SHA Block(without V ) | $0$ |
| Attn Block | $0$ |
| FFN Block | $0$ |

Table 16: Gradient covariance (along sequence length) propagation through the components of transformer model.

| Component | $\mathrm{Cov}^l_{g_{\text{in}}}$ |
|---|---|
| Embeddings | - |
| FFN $(d_1.d_2)$ | $d_2 \sigma_w^2 \mathrm{Cov}^l_{g_{\text{out}}}$ |
| ReLU | $\left(\dfrac{1}{4} + \dfrac{\sin^{-1}(r^l_{x_{\text{in}}})}{2\pi}\right)\mathrm{Cov}^l_{g_{\text{out}}}$ |
| GeLU | $\left[\dfrac{1}{4} + \dfrac{1}{2\pi}\sin^{-1}\left(\dfrac{r^l_{x_{\text{in}}}\sigma^2_{x_{\text{in}}}}{\sigma^2_{x_{\text{in}}}+1}\right) + \dfrac{r^l_{x_{\text{in}}}\sigma^2_{x_{\text{in}}}((2\sigma^2_{x_{\text{in}}}+3)(\sigma^2_{x_{\text{in}}}+1)-2(r^l_{x_{\text{in}}}\sigma^2_{x_{\text{in}}})^2)}{2\pi(\sigma^2_{x_{\text{in}}}+1)((\sigma^2_{x_{\text{in}}}+1)^2-(r^l_{x_{\text{in}}}\sigma^2_{x_{\text{in}}})^2)^{\frac{3}{2}}}\right] r^l_{g_{\text{out}}} \sigma^2_{g_{\text{out}}}$ |
| LayerNorm $(d)$ | $\dfrac{\mathrm{Cov}^l_{g_{\text{out}}}}{\sigma^2_{x_{\text{in}}}}$ |
| Dropout $(p)$ | $\mathrm{Cov}^l_{g_{\text{out}}}$ |
| SHA Block (without V) | $\dfrac{d_{\text{in}}\sigma^2_{g_{\text{out}}}}{L}(1 + (L-1)r^l_{g_{\text{out}}})$ |
| Attn Block (Approx) | $\dfrac{d^2_{\text{in}}\sigma^2_{g_{\text{out}}}\sigma^2_v\sigma^2_o}{L}(1 + (L-1)r^l_{g_{\text{out}}})$ |
| FFN Block | $4d^2_{\text{in}}\sigma^2_{w_1}\sigma^2_{w_2}\mathrm{Cov}^l_{g_{\text{out}}}\left(\dfrac{1}{4} + \dfrac{\sin^{-1}(r^l_{x_{\text{in}}})}{2\pi}\right)$ |

## F  NUMERICAL VERIFICATION

We perform numerical verification for the formulae reported in Table 11, Table 12, Table 13, Table 14, Table 15 and Table 16. The parameter ranges have been provided in Table 18. For each parameter, 3-5 values were sampled uniformly (or log uniformly) across the range for numerical simulation. Table 17 provides the percentage error corresponding to the $50_{th}$, $90_{th}$ and $99_{th}$ percentile. These simulation results are all fully reproducible using our code released as supplementary material. Even at 99 percentile, no error (other than SHA backwards) is larger than $10\%$, verifying our assumptions.

Table 17: Percentage Errors [50th, 90th, 99th percentile] for the theoretical formulas corresponding to forward and backward pass through components of the transformer model.

| Component | $\mu_{x_{\text{out}}}$ | $\sigma^2_{x_{\text{out}}}$ | $\sigma^2_{g_{\text{in}}}$ | $\mathrm{Cov}^l_{x_{\text{out}}}$ | $\mathrm{Cov}^l_{g_{\text{in}}}$ |
|---|---|---|---|---|---|
| FFN | [0.0, 0.4, 1.3] | [0.4, 1.4, 2.8] | [0.2, 1.0, 2.2] | [0.4, 1.4, 2.8] | [0.2, 1.0, 2.2] |
| ReLU | [0.3, 1.3, 2.3] | [0.5, 1.9, 3.4] | [0.6, 1.5, 2.6] | [0.3, 1.6, 3.1] | [0.2, 1.1, 2.3] |
| GeLU | [0.1, 1.0, 2.4] | [0.2, 0.6, 1.3] | [0.2, 0.6, 1.1] | [0.1, 0.5, 1.2] | [0.1, 0.4, 0.9] |
| LayerNorm | [0.0 , 0.0, 0.0] | [0.0, 0.0, 0.0] | [0.4, 1.5, 3.2] | [0.1, 0.5, 1.0] | [0.2, 0.9, 2.2] |
| Dropout | [0.0, 0.1, 0.5] | [0.1, 0.5, 1.5] | [0.1, 0.7, 1.5] | [0.0, 0.4, 1.3] | [0.1, 0.5, 1.2] |
| Softmax | [0.0 , 0.0, 0.0] | [0.2, 0.9, 4.0] | [0.1, 0.6, 4.5] | - | - |
| Single-Head Atten. | [0.2, 1.0, 2.5] | [1.4, 4.1, 7.9] | [2.2, 13.3, 44.5] | [1.3, 3.9, 7.5] | [1.6, 4.5, 8.2] |

Table 18: Range of input variance/correlations used for theoretical formula verification reported in Table 17 for the theoretical formulas corresponding to forward and backward pass through components of the transformer model. The dropout probability range was $[0, 1)$ for Dropout and Single-Head Attention, and $\sigma_w^2$ for FFN was $[10^{-2}, 10^2]/d_{\text{in}}$.

| Component | $\mu_{x_{\text{in}}}$ | $\sigma_{x_{\text{in}}}^2$ | $\sigma_{g_{\text{out}}}^2$ | $\text{Corr}_{x_{\text{in}}}^l$ | $\text{Corr}_{g_{\text{out}}}^l$ | $d_{\text{in}}$ | $d_{\text{out}}$ | $L$ |
|---|---|---|---|---|---|---|---|---|
| FFN | [-10, 10] | [0.1, 10] | [0.1, 10] | [0, 1.0) | [0, 1.0) | $[10^1, 10^3]$ | $[10^1, 10^3]$ | $[10^2, 10^3]$ |
| ReLU | [0] | [0.1, 10] | [0.1, 10] | [0, 1.0) | [0, 1.0) | - | - | $[10^2, 10^3]$ |
| GeLU | [0] | [0.1, 10] | [0.1, 10] | [0, 1.0) | [0, 1.0) | - | - | $[10^2, 10^3]$ |
| LayerNorm | [-10, 10] | [0.1, 10] | [0.1, 10] | [0, 1.0) | [0, 1.0) | $[10^2, 10^3]$ | - | $[10^2, 10^3]$ |
| Dropout | [-10, 10] | [0.1, 10] | [0.1, 10] | [0, 1.0) | [0, 1.0) | $[10^2, 10^3]$ | - | $[10^2, 10^3]$ |
| Softmax | [0] | $[10^{-4}, 1]$ | [0.1, 10] | [0, 1.0) | - | - | - | $[300, 10^4]$ |
| Single-Head Atten. | [0] | [1] | [0.1, 10] | [0, 1.0) | [0, 1.0) | $[10^2, 10^3]$ | [32, 64, 128, 256] | $[300, 10^4]$ |

## G    RANK COLLAPSE AND CORRELATION ANALYSIS

In the previous sections, we derived the formulas that determine how the correlation will change through the Attention and FFN blocks both for forward and backward pass. Both attention and FFN blocks modify the correlation as shown in the Table 19.

Table 19: Approximate Correlation Propagation during forward and backward pass through the blocks of a transformer layer.

| Component | $\sigma_{\mathbf{x_{out}}}^2$ | $\mathbf{r_{x_{out}}^l}$ | $\sigma_{\mathbf{g_{in}}}^2$ | $\mathbf{r_{g_{in}}^l}$ |
|---|---|---|---|---|
| Attention Block | $\dfrac{d^2\sigma_o^2\sigma_v^2\sigma_{x_{\text{in}}}^2 * r_{x_{\text{in}}}^l}{(1-p)}$ | $1-p$ | $\dfrac{d^2\sigma_o^2\sigma_v^2 * \sigma_{g_{\text{out}}}^2}{(1-p)}r_{g_{\text{out}}}^l$ | $1-p$ |
| FFN Block | $\dfrac{2d^2\sigma_{w_1}^2\sigma_{w_2}^2\sigma_{x_{\text{in}}}^2}{(1-p)}$ | $(1-p)(\dfrac{1}{\pi}+\dfrac{r_{x_{\text{in}}}^l}{2}+(\dfrac{1}{2}-\dfrac{1}{\pi})r_{x_{\text{in}}}^l{}^2)$ | $\sigma_{x_{\text{out}}}^2 * \sigma_{g_{\text{out}}}^2$ | $(1-p)(\dfrac{1}{2}+\dfrac{\sin^{-1}(r_{x_{\text{in}}}^l)}{\pi})r_{g_{\text{out}}}^l$ |

Simplifying the formulae in the table above, we rewrite the output variance for the attention block as $\sigma_{x_{\text{attn}}}^2 = C_1 * r_{x_{\text{in}}}^l * \sigma_{x_{\text{in}}}^2$, and the output of the FFN block is $\sigma_{x_{\text{ffn}}}^2 = C_2 * \sigma_{x_{\text{in}}}^2$, where $C_1$ and $C_2$ are defined as follows.

$$C_1 = \frac{d^2\sigma_o^2\sigma_v^2}{(1-p)}, C_2 = \frac{2d^2\sigma_{w_1}^2\sigma_{w_2}^2}{(1-p)},$$

This also helps us to rewrite the backward pass as the $\sigma_{g_{\text{attn}}}^2 = C_1 * r_{g_{\text{out}}}^l * \sigma_{g_{\text{out}}}^2$ and $\sigma_{g_{\text{ffn}}}^2 = C_2 * \sigma_{g_{\text{out}}}^2$.

Specifically in case of Xavier initialization with 0.1 dropout, $C_1 = 2.2, C_2 = 0.4$.

Assuming a dropout of 0.1, the FFN block (with the ReLU) will reduce the correlation if it rises above 0.64 (where $r_{x_{\text{out}}}^l < r_{x_{\text{in}}}^l$ for FFN block). And the attention block will never output a correlation higher than 0.9. Hence correlation will never reach 1, but rather a steady, stable value between ReLU's maximum correlation and that of the attention block. Dropout's effect in preventing rank collapse was also observed in Rong et al. (2019).

We can approximate the stable value of correlation after many layers based on the weightage average of the correlation in the Attention output and FFN output. When the attention output is added to the skip connection, the new correlation will be a weighted (by variance) average of the correlation among the tokens of attention output and among the tokens in the skip connection. And the same will happen after the FFN block.

A weighted average of the correlations of FFN and attention blocks gives the stable asymptotic correlation $r^l_{x_{\max}}$

$$r^l_{x_{\max}} = \frac{C_1 * (1 - p) + C_2 * (1 - p)(\frac{1}{\pi} + \frac{r^l_{x_{\max}}}{2} + (\frac{1}{2} - \frac{1}{\pi}){r^l_{x_{\max}}}^2)}{C_1 + C_2}$$

Specifically for the case of xavier initialization, solving the above equation with $C_1 = 2.2, C_2 = 0.4$, gives $r^l_{x_{\max}} \approx 0.88$.

Similarly, the correlation for backward gradient will also converge at a stable value $r^l_{g_{\max}}$, obtained by solving the below equation -

$$r^l_{g_{\max}} = \frac{C_1 * (1 - p) + C_2 * (1 - p)(\frac{1}{2} + \frac{\sin^{-1}(r^l_{x_{\max}})}{\pi})r^l_{g_{\max}}}{C_1 + C_2}$$

Specifically for the case of xavier initialization, this gives $r^l_{g_{\max}} = 0.87$. Note how $r^l_{g_{\max}} \approx r^l_{x_{\max}}$.

**Discussion on Noci et al. (2022)** Noci et al. (2022) focuses primarily on linear activation, we theoretically analyze the change in output correlation caused by ReLU. We find that ReLU (or any asymmetric non-linearity in general) critically affects correlation. As our closed form expressions suggest, both FFN block (because of ReLU) and dropout reduce the correlation. While Noci et al. (2022) mentions the use of dropout, as we show above and observe empirically in Figure 6, rank will not collapse with dropout, and perhaps Noci et al. (2022) did not use dropout. Further, we observed that Figure 10 of the supplementary of Noci et al. (2022) shows a correlation above 1, which is impossible.

We replicated the experimental settings of Noci et al. (2022) without dropout, and observed that the rank collapse occurs due to incorrect initialization. They use a rather non-standard version of xavier initialization - instead of $\frac{2}{fan_{in} + fan_{out}}$, they use $\frac{1}{fan_{out}}$. Hence, they initialize a much higher value for V as $fan_{in}$ is much greater than $fan_{out}$ ("Number of heads" times greater), and this results in variance of the output of the attention block $C1$ being much higher than FFN $C2$. As attention block outputs a much higher correlation than the FFN block, increasing its output variance without using dropout will result in rank collapse. This highlights the criticality of correct initialization, as well as the explainability power of our theoretical framework proposed in the paper.

## H MOMENT PROPAGATION THROUGH THE ENTIRE TRANSFORMER MODEL

### H.1 VANILLA PRE-LN

We will use the approximations listed in Table 2 here.

#### H.1.1 FORWARD PASS

For forward pass, a Transformer Pre-LN has LayerNorm followed by the Attention block, residual connection, LayerNorm, and then the FFN block. Let $\sigma^2_{\text{layer}}$ be the output variance after 1 such layer, and $\sigma^2_{\text{model}}$ be the output variance after the entire model of $N$ layers.

$$\sigma^2_{x_{\text{attn}}} = \frac{d^2 \sigma^2_o \sigma^2_v * r^l_{x_{\text{in}}}}{(1 - p)}$$

$$\sigma^2_{x_{\text{ffn}}} = \frac{2d^2 \sigma^2_{w_1} \sigma^2_{w_2}}{(1 - p)}$$

$$\sigma^2_{x_{\text{layer}}} = \sigma^2_{x_{\text{in}}} + \sigma^2_{x_{\text{attn}}} + \sigma^2_{x_{\text{ffn}}}$$

$$= \sigma^2_{x_{\text{in}}} + \frac{d^2 \sigma_o^2 \sigma_v^2 * r^l_{x_{\text{in}}}}{(1-p)} + \frac{2d^2 \sigma^2_{w_1} \sigma^2_{w_2}}{(1-p)}$$

$$\text{Let, } C_1 = \frac{d^2 \sigma_o^2 \sigma_v^2}{(1-p)}, C_2 = \frac{2d^2 \sigma^2_{w_1} \sigma^2_{w_2}}{(1-p)bu},$$

$$\text{Then, } \sigma^2_{x_{\text{layer}}} = \sigma^2_{x_{\text{in}}} + C_1 * r^l_{x_{\text{in}}} + C_2$$

As we discuss in Section 3.4, the correlation $r^l_{x_{\text{in}}}$ quickly reaches a stable constant maximum value $r^l_{x_{\text{max}}}$, which can be found using the calculations in Appendix G. Let $r^l_{x_{\text{min}}} > 0$ be the minimum value of this correlation, let $C_3 = C_1 * r^l_{x_{\text{max}}} + C_2$, and $C_4 = C_1 * r^l_{x_{\text{min}}} + C_2$. Then,

$$\sigma^2_{x_{\text{in}}} + C_4 \le \sigma^2_{x_{\text{layer}}} \le \sigma^2_{x_{x_{\text{in}}}} + C_3$$

Hence after N layers,

$$\sigma^2_{x_{\text{in}}} + N * C_4 \le \sigma^2_{x_{\text{model}}} \le \sigma^2_{x_{\text{in}}} + N * C_3$$

$$\implies \sigma^2_{x_{\text{model}}} = \Theta(N) \tag{2}$$

This shows that output variance of Pre-LN will increase linearly with number of layers $N$.

In practice, because the correlation quickly reaches $r^l_{x_{\text{max}}}$, the variance of the entire model $\sigma^2_{x_{\text{model}}} \approx \sigma^2_{x_{\text{in}}} + N * C_3$.

**Discussion:** This has the effect that transformer blocks near the output can affect the model output much less, as the skip connection variance increases but block output variance is constant. We conjecture that parameters in these are hence not being utilized to their full potential. Specifically in case of Xavier initialization, $C_1 = 2.2, C_2 = 0.4, r^l_{x_{\text{max}}} = 0.85$. For large $d$, $\sigma^2_{x_{\text{in}}}$ will be negligibly small compared to $\sigma^2_{x_{\text{layer}}}$, so we have -

$$\sigma^2_{x_{\text{model}}} \approx C_3 * N \approx (2.2 * 0.85 + 0.4)N \approx 2.2N$$

### H.1.2 BACKWARD PASS

For the backward pass, a Transformer Pre-LN gradient will first backpropagate through the FFN block, then gets rescaled by Layernorm, and added with the skip connection. It then backpropagates through the Attention block, gets rescaled by Layernorm, and finally added with the skip connection. Let $\sigma^2_{g,n}$ be the gradient variance backpropagating from the $n^{th}$ layer, and $\sigma^2_{g_{\text{model}}}$ be the gradient variance after the entire model of $N$ layers.

For the Attention block, let $\sigma^2_{g_{\text{attn}},n-1}$ be the gradient backpropagating from the block. Then for long sequence length $L$ we have -

$$\sigma^2_{g_{\text{attn}},n-1} = \frac{d^2 \sigma_o^2 \sigma_v^2 * \sigma^2_{g_{\text{out}},n}}{L(1-p)} \left(1 + (L-1)r^l_{g_{\text{out}},n}\right)$$

$$\approx \frac{d^2 \sigma_o^2 \sigma_v^2 * r^l_{g_{\text{out}},l} * \sigma^2_{g_{\text{out}},n}}{(1-p)}$$

$\sigma^2_{g_{\text{attn}},n-1}$ is then rescaled by the Layernorm to give $\sigma^2_{g_{\text{attn-layernorm}},n-1}$. As Layernorm scales gradient by the inverse of the input variance $\sigma^2_{x_{\text{in}},n-1}$, which from the section above, we know is approximately $\sigma^2_{x_{\text{in}},n-1} = C_3 * (n-1)$. Then

$$\sigma^2_{g_{\text{attn}},n-1} = C_1 * r^l_{g_{\text{out}},n} * \sigma^2_{g_{\text{out}},n}$$

$$\sigma^2_{g_{\text{attn-layernorm}},n-1} = \frac{C_1 * r^l_{g_{\text{out}},n} * \sigma^2_{g_{\text{out}},n}}{\sigma^2_{x_{\text{in}},n-1}}$$

$$\approx \frac{C_1 * r^l_{g_{\text{out}},n} * \sigma^2_{g_{\text{out}},n}}{C_3 * (n-1)}$$

Therefore, the final gradient $\sigma^2_{g_{\text{attn-layer}},n-1}$ after addition with the skip connection is

$$\sigma^2_{g_{\text{attn-layer}},n-1} = (1 + \frac{C_1 * r^l_{g_{\text{out}},n}}{C_3 * (n-1)})\sigma^2_{g_{\text{out}},n}$$

Similarly, we can get $\sigma^2_{g_{\text{ffn-layer}},n-1}$ for the ffn block. Then to get the gradient backpropagated through the entire layer $\sigma^2_{g_{\text{out}},n-1}$, we have,

$$\sigma^2_{g_{\text{ffn-layer}},n-1} = (1 + \frac{C_2}{C_3 * (n-1)})\sigma^2_{g_{\text{out}},n}$$

$$\sigma^2_{g_{\text{out}},n-1} = (1 + \frac{C_1 * r^l_{g_{\text{out}},n}}{C_3 * (n-1)})(1 + \frac{C_2}{C_3 * (n-1)})\sigma^2_{g_{\text{out}},n}$$

$$\sigma^2_{g_{\text{out}},n-1} \approx (1 + \frac{C_1 * r^l_{g_{\text{out}},n}}{C_3 * (n-1)} + \frac{C_2}{C_3 * (n-1)})\sigma^2_{g_{\text{out}},n}$$

$$= (1 + \frac{C_1 * r^l_{g_{\text{out}},n} + C_2}{C_3 * (n-1)})\sigma^2_{g_{\text{out}},n}$$

$$= (1 + \frac{C_1 * r^l_{g_{\text{out}},n} + C_2}{(C_1 * r^l_{x_{\text{in}},n} + C_2) * (n-1)})\sigma^2_{g_{\text{out}},n}$$

Where, ignore higher order terms for large n. As we discuss in the main paper, the correlation $r^l_{g_{\text{out}},n}$ quickly reaches a stable constant maximum value $r^l_{g_{\text{max}}}$, which is approximately equal to (but slightly less than) $r^l_{x_{\text{max}}}$. Hence, we can approximately replace the correlations with their maximum values. Note that while $r^l_{g_{\text{out}},n}$ and $r^l_{x_{\text{in}},n}$ will have slightly different stable values, the term $n-1$ will dominate in the following equations. Hence,

$$\sigma^2_{g_{\text{out}},n-1} = (1 + \frac{C_1 * r^l_{g_{\text{out}},n} + C_2}{(C_1 * r^l_{x_{\text{in}},n} + C_2) * (n-1)})\sigma^2_{g_{\text{out}},n}$$

$$\approx (1 + \frac{C_1 * r^l_{g_{\text{max}}} + C_2}{(C_1 * r^l_{x_{\text{max}}} + C_2) * (n-1)})\sigma^2_{g_{\text{out}},n}$$

$$\approx (1 + \frac{C_{g_{pre}}}{n-1})\sigma^2_{g_{\text{out}},n}$$

Since $C_{g_{pre}} > 0$, we will witness an increase in gradient going backward. Applying the above equation repeatedly until the final layer $N$, this recurrence can be approximately solved by treating $\sigma^2_{g_{\text{out}},n}$ as a continuous function of $n$, taking logarithm of both sides, and integrating. This gives the following solution for $\sigma^2_{g_{\text{out}},n}$:

$$\sigma^2_{g_{\text{out}},n} = \sigma^2_{g_{\text{out}},N} * (\frac{N}{n})^{C_{g_{pre}}}$$

If $C_{g_{pre}} \approx 1$, we get hyperbolic growth as shown below

$$\sigma^2_{g_{\text{out}},n} = \sigma^2_{g_{\text{out}},N} * \left(\frac{N}{n}\right)$$

This shows that gradient variance of Pre-LN will increase hyberbolically with number of layers $N$ while going backwards.

**Discussion:** This has the effect that much lower learning rate is required for the entire model, because the gradients near the input layers are much higher, slowing down learning and making the model unstable.

## H.2 VANILLA POST-LN

### H.2.1 FORWARD PASS

The forward pass of Post-LN is trivially always 1 at initialization, because the skip connection does not cross the LayerNorm.

### H.2.2 BACKWARD PASS

Following an analysis similar to that for Pre-LN, we get

$$\sigma^2_{g_{\text{ffn-layer}},n-1} = \frac{1 + C_2}{1 + C_1 * r^l_{x_{\text{out}},n-1}} \sigma^2_{g_{\text{out}},n}$$

$$\sigma^2_{g_{\text{attn-layer}},n-1} = \frac{1 + C_1 * r^l_{g_{\text{out}},n}}{1 + C_2} \sigma^2_{g_{\text{out}},n}$$

$$\sigma^2_{g_{\text{out}},n-1} = \frac{1 + C_1 * r^l_{g_{\text{out}},n}}{1 + C_2} * \frac{1 + C_2}{1 + C_1 * r^l_{x_{\text{out}},n-1}} * \sigma^2_{g_{\text{out}},n}$$

$$= \frac{1 + C_1 * r^l_{g_{\text{out}},n}}{1 + C_1 * r^l_{x_{\text{out}},n-1}} \sigma^2_{g_{\text{out}},n}$$

Let $C_{5,n} = \frac{1 + C_1 * r^l_{g_{\text{out}},n}}{1 + C_1 * r^l_{x_{\text{out}},n-1}}$. As we discuss in Appendix G, the correlations both quickly reach a maximum stable value. But the $r^l_{g_{\text{out}},n}$'s maximum value $r^l_{g_{\max}}$ is slightly different than $r^l_{x_{\max}}$. Let $C_5 = \frac{1 + C_1 * r^l_{g_{\max}}}{1 + C_1 * r^l_{x_{\max}}}$, then $C_5$ can be either greater or smaller than 1. Hence, we get

$$\sigma^2_{g_{\text{attn-layer}},n-1} = C_{5,n} \sigma^2_{g_{\text{out}},n}$$

$$= \prod_{i=n}^{N} C_{5,i} \sigma^2_{g_{\text{out}},N}$$

$$\approx C_5^{(N-n)} \sigma^2_{g_{\text{out}},N}$$

$$\sigma^2_{g_{\text{attn-layer}},n-1} = C_5^{(N-n)} \sigma^2_{g_{\text{out}},N} \tag{3}$$

This shows that gradient variance of Post-LN will decrease/increase exponentially with number of layers $N$ while going backwards. Even very slightly different value of $C_5$ from 1, such as 0.96, will cause a $2000x$ fall in gradient after 200 layers.

**Discussion:** This shows why Post-LN transformer is much more difficult to train for deeper models than Pre-LN. While for Pre-LN the backwards gradient increases hyber-bolically to a maximum of $N$, in Post-LN the gradient can increase or decrease exponentially, stopping the model from converging.

### H.3 DeepScaleLM Pre-LN

#### H.3.1 Forward Pass

In DeepScaleLM, the weight initialization are chosen specifically so that $\sigma^2_{x_{attn}}$ and $\sigma^2_{x_{ffn}}$ are both equal to 1 for all layers, by iteratively calculating $r^l_{x_{in}}$ as detailed in Appendix M. Also, the embeddings are initialized so that $\sigma^2_{x_{in}}$ is also 1. Hence,

$$\sigma^2_{layer} = \lambda^2 * \sigma^2_{skip} + \beta^2 * \sigma^2_{block}$$
$$= \lambda^2 + \beta^2 = 1$$

Hence the forward pass variance remains 1 throughout the model.

#### H.3.2 Backward Pass

For the FFN-block, we have $\sigma^2_{x_{in},n-1} = \sigma^2_{x_{out},n-1} = 1$, as per equations in Table 2 of the main paper. Similar to Vanilla-PreLN, we arrive at

$$\sigma^2_{g_{attn-layernorm},n-1} = \frac{C_1 * r^l_{g_{out},n} * \sigma^2_{g_{out},n}}{\sigma^2_{x_{in},n-1}}$$

Here, $\sigma^2_{x_{in},n-1} = 1$ as shown above, and since weights are initialized so that $C1 * r^l_{x_{in}} = 1$. Let $C_{6,n} = \frac{r^l_{g_{out},n}}{r^l_{x_{out},n-1}}$, and $C_6 = \frac{r^l_{g_{max}}}{r^l_{x_{max}}}$. As we show in the main paper, $C_6 \approx 1$. Similarly to previously, we use the maximum values of these correlations instead to get -

$$\sigma^2_{g_{attn-layernorm},n-1} = \frac{r^l_{g_{out},n}}{r^l_{x_{in},n-1}} * \sigma^2_{g_{out},n}$$
$$= C_{6,n} * \sigma^2_{g_{out},n}$$

Therefore, assuming no covariance between block gradients and skip connection (which will be true at initialization), the final gradient $\sigma^2_{g_{attn-layer},n-1}$ after addition with the skip connection is

$$\sigma^2_{g_{attn-layer},n-1} = \lambda^2 \sigma^2_{g_{out},n} + \beta^2 \sigma^2_{g_{attn-layernorm},n-1}$$
$$= \lambda^2 \sigma^2_{g_{out},n} + \beta^2 C_{6,n} \sigma^2_{g_{out},n}$$
$$= (\lambda^2 + \beta^2 C_{6,n}) * \sigma^2_{g_{out},n}$$
$$= (1 + \frac{C_{6,n} - 1}{N}) * \sigma^2_{g_{out},n}$$

Similarly for the FFN layer, $\sigma^2_{g_{ffn-layer},n-1} = \sigma^2_{g_{out},n}$, as $\sigma^2_{x_{in},n-1} = \sigma^2_{x_{out},n-1} = 1$.

Hence,

$$\sigma^2_{g_{out},n-1} = (1 + \frac{C_{6,n} - 1}{N}) * \sigma^2_{g_{out},n},$$
$$\sigma^2_{g_{out},1} = \prod_{i=1}^{N}(1 + \frac{C_{6,n} - 1}{N}) * \sigma^2_{g_{out},N},$$
$$\approx \prod_{i=1}^{N}(1 + \frac{C_6 - 1}{N}) * \sigma^2_{g_{out},N},$$
$$\approx (1 + \frac{C_6 - 1}{N})^{N-1} * \sigma^2_{g_{out},N},$$
$$= e^{C_6 - 1} * \sigma^2_{g_{out},N}$$

$$\approx \sigma_{g_{\text{out}},N}^2$$

, where we applied $(1 - \frac{k}{N})^N \approx e^{-k}$, and $C_6 \approx 1$.

**Discussion:** Hence for DeepScaleLM, the backward variance of gradient remains constant (bounded by a constant almost 1) across all layers.

### H.4 DEEPSCALELM POST-LN

#### H.4.1 FORWARD PASS

Same as vanilla Post-LN, this will remain preserved at 1.

#### H.4.2 BACKWARD PASS

Following an analysis similar to that for Vanilla Post-LN, we get

$$\sigma_{g_{\text{ffn-layer}},n-1}^2 = \sigma_{g_{\text{out}},n}^2$$
$$\sigma_{g_{\text{attn-layer}},n-1}^2 = (\lambda^2 * 1 + \beta^2 * C_1 * r_{g_{\text{out}},n}^l)\sigma_{g_{\text{out}},n}^2$$
$$= (\lambda^2 + \beta^2 * \frac{r_{g_{\text{out}},n}^l}{r_{x_{\text{in}},n}^l})\sigma_{g_{\text{out}},n}^2$$
$$\sigma_{g_{\text{out}},n-1}^2 = (\lambda^2 + \beta^2 * \frac{r_{g_{\text{out}},n}^l}{r_{x_{\text{in}},n}^l})\sigma_{g_{\text{out}},n}^2$$

Similar to Pre-LN, we use the maximum value of these correlations, and assume $C_6 = 1$. We get

$$\sigma_{g_{\text{out}},n-1}^2 = (\lambda^2 + \beta^2 * \frac{r_{g_{\text{max}}}^l}{r_{x_{\text{max}}}^l})\sigma_{g_{\text{out}},n}^2$$
$$= (\lambda^2 + \beta^2 C_6)\sigma_{g_{\text{out}},n}^2$$
$$\approx (\lambda^2 + \beta^2)\sigma_{g_{\text{out}},n}^2$$
$$= \sigma_{g_{\text{out}},n}^2$$

Hence for DeepScaleLM, the backward variance of gradient remains constant across all layers.

**Discussion:** Similar to DeepScale-LM Pre-LN, the assumption $C_6 = 1$ is not required, and yields the same constant bound if we do not assume it to be 1.

### H.5 DEEPSCALELM (SIMPLIFIED) PRE-LN

#### H.5.1 FORWARD PASS

For simplified DeepScaleLM, the initialization for the FFN block does not change, so its output remains 1 same as DeepScaleLM. For the Attention block, we changed its initialization to mimic that of the FFN block. We will show that initially, simplified DeepScaleLM's forward pass is bounded.

$\sigma_{x_{\text{ffn}}}^2 = 1$ as DeepScaleLM, $\sigma_{x_{\text{attn}}}^2 = \frac{r_{x_{\text{in}}}^l}{2}$. Therefore, the output variance after layer $n$ will be

$$\sigma_{x_{\text{attn-skip}},n}^2 = \lambda^2 * \sigma_{x_{\text{layer}},n-1}^2 + \beta^2 * \sigma_{x_{\text{attn}}}^2$$
$$= (1 - \frac{2}{N}) * \sigma_{x_{\text{layer}},n-1}^2 + \frac{1}{N} * r_{x_{\text{in}}}^l$$

Similarly after the FFN block, the output skip will be -

$$\sigma_{x_{\text{layer}},n}^2 = \lambda^2 * \sigma_{x_{\text{attn-skip}},n}^2 + \beta^2 * \sigma_{x_{\text{ffn}}}^2$$

$$= (1 - \frac{2}{N}) * ((1 - \frac{2}{N}) * \sigma^2_{x_{\text{layer}, n-1}} + \frac{1}{N} * r^l_{x_{\text{in}}}) + \frac{2}{N} * 1$$

$$= (1 - \frac{2}{N})^2 * \sigma^2_{x_{\text{layer}, n-1}} + (1 - \frac{2}{N}) * \frac{1}{N} * r^l_{x_{\text{in}}} + \frac{2}{N}$$

As correlation coefficient $r^l_{x_{\text{in}}} \leq 1$, we get,

$$\sigma^2_{x_{\text{layer}, n}} \leq (1 - \frac{2}{N})^2 * \sigma^2_{x_{\text{layer}, n-1}} + (1 - \frac{2}{N}) * \frac{1}{N} * 1 + \frac{2}{N}$$

$$= (1 - \frac{2}{N})^2 * \sigma^2_{x_{\text{layer}, n-1}} + \frac{3}{N} - \frac{2}{N^2}$$

$$\leq (1 - \frac{2}{N})^2 * \sigma^2_{x_{\text{layer}, n-1}} + \frac{3}{N}$$

Applying the above recurrence equation $N$ times, we get

$$\sigma^2_{x_{\text{layer}, N}} \leq (1 - \frac{2}{N})^{2N} * \sigma^2_{x_{\text{layer}, 0}} + \frac{3}{N} * \sum_{i=0}^{N} (1 - \frac{2}{N})^{2i}$$

$$= (1 - \frac{2}{N})^{2N} * \sigma^2_{x_{\text{layer}, 0}} + \frac{3}{N} * \frac{1 - (1 - \frac{2}{N})^{2N}}{1 - (1 - \frac{2}{N})^2}$$

Since $\lambda^2 + \beta^2 = 1$ and $\beta^2$ is small for large N. We can rewrite the above equations completely in terms of $\beta$ as follows

$$\sigma^2_{x_{\text{layer}, N}} = (1 - \beta^2)^{2N} * \sigma^2_{x_{\text{layer}, 0}} + \frac{3}{2}\beta^2 * \frac{1 - (1 - \beta^2)^{2N}}{1 - (1 - \beta^2)^2} \tag{4}$$

$$\approx (1 - \beta^2)^{2N} * \sigma^2_{x_{\text{layer}, 0}} + \frac{3}{4}(1 - (1 - \beta^2)^{2N}) \tag{5}$$

For large $N$, we know $(1 - \frac{k}{N})^N \approx e^{-k}$. So the above becomes -

$$\sigma^2_{x_{\text{layer}, N}} \approx e^{-4} * \sigma^2_{x_{\text{layer}, 0}} + \frac{3}{N} * \frac{1 - e^{-4}}{\frac{4}{N} - \frac{4}{N^2}}$$

$$\leq e^{-4} * \sigma^2_{x_{\text{layer}, 0}} + \frac{3}{N} * \frac{1 - e^{-4}}{\frac{4}{N}}$$

$$= e^{-4} * 1 + \frac{3}{4} * (1 - e^{-4})$$

$$= \frac{3}{4} + \frac{1}{4e^4}$$

This gives us an upper bound on the output variance after $N$ layers. By setting $r^l_{x_{\text{in}}} = 0$ instead of 1 in the equation above, and proceeding similarly, we can also arrive at a lower bound of $\frac{1}{2} + \frac{1}{2e^4}$.

$$\frac{1}{2} + \frac{1}{2e^4} \leq \sigma^2_{x_{\text{layer}, N}} \leq \frac{3}{4} + \frac{1}{4e^4} \tag{6}$$

**Discussion** Informally, this is because the attention block output variance will be between 0 and 0.5, and ffn block output always 1. Because of our $\lambda, \beta$ scaling, the output will slowly converge to be in between the two outputs.

Note that the above derivation assumes no correlation between the block output and the skip connection. As we mentioned in our main paper, we do observe correlation between the input and the output. As such, theoretically, after every block, the variance $\sigma^2_{x_{\text{layer}, n}}$ can increase by $\sigma^2_{x_{\text{block}}} + \sqrt{\sigma^2_{x_{\text{layer}, n}}}$.

This will cause the final output variance to increase by factors of $2 * \sqrt{N}$. In practice however, we observe the output variances to not grow too large.

### H.5.2    BACKWARD PASS

Similar to DeepScaleLM Pre-LN, we arrive at

$$\sigma^2_{g_{\text{attn-layernorm}},n-1} = \frac{C_1 * r^l_{g_{\text{out}},n} * \sigma^2_{g_{\text{out}},n}}{\sigma^2_{x_{\text{in}},n-1}}$$

$$\approx \frac{0.5 * C_6}{\sigma^2_{x_{\text{in}},n-1}} * \sigma^2_{g_{\text{out}},n}$$

$$\sigma^2_{g_{\text{attn-layer}},n-1} = \lambda^2 \sigma^2_{g_{\text{out}},n} + \beta^2 \sigma^2_{g_{\text{attn-layernorm}},n-1}$$

$$= (\lambda^2 + \beta^2 * \frac{0.5 * C_6}{\sigma^2_{x_{\text{in}},n-1}}) * \sigma^2_{g_{\text{out}},n}$$

$$= (1 + \frac{2}{N} * (\frac{0.5 * C_6}{\sigma^2_{x_{\text{in}},n-1}} - 1)) * \sigma^2_{g_{\text{out}},n}$$

Similarly, for the FFN layer, we get

$$\sigma^2_{g_{\text{ffn-layer}},n-1} = (1 + \frac{2}{N} * (\frac{1}{\sigma^2_{x_{\text{in}},n-1}} - 1)) * \sigma^2_{g_{\text{out}},n}$$

Multiplying these, we get

$$\sigma^2_{g_{\text{out}},n-1} = (1 + \frac{2}{N} * (\frac{0.5 * C_6}{\sigma^2_{x_{\text{in}},n-1}} - 1)) * (1 + \frac{2}{N} * (\frac{1}{\sigma^2_{x_{\text{in}},n-1}} - 1)) * \sigma^2_{g_{\text{out}},n}$$

$$\approx (1 + \frac{2}{N} * (\frac{0.5 * C_6}{\sigma^2_{x_{\text{in}},n-1}} + \frac{1}{\sigma^2_{x_{\text{in}},n-1}} - 2)) * \sigma^2_{g_{\text{out}},n}$$

As $0.5 \leq \sigma^2_{x_{\text{in}},n-1}$, we get $-4 \leq (\frac{C_6}{\sigma^2_{x_{\text{in}},n-1}} + \frac{2}{\sigma^2_{x_{\text{in}},n-1}} - 4) \leq 2C_6 + 2$. Hence, on applying the above recurrence N times, we get

$$e^{-4} * \sigma^2_{g_{\text{out}},N} \leq \sigma^2_{g_{\text{out}},n-1} \leq e^{2C_6+2} * \sigma^2_{g_{\text{out}},N}$$

Hence, we show that even for simplified DeepScaleLM Pre-LN, the maximum relative increase/fall in gradient variance is bounded across layers.

**Discussion:**    The above derivations will also be valid if there is correlation in the input. Correlation will cause $\sigma^2_{x_{\text{in}},n-1}$ to increase, effectively decreasing the backpropagated gradient through the block to decrease (as Layernorm will scale by inverse of $\sigma^2_{x_{\text{in}},n-1}$). However, even in that case, our gradient will still be bounded by the above lower-bound.

Intuitively, as the gradient can flow freely through the skip connection, hence, $\sigma^2_{g_{\text{out}},n-1} \geq \lambda^4 * \sigma^2_{g_{\text{out}},n}$, which when applied $N$ times, yields $\sigma^2_{g_{\text{out}},1} \geq e^{-4} * \sigma^2_{g_{\text{out}},N}$

### H.6    DEEPSCALELM (SIMPLIFIED) POST-LN

### H.6.1    FORWARD PASS

The forward pass variance for Post-LN is trivially bounded.

### H.6.2    BACKWARD PASS

Following an analysis similar to that for DeepScaleLM Post-LN, we get

$$\sigma_{g_{\text{out}},n-1}^2 = \frac{\lambda^2 + 0.5 * \beta^2 * r_{g_{\text{out}},n}^l}{\lambda^2 + 0.5 * \beta^2 * r_{x_{\text{in}},n}^l} \sigma_{g_{\text{out}},n}^2$$

$$= \frac{1 + \frac{2}{N}(0.5 r_{g_{\text{out}},n}^l - 1)}{1 + \frac{2}{N}(0.5 r_{x_{\text{in}},n}^l - 1)} \sigma_{g_{\text{out}},n}^2$$

Applying taylor expansion, we get,

$$\sigma_{g_{\text{out}},n-1}^2 \approx (1 + \frac{2}{N}((0.5 r_{g_{\text{out}},n}^l - 1) - (0.5 r_{x_{\text{in}},n}^l - 1)))\sigma_{g_{\text{out}},n}^2$$

$$= (1 + \frac{1}{N}(r_{g_{\text{out}},n}^l - r_{x_{\text{in}},n}^l))\sigma_{g_{\text{out}},n}^2$$

The above equation can be rewritten in terms of $\beta$ as follows

$$\sigma_{g_{\text{out}},n-1}^2 = (1 + \frac{\beta^2}{2}(r_{g_{\text{out}},n}^l - r_{x_{\text{in}},n}^l))\sigma_{g_{\text{out}},n}^2 \qquad (7)$$

As $-2 \leq (r_{g_{\text{out}},n}^l - r_{x_{\text{in}},n}^l) \leq 2$, applying the above recurrence $N$ times we get

$$e^{-2} * \sigma_{g_{\text{out}},N}^2 \leq \sigma_{g_{\text{out}},n-1}^2 \leq e^2 * \sigma_{g_{\text{out}},N}^2$$

**Discussion:** The above derivations assume no correlation in the input, and hence is only correct at initialization. However, if there is correlation between the block output and skip connection ($r_x$), the layernorm will cause $\sigma_{g_{\text{out}},n-1}^2$ to be down-scaled by a factor of $1 + \frac{2*r_x}{\sqrt{N}}$, where c is some constant, as opposed to $1 + \frac{2}{N}$ above. However, if there is also correlation in the gradients of the block and skip connection ($r_g$), the numerator in the equations above for $\sigma_{g_{\text{out}},n-1}^2$ will also be increased, by a factor of $1 + \frac{2*r_g}{\sqrt{N}}$. Hence if the correlations among the gradients and among the output are similar, the above bounds will remain. If $\beta^2$ is set as $\frac{1}{N^2}$, then even if input correlations exist, the backward gradient will be bounded, following a similar derivation as above. However, we conjecture that this decreases the ability of the transformer layers to modify the skip connection too strongly, decreasing the "expressivity" of the model. This is similar to the approach of DSInit, which we show in our main paper does indeed decrease model performance.

## I    MODEL MOMENT FIGURES

### I.1    VALIDITY OF THEORETICAL PREDICTIONS AND EXPLODING GRADIENTS EVEN AFTER TRAINING

When training a 64-layer Language Model, we observed repeated gradient explosions causing model divergence. Also, our 64-layer model initially performed worse than a smaller 48-layer model. We observed that (1) The norm of the model output was increasing going forwards (2) The backprop-agated gradient was increasing going backward. These issues were more pronounced in the deeper 64-layer model compared to the 48-layer model. We applied our formulae to understand these in-stabilities.

In Figure 9, we can see the observed growth in the output of a 48-layer PreLN model after 100k training steps. The observed growth across 48 layers remains very close to our predicted values. We observe a similar trend for the backward pass of our 64-layer PreLN model after it was trained for 150k steps. Figure 8 shows the observed gradient explosion hows the observed gradient explosion vs. our hyperbolic growth estimation. Interestingly, our theoretical estimates hold approximately even after the models have been trained for a large number of steps. The model stays in the regime it is initialized with, highlighting the importance of correct initialization.

### I.2    IMPORTANCE OF RESIDUAL SCALING

Any value for $\lambda$ or $\beta$ such that $\lambda^2 + \beta^2 = 1$ is sufficient to preserve the output (similar to Liu et al. (2020b)) and gradients at initialization. However, we observe that for high $\beta$, as training progresses,

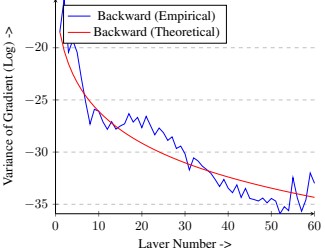

Figure 8: Gradient explodes backward for 64-layer pre-LN, increasing hyperbolically with number of layers $N$, after 150k training steps. Our theoretical models still hold - the model never escapes the regime it is initialized in.

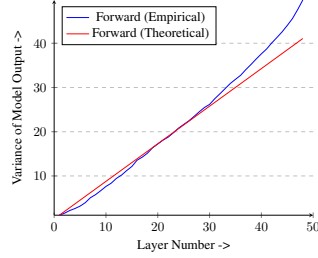

Figure 9: Linear growth in the forward pass for a 48-layer 1024-d PreLN model after training for 100k steps. Our theoretical models still hold well.

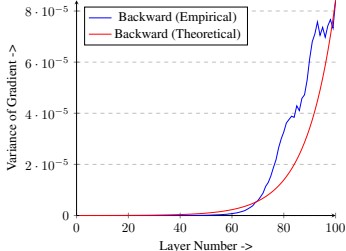

Figure 10: Gradient vanishes for Deep-ScaleLM, decreasing exponentially with layers $N$, using fixed $\lambda^2 = 0.9$ and $\beta^2 = 0.1$, after 50k training steps.

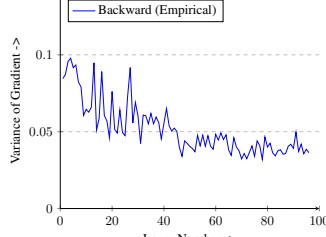

Figure 11: Gradient does not vanish for DeepScaleLM, using fixed $\lambda^2 = 1 - \frac{1}{N}$ and $\beta^2 = \frac{1}{N}$, after 50k training steps.

the gradient starts to vanish, as shown in Figure 10. This is because $\mathrm{Cov}(\mathrm{residual}, \mathrm{block})$ is no longer zero, which causes the forward output to grow across layer and the gradient to vanish.

On the contrary when we choose $\beta^2 = \frac{1}{N}$, gradient is conserved even after 50k steps of training as shown in Figure 11.

## J COMPUTE

### J.1 THEORETICAL COMPUTE

Table 20 provides the exact compute for the models reported in Table 4. We follow the code provided by Electra (Clark et al., 2020) to calculate the each model's compute (FLOPs). We observe that up to 200 layers, the extra compute is within $6 - 7\%$ of the original shallow model.

Table 20: Model compute with increasing depth (keeping $Nd^2$ constant).

| Layers(N) | d | Compute (Flops) | % Extra |
|---|---|---|---|
| *165M* | | | |
| 12 | 1024 | 1.06e20 | - |
| 48 | 512 | 1.03e20 | -2.5% |
| 192 | 256 | 1.12e20 | 6.3% |
| 784 | 128 | 1.38e20 | 30.6% |
| *330M* | | | |
| 24 | 1024 | 1.92e20 | - |
| 96 | 512 | 1.96e20 | 2.3% |
| 384 | 128 | 2.19e20 | 14.5% |

### J.2 WALL CLOCK TIMES

We also compared wall clock time overheads, and found them to not be too large. For example, the 48-layer-512-d model has only 9.8% overhead in wall clock time compared to 12-layer-1024-d model. Even when larger number of layers, such as 96-layer-512-d, the overhead is only 14.9% compared to 24-layer-1024-d model. Profiling revealed majority of the overhead was due to extra latency of added GPU kernel launches. Hence, approaches such as cudaGraphs (which batches kernel launches together) or graph compilation techniques may decrease this overhead further.

This overhead will decrease the bigger the original model size, and becomes much smaller. For example, for a 5B params model with 24-Layers-4096d (a reasonable shape in contemporary models, for example, LLaMa 7B has 32L-4096D) has much less compute overhead - only 6.6% overhead at 96 layers, and 13.6% overhead at 192 layers.

Despite this wall-clock time overhead, due to large performance gains from increasing depth, the 165M params 192-L model from Table 4 outperforms the vanilla 330M bert-large 24-L model with $2x$ more params, even at equal wall times.

Furthermore, a large fraction of the perplexity improvements mentioned happen when increasing the number of model layers by $4x$ - and as shown above, the wall clock time overhead is minimal. Making standard models $4x$ more deep to $50 - 100$ layers, will provide a large fraction of performance gains without much overhead.

### K DISCUSSION OF RELATIVE STRENGTH

In Equation 4, we discussed that the backward recurrence equation for PreLN can be written as

$$\sigma^2_{x_{\text{layer}},N} \approx (1 - \beta^2)^{2N} * \sigma^2_{x_{\text{layer}},0} + \frac{3}{4}(1 - (1 - \beta^2)^{2N})$$

Replacing $\beta^2 = \frac{k}{N^\alpha}$ and using $(1 + \frac{k}{N^\alpha})^N = e^{kN^{1-\alpha}}$, we get

$$\sigma^2_{x_{\text{layer}},N} \approx e^{2cN^{1-\alpha}} * \sigma^2_{x_{\text{layer}},0} + \frac{3}{4}(1 - e^{2cN^{1-\alpha}})$$
$$= e^{2cN^{1-\alpha}} * (\sigma^2_{x_{\text{layer}},0} - \frac{3}{4}) + \frac{3}{4}$$

Hence, the fall in gradient for $\beta^2 = \frac{k}{N^\alpha}$ is $\mathcal{O}(e^{kN^{1-\alpha}})$.

Similarly for PostLN, we can use Equation 7

$$\sigma^2_{g_{\text{out}},n-1} = (1 + \frac{\beta^2}{2}(r^l_{g_{\text{out}},n} - r^l_{x_{\text{in}},n}))\sigma^2_{g_{\text{out}},n}$$

$$(1 - \beta^2) * \sigma^2_{g_{\text{out}}, N} \leq \sigma^2_{g_{\text{out}}, n-1} \leq (1 + \beta^2) * \sigma^2_{g_{\text{out}}, N}$$

Hence, for N layers, the gradient fall/growth is again $\mathcal{O}(e^{\pm k N^{1-\alpha}})$.

## L  DISCUSSION OF APPROXIMATIONS AND ASSUMPTIONS

### L.1  ILLUSTRATIVE APPROXIMATIONS OF FULL FORMULAE IN MAIN PAPER

Some values listed in Table 1 are approximations/illustrative simplifications of their full closed forms in Appendix E and Appendix C. We discuss all of these below.

For ReLU forward correlation, we used a simple polynomial regression of the closed form formula. This simple regression is a remarkably good fit, as shown in figure Figure 12.

For layernorm, we ignored the factor of 1 compared to $d$, or $1/d$ compared to 1, assuming large enough hidden dimension $d$.

For SHA without V, we used the final simplified formulae for $\sigma^2_{x_{\text{out}}}$ and output correlation from Appendix C.8. For the gradient, we further simplified the formulae in Appendix C.8, assuming $L \approx L - 1$.

In Table 2, we applied a similar approximation as above for ReLU, from the full formula in Appendix E for output correlation. This polynomial approximation is also a very good fit, as shown in Figure 13.

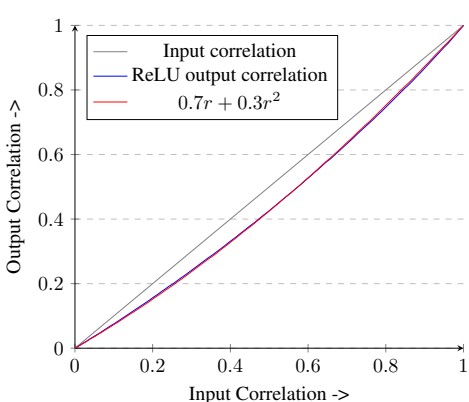

Figure 12: Approximation of the Relu forward correlation formula

Figure 13: Approximation of the FFN forward correlation formula, without dropout. Dropout will reduce the above correlation by $1 - p$.

### L.2  ASSUMPTIONS AND APPROXIMATIONS IN DERIVATIONS

Except for attention and softmax all other derivations of transformer components - Embeddings, FFN, ReLU/GeLU, Dropout, FFN Block are fully exact, assuming only normal distribution of inputs, weights and gradients. Furthermore for LayerNorm and softmax, we only add the assumption that the sequence length/hidden dimension is large.

For simplification of these formulae when doing empirical analysis, we used additional assumptions. For embeddings, we assumed Zipf's law to calculate initial input correlation in tokens, as well as assumed uniform distribution for segment lengths for next sentence prediction task of BERT. Note that this assumption is not strictly required, and can also be empirically observed and given as input to our method.

For LayerNorm, we assume the hidden dimension is large, $d >> 1$ (so that we can ignore factors related to $d - 1$ or $1/d$). For softmax, we assume sequence length $L$ is large so that the sum of

log-normals can be written as another log-normal. The single-head attention requires some more assumptions as are listed in Appendix C.8.

## M    DEEPSCALELM PSEUDOCODE

```
## Define constants for scaling residual and output
λ² = 1 − 2/N  ;  β² = 2/N
## Define constants for embedding and FFN block
σ²_e = (1−p)/3  ;  σ²_f = 1/d * √((1−p)/2)

## Scale skip connection and block output
def add_skip(x, f(x)):
    return λ * x + β * f(x)

## Find layerwise input correlation upto N layers
def corr_input_layerwise(r, N):
    r_N = []
    for i in range(N):
        r = λ² . r + β²(1 − p)
        r = λ² . r + β²(1 − p)(r^l_{x_in} + ((1 − (r^l_{x_in})²)^{0.5})/π − (r^l_{x_in} cos^{-1}(r^l_{x_in}))/π)
        r_N.append(r)
    return r_N

## Define constants for attention block
σ²_{l,o} = 1/d * √((1−p)/r^{l,n}_{x_in}) ;  σ²_{qk} = 1/d  ;  r = r^l_{x_in}
where r^{l,n}_{x_in} = corr_input_layerwise(r, N)[n]

## Stable initialization of weights
def dslm_init(w, l):
    if w is ['ffn']:
        nn.init.normal_(w, gain = σ_f)
    elif w is ['v_proj', 'out_proj']:
        nn.init.normal_(w, gain = σ_{l,o})
    elif w is ['q_proj', 'k_proj']:
        nn.init.normal_(w, gain = σ_{qk})
    elif w is ['embd']:
        nn.init.normal_(w, gain = σ_e)
```

Figure 14: Pseudo-code for our proposed method DeepScaleLM: We scale the block output and the skip connection before adding, and keep track of correlation across layers. We appropriately initialize the weights. ($N$: num of layers, $d$: model hidden dimension, $p$: dropout probability, $r^l_{x_{in}}$ is calculated based on expressions provided in subsection C.1.)

```
## Define constants of DeepScaleLM
λ² = 1 − 2/N  ;  β² = 2/N
σ²ₑ = (1−p)/3  ;  σ²qk = 1/d  ;  σ²f = 1/d * √((1−p)/2)

## Scale skip connection and block output
def add_skip(x, f(x)):
    return λ * x + β * f(x)

## Stable initialization of weights
def init(w):
    if w is ['ffn', 'v_proj', 'out_proj']:
        nn.init.normal_(w, gain = σf)
    elif w is ['q_proj', 'k_proj']:
        nn.init.normal_(w, gain = σqk)
    elif w is ['embd']:
        nn.init.normal_(w, gain = σe)
```

Figure 15: Pseudo-code for simplified version of our DeepScaleLM method.

## N  HYPER-PARAMETERS

We used Megatron-LM's default BertWordPieceLowerCase tokenizer, with the original BERT lower-cased vocab, and with trainable position embeddings. The same hyper-parameters (including LR schedule, warmup) were used for all models, and LR search over the range below was performed for all models. The final best models always had optimal LR within the range and not at the boundary of the LR range for all of our experiments.

Table 21: Training Hyper-Parameters. We use all original hyper-parameters of BERT, except for learning-rate(LR).

| Parameters | Values |
|---|---|
| Optimizer | Adam |
| $\beta_2, \beta_2$ | 0.9, 0.999 |
| Effective Batch Size | 256 |
| Drop-out ($p$) | 0.1 |
| Sequence Length | 256 |
| Train Iters | 100,000 |
| Num GPUs | 8 |
| Learning rate | [1, 3, 5, 7, 10]*$10^{-4}$ |
| Schedule | Linear |
| LR Decay Iterations | 98% |
| Warmup steps | 1% |
| Min LR | $1 * 10^{-5}$ |
| Gradient clipping | 1.0 |
| Batch Size / GPU | 2 |
| Grad Accum Steps | 16 |

