# OpenReview forum: "Transformers Get Stable: An End-to-End Signal Propagation Theory for Language Models"
_ICLR.cc/2024/Conference — Submitted to ICLR 2024_

### Official Review · Reviewer_PFEM · 2023-10-18

**Soundness:** 3 good
**Presentation:** 2 fair
**Contribution:** 1 poor
**Rating:** 3
**Confidence:** 4

**Summary:**

Despite their success, scaling transformer models in depth remains challenging. This work introduces formulas governing signal moments in transformers, offering a unified signal propagation theory. In this paper, the proposed framework aids in addressing issues like vanishing/exploding gradients, rank collapse, and instability from high attention scores. We also propose DeepScaleLM, an initialization and scaling method conserving output/gradient moments, enabling deep model training. The proposed method improve deep narrow Bert's perplexity by 1.0 point and downstream task performance by 2.2 points compared to shallow models across various sizes, even outperforming larger shallow models with half the parameters.

**Strengths:**

1. Simple idea and it makes intuitive sense.
2. Creative combinations of various existing techniques.

**Weaknesses:**

1. Lack of novelty: scaling Residual/Skip-Connection is a well-known trick to stabilize training of deep neural networks, very similar ideas can be found at [1, 2]
2. Similar signal propagation idea is presented in [3]
3. The results are not surprising [4] shows exactly the same conclusion with similar model configs.
4. The idea of preventing rank collapse has been thoroughly explored in [5,6]
5. Experiments on downstream tasks seem lack of diversity. More downstream tasks with different characteristics should be included.
6. Results on more modern architectures beyond Bert should be included to present a convincing argument.

Overall, most of the tricks are already well-studied and published, the paper generally feels incremental and results are not surprising. In combination with weak empirical studies on models with trivial sizes, this paper doesn't seem to be significant enough to be presented at the ICLR venue.

[1] Kai, Hu, et al. "Is normalization indispensable for training deep neural networks?." (Neurips 2020)
[2] Bachlechner, Thomas, et al. "Rezero is all you need: Fast convergence at large depth." Uncertainty in Artificial Intelligence. PMLR, 2021.
[3] He, Bobby, et al. "Deep transformers without shortcuts: Modifying self-attention for faithful signal propagation." arXiv preprint arXiv:2302.10322 (2023).
[4] Xue, Fuzhao, et al. "A Study on Transformer Configuration and Training Objective." (ICML 2023).
[5] Zhai, Shuangfei, et al. "Stabilizing transformer training by preventing attention entropy collapse." International Conference on Machine Learning. PMLR, 2023.
[6] Zhou, Daquan, et al. "Deepvit: Towards deeper vision transformer." arXiv preprint arXiv:2103.11886 (2021).

**Questions:**

1. What happens when the same method is applied on decoder only architecture?
2. Does the shallow network with same parameters run faster or slower in terms of wall time? What is the benefit of using deeper and narrow config beyond marginal improvement in perplexity?
3. The results of the pretraining experiments seem off, can you please cite credible sources on numbers with similar config pretrained on Pile-CC?
4. On the parameter counts, do you count the embedding parameter size when reshaping the networks in the experiments, which could potentially be an unfair comparison?

---

> ### Author Response · Authors · 2023-11-20
> **Response to Reviewer PFEM: Part 1/2**
>
> ### **For more Finetuning Experiments, modern architectures and new results**:
> Please see common response.
>
> ### **Regarding rank collapse work in [5,6]**:
> We discussed in section 3.4 and B.4 that [5] ($\sigma$Reparam) is one of the many works that have reported/studied rank collapse, and that they propose scaling all the linear weight matrices in the model by their spectral norm. Our method theoretically prevents rank collapse at initialization, whereas [5] empirically finds that bounded weights of attention will prevent rank collapse later during training. Similarly, [6] is also one of the papers we mentioned in 3.4 and B.4, and is again a method that targets attention collapse that occurs dynamically during training, and not at initialization.
>
> Both these methods will not prevent rank collapse at initialization caused by the very structure of the transformer model, in particular increase in correlation caused by both attention and ReLU/GeLU as shown by our formulae. This is clearly demonstrated by the fact that [5] was unable to train a 200L post-LN model with $\sigma$ Reparam.
>
> Both these methods are orthogonal to our method, to further stabilize training dynamics of the model - though in none of the training of our models did we observe divergence, even upto 768 layers (which is significantly deeper than the maximum depths studied in these works).
>
>
> ### **Regarding scaling in prior works**:
> Scaling the residual/skip connection is one of the features we use in DSLM to stabilize the model, and it indeed has a long history, as we discuss in B.3. However, as our ablations of our initialization in Table 10, and comparisons to DSInit (Table 5) shows, it is crucial to account for the exact constants in model variance propagation.
>
> Without these constants derived using our formulae (Table 1-2, 11-16), the models perform significantly worse.
>
> ### **Regarding ‘similar signal propagation idea presented in [3]’**:
> As we discuss in our related works (Section B.2 Paragraph 3), [3] (Deep transformers without shortcuts) assumes MLP to be linear in the effect it has on attention. Our analysis does not make any such assumptions about the interactions between various model components, and in particular, we use closed-form expression for the output correlation of GeLU (the nonlinearity used in [3]). In particular, our formulae can be used to show that an MLP block with GeLU will also increase correlation, in the absence of dropout (the same setting as used in [3]). As such, at deeper depths, the method in [3] will still exhibit rank collapse.
>
> Furthermore, their 72 and 108 layer models underperform compared to a 36 layer model with same hidden dimension, in spite of having many times extra parameters - this clearly highlights deficiencies in their modeling of signal propagation, in particular the impact of non-linearities.
>
> ### **Regarding similar observations as in [4]**:
> [4] (A Study on Transformer Configuration and Training Objective (Bamboo)) shows that Masked autoencoder objective may help stabilize a vanilla narrow-but-deep transformer model. Their configurations are vanilla transformers without any modifications, similar to our “baseline” models. The performance of their model falls at 96 layers, and they found 48 layers to be optimal.
>
> As can be seen in Table 4, we also observe this drop in performance, for both vanilla Pre-LN and Post-LN models for our experiments on 165M models, where deeper models (such as 192 layers) underperform. Using our method however, one can train much deeper models (such as 192 or 384 layer models) which not only converge, but outperform the vanilla transformers of [4]. These differences are even more pronounced for larger models, where vanilla post-LN BERT from [4] did not converge for deeper models.
>
>
> ### **Regarding Wall Clock Time and  benefit of using deeper and narrow config**:
> As we detailed in Section 4.4 and Appendix J, our deeper models have small overheads in compute and wall clock times compared to shallow models. The benefits of using deeper and narrower configs with our method are -
> 1. Improved perplexity - allowing 165M model to outperforms the baseline 330M model with twice the params. At equal params, we achieve 1.0 improvements in perplexity.
> 2. Large improvements in downstream performance - Our deeper configs outperform original models by 2.2 points on RACE and 1.0 points on MNLI. The 165M model again outperforms original model with twice the params
> 3. Improved model quantization - Similar to Unit Scaling [1], our method results in models which lose much less performance, when quantized (via direct casting) to FP8 precision compared to original models, allowing our model to be served with much less GPU VRAM requirements.

---

> ### Author Response · Authors · 2023-11-20
> **Response to Reviewer PFEM: Part 2/2**
>
> ### **Regarding results of the pretraining experiments compared to other works**:
> See common response to all reviewers regarding model performance.
> Regarding counting the embedding parameter size when reshaping the networks:
> We do not count the embedding parameter size when reshaping the model for fair comparison. As our thinner-deeper models have smaller hidden dim, their embeddings are much smaller - for example, the 192L-256D model has 25M fewer params than the baseline 12L-1024D model, and still outperforms them.
>
> ### **Regarding pre-training performance**:
>
> Original BERT model was trained for 128B tokens, and its performance was reported in Table 6 in Devlin et al. Our models are trained for Chinchilla optimal 3B/6B tokens, 40-20x fewer than the original BERT pre-training tokens. Also, the original Devlin et al models are trained on Book Corpus and Wikipedia, which are much more cleaner datasets than Pile-CC, which is derived from Common Crawl.
>
> Our baseline model are trained for 1e-19 flops, and 165M models for 2.5e-18 flops. As can be seen in figure A5 of Chinchilla paper, their experiments predict a loss of approximately 3.2 to 2.8 given this compute budget - which our models clearly outperform.  Our GPT models achieve better performance than this at a loss of $ln(11.6)=2.45$, our 330M BERT achieves a loss of $ln(13.2)=2.58$, and the 165M model achieves $ln(14.2)=2.65$. Furthermore in [1], for a 32-layer 1024D model trained on C4 (another common crawl derived dataset) for 800M tokens, they report a perplexity of 24.7 for vanilla pre-LN transformer. At the same number of tokens, our baseline GPT 24L 1024D model was at 11.8 PPL.
>
> We would like to reiterate that we used all the original hyper-parameters of BERT/GPT, and our baseline models directly used the original Megatron-LM codebase. Furthermore, we did a sweep of LR to find the best LR. We will release the training scripts to enable direct reproduction/comparison of our training.
>
> [1] He, Bobby, et al. "Deep Transformers without Shortcuts: Modifying Self-attention for Faithful Signal Propagation." The Eleventh International Conference on Learning Representations. 2022.
>
> ### **Regarding Downstream performance**:
> We add the performance of our model on the MNLI task as well (see common response). Figure 5 in Devlin et al. provides the expected MNLI dev accuracy based on the number of pretraining steps. From Figure 5 of BERT paper, this would correspond to around 25k steps and the expected accuracy is under 80% for 110M Bert Base. Our models achieve better accuracy than expected based on this Figure.

---

> ### Author Response · Authors · 2023-11-22
> **Request for Rebuttal Feedback**
>
> Dear Reviewer,
>
> We would like to thank you for reviewing our paper. With less than 1 day left, we would appreciate if you could provide feedback regarding whether our response has addressed your concerns. Kindly let us know if there are any other details you would like us to clarify!
>
> Best regards,
> The Authors

---

> > ### Comment · Reviewer_PFEM · 2023-11-22
> > **Thank you for your responses**
> >
> > I have read the responses and do not have further questions. I still found the experiment results not convincing and in combination with lack of novelty, I can not recommend acceptance.

---

### Official Review · Reviewer_wRZy · 2023-10-30

**Soundness:** 3 good
**Presentation:** 3 good
**Contribution:** 3 good
**Rating:** 6
**Confidence:** 2

**Summary:**

The author introduces a theory to understand unstable issues in deep transformers and suggests a solution called DeepScaleLM. Their experiments show the effectiveness and superior performance of this method.

**Strengths:**

1. The authors present a novel theoretical analysis concerning the moments of transformer models.
3. Their development of an effective, theory-driven approach is both sound and provides valuable insights.
3. They conducted comprehensive experimental exploration to support their theory.

**Weaknesses:**

While I am not well-versed in the experimental section, I would like to point out certain aspects that I found challenging or unclear during my reading.

1. In Figure 2, there seems to be a discrepancy. The author mentions that the backward gradient variance rises hyperbolically with N, but the depicted curve suggests a decline as N grows. This is somewhat perplexing.
2. The representation and caption for Figure 5 lack clarity, making it challenging to decipher the conveyed information.
3. For Table 4 and Figure 7, it would be beneficial if the author could elucidate why the thinnest and deepest transformers utilizing DSLM yield the most optimal results.

**Questions:**

See weakness

---

> ### Author Response · Authors · 2023-11-20
> **Response to Reviewer wRZy**
>
> ### **Regarding gradient variance growth discrepancy in Figure 2**:
> We would like to clarify Figure1-4. These figures depict the output (forward) and gradient (backward) variance after the $N^{th}$ layer, for a single model with 192 layers. The layers with higher numbers are near the output, lowest numbers are near the input, with $N$ denoting the current layer. As can be seen from Figure 2, as the gradient is backpropagated to shallower layers, it increases hyperbolically as a function of the current layer number $N$.
>
> We should have used $n$ in Figure 1-4 to ensure consistency with our derivation in Appendix H. We will add the above clarifications to Figure 1-4. We will release the scripts to reproduce these figures 1-4.
>
> ### **Regarding lack of clarity in Figure 5**:
> Figure 5 depicts the input correlation $r^l_x$ (along x axis), vs the output correlation (along y axis) of a transformer layer, for both FFN block (Red line) and Attention Block (Blue line). The grey line is the $y=x$ line, added to show more clearly that the FFN block decreases correlation (compared to its input) after  $r^l_x > 0.65$, and that the Attention block always outputs correlation 0.9. These output correlations are plotted using Formulae for $r^l_{x_{out}}$ from Table 2.
>
> This figure shows that with dropout, rank collapse (i.e., a situation where $r^l_x \approx 1$) will not occur in a standard transformer. This result is then empirically verified in Figure 6, where it can be clearly seen that no rank collapse is observed, contrary to the findings of Noci. et. al. 2022, as further discussed in Appendix G. We will explain these figures more clearly in the manuscript.
>
> ### **Regarding explanation about why the thin/deep transformers utilizing DSLM yield the most optimal results (Table 4 and Figure 7)**:
> Table 4 and Figure 7 show that standard transformers should be much more deep, at the expense of width - for example for 165M params, 48-192 layers deep, and for 330M params, 96-384 layers deep. Montúfar et al. (2014); Raghu et al. (2017) show that the complexity of Deep Neural Nets increases polynomially with width and exponentially with depth. Given a fixed parameter budget, there is a tradeoff between having richer representations from more deep layers stacked one after another, versus the representation capacity of model width.
>
> Note that the 768 layer model underperforms compared to the 192 model - after a very large depth, the width becomes too narrow for the model to have enough representation capacity. The thinnest and deepest model (768 layer) does not perform the best - But our experiments show that, with correct initialization and scaling, the optimal depth is much more deeper (and the width narrower) than is usually used in standard models.

---

> ### Author Response · Authors · 2023-11-22
> **Request for Rebuttal Feedback**
>
> Dear Reviewer,
>
> We would like to thank you for reviewing our paper. With less than 1 day left, we would appreciate if you could provide feedback regarding whether our response has addressed your concerns. Kindly let us know if there are any other details you would like us to clarify!
>
> Best regards,
> The Authors

---

> > ### Comment · Reviewer_wRZy · 2023-11-22
> > **Response to Rebuttal**
> >
> > Thanks for your response. After reading the rebuttal and the other reviews,  I decided to keep my original score.

---

### Official Review · Reviewer_eeiy · 2023-11-07

**Soundness:** 3 good
**Presentation:** 3 good
**Contribution:** 2 fair
**Rating:** 5
**Confidence:** 4

**Summary:**

This manuscript studies signal propagation in transformer networks
to study difficulties in training deep transformer networks.
From the analysis, an initialisation method is proposed to facilitate learning.
The theoretical results are verified empirically on language modelling tasks.

**Strengths:**

- (significance) Enabling the training of deeper architectures typically leads to improved performance on a variety of tasks.
   At least historically, this kind of result has proven extremely impactful.
 - (clarity) The paper is well written and easy to follow.
   I especially liked how Table&nbsp;1 provides an overview over signal propagation through the different building blocks.
 - (originality) This is the first work that performs such a thorough signal propagation analysis for transformer models.
 - (quality) The derivations in the appenix provide evidence for the theoretical claims.

**Weaknesses:**

- (originality) A very similar idea for regular networks has been presented in (Arpit et al., 2016).
   The connections with this work should definitely be discussed.
 - (originality) By moving all related work to the appendix and not citing much in the main text it becomes unclear which concepts are new and which already exist.
   I did notice the references to the appendix section, but I would argue that a citation puts more emphasis on the fact that something already exists and is not a contribution of this work.
   Citing one (seminal) paper for each aspect should suffice to counter this issue.
 - (clarity) In Section&nbsp;4.4 an initialisation that renders a model more linear is claimed to be bad for performance.
   However, there is quite a bit of work where this kind of linearity is considered desirable (e.g. Hardt &amp; Moritz, 2017; Zhang et al., 2019)
 - (clarity) I am not sure if it makes sense to report perplexity in the context of masked language modelling.
   I am no expert in NLP, but I thought perplexity is only meaningful for autoregressive language models.
 - (quality) The baseline performances seem to be remarkably weak.
   Table&nbsp;6 in (Delvin et al., 2019) reports perplexities in the range 3-5.
   Results for GPT models on the pile go below 1.
   Also, Yang et al. (2019) report results in the range of 70-76% accuracy for BERT on RACE.
 - (significance) The results in this paper can not be directly applied to vision transformers.
   It could be emphasised a little stronger in the main paper that the analysis focuses on text inputs.
 - (significance) There are too little direct comparisons with competing methods (e.g.&nbsp;Zhang et al., 2019; Noci et al., 2022; Wang et al., 2022a; He et al., 2023).
   Apart from Table&nbsp;5, these alternative methods seem to be ignored completely.
   Also, it would be easier to compare if some of the experiments from these papers would have been adopted.
 - (significance) The experiments all seem to use the same architecture.
   Ideally, an initialisation method works for a variety of architectures.
   This is never properly tested.
 - (quality) Experiments do not have error bars.
   Especially for a random initialisation strategy, error bars would be helpful to assess how consistent the improvements are.
   If error bars would make the experiments prohibitively expensive, it would be nice to include at least one small-scale experiment to provide some insights on the variability of the proposed method.

### Appendix

 - (quality) The propagation of correlation between samples was introduced by (Poole et al., 2016), not by (Schoenholz et al., 2017).
 - (quality) It seems like two fundamental signal propagation papers are missing in the related work.
   The foundations for signal propagation analysis can be found in (Neal, 1996) and a popular reference is (LeCun et al., 1998).
 - (quality) In Section&nbsp;B.2§1, it is claimed that this work considers expectations over inputs, but the expectations are over inputs AND weights (cf.&nbsp;Glorot &amp; Bengio).
 - (clarity) Section&nbsp;B.2§4 states that non-IID inputs are "also" accounted for, but I would argue that this is the only case that is accounted for.
 - (clarity) It is unclear what computation is contained in the embedding component.
   Originally, I suspected this to be only about the token embeddings, but it seems to include other computations as well.
 - (quality) The approximation for the correlation in the embedding layer seems to be missing the Euler constant:
   $$\sum_{i=1}^{|V|} p_i^2 = \frac{\sum_{i=1}^{|V|} 1 / i^2}{\Big(\sum_{i=1}^{|V|} 1 / i\Big)^2} \approx \frac{\zeta(2)}{(\ln |V| + \gamma)^2}.$$
   Without the constant, the approximation is pretty bad.
   Also, it could be made clearer in the derivation where this approximation comes from and that the final simplification step assumes large $L$.
 - (originality) It should be more clearly stated for each derivation where it can be found in literature.
   In its current form, it is hard to distinguish which derivations are really new.
 - (clarity) Instead of using the identity $\pi - \arccos(x) = \frac{\pi}{2} + \arcsin(x)$, it would be better to use $\pi - \arccos(x) = \arccos(-x)$ to stay closer to the formulation from (Cho &nbsp; Saul, 2009; Daniely et al., 2016).
   Similarly, the GELU variance can be further reduced to $$\frac{\sigma^2}{2 \pi} \bigg(\arccos\Bigl(\frac{-\sigma^2}{1 + \sigma^2}\Bigr) + \frac{2 \sigma^2}{(1 + \sigma^2) \sqrt{2 \sigma^2 + 1}} - \frac{\sigma^2}{1 + \sigma^2}\bigg).$$
 - (clarity) The derivation of LayerNorm could use some more explanation.
   Also, the dependencies between samples, sample mean and sample variance should be discussed a bit more.
   Finally, the affine transformation that is typically included after each normalisation layer is not addressed at all.
 - (clarity) It should be more clear what the limitations of the different approximations for the softmax derivation are.
   A quick numerical check verifies that the approximation breaks down quite quickly for larger variances.
   Although this is probably not practically relevant, it would be nice to provide some insights when the analysis breaks down.
 - I noticed that the derivation for scaled dot-product attention starts with some rough shortcuts.
   Also, a completely different approach is used to handle the softmax.
   This seems suspicious.
   I do not have time to check the math any further, but the numerical results should provide enough proof that these results, if not correct, are at least useful.

### Minor Comments

 - Citations could be polished a bit more (e.g. "Deep Information Propagation" has been published at ICLR, "ReZero is all you need" has been published at ICML, ...).
   I also noticed not some references have a link while others do not have an URL.
 - possible typos in Section&nbsp;1§5: "issues with very deep transformerS",
   in Section&nbsp;1§6: "ensures the moments of outputs and gradients (to) remain fully conserved"

### References

 - Neal, R. M. (1996).
   Bayesian Learning for Neural Networks.
 - LeCun, Y., Bottou, L., Orr, G. B., & Müller, K.-R. (1998).
   Efficient BackProp.
   Neural Networks: Tricks of the Trade (1st ed., pp. 9–50).
 - Cho, Y. &amp; Saul, L. (2009).
   Kernel methods for deep learning.
   Advances in neural information processing systems, 22.
 - Arpit, D., Zhou, Y., Kota, B., & Govindaraju, V. (2016).
   Normalization Propagation: A Parametric Technique for Removing Internal Covariate Shift in Deep Networks.
   Proceedings of The 33rd International Conference on Machine Learning, 48.
 - Daniely, A., Frostig, R., &amp; Singer, Y. (2016).
   Toward Deeper Understanding of Neural Networks: The Power of Initialization and a Dual View on Expressivity.
   Advances in Neural Information Processing Systems, 29.
  - Poole, B., Lahiri, S., Raghu, M., Sohl-Dickstein, J., &amp; Ganguli, S. (2016).
   Exponential expressivity in deep neural networks through transient chaos.
   Advances in Neural Information Processing Systems, 29.
 - Hardt, M., &amp; Ma, T. (2017).
   Identity Matters in Deep Learning.
   International Conference on Learning Representations, 5.
 - Yang, Z., Dai, Z., Yang, Y., Carbonell, J., Salakhutdinov, R. R., &amp; Le, Q. V. (2019).
   Xlnet: Generalized autoregressive pretraining for language understanding.
   Advances in neural information processing systems, 32.

**Questions:**

1. Please, add citations to make clear which parts of this work are not part of the contribution.
 2. Please, connect the proposed initialisation to the work from (Arpit et al., 2016).
 3. How is perplexity computed for masked language modelling and why does it make sense as a metric?
 4. Do you have any references or other explanation for the seamingly poor baseline results?
 5. How easy or difficult would it be to apply the analysis to vision transformers or transformers that do not work with language/embeddings?
 6. How does deepScaleLM compare to other (non-standard) initialisation strategies and deep transformers?
 7. How does deepScaleLM perform on different architectures (e.g. GPT, long-context transformers, ...)?
 8. Would it be feasible to include error bars for some of the results?
 9. How can the numerical results be so good if the approximation for the Embedding layer correlations is significantly off?
    Is correlation not that important after all?
 10. Why are the insights from Section&nbsp;C.7 not reused for the analysis in Section&nbsp;C.8?

---

> ### Author Response · Authors · 2023-11-20
> **Response to Reviewer eeiy : Part 1/4**
>
> ### **Regarding applying DSLM to Vision Transformers**:
> Kindly refer to common response.
>
> ### **Regarding using the same architecture for the initialisation method**:
> Please see common responses for results with decoder-only (GPT) architecture.
>
> ### **Regarding comparison with Arpit et al., 2016**:
> Normalization Propagation (Arpit et al., 2016), proposes theoretical value for the batch statistics of mean and variance after ReLU, and uses these to normalize the output after each Relu. Their theoretical formulae for $\mu_x$, $\sigma^2_x$, and $\sigma^2_g$ are the same as in this work. This normalization successfully conserves the mean and variance of the forward propagation of the model as zero mean and unit variance.
>
> However, this results in the backward gradient variance increasing by $0.5*\frac{1}{\sigma^2_x} = 1.47$, and to counteract this, they scale the inputs before applying relu by $\sqrt{\frac{1}{1.47}}$, and this successfully conserves the gradient as unit variance.
>
> If our understanding of Arpit et al 2016 is correct, this scaling operation of the input results in the mean and the variance no longer being conserved - something seemingly ignored in Arpit et al. In fact, the variance falls to $30\%$ of initial value after just 4 layers. Applying an analysis similar to Klambauer et al. 2017 shows that Arpit et al.’s method converges to a fixed stable point of approximately $\mu=-0.34$ and $\sigma^2_x=0.24$, as we confirmed using numerical simulations. While we skip this proof here for brevity, kindly let us know if you would like us to expand on its proof and simulation results.
>
> Furthermore, Arpit et. al does not handle attention, which we show requires careful consideration of covariances.
>
> ### **Regarding reformatting the Related Works section**:
> Since our work uses proper initialization, residual scaling and signal propagation, we believed a detailed discussion of related works covering all these topics was essential. We felt that a short related works section in the main paper would not do justice to the large number of prior works we wanted to discuss. As such, we dedicated 2 pages to the related works, and it was difficult to have this entire section in the main paper due to space constraints. Instead, throughout our main paper, we tried to refer to prior works in-line. Being cognizant of the importance of this section, we made it the first section of our appendix. We will add a summarized related works section in the main paper as suggested. We had missed Poole et al., 2016 and the seminal references (Neal, 1996, & LeCun et al., 1998), thank you for pointing it out. We will add these to our manuscript.
>
>
> ### **Regarding “linearity” being bad for performance**:
> Hardt & Moritz, 2017 suggests that a network should be able to express the identity transformation, in order to optimize well. We would like to clarify that by a “more linear” network, we are referring to a network which behaves almost like a single linear projection during training, because the residuals are negligibly small. As a thought experiment, if $\beta$ is reduced to a very small value (such as 1e-20), and assuming layernorm gain is fixed to 1, the transformer network will return the same output as input - and even training for a long time will not affect the model’s output significantly. As [1] shows, there is an expressivity-trainability tradeoff in training deep networks, and while having lower $\beta$ will result in networks whose gradients don’t explode/vanish, they will converge slowly/suboptimally.
>
>
> [1] Yang, Ge, and Samuel Schoenholz. "Mean field residual networks: On the edge of chaos." Advances in neural information processing systems 30 (2017).
>
> ### **Regarding pre-training performance**:
>
> Original BERT model was trained for 128B tokens, and its performance was reported in Table 6 in Devlin et al. Our models are trained for Chinchilla optimal 3B/6B tokens, 40-20x fewer than the original BERT pre-training tokens. Also, the original Devlin et al models are trained on Book Corpus and Wikipedia, which are much more cleaner datasets than Pile-CC, which is derived from Common Crawl.
>
> Our baseline model are trained for 1e-19 flops, and 165M models for 2.5e-18 flops. As can be seen in figure A5 of Chinchilla paper, their experiments predict a loss of approximately 3.2 to 2.8 given this compute budget - which our models clearly outperform.  Our GPT models achieve better performance than this at a loss of $ln(11.6)=2.45$, our 330M BERT achieves a loss of $ln(13.2)=2.58$, and the 165M model achieves $ln(14.2)=2.65$.
>
> We would like to reiterate that we used all the original hyper-parameters of BERT/GPT, and our baseline models directly used the original Megatron-LM codebase. Furthermore, we did a sweep of LR to find the best LR. We will release the training scripts to enable direct reproduction/comparison of our training.

---

> ### Author Response · Authors · 2023-11-20
> **Response to Reviewer eeiy : Part 2/4**
>
> ### **Regarding Downstream performance**:
> We add the performance of our model on the MNLI task as well (see common response). Figure 5 in Devlin et al. provides the expected MNLI dev accuracy based on the number of pretraining steps. From Figure 5 of BERT paper, our models would correspond to around 25k steps and the expected accuracy is under 80% for 110M Bert Base. Our models achieve better accuracy than expected based on this Figure.
>
> ### **Regarding Perplexity**:
> The confusion regarding weak performance arose from two different definitions of perplexity. We calculate perplexity as the exponential of the model’s loss, (as provided in Megatron-LM paper[1] and code [link](https://github.com/NVIDIA/Megatron-LM/blob/443ce9f3f98fdc5a53c6b480c6e21b79944d198e/megatron/training.py#L975). Regarding GPT perplexity < 1, we estimate that the reviewer is referring to bits-per-byte as perplexity, whereas we calculate on a token level. In hindsight, we should have clarified this in the main paper. The chinchilla paper (in Figure A5) shows expected loss given our compute limit of 1e-19 flops at approx 2.8, and our GPT models achieve better performance than this.
>
> Furthermore in [2], for a 32-layer 1024D model trained on C4 (another common crawl derived dataset) for 800M tokens, they report a perplexity of 24.7 for vanilla pre-LN transformer. At the same number of tokens, our baseline 24L 1024D model was at 11.8 PPL.
>
> [1] Shoeybi, Mohammad, et al. "Megatron-lm: Training multi-billion parameter language models using model parallelism." arXiv preprint arXiv:1909.08053 (2019).
>
> [2] He, Bobby, et al. "Deep Transformers without Shortcuts: Modifying Self-attention for Faithful Signal Propagation." The Eleventh International Conference on Learning Representations. 2022.
>
>
> ### **Regarding consistency of improvements and error bars**:
> In our initial experiments, we observed very little variation in performance across different runs - we conjecture that the model is trained on a large enough number of tokens for differences in initialization/data seed to not matter. We provide numbers below for 12L-1024D Post-LN and DSLM models from Table 4 below -
>
> | Model | Mean | Standard Error |
> | ------ | ------ |  ------ |
> |Post-LN Baseline |14.33 | 0.14 |
> | DSLM | 15.56 | 0.08  |
>
> As the variation was so small, and due to compute limitations, we did not run multiple runs for other experiments thereafter. We also reported the best score for Baseline Post-LN, and the worst score for DSLM for the 12L-1024D models Table 4 for a conservative comparison. We will add these details in the paper.
>
> ### **Regarding non-IID inputs being the only case that is accounted for in  B.2**:
> Our exact formulae for blocks and components also account for IID cases - as can be verified by our simulations, in which we do cover cases IID inputs with exactly 0 correlation, as noted in $Corr^l_{x_{in}}$ column in Table 18. In the simplified formulae for Table 19, and in DeepScaleLM initialization and model, we simplified our formulae so that they only remain accurate for non-IID inputs. This was because of three considerations:
> 1. In NLP domain, most text will inevitably be non-IID due to repeated common words. This was encountered in all our experiments.
> 2. Even in Vision, for ViT in particular, there will be correlation among pixel intensities across patch embeddings, as discussed in common response section.
> 3. Even if there is exactly 0 correlation in input, the very first attention layer and the first FFN layer in particular, will add correlations to the output, ensuring our simplified formulae hold reasonably accurately.
>
>
> ### **Regarding the computation contained in the embedding component**:
> We will update our paper with clear description and details of the embedding component. Our embedding component is same as that of the original BERT model, except for position embeddings. The embeddings component of BERT consists of 3 look up tables - token embeddings, position embeddings, and segment embeddings. For a given input token, each of these 3 embeddings are added before being passed to the transformer model. We describe each of these below -
>
> - Word Embeddings - A lookup table mapping input word ids, of size (Vocabulary X Hidden_dim). Output denoted by $x_{out_{we}}$ in Section 3.1.
>
> - Position Embeddings -  A lookup table mapping input positions, of size (Sequence_length X Hidden_dim). We use the more commonly used trainable position embeddings (compared to BERT’s original sinusoidal embeddings)
>
> - Segment Embeddings - A lookup table mapping segments (denoting Sentence A or Sentence B for Next Sentence Prediction task of BERT), of size (2 X Hidden_dim). Output denoted by $x_{out_{se}}$ in Section 3.1.

---

> ### Author Response · Authors · 2023-11-20
> **Response to Reviewer eeiy : Part 3/4**
>
> ### **Regarding missing the Euler constant in the approximation for the correlation in the embedding layer**:
>
> We would like to apologize for a minor error here, where we wrote our predicted $r^l_{x_{in}} = 0.247$. In writing this value for the paper, we had mistakenly used Log to the base 10, instead of natural logarithm. Using natural logarithm gives a predicted value of 0.2273 (instead of the 0.247), which is within 3% of the empirical correlation 0.221. Adding the euler’s constant term gives an even closer value of 0.2267, but not significantly different from our simplified version. In our experiments, we had used the natural logarithm value of 0.227.
>
> We had skipped the Euler constant, as it was relatively small (0.58) compared to $Log_e(|V|)=10.37$ for our vocabulary of size 32000. We will clarify this, as well as mention the simplification from large L.
>
> ### **Regarding stating which formulae can be found in literature**:
> We will more clearly specify which formulae (or equivalent forms) cannot be found in prior literature, and also color-code Tables 1, 11-16 for the same.
>
> ### **Regarding more explanation in the derivation of LayerNorm**:
> We will add further explanations to the LayerNorm derivations. The affine transformation for layernorm are typically initialized with 1 scale and 0 bias, hence do not change any of our derivations and were hence ignored. We will clarify this in our paper.
>
> ### **Regarding limitations of different approximations**:
> We will add a discussion on the effective range of softmax approximation. Our approximations become inaccurate for larger variances, particularly when softmax starts to “degenerate” to a single dimension 1 and others 0. It is reasonably accurate in more realistic ranges however.
>
> ### **Regarding the proof of scaled dot-product attention being different from softmax**:
> We had initially attempted a much more simpler proof, where one assumed attention scores to be independent of values. This then enabled the use of our previous LogNormal-based softmax derivation to easily derive the forward variances. But the theoretically predicted values strongly disagreed with empirical values from simulations. This is because SHA is -
> $\mathrm{Dropout}(\mathrm{SoftMax}(\frac{\mathbf{X_{\text{in}}}\mathbf{W_Q}\mathbf{{W_K}^T}\mathbf{X^T_{\text{in}}}}{\sqrt{d_k}}))\mathbf{X_{\text{in}}}\mathbf{W_V}$ , and the $\mathbf{{W_K}^T}\mathbf{X^T_{\text{in}}}$ term cannot be treated independently of the $\mathbf{X_{\text{in}}}\mathbf{W_V}$ term.
>
> A simple verification of this can be checked by simply simulating $(XW)^T*X$, and verifying that the variances of the results do not match that of $L * \sigma^2(XW)$, but do if the second X is replaced by another random tensor.
>
> This necessitates an alternate methodology to derive SHA, where the components are treated as a unified whole.
>
> ### **Regarding using perplexity for masked language modeling**:
> The perplexity definition used here is simply exponential of the pre-training test-set loss, and hence a direct measure of model pre-training performance. Perplexity for MLM has been used in other works, such as Roberta [1], Deberta[2], Megatron LM[3], Scaling Laws vs Model Architectures[4], or a perplexity variant in [5], and in [6].
>
> [1] Liu, Yinhan, et al. "Roberta: A robustly optimized bert pretraining approach." arXiv preprint arXiv:1907.11692 (2019).
>
> [2] He, Pengcheng, et al. "DEBERTA: DECODING-ENHANCED BERT WITH DISENTANGLED ATTENTION." International Conference on Learning Representations. 2020.
>
> [3] Shoeybi, Mohammad, et al. "Megatron-lm: Training multi-billion parameter language models using model parallelism." arXiv preprint arXiv:1909.08053 (2019).
>
> [4] Tay, Yi, et al. "Scaling laws vs model architectures: How does inductive bias influence scaling?." arXiv preprint arXiv:2207.10551 (2022).
>
> [5] Salazar, Julian, et al. "Masked Language Model Scoring." Proceedings of the 58th Annual Meeting of the Association for Computational Linguistics. 2020.
>
> [6] Lu, Jinghui, et al. "What Makes Pre-trained Language Models Better Zero-shot Learners?." Proceedings of the 61st Annual Meeting of the Association for Computational Linguistics (Volume 1: Long Papers). 2023.

---

> ### Author Response · Authors · 2023-11-20
> **Response to Reviewer eeiy : Part 4/4**
>
> ### **Regarding accuracy of numerical results**:
> Kindly refer to the response regarding 'approximation for the correlation in the embedding layer'. Our correlation estimates were within 3% of the ground truth value. Furthermore, as can be clearly seen from our formulae and from simulations, correlation is indeed very critical to correctly model signal propagation in transformers.
>
>
> ### **Regarding comparisons with competing methods**:
>
> We further discuss the methods the reviewer mentioned:
>
> **Zhang et al 2019**: We discuss DSInit’s signal propagation in 3.3, their scaling in B.3, and compare performance for two deep models in Table 5. Crucially, DSInit’s initialization fails to converge for the 96-layer model despite a more conservative scaling of $O(N^{-2})$, highlighting the impact of carefully considering the constants (from initialization) hidden away in asymptotic formulations.
>
> **Noci et al 2022**: We provide a detailed discussion of the Rank Collapse of Noci. et. al. in Section 3.4,  Appendix G highlighting some issues with their rank collapse. We also discuss some deficiencies in their signal propagation in 3.3 and B.2 - By not assuming constant attention, we arrive at exponential back-propagated gradients to Q,K, which we show can affect model stability, both theoretically and empirically. We also refer to their scaling in 3.4 and B.3. Their scaling corresponds to $\lambda=1$ and $\beta=k/N$. As we mentioned in B.3, we used $lambda^2 +\beta^2 = 1$ instead, as He. et al 2023 showed fully normalized residual connections often result in better performance. Our initial experiments also showed lower scores with $\lambda=1$.
>
> **Wang et al. 2022a**: We discuss DeepNorm’s signal propagation in 3.3, and compare performance for two deep models in Table 5. As we further discuss in B.3, DeepNorm shows performance improvements on making the model deeper but keeping the hidden dimension constant, with a larger model - whereas our method shows performance improvements on making the model deeper while keeping the parameters constant.
>
> **He et al. 2023**: We discuss the signal propagation theory of He et al in B.2, scaling in B.3. While He et al. considers the MLP to be linear, but by also considering the non-linearity in MLP we can correctly model how the nonlinearity of MLP strongly affects attention via correlation. We arrive at formulae which model the propagation accurately across the entire model, which cannot be done without considering the non-linearity. Their 72 and 108 layer models underperform compared to a 36 layer model with same hidden dimension, in spite of having many times extra parameters - compared to our method where deeper models outperform standard models without extra params, we hence did not benchmark this method. [2] also found He et al to underperform compared to baselines.
>
> We would like to further discuss 3 more methods for deeper models, which we investigated:
>
> **Admin (Liu et. al. 2020a)**: Admin requires additional “profiling” passes over the dataset to achieve unit variance for forward propagation, as we discuss in B.1. - while we can do this fully theoretically. We attempted to train our 192L-256D models of Table 5 with Admin, but all our experiments failed to converge, even after decreasing LR by 20x. As Admin paper only trained upto 72-layer models, we did not consider it for inclusion in Table 5.
>
> **SkipInit**: SkipInit (De & Smith, 2020) underperforms compared to baseline Pre-LN models for Language Modelling, as observed by He et. al. 2023. As our method outperforms baseline Pre-LN model, we did not perform comparisons to this method.
>
> **Shaped Attention**:  As per [2], Shaped Attention[1] does not perform as well as baseline models, resulting in significantly worse performance (Figure 2, 3, 5).
>
> [1] Noci, Lorenzo, et al. "The shaped transformer: Attention models in the infinite depth-and-width limit." arXiv preprint arXiv:2306.17759 (2023).
>
> [2] Anonymous, Simplifying Transformer Blocks, 2023. https://openreview.net/forum?id=RtDok9eS3s

---

> > ### Comment · Reviewer_eeiy · 2023-11-22
> > **Rebuttal acknowledgement**
> >
> > I thank the authors for their elaborate rebuttal and will consider it for my final decision.
> > I also like the color-coding idea to make clear which results are novel!
> >
> > A few questions remain after a quick read-through of the rebuttal:
> >  - The perplexity computations as detailed in appendix E of the megatron paper that you refer to seems to confirm that it is a metric for autoregressive predictions.
> >    In masked language modelling, you do not necessarily have all of the previous tokens if they have been masked.
> >   As a result, it is still not clear why perplexity makes sense in a masked language modelling setting.
> >   Does any of the listed papers address this discrepancy?
> >  - The rebuttal emphasises the need for chinchilla scaling laws to compare results from other works.
> >    I am aware of these scaling laws, but I have never seen them being used in this way.
> >    Is this a common way to compare models? If yes, I would expect at least one reference that reports similar results.
> >    Maybe the other reviewers might also be able to clear this for me, but the performance is so much worse than other reported results and I am not entirely convinced (yet) that these scaling laws can be used in this way.
> >  - Concerning the comparison with competing models: I would have been interested in an empirical comparison, but I understand if this is not feasible.

---

> > > ### Author Response · Authors · 2023-11-22
> > > **Regarding Questions from Rebuttal Acknowledgement**
> > >
> > > Thank you for your response! We would like to address your remaining concerns below -
> > >
> > > 1. The perplexity we use is simply the exponential of the model's loss on the validation data, as calculated in Megatron LM's default code [here](https://github.com/NVIDIA/Megatron-LM/blob/443ce9f3f98fdc5a53c6b480c6e21b79944d198e/megatron/training.py#L975). Calling this measure "perplexity" is indeed an abuse of notation, as previous words which are masked are not available, and future words are. However, this measure is still used in Roberta and Deberta (two very popular BERT-variants). Furthermore, [1] in Figure 5 shows MLM loss is well-correlated with downstream performance, and hence we use perplexity (exponential of MLM loss) as a measure to compare models.
> > > 2. We used the scaling laws in the rebuttal to give a ballpark estimate that our experimental losses are within the expected range. We will release the code, as well as exact scripts to recreate our models on public datasets, to allow others to verify that this is indeed the performance expected given the model and training steps. Another work, [1] (in Figure 1) shows that at 3.3B tokens (chinchilla optimal), their models/variants achieve MLM loss of approx 1.9-2.4 with optimized hyperparameters. Our models use much more noisier data (Common Crawl vs Wikipedia), and use original BERT hyper-parameters, to arrive at an MLM loss of 2.65.
> > > 3. Concerning empirical comparisons to other deep methods, we would clarify that in addition to DeepNet and DSInit, empirically we have Admin diverging, and our initial experiments showed Noci et al underperform. These results were omitted from the paper for brevity, which we provide below for Bert-Large 330M params 24L-1024D -
> > > Noci et al.’s Scaling - 14.9
> > > DSLM Scaling - 14.0
> > >
> > > [1] Geiping, Jonas, and Tom Goldstein. "Cramming: Training a Language Model on a single GPU in one day." International Conference on Machine Learning. PMLR, 2023.

---

### Official Review · Reviewer_vaGo · 2023-11-10

**Soundness:** 3 good
**Presentation:** 2 fair
**Contribution:** 3 good
**Rating:** 6
**Confidence:** 3

**Summary:**

This paper derives a closed form signal propagation formula, i.e., mean, variance, and input-output correlation, for each component in Transformer, including both forward and backward passes. It helps explain gradient vanishing or explosion, rank collapse, and training instability of Transformer. Leveraging these formulas, the paper then proposes a new initialization scheme, DeepScaleLM, that stabilizes the signal propagation across training. Experimental results show that with the proposed initialization scheme, Transformer with 100s of layers can be trained and obtains better results than a shallow counterpart.

**Strengths:**

The paper studies a fundamental problem of scaling neural nets in depth. The study follows the direction of signal propagation, which is promising. The derived mean and variance formulas are shown to be matching the empirical simulation. Experimental results are also encouraging, showing depth is indeed helpful for Transformer as well.

**Weaknesses:**

1. The main issue probably comes from the model selection in the experiment section. Although BERT experiments are good, an additional evaluation using an encoder-decoder Transformer or decoder-only Transformer would provide more interesting spikes and make the justification more compelling.

2. The concrete picture of the adjusted model architecture is missing to the reader. The proposed DeepScaleLM scheme scales residual connections, and seems like it adds extra dropout layers. In which component are these adjustments applied? Part of a layer or every component in a layer? Partial layers or every layer?

**Questions:**

1. Plotting log-scale in Figure 3 does not help the reader understand deeply about the gap between empirical simulation and theoretical prediction. Providing statistics such as min, max, mean, variance of the gap would be better.

2. In Section 2, the paper assumes normal distribution of inputs, weights, and gradients while deriving the formulas in Table 1. To what extent this assumption holds since in Section 1 paragraph 2 the paper mentioned that some of the assumptions in prior works broke down on real world data?

---

> ### Author Response · Authors · 2023-11-20
> **Response to Reviewer vaGo**
>
> ### **Regarding more experiments with decoder-only Transformer, and new finetuning datasets**:
> Kindly refer to the common response section to find new results.
>
> ### **Regarding details about adjusted model architecture**:
> We do not add any new dropout over the original standard transformer architecture of BERT - we show the critical importance of dropout (which are already present in standard transformer) in stabilizing very deep transformers by preventing rank collapse. The only changes we made were
>
> 1) Scaling of residual and skip connection, and
> 2) Changing the standard deviation of initialization of weights of model params.
>
> ### **Regarding clarity and statistics for Figure 3**:
> To clarify, Figure 3 shows the variance of gradient across different layers for a given model with 192 layers- it shows the gradient flowing backwards decreases exponentially. We will clarify this in the paper. We provide percentage relative error of our predicted gradient variance compared to the empirical values below:
>
> | Mean Error | Median Error | Std Error |
> | ------------------|---------------------|-----------------------------------|
> |    6.8\% | 5.2\% | 7.8\% |
>
> Furthermore,  we also measure the $R^2$ of our predicted gradient as a goodness of fit measure, and we find and $R^2$ of 0.998 - as can be seen, our predictions provide a good fit. We will add these goodness of fit measures to the main paper.
>
> ### **Regarding the assumption of normally distributed inputs, weights, and gradients**:
> **Inputs**: As the embeddings are lookup tables of token-ids, and embedding weights are initialized from normal distribution in xavier, the inputs to the transformer are normally distributed. We will state this clearly in the paper. However, while the inputs are normal, due to the repetition of the tokens / segment ids, the inputs are not IID, as many previous works have assumed - our work handles this explicitly.
>
> **Weights**: Weights are initialized from normal distribution in xavier, and are hence normal.
>
> **Gradients**: As the model outputs are gaussian, the softmax of the classification head results in a Log-Normal distribution for probabilities $p$, as shown in C.7. Since the cross-entropy loss is $-log(p)$, this results in the loss (and hence the final gradient being back-propagated) being log(Log-normal distribution), which is normal distribution.
>
> We further verify this empirically by checking the normality of the backpropagated gradients to the deepest transformer layer, and the gradients match the best-fit normal distribution with an $R^2$ of $0.999$, showing that the gradients are indeed normally distributed. We will add this discussion on normality to the main paper.

---

> ### Author Response · Authors · 2023-11-22
> **Request for Rebuttal Feedback**
>
> Dear Reviewer,
>
> We would like to thank you for reviewing our paper. With less than 1 day left, we would appreciate if you could provide feedback regarding whether our response has addressed your concerns. Kindly let us know if there are any other details you would like us to clarify!
>
> Best regards,
> The Authors

---

### Author Response · Authors · 2023-11-20
**Common Response to all the Reviewers**

We would like to thank all reviewers for detailed feedback. Based on the reviewer suggestions, we conducted experiments applying our method to new architectures, modalities, and downstream tasks. Additionally, we also report the effects of our method on model quantization.

### **More Finetuning Experiments - MNLI Results**:

As suggested by reviewers, we run further finetuning experiments on the MNLI benchmark. We finetune the best baseline and best DSLM models for each model size.

| Model Size | Baseline  | DSLM |
| ------ | ------ |  ------ |
| 165M | 79.8 | 81.1 |
| 330M | 80.6 | 81.6 |

### **Decoder-only architecture Experiments - GPT Results**:

We applied DSLM to the decoder only GPT model, trained for 8B tokens (slightly more than chinchilla-optimal). Similar to Bert, we increased the number of layers. Deeper models trained with DSLM provide a improvement in performance. We have searched LR for the baseline models, but did not do LR hyper-parameter search for DSLM due to compute and time limitations.

| Model Size | Baseline Shape | DSLM Shape| Baseline | DSLM |
| ------ | ------ | ------ |  ------ |  ------ |
| 165M | 12L-1024D | 48L-512D | 11.6 | 11.3 |
| 330M | 24L-1024D | 96L-512D | 10.4 | 10.1 |


### **Additional Modality - Vision Transformers**:

We applied our method to a 192 layer ViT model. Our method successfully stabilizes vision transformer with 100s of layers (for 192 layers, the forward and backward variances remain constrained). We plotted the figure for ViT corresponding to Figure 4 for 192 layer model using ImageNet data. The figure can be found at this [link](https://github.com/AnonNoNameAccount/Transformers_Get_Stable/blob/main/Figure_1.png).


Our method can be directly applied to vision transformers, and preserve forward and backward for ViT. All our derivations of transformer components, blocks, layers, and even entire model are fully agnostic of the modality of the input. Applying our method to vision transformers (for eg. ViT or DeiT) will only require handling the input embeddings section (C.1) - For ViT, this is a simple linear projection. Given normalized image inputs, our Linear section (C.2) provides formulae to calculate the variance and correlation of the embeddings which are input to the model.


We empirically verified that for images from ImageNet, the embeddings after the linear projection (for normalized images) do indeed follow the normal distribution, with an $R^2$ of 0.95. Furthermore, as normalized images have approximately unit variance and zero mean, given linear weights initialized by $\sqrt{\frac{1}{d}}$, the output variance was observed as 1.02 (within 2% error). While we used Zipf’s law to estimate input embedding correlation for text, this could simply be empirically measured for vision after the embedding layer - we measured this to be 0.258, very similar to the value of 0.22 for NLP inputs for BERT.


### **Model Quantization with DSLM**:

Similar to Unit Scaling [1], our method results in models which lose much less performance, when quantized (via direct casting) to FP8 precision compared to original models. We apply 8 bit quantization to the Bert baseline model and the model trained with DSLM. The table below provides the performance corresponding to the full precision inference and fp8 inferences (two different fp8 standards E5M2 and E4M3). DSLM model can be compressed to ¼ the original size with significantly lower performance loss.

| Model | fp32 Score     | fp8 E5M2 Score | fp8 E5M2 $\Delta$ | fp8 E4M3 Score | fp8 E4M3 $\Delta$ |
| ------ | ------ | ------ |  ------ | ------ |  ------ |
|Bert Baseline | 14.8 | 42.5 | 27.7 | 16.5 | 1.7 |
| DSLM Model | 13.1 |  21.4   | 8.3 | 13.9 | 0.8 |

### **Architectural changes of DSLM**:

The scaling parameters introduced by DSLM can be fully absorbed into the model checkpoint weights and  do not require any changes to the inference code. As inference code maybe highly optimized / hardware-specific, compatibility with vanilla transformers during inference will make deployment of DSLM method easy. The scaling parameters are absorbed into the checkpoints by scaling layernorm gain and output linear weights.

---

### Meta-Review · Area_Chair_vvfx · 2023-12-06

**Metareview:**

The paper presents an analysis of signal propagation in transformer networks, introducing a new initialization scheme, DeepScaleLM, which enables training of deeper transformer models with improved performance. The work combines theoretical insights with empirical validation, showing advantages in training stability and performance.

Strengths:
- Tries to address a fundamental issue in scaling neural networks, providing an analysis of signal propagation in transformers.
- Introduces DeepScaleLM, a novel initialization scheme.

Weaknesses:
- Overall novelty is fairly low.
- Lacks comprehensive model evaluation, particularly missing assessments on encoder-decoder and decoder-only transformers.
- Fails to provide a clear picture of the model architecture adjustments, particularly regarding the application of DeepScaleLM.
- Some methodological aspects, such as the use of perplexity in masked language modeling and baseline performance comparisons, are questionable.

**Justification For Why Not Higher Score:**

N/A

**Justification For Why Not Lower Score:**

N/A

---

### Decision · Program_Chairs · 2024-01-16

Reject